

# Generalized charges, part II: Non-invertible symmetries and the symmetry TFT

**Lakshya Bhardwaj and Sakura Schäfer-Nameki**

Mathematical Institute, University of Oxford,
Andrew-Wiles Building, Woodstock Road, Oxford, OX2 6GG, UK

## Abstract

Consider a $d$-dimensional quantum field theory (QFT) $\mathfrak{T}$, with a generalized symmetry $\mathcal{S}$, which may or may not be invertible. We study the action of $\mathcal{S}$ on generalized or $q$-charges, i.e. $q$-dimensional operators. The main result of this paper is that $q$-charges are characterized in terms of the topological defects of the Symmetry Topological Field Theory (SymTFT) of $\mathcal{S}$, also known as the "Sandwich Construction". The SymTFT is a $(d+1)$-dimensional topological field theory, which encodes the symmetry $\mathcal{S}$ and the physical theory in terms of its boundary conditions. Our proposal applies quite generally to any finite symmetry $\mathcal{S}$, including non-invertible, categorical symmetries. Mathematically, the topological defects of the SymTFT form the Drinfeld Center of the symmetry category $\mathcal{S}$. Applied to invertible symmetries, we recover the result of Part I [1] of this series of papers. After providing general arguments for the identification of $q$-charges with the topological defects of the SymTFT, we develop this program in detail for QFTs in 2d (for general fusion category symmetries) and 3d (for fusion 2-category symmetries).

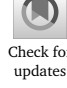

## Contents

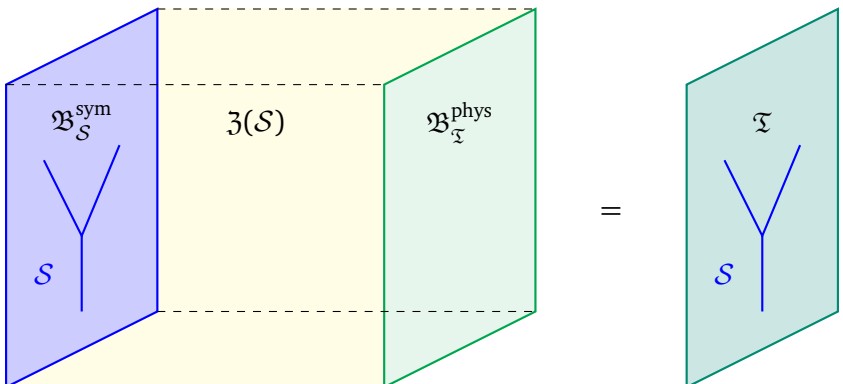

Figure 1: The sandwich construction: a theory $\mathfrak{T}$ in $d$ dimensions with global symmetry $\mathcal{S}$ can be obtained as an interval compactification of a $d+1$ dimensional SymTFT $\mathfrak{Z}(\mathcal{S})$ with two boundary conditions: $\mathfrak{B}_{\mathcal{S}}^{\mathrm{sym}}$ is the topological symmetry boundary, where all the symmetry structure is localized, and $\mathfrak{B}_{\mathfrak{T}}^{\mathrm{phys}}$ is the physical boundary. After the interval compactification, the topological defects (drawn as blue lines) of $\mathfrak{B}_{\mathcal{S}}^{\mathrm{sym}}$ become topological defects of the theory $\mathfrak{T}$ that generate the $\mathcal{S}$ symmetry of $\mathfrak{T}$.

# 1 Introduction and summary of main results

Consider a $d$-dimensional QFT $\mathfrak{T}$. As is by now very well known and established, the generalized global symmetries of $\mathfrak{T}$ are described in terms of topological defects of various dimensions in $\mathfrak{T}$ [2]. Let $\mathfrak{T}$ admit a (generalized) symmetry[1] $\mathcal{S}$. Throughout this paper we assume that $\mathcal{S}$ is a discrete, as opposed to a continuous, symmetry. This symmetry can be a higher-form symmetry, a higher-group, or a non-invertible symmetry, with a well-defined set of rules how to compose symmetry generators. Although, mathematically speaking, $\mathcal{S}$ has the structure of a fusion $(d-1)$-category, which captures these properties of topological defects of $\mathfrak{T}$, we will make an effort to make this language well-motivated from a physics point of view.

The goal of this paper is to describe the charges under generalized symmetries. Put differently, we characterize the physical operators in the theory $\mathfrak{T}$, which transform under the symmetry $\mathcal{S}$.

For ordinary symmetries that form groups, the point-like operators that they act on should transform in representations. Likewise, the analog extension to $p$-form symmetries reads, that $p$-dimensional operators transform in representations of the $p$-form symmetry group. However, already for such group-like (invertible) 0-form and higher-form symmetries, we have shown in Part I [1], see also [3], of this series of papers that representations comprise only a subset of possible charges. Generically, there can be charges for $p$-form symmetries that are of dimension $q \geq p$.

In this paper we extend the scope and allow also for non-invertible symmetries, whose study in particular in $d \geq 3$ has recently been initiated in [4–10] from various perspectives. At this point we will simply allow the symmetry $\mathcal{S}$ to be any fusion higher-category (that describes non-invertible symmetries in $d \geq 3$).

It is useful to introduce some terminology. Generalized charges describe the possible ways generalized global symmetries can act on operators in a QFT. We further characterize generalized charges as follows [1]:

---

[1]We will frequently drop the adjectives "generalized" and "global" in this paper, and refer to "generalized global symmetries" simply as "symmetries".

---

**Definition 1.1: Definition**

Generalized charges of $q$-dimensional operators are referred to as **$q$-charges**.

---

In the current paper, which is Part II in this series, we discuss generalized charges for non-invertible symmetries. Whilst in Part I it was possible to easily derive the $q$-charges from first principles using the structure of symmetry defects, in the non-invertible realm it will be extremely useful to utilize the so-called Symmetry TFT (SymTFT) [11] (and [12–14] for earlier discussions), which is a $d + 1$-dimensional topological field theory, whose precise properties we will discuss in detail. The SymTFT has two boundary conditions: a topological one, which encodes the symmetries, and a physical one. This is also known as the "sandwich construction".

The main statement that we will put forward in the current paper and substantiate from multiple vantage points is the following:

---

**Main Statement: Generalized charges from SymTFT**

The $q$-charges, with $0 \leq q \leq d - 2$, of a symmetry $\mathcal{S}$ are $(q + 1)$-dimensional topological operators

$$Q_{q+1}, \tag{1}$$

genuine or non-genuine, of the Symmetry TFT $\mathfrak{Z}(\mathcal{S})$.
Put differently, the $q$-charges are the topological defects of the Drinfeld center of the symmetry $\mathcal{S}$.

---

Thus, as with any sandwich (SymTFT), the flavor (charges) comes from the (Drinfeld) center. The remainder of this introduction is an overview of each of the concepts in this statement, and illustrative examples.

**The SymTFT.** Given a theory $\mathfrak{T}$ with symmetry $\mathcal{S}$, there is a SymTFT $\mathfrak{Z}(\mathcal{S})$ [11–14], which is a $(d+1)$-dimensional TQFT. Its interval compactification reduces back to the original theory $\mathfrak{T}$. See figure 4. Conventionally, the left boundary condition $\mathfrak{B}_{\mathcal{S}}^{\text{sym}}$ is topological, while the right boundary condition $\mathfrak{B}_{\mathfrak{T}}^{\text{phys}}$ is either topological or non-topological depending on whether the QFT $\mathfrak{T}$ is topological or non-topological respectively. The topological defects living inside the worldvolume of $\mathfrak{B}_{\mathcal{S}}^{\text{sym}}$, unattached to any topological defects of the bulk TQFT $\mathfrak{Z}(\mathcal{S})$, describe the symmetry $\mathcal{S}$.

The proposal can be summarized in terms of the following statement (we will provide a more refined statement of this "sandwich" construction later on in statement 2.1):

After the interval compactification, the topological defects of $\mathfrak{B}_{\mathcal{S}}^{\text{sym}}$ become precisely the topological defects of $\mathfrak{T}$ of the symmetry $\mathcal{S}$. See figure 5. Let us note the following points:

- The symmetry $\mathcal{S}$ determines the pair $(\mathfrak{Z}(\mathcal{S}), \mathfrak{B}_{\mathcal{S}}^{\text{sym}})$. Importantly, this is independent of the theory $\mathfrak{T}$ that we start with. That is, this pair is same if the symmetry $\mathcal{S}$ is realized in any other $d$-dimensional QFT $\mathfrak{T}'$.

- On the other hand, the boundary condition $\mathfrak{B}_{\mathfrak{T}}^{\text{phys}}$ depends on the identity of the $d$-dimensional QFT $\mathfrak{T}$. That is, we have

$$\mathfrak{B}_{\mathfrak{T}}^{\text{phys}} \neq \mathfrak{B}_{\mathfrak{T}'}^{\text{phys}}, \tag{2}$$

if $\mathfrak{T} \neq \mathfrak{T}'$.

We will adopt the following terminology for the TQFT $\mathfrak{Z}(\mathcal{S})$, which has become quite common in the community, which depends solely on the symmetry $\mathcal{S}$:

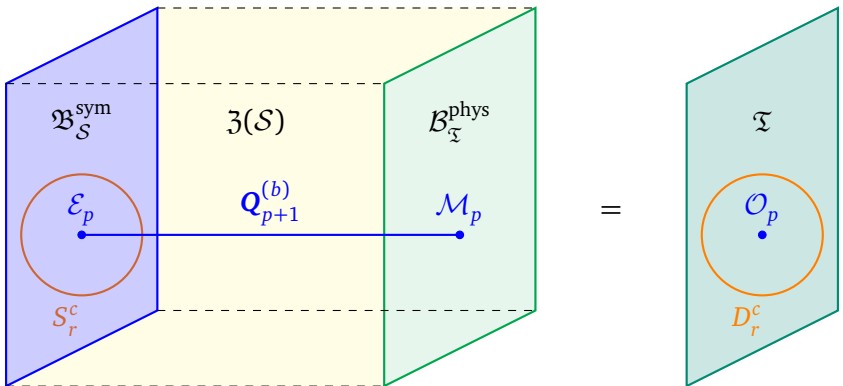

Figure 2: SymTFT for BF type theories with $p$-form symmetry. The bulk topological defect $\boldsymbol{Q}_{p+1}^{(b)}$ has Dirichlet boundary condition on the symmetry boundary and ends in a $p$-dimensional operator $\mathcal{E}_p$, and $\boldsymbol{Q}_{r=d-p-1}^{(c)}$ has Neumann boundary condition, as a consequence of which it projects onto the symmetry boundary to become a non-trivial topological defect $S_r^c$. After interval compactification, the above picture shows precisely the non-trivial linking between the charge $\mathcal{O}_p$ and the symmetry defect $D_r^c$.

---

**Definition 1.2: Definition: SymTFT**

**Symmetry TFT** (or **SymTFT** for short) associated to a symmetry $\mathcal{S}$ is the TQFT $\mathfrak{Z}(\mathcal{S})$ described above.

An inherent definition of the $(d+1)$-dimensional symmetry TFT $\mathfrak{Z}(\mathcal{S})$ is that it admits a $d$-dimensional topological boundary condition $\mathfrak{B}_{\mathcal{S}}^{\text{sym}}$, whose topological operators form $\mathcal{S}$, a fusion $(d-1)$-category. In the language of [8, 15], $\mathcal{S}$ is the **symmetry category** of $\mathfrak{B}_{\mathcal{S}}^{\text{sym}}$.

---

We re-emphasize that SymTFT is defined without reference to the theory $\mathfrak{T}$. An intuitive – and sometimes also constructive – way to think about the SymTFT is as a type of "gauging of $\mathcal{S}$ in $d+1$ dimensions".

---

**Example 1.1: SymTFT as a BF-theory**

Lets make this precise in the case of an abelian $p$-form symmetry $G^{(p)} = \mathbb{Z}_N$: in this instance we couple the theory to a $(p+1)$-form gauge field of a $\mathbb{Z}_N$ gauge theory in $d+1$ dimensions. This can be formulated in two ways. In terms of a continuous, $U(1)$-valued $(p+1)$-form field $b_{p+1}$ and its dual $c_{d-p-1}$, the SymTFT has action

$$S_{\mathfrak{Z}(\mathcal{S})} = \frac{i}{2\pi} N \int_{M_{d+1}} b_{p+1} \wedge dc_{d-p-1} \,. \tag{3}$$

This is a Dijkgraaf-Witten theory for $\mathbb{Z}_N$ or a BF-theory. We will often also use the formulation using cochains:[2] A cochain $b_{p+1} \in C^{p+1}(M_{d+1}, \mathbb{Z}_N)$ and its dual $c_{d-p-2}$ with action

$$S_{\mathfrak{Z}(\mathcal{S})} = \int_{M_{d+1}} b_{p+1} \cup \delta c_{d-p-1} \,. \tag{4}$$

---

The topological defects of this SymTFT are

$$Q_{p+1}^{(b)} = e^{i \int_{M_{p+1}} b_{p+1}}, \qquad Q_{d-p-1}^{(c)} = e^{i \int_{M'_{d-p-1}} c_{d-p-1}}. \tag{5}$$

The topological boundary condition, which yields the $\mathbb{Z}_N^{(p)}$ $p$-form symmetry of the original theory, is

$$
\begin{aligned}
b_{p+1}: & \quad \text{Dirichlet}, \\
c_{d-p-1}: & \quad \text{Neumann}.
\end{aligned} \tag{6}
$$

In particular this means that the topological operator $Q_{p+1}^{(b)}$ ends on the symmetry boundary $\mathfrak{B}_{\mathcal{S}}^{\text{sym}}$ whereas $Q_{d-p-1}^{(c)}$ projects to a topological defect $S_{d-p-1}$ in $\mathfrak{B}_{\mathcal{S}}^{\text{sym}}$, which will after the interval compactification, generate the $p$-form symmetry.

Due to the BF-coupling, these operators are not commutative, but satisfy

$$Q_{p+1}^{(b)}(M) Q_{d-p-1}^{(c)}(M') = \exp\left(\frac{2\pi i L(M, M')}{N}\right) Q_{d-p-1}^{(c)}(M') Q_{p+1}^{(b)}(M). \tag{7}$$

This in turn means that on the symmetry boundary, the operator $S_{d-p-1}^c$ will link non-trivially with the end of the bulk topological operator, $Q_{p+1}^{(b)}$, which is a $p$-dimensional defect $\mathcal{E}_p$. This is depicted in figure 2. After interval compactification, we obtain the theory $\mathfrak{T}$ with symmetry generated by $D_{d-p-1}^c$ (which is the image of $S_{d-p-1}^c$ under the interval compactification), and $p$-charge $\mathcal{O}_p$.

Although the above BF-action is familiar, e.g. from QFT, holography or even string theory, it is only a very special example of a SymTFT. In particular the simple recovery of the symmetry $\mathcal{S} = \mathbb{Z}_N^{(p)}$ in terms of imposing Dirichlet/Neumann b.c. on the fields $b$ and $c$ is in general not as straight-forward. We will illustrate this now.

---

**Example 1.2: SymTFT for 2d theories with $G^{(0)} = S_3$**

Consider a 2d theory, with non-anomalous 0-form symmetry, given by a non-abelian, finite group $G^{(0)} = S_3$ the permutation group of three elements. This is parameterized by

$$G^{(0)} = S_3 = \{\text{id}, a, a^2, b, ab, a^2 b\}, \tag{8}$$

with relationships

$$a^3 = b^2 = \text{id}, \qquad ba = a^2 b. \tag{9}$$

To construct the SymTFT we also require the knowledge of its representations, which are denoted by $+$ (trivial), $-$ (sign), $E$ (2d representations). As this group is a semi-direct product $S_3 = \mathbb{Z}_3 \rtimes \mathbb{Z}_2$ we can write a BF-type action for $\mathbb{Z}_3$ and then implement the outer automorphism $\mathbb{Z}_2$ action. However, for a general non-abelian finite group this is not possible, so we shall discuss the SymTFT from a different perspective.

Mathematically, starting with a symmetry $\mathcal{S}$ we can directly compute the topological defects of the SymTFT (without constructing the theory itself).

For an **abelian** group-like symmetry, the topological defects of the SymTFT will be labeled by a group element $g \in G$ and a representation $R$. For 2d theories, these are

---

[2] For a discussion how to map between these, see [16].

topological lines in the 3d SymTFT

$$Q_1^{(g,R)}.\tag{10}$$

In this abelian case, we can recover the original symmetry $\mathcal{S} = G^{(0)}$ by restricting to the lines $R = 1$, as the boundary conditions along the symmetry boundary are

$$
\begin{aligned}
Q_1^{(1,R)}: & \quad \text{Dirichlet,} \\
Q_1^{(g,1)}: & \quad \text{Neumann.}
\end{aligned}
\tag{11}
$$

If the lines $Q_1^{(1,R)}$ end on physical boundary, they give rise to genuine local operators (genuine 0-charges) transforming in a representation $R$. If the lines $Q_1^{(g\neq\mathrm{id},R)}$ end on physical boundary, they give rise to twisted sector local operators which are attached to $g$-topological lines and additionally transform in representation $R$. These considerations are in complete agreement with the BF-theory analysis in example 1.1.

However, for **non-abelian** $G^{(0)}$ in 2d, the topological defects of the SymTFT will be shown to be labeled by

- conjugacy classes $[g]$,

- representations of the stabilizer group $H_g$ of $g \in [g]$.

For $G = S_3$ there are three conjugacy classes

$$
\begin{aligned}
[\mathrm{id}], & \quad H_{\mathrm{id}} = S_3, \\
[a], & \quad H_a = \{\mathrm{id}, a, a^2\} = \mathbb{Z}_3, \\
[b], & \quad H_b = \{\mathrm{id}, b\} = \mathbb{Z}_2.
\end{aligned}
\tag{12}
$$

The representations of $H_a = \mathbb{Z}_3$ are 1d and characterized by third roots of unity 1, $\omega = e^{2\pi i/3}$, $\omega^2$. Likewise the representations of $H_b = \mathbb{Z}_2$ are labelled by $\pm$.

The topological lines of the SymTFT are then

$$
\begin{aligned}
Q_1^{([\mathrm{id}],R)}: & \quad R = 1, -, E, \\
Q_1^{([a],R)}: & \quad R = 1, \omega, \omega^2, \\
Q_1^{([b],R)}: & \quad R = \pm.
\end{aligned}
\tag{13}
$$

The first observation here is that the original symmetry $\mathcal{S}$ is not obviously a straightforward b.c. on a subset of lines, unlike the abelian case. We will see that this requires studying so-called algebras and their bimodules. This example demostrates very nicely, that we need to consider a formulation of the SymTFT beyond abelian BF-type theories.

**Generalized charges.** The main result of this paper is the identification of the generalized charges under the symmetry $\mathcal{S}$ with the topological defects of the SymTFT. We will now add some more substance to the main claim of the paper, statement 1.

Let us consider here the case of a $q$-charge of $\mathcal{S}$ associated to a genuine $(q+1)$-dimensional topological operator $Q_{q+1}$ of $\mathfrak{Z}(\mathcal{S})$. The generalization to non-genuine $Q_{q+1}$ is similar. Operators in $\mathfrak{T}$ transforming in the $q$-charge $Q_{q+1}$ are constructed in the SymTFT setup by performing an interval compactification of $Q_{q+1}$ as shown in figure 3. Let us pick:

$$
\begin{aligned}
\mathcal{E}_q = & \text{a topological } q\text{-dimensional operator at the end of } Q_{q+1} \text{ along } \mathfrak{B}_{\mathcal{S}}^{\mathrm{sym}}, \\
\mathcal{M}_q = & \text{a non-topological } q\text{-dimensional operator at the end of } Q_{q+1} \text{ along } \mathfrak{B}_{\mathfrak{T}}^{\mathrm{phys}}.
\end{aligned}
\tag{14}
$$

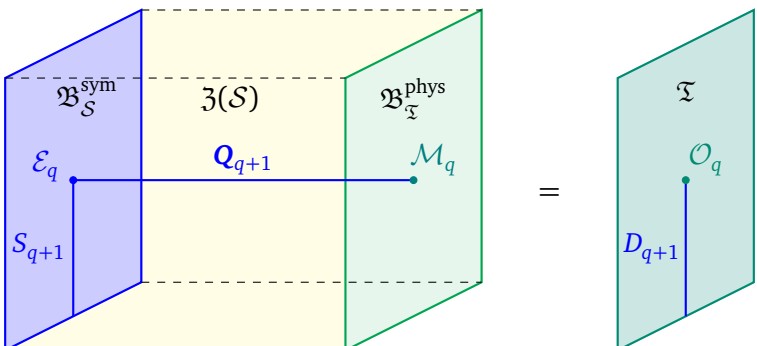

Figure 3: Construction of the $q$-charges from bulk topological operators $\mathbf{Q}_{q+1}$: The topological operator of the SymTFT $\mathbf{Q}_{q+1}$ ends on the symmetry boundary $\mathfrak{B}^{\text{sym}}_{\mathcal{S}}$ as well as on the physical boundary $\mathfrak{B}^{\text{phys}}_{\mathfrak{T}}$: in a $q$-dimensional topological operator $\mathcal{E}_q$ and a non-topological operator $\mathcal{M}_q$, respectively. After interval compactification, this results in a $q$-charge $\mathcal{O}_q$. The operator can also be a twisted sector for the symmetry $\mathcal{S}$, i.e. attached to topological defects $D_{q+1}$ in $\mathcal{S}$. This occurs whenever $\mathcal{E}_q$ in $\mathfrak{B}^{\text{sym}}_{\mathcal{S}}$ forms a junction between $\mathbf{Q}_{q+1}$ and $S_{q+1}$, which is a topological defect in $\mathcal{S}$.

If there does not exist an end $\mathcal{M}_q$ of $\mathbf{Q}_{q+1}$ along $\mathfrak{B}^{\text{phys}}_{\mathfrak{T}}$, then the $q$-charge $\mathbf{Q}_{q+1}$ is not carried by any $q$-dimensional operator in the theory $\mathfrak{T}$. The operator $\mathcal{E}_q$ may be a twisted sector operator, which is attached to a symmetry defect $S_{q+1} \in \mathcal{S}$. Upon interval compactification, this results in a $q$-charge $\mathcal{O}_q$ (which may be a twisted sector operator, if $\mathcal{E}_q$ was attached to $S_{q+1}$), see figure 3.

The topological operators of the SymTFT of dimension less than or equal to $p$ form a $p$-category. This immediately leads to the following conclusion:

---

**Statement 1.1: Higher-categorical structure of generalized charges**

All $q$-charges (for $0 \leq q \leq d-2$) of a symmetry $\mathcal{S}$ combine to form the structure of a $(d-1)$-category $\mathcal{Z}(\mathcal{S})$ of topological defects of codimension greater than or equal to 2 of the SymTFT $\mathfrak{Z}(\mathcal{S})$.

Mathematically, the $(d-1)$-category $\mathcal{Z}(\mathcal{S})$ of topological defects of the SymTFT $\mathfrak{Z}(\mathcal{S})$ is known as the **Drinfeld center** of the symmetry fusion $(d-1)$-category $\mathcal{S}$.

---

We will now consider the two examples from before: the abelian BF-theory as well as the 2d theory with 0-form symmetry $S_3$, to illustrate these statements.

---

**Example 1.3: Generalized charges for BF-theories**

The topological defects of the SymTFT for a $\mathbb{Z}^{(p)}_N$-form symmetry in (5). To see whether these are all the defects, lets restrict to $d=3$ and $p=0$ so the SymTFT is

$$S_{3(\mathcal{S})} = \int_{M_4} b_1 \cup \delta c_2 \,. \tag{15}$$

The standard topological defects are

$$\mathbf{Q}^{(b)}_1 = e^{i \int_{M_1} b_1} \,, \qquad \mathbf{Q}^{(c)}_2 = e^{i \int_{M_2} c_2} \,. \tag{16}$$

---

In addition we can also consider topological defects that are condensation defects, or theta-defects, which correspond to inserting a mesh of lines $Q_1^{(b)}$ on a 2d surfaces

$$C\left(Q_2^{(\mathrm{id})}(M_2), Q_1^{(b)}\right) = \frac{1}{\sqrt{|H_1(M_2, \mathbb{Z}_N)|}} \sum_{M_1 \in H_1(M_2, \mathbb{Z}_N)} Q_1^{(b)}(M_1) Q_2^{(\mathrm{id})}(M_2). \quad (17)$$

The Drinfeld center for this symmetry contains both this defect as well as the defect where the condensation is applied to $Q_2^{(c)}$

$$C\left(Q_2^{(c)}(M_2), Q_1^{(b)}\right) = \frac{1}{\sqrt{|H_1(M_2, \mathbb{Z}_N)|}} \sum_{M_1 \in H_1(M_2, \mathbb{Z}_N)} Q_1^{(b)}(M_1) Q_2^{(c)}(M_2). \quad (18)$$

We will see in this paper from a different perspective (using properties of fusion 2-categories) that these defects are the complete set of simple objects in the Drinfeld center of $\mathcal{S}$.

That the condensation defects have to be present in the bulk SymTFT is also clear from the perspective of the dual boundary conditions, when we have a $\mathbb{Z}_N^{(1)}$ 1-form symmetry: $b_1$ Neumann and $c_2$ Dirichlet. With this boundary condition we have symmetry generators, that are projections of the lines $Q_1^{(b)}$ parallel to the boundary. The 3d theory with these boundary conditions has a 1-form symmetry, and we can form the condensation defects in the 3d theory by gauging the 1-form symmetry on a 2d surface. The above SymTFT condensation defects project precisely to the condensation defects in the 3d theory.

An even more manifest way to include these defects is to revisit the construction of the SymTFT: we started off with a 3d theory $\mathfrak{T}$ with 0-form symmetry $\mathbb{Z}_N^{(0)}$. The SymTFT is obtained by coupling the theory to a 4d $\mathbb{Z}_N$-gauge theory, and gauging the symmetry $\mathcal{S} = \mathbb{Z}_N^{(0)}$. Before gauging we can stack the theory with $\mathcal{S}$-symmetric TQFTs [9]. Including these in the construction of the SymTFT means we get additional couplings, which are localized on submanifolds

$$S_{3(\mathcal{S})}^{\mathrm{theta}} = \left( \int_{M_4} b_1 \cup \delta c_2 + \int_{M_2} a_1 \cup \delta a_0' + b_1 \cup a_1 \right). \quad (19)$$

This corresponds to coupling the BF-theory in addition to localized $\mathbb{Z}_N^0$-form gauge theories on 2d submanifold $M_2$ in the 4d worldvolume of the SymTFT. After the coupling we obtain precisely the condensation defects of the 4d SymTFT discussed above.

### Example 1.4: Generalized charges for $G^{(0)} = S_3$ in 2d.

The topological defects in the SymTFT for the 0-form symmetry $S_3$ in 2d are listed in (13). The charges of the form $Q_1^{([\mathrm{id}], R)}$ end on symmetry boundary and give rise after interval compactification to genuine local operators in the representation $R$. The lines with non-trivial conjugacy classes $Q_1^{[g], R}$ give rise to twisted sector operators after the interval compactification: such operators are attached to topological line operators generating $S_3$ symmetry.

**Gauging and the Drinfeld center.** Gauging a (non-anomalous) part of the symmetry $\mathcal{S}$ results in another symmetry $\mathcal{S}'$. We will see that gauging will not change the SymTFT and its topological defects (i.e. the Drinfeld center), i.e. if a theory $\mathfrak{T}$ with symmetry $\mathcal{S}$ maps under a gauging operation to a theory with symmetry $\mathcal{S}'$ then

$$\mathfrak{Z}(\mathcal{S}) = \mathfrak{Z}(\mathcal{S}') \,. \tag{20}$$

Furthermore this means that the generalized charges do not change! The only change in the sandwich is

$$\mathfrak{B}_{\mathcal{S}}^{\text{sym}} \;\rightarrow\; \mathfrak{B}_{\mathcal{S}'}^{\text{sym}} \,, \tag{21}$$

whereas we keep the SymTFT and the physical boundary $\mathfrak{B}_{\mathfrak{T}}^{\text{phys}}$ unchanged. The squeezed sandwich gives of course a different physical theory, related by gauging to the original theory. Even though the generalized charges are unchanged, what does change due to (21) is the separation into twisted and untwisted sector charges.

Lets revisit this briefly in the case of the abelian BF-theory. Changing $\mathfrak{B}^{\text{sym}}$ means we change the boundary conditions on the finite gauge fields $b$ and $c$. E.g. we considered the boundary conditions (6) with $b_{p+1}$ Dirichlet and $c_{d-p-1}$ Neumann realizing theory with $\mathbb{Z}_N^{(p)}$ $p$-form symmetry. Gauging the latter is simply realized in terms of a change in boundary conditions and results in a $\widehat{\mathbb{Z}}_N$ $(d-p-2)$-form symmetry

$$\widehat{\mathbb{Z}}_N^{(d-p-2)} : \qquad \begin{cases} b_{p+1} \,, & \text{Neumann,} \\ c_{d-p-1} \,, & \text{Dirichlet.} \end{cases} \tag{22}$$

**Plan of the paper.** After this introduction and overview of the main results, we will now proceed with the main text. In section 2 we discuss general aspects of symmetries, and define the Symmetry TFT. Furthermore a detailed exposition of the sandwich conjecture is provided. We exemplify the SymTFT for invertible and non-invertible symmetries. We also show that we can use the SymTFT to classify all $\mathcal{S}$-symmetric TQFTs in section 2.8, which will have important applications in [17, 18].

Section 3 contains the main result of the paper: the identification of the generalized charges (or $q$-charges) with the topological defects of the SymTFT. The latter are mathematically characterized in terms of the Drinfeld center of the symmetry category $\mathcal{S}$. We discuss how the bulk topological defects can be identified with charges, and how to compute the action of the symmetry on such charges. Again, we provide examples, invertible and non-invertible to illustrate the general results. As a corollary we rederive the results of Part I [1] for invertible symmetries and higher-representations.

Section 4 is devoted to the case of 2d theories, and their symmetries, SymTFTs, Drinfeld centers and so the generalized charges. This is particularly insightful for symmetries derived from non-abelian finite groups, as well as the intrinsically non-invertible symmetries, such as the Ising symmetries.

Section 5 discusses the 3d theories and their fusion 2-category symmetries. The main focus here is on symmetries of the type 2-$\text{Vec}_G$ and gauged versions thereof, which in fact are (up to symmetries that are module categories of fusion categories) the most general fusion 2-categories. We determine the Drinfeld center and discuss the generalized charges in several examples.

## 2 General aspects of symmetries and the SymTFT

At the most basic level, symmetries correspond to topological defects of a theory. However, one can also abstract out a symmetry $\mathcal{S}$ and study it independently of a specific theory admitting

the symmetry $\mathcal{S}$. The generalized charges of a symmetry $\mathcal{S}$ can be similarly studied without referring to a specific $\mathcal{S}$-symmetric theory, which may or may not admit operators realizing all the possible generalized charges. This abstraction is akin to the study of groups and their representations, irrespective of a specfic theory, where they are realized.

It is for these reasons that we begin with a general discussion of symmetries, delineating clearly what it means for a theory $\mathfrak{T}$ to admit a symmetry $\mathcal{S}$. This sets the stage for the discussion of generalized charges in the subsequent sections.

As a preparation for discussing generalized charges, we discuss the notion of a symmetry TFT $\mathfrak{Z}(\mathcal{S})$ associated to a symmetry $\mathcal{S}$, and how the interval compactifications (which we refer to as the sandwich constructions) of $\mathfrak{Z}(\mathcal{S})$ realize $\mathcal{S}$-symmetric theories.

## 2.1 Topological defects and symmetry category

The fundamental notion in the discussion of symmetries is that of topological defects. Topological defects of various dimensions can coexist in a given theory, and in general form a layering structure, i.e. there can be topological defects embedded within other topological defects. As a consequence, the different topological defects of a $d$-dimensional theory $\mathfrak{T}$ form a $(d-1)$-category $\mathcal{S}(\mathfrak{T})$. We will denote

$$D_p^{(a)}: \quad \text{a topological defect of dimension } p \text{ in a theory,} \tag{23}$$

where $a$ takes values in a label set for the symmetry. In a $(d-1)$-category, we have topological defects of dimensions $0 \leq p \leq d-1$.

In more detail, following [8], we have the following identification:

1. Topological defects of codimension-1 of $\mathfrak{T}$ describe objects of $\mathcal{S}(\mathfrak{T})$.

2. Topological defects of codimension-2 of $\mathfrak{T}$ converting a codimension-1 topological defect $D_{d-1}^{(a)}$ to another codimension-1 topological defect $D_{d-1}^{(b)}$ describe 1-morphisms of $\mathcal{S}(\mathfrak{T})$ from the object corresponding to $D_{d-1}^{(a)}$ to the object corresponding to $D_{d-1}^{(b)}$. Denote the collection of these 1-morphisms by $\text{Hom}(D_{d-1}^{(a)}, D_{d-1}^{(b)})$.

3. Pick two codimension-2 topological defects $D_{d-2}^{(a,b;A)}$ and $D_{d-2}^{(a,b;B)}$ in $\text{Hom}(D_{d-1}^{(a)}, D_{d-1}^{(b)})$. Topological defects of codimension-3 of $\mathfrak{T}$ converting $D_{d-2}^{(a,b;A)}$ to $D_{d-2}^{(a,b;B)}$ describe 2-morphisms of $\mathcal{S}(\mathfrak{T})$ from the 1-morphism corresponding to $D_{d-2}^{(a,b;A)}$ to the 1-morphism corresponding to $D_{d-2}^{(a,b;B)}$.

4. One can continue iteratively in this fashion until one reaches codimension-$d$ topological defects of $\mathfrak{T}$ that correspond to $(d-1)$-morphisms of $\mathcal{S}(\mathfrak{T})$.

Moreover, the topological defects can be fused together to give rise to other topological defects. This converts $\mathcal{S}(\mathfrak{T})$ into a **multi-tensor** $(d-1)$-category. Following the language used in [8, 15], we refer to $\mathcal{S}(\mathfrak{T})$ as the **symmetry category of the theory** $\mathfrak{T}$.

> **Definition 2.1: Symmetry category**
>
> The **symmetry category** $\mathcal{S}(\mathfrak{T})$ of a $d$-dimensional theory $\mathfrak{T}$ is the multi-tensor $(d-1)$-category formed by topological defects of $\mathfrak{T}$.

There are a few points to be made here:

1. We obtain a multi-tensor rather than a tensor $(d-1)$-category in general, because we allow the theory $\mathfrak{T}$ to have topological local operators other than multiples of identity local operator. If the only topological local operators of $\mathfrak{T}$ are multiples of the identity, i.e. if $\mathfrak{T}$ is a **simple** theory, then $\mathcal{S}(\mathfrak{T})$ is a **tensor** $(d-1)$-category.

2. If $\mathfrak{T}$ only admits finite symmetries, then $\mathcal{S}(\mathfrak{T})$ is a **multi-fusion** $(d-1)$-category. If additionally the only topological local operators of $\mathfrak{T}$ are multiples of identity, then $\mathcal{S}(\mathfrak{T})$ is a **fusion** $(d-1)$-category.

3. $\mathcal{S}(\mathfrak{T})$ is Karoubi- or condensation-complete in the sense of [19, 20], because it contains all kinds of condensation defects that can be obtained by gauging/condensing other topological defects.

## 2.2 Symmetry: Definition independent of a theory

The above notion of symmetry category is tied to a particular theory. But it is often useful to abstract out the notion of symmetries and study them independently of a theory.

We imagine a collection of abstract topological defects living in $d$ spacetime dimensions which has all the properties that could make this collection of abstract topological defects the topological defects of a $d$-dimensional theory. We can refer to such a collection of abstract topological defects as a symmetry. Mathematically, we can define symmetries as follows:

---

**Definition 2.2: Symmetry**

A **symmetry** $\mathcal{S}$ is a Karoubi (or condensation)-complete multi-tensor $(d-1)$-category.

---

In this paper, we study only finite symmetries, in which case $\mathcal{S}$ is a multi-fusion $(d-1)$-category. We moreover restrict to the case in which $\mathcal{S}$ is a fusion $(d-1)$-category for simplicity. We will denote the topological defects in an abstract symmetry category by

$$S_p^{(a)}: \quad \text{a topological defect of dimension } p \text{ in } \mathcal{S}, \tag{24}$$

distinguishing them at a notational level from the topological defects of a $d$-dimensional theory. Compare (24) with (23).

---

**Example 2.1: Non-anomalous (higher-)group symmetries**

The simplest examples of symmetries are the standard 0-form global symmetries described by a group $G^{(0)}$. If the 0-form symmetry is non-anomalous (i.e. does not carry any 't Hooft anomaly), its associated $(d-1)$-category is

$$\mathcal{S} = \mathcal{S}_{G^{(0)}} := (d-1)\text{-Vec}_{G^{(0)}}, \tag{25}$$

namely the $(d-1)$-category formed by $G^{(0)}$-graded $(d-1)$-vector spaces. This includes three types of topological defects:

1. Topological codimension-1 defects

$$S_{d-1}^{(g)}, \qquad g \in G^{(0)}, \tag{26}$$

generating the 0-form symmetry.

---

2. $p$-dimensional TQFTs[3] for $p \leq d-1$ that can be constructed by performing a (possibly generalized) gauging of the trivial $p$-dimensional TQFT. Equivalently, these are $p$-dimensional TQFTs admitting a topological boundary condition.

3. Combinations (i.e. fusion products) of 0-form symmetry generators with the above TQFTs.

Note that the topological defects of the first type are invertible, but the topological defects of the second and third types are non-invertible.

The above description generalizes to arbitrary non-anomalous invertible symmetries, that includes higher-form and higher-group symmetries. Such a symmetry is described by a $p$-group $\mathbb{G}^{(p)}$ for $1 \leq p \leq d-1$, and the associated $(d-1)$-category is

$$\mathcal{S} = \mathcal{S}_{\mathbb{G}^{(p)}} := (d-1)\text{-Vec}_{\mathbb{G}^{(p)}}, \tag{27}$$

which contains combinations of

1. The symmetry generators for the $p$-group $\mathbb{G}^{(p)}$. This includes within it $r$-form symmetry groups $G^{(r)}$ for $0 \leq r \leq p-1$. Correspondingly we have symmetry generators

$$S_{d-r-1}^{(g)}, \qquad g \in G^{(r)}. \tag{28}$$

2. Condensation defects, which are themselves of three types:

   - $q$-dimensional TQFTs for $q \leq d-1$ that can be constructed by performing a (possibly generalized) gauging of the trivial $q$-dimensional TQFT.

   - Condensation defects of the type discussed in [7] that can be obtained by gauging the symmetry generators $S_{d-r-1}^{(g)}$ on $q$-dimensional sub-manifolds in spacetime for $q \leq d-1$.

   - Mixtures of the above two.

   It should be noted that, following [19], we call any defect obtained by any kind of gauging as a **condensation defect** in this paper.

---

**Example 2.2: (Higher-)representation symmetries**

Another natural class of fusion $(d-1)$-categories is

$$\mathcal{S} = (d-1)\text{-Rep}(G^{(0)}), \tag{29}$$

for finite groups $G^{(0)}$, namely the $(d-1)$-categories formed by $(d-1)$-representations of groups $G^{(0)}$. The corresponding symmetries are generally non-invertible and hence such higher-representations provide natural examples of non-invertible symmetries. In concrete theories, (29) arises as symmetry after one gauges a $G^{(0)}$ non-anomalous 0-form symmetry.

---

[3]More precisely, two $p$-dimensional TQFTs related by stacking an invertible $p$-dimensional TQFT are associated to the same $p$-dimensional defect in $(d-1)$-Vec$_{G^{(0)}}$. Reference [20] refers to $p$-dimensional TQFTs modulo invertible ones as "$p$-dimensional non-anomalous topological orders".

If $G^{(0)}$ is abelian, then after gauging we obtain a $(d-2)$-form symmetry group [2]

$$G^{(d-2)} = \widehat{G}^{(0)}, \tag{30}$$

where $\widehat{G}^{(0)}$ is Pontryagin dual of $G^{(0)}$. Correspondingly, we have the identification

$$(d-1)\text{-Rep}(G^{(0)}) = (d-1)\text{-Vec}_{\mathbb{G}^{(d-1)}_{G^{(d-2)}}}, \tag{31}$$

where $\mathbb{G}^{(d-1)}_{G^{(d-2)}}$ is a $(d-1)$-group whose only non-trivial component is the $(d-2)$-form symmetry group $G^{(d-2)}$. Thus the two types of symmetries that we have discussed are not mutually exclusive. The higher-representation symmetries of type (29) are not of higher-group type iff $G^{(0)}$ is a non-abelian group.

Similarly, gauging a non-anomalous $\mathbb{G}^{(p)}$ $p$-group symmetry leads to a generally non-invertible symmetry described by a fusion $(d-1)$-category

$$\mathcal{S} = (d-1)\text{-Rep}(\mathbb{G}^{(p)}), \tag{32}$$

namely the $(d-1)$-category formed by $(d-1)$-representations of the $p$-group $\mathbb{G}^{(p)}$. See [9,10,21] for more details. Again, for special $p$-groups $\mathbb{G}^{(p)}$, the associated higher-representation symmetries are of higher-group type. If $\mathbb{G}^{(p)}$ is a product of higher-form symmetry groups without any mixing between them

$$\mathbb{G}^{(p)} = \prod_{r=0}^{p-1} G^{(r)}, \tag{33}$$

and if the 0-form symmetry group $G^{(0)}$ in $\mathbb{G}^{(p)}$ is abelian, then after the gauging we obtain a dual $(d-1)$-group

$$\mathbb{G}^{(d-1)}_{\text{dual}} = \prod_{r=d-p-1}^{d-2} G^{(r)}_{\text{dual}}, \tag{34}$$

which splits into a product of dual higher-form symmetry groups, which are

$$G^{(r)}_{\text{dual}} = \widehat{G}^{(d-r-2)} = \text{Hom}\left(G^{(r)}, U(1)\right), \tag{35}$$

where the hat denotes the Pontryagin dual group. Correspondingly, in such a case we have

$$(d-1)\text{-Rep}(\mathbb{G}^{(p)}) = (d-1)\text{-Vec}_{\mathbb{G}^{(d-1)}_{\text{dual}}}. \tag{36}$$

However, in general higher-representation type symmetries differ from higher-group symmetries.

## 2.3 Equipping a theory with a symmetry

Since we have abstracted out the notion of a symmetry $\mathcal{S}$, we can now ask when a $d$-dimensional theory $\mathfrak{T}$ admits the symmetry $\mathcal{S}$. That is, we want to identify the abstract collection of topological defects comprising $\mathcal{S}$ as topological defects realized in the theory $\mathfrak{T}$. Note that, in general, we can identify two different defects in $\mathcal{S}$ as the same topological defect of $\mathfrak{T}$. In other words, we are seeking a map

$$\sigma: \quad \mathcal{S} \rightarrow \mathcal{S}(\mathfrak{T}). \tag{37}$$

This map should respect the full fusion higher-categorical structure. Thus, mathematically $\sigma$ should be what is known as a **tensor functor**. We are thus led to following definition:

**Definition 2.3: Theory Admitting a Symmetry**

We call $\mathcal{S}$ **a symmetry of** $\mathfrak{T}$ if and only if there exists a tensor functor

$$\sigma: \quad \mathcal{S} \to \mathcal{S}(\mathfrak{T}), \tag{38}$$

from the symmetry $\mathcal{S}$ to the symmetry category $\mathcal{S}(\mathfrak{T})$ of $\mathfrak{T}$.

Note that given a theory $\mathfrak{T}$, there may be multiple ways of realizing the symmetry $\mathcal{S}$ in $\mathfrak{T}$. That is, in general there may be multiple choices for the functor $\sigma$. If this is the case, then it means that there are multiple collections of topological defects of $\mathfrak{T}$ that behave like defects captured by the category $\mathcal{S}$. This leads to the following definition:

**Definition 2.4: $\mathcal{S}$-symmetric Theories**

The tuple

$$\mathfrak{T}_\sigma \equiv (\mathfrak{T}, \sigma), \tag{39}$$

is referred to as an $\mathcal{S}$-**symmetric $d$-dimensional theory** whose underlying theory is $\mathfrak{T}$.

Following the above definition 2.3, it is clear that the symmetry category $\mathcal{S}(\mathfrak{T})$ is canonically a symmetry of the theory $\mathfrak{T}$, by simply choosing $\sigma$ to be the identity functor

$$\sigma = \mathrm{id}. \tag{40}$$

In other words, there is a canonical $\mathcal{S}(\mathfrak{T})$-symmetric theory $\mathfrak{T}_{\mathrm{id}}$ whose underlying theory is $\mathfrak{T}$.

The same underlying theory $\mathfrak{T}$ may lead to multiple $\mathcal{S}$-symmetric theories $\mathfrak{T}_\sigma$ for a fixed $\mathcal{S}$. Let us discuss some examples:

**Example 2.3: Equipping a 2d theory with a 0-form symmetry**

We now give an example of non-trivial choices of $\sigma$. Consider a 2d theory $\mathfrak{T}$, which admits topological line defects

$$D_1^{(g)}, \qquad g \in G, \tag{41}$$

that are all distinct from each other. Here $G$ is a finite group and the fusion of $D_1^{(g)}$ obeys group multiplication of $G$. Moreover, assume that we can pick some topological local operators

$$D_0^{(g,g')}: \quad D_1^{(g)} \otimes D_1^{(g')} \to D_1^{(gg')}, \tag{42}$$

living at the junctions of these topological line defects in $\mathfrak{T}$, such that these lines and local operators obey associativity:

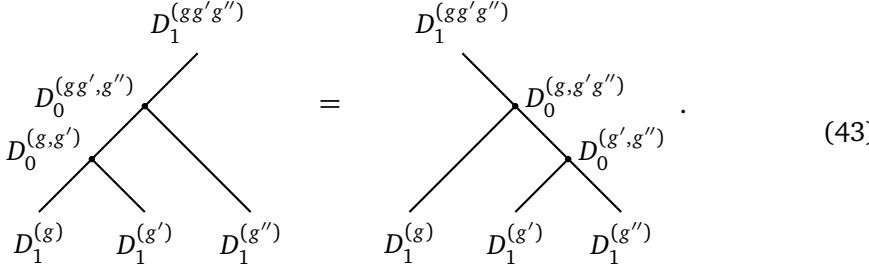

$$\tag{43}$$

Then, it means that these topological line defects and topological junction local operators

generate a subcategory

$$\mathcal{S}_G \subseteq \mathcal{S}(\mathfrak{T}),\tag{44}$$

of the full symmetry category $\mathcal{S}(\mathfrak{T})$ of $\mathfrak{T}$.

We can consider making $\mathfrak{T}$ into a $G^{(0)}$-symmetric theory where $G^{(0)}$ is some 0-form symmetry group. That is, we are looking for tensor functors

$$\sigma: \quad \mathcal{S}_{G^{(0)}} \to \mathcal{S}(\mathfrak{T}).\tag{45}$$

Here $\mathcal{S}_{G^{(0)}}$ comprises of abstract topological line operators

$$S_1^{(g)}, \qquad g \in G^{(0)},\tag{46}$$

and abstract topological junction local operators

$$S_0^{(g,g')}: \quad S_1^{(g)} \otimes S_1^{(g')} \to S_1^{(gg')},\tag{47}$$

such that these lines and local operators also obey associativity.

Let us restrict our attention to functors $\sigma$ whose image lies in the subcategory $\mathcal{S}_G$ of $\mathcal{S}(\mathfrak{T})$. To specify such a functor, we need to first specify a group homomorphism

$$\rho: \quad G^{(0)} \to G,\tag{48}$$

which specifies $\sigma$ at the level of topological lines

$$\sigma(S_1^{(g)}) \cong D_1^{(\rho(g))}.\tag{49}$$

At the level of topological local operators, we can have

$$\sigma(S_0^{(g,g')}) = \alpha(g,g') D_0^{(\rho(g),\rho(g'))},\tag{50}$$

where $\alpha(g,g') \in \mathbb{C}^\times$. That is, $S_0^{(g,g')}$ may in general only be proportional to $D_0^{(\rho(g),\rho(g'))}$. Imposing associativity on $\sigma(S_1^{(g)})$ and $\sigma(S_0^{(g,g')})$ leads to the condition

$$\delta\alpha = 1,\tag{51}$$

i.e. $\alpha$ is a 2-cocycle on $G^{(0)}$ valued in $\mathbb{C}^\times$. Actually $\alpha$ can be shifted

$$\alpha \to \alpha + \delta\beta,\tag{52}$$

where $\beta$ is a 1-cochain on $G^{(0)}$ valued in $\mathbb{C}^\times$. This corresponds to changing the choice of isomorphism between $\sigma(S_1^{(g)})$ and $D_1^{(\rho(g))}$ that allows us to write the equation (50).

Thus various such functors $\sigma$ can be represented as tuples

$$\sigma = (\rho, [\alpha]),\tag{53}$$

where $\rho$ is a group homomorphism and

$$[\alpha] \in H^2\big(G^{(0)}, \mathbb{C}^\times\big).\tag{54}$$

These functors specify all the possible $G^{(0)}$-symmetric theories $\mathfrak{T}_\sigma$ whose underlying theory is the same theory $\mathfrak{T}$.

Some special cases are noteworthy:

1. If $\rho$ is an injective homomorphism, then we are choosing a subgroup $G^{(0)} \subseteq G$ as the 0-form symmetry to study. The choice $[\alpha]$ describes a choice of coupling $\mathfrak{T}$ to background fields for the $G^{(0)}$ 0-form symmetry.

2. Let $\mathfrak{T}$ be the trivial 2d theory. Then $G = \{\text{id}\}$, and the only choice for $\rho$ is the identity homomorphism. The tensor functors $\sigma$ are classified by classes $[\alpha]$ in second group cohomology of $G^{(0)}$. The resulting $\mathcal{S}$-symmetric theories $\mathfrak{T}_\sigma$ are known as 2d **SPT phases** protected by $G^{(0)}$ 0-form symmetry.

In the above example, the symmetries corresponding to the kernel $\ker(\rho) \subseteq G^{(0)}$ of the homomorphism $\rho$ are all implemented by the identity line defect $D_1^{(\text{id})}$ in $\mathfrak{T}$

$$\sigma(S_1^{(g)}) = D_1^{(\text{id})}, \qquad g \in \ker(\rho) \subseteq G^{(0)}. \tag{55}$$

Such symmetries are sometimes referred to as **non-faithfully acting symmetries**. More generally, we have:

---

**Definition 2.5: Non-faithful symmetries**

Consider an $\mathcal{S}$-symmetric theory $\mathfrak{T}_\sigma$. A symmetry in $\mathcal{S}$ acts **non-faithfully** on $\mathfrak{T}$ if it is described by some abstract $p$-dimensional defect $S_p \in \mathcal{S}$, which is mapped to the identity $p$-dimensional defect $D_p^{(\text{id})}$ of the theory $\mathfrak{T}$ by the corresponding tensor functor $\sigma$

$$\sigma(S_p) \cong D_p^{(\text{id})} \in \mathcal{S}(\mathfrak{T}). \tag{56}$$

---

The consideration of non-faithful symmetries is physically important because **faithfulness of a symmetry is not an RG-invariant** notion. One can begin with a faithfully acting symmetry in the UV that flows to a non-faithfully acting symmetry in the IR. An example is provided below.

---

**Example 2.4: Non-faithful symmetries and confinement**

Consider $\mathbb{Z}_2^{(1)}$ electric 1-form symmetry of 4d $\mathcal{N} = 1$ $SU(2)$ supersymmetric Yang-Mills (SYM). This symmetry is faithful in the UV as it acts non-trivially on Wilson lines, while the identity surface acts trivially on all line operators.

In the IR there are two massive/gapped vacua. Both of these vacua exhibit confinement, which means that there are no line operators in the IR on which $\mathbb{Z}_2^{(1)}$ can act non-trivially. In fact, in each vacuum, the IR theory is actually trivial and the 1-form symmetry is generated by the identity surface operator of the trivial theory. The $\mathbb{Z}_2^{(1)}$ symmetry is thus non-faithful in the IR.

Even though the underlying IR theories in both vacua are same (i.e. the trivial theory), the $\mathbb{Z}_2^{(1)}$-symmetric IR theories are different. This corresponds to two different choices for the functor $\sigma$ for the same source and target 3-categories

$$\mathcal{S} = \mathcal{S}_{\mathbb{Z}_2^{(1)}}, \qquad \mathcal{S}(\mathsf{T}) = 3\text{-Vec} \equiv \mathcal{S}_{G^{(0)}=\{\text{id}\}}, \tag{57}$$

where the definition of $\mathcal{S}_{\mathbb{Z}_2^{(1)}}$ is discussed around (27) and $(d-1)$-Vec is the fusion $(d-1)$-category which is the symmetry category of the trivial theory in $d$-dimensions. Let

$$S_2^{(-)} \in \mathcal{S}, \tag{58}$$

---

be the generator for $\mathbb{Z}_2^{(1)}$, then for both vacua we have

$$\sigma(S_2^{(-)}) = D_2^{(\text{id})} \in \mathcal{S}(\mathfrak{T}), \qquad (59)$$

where $D_p^{(\text{id})}$ denotes the identity $p$-dimensional operator. To distinguish the two vacua let $S_0^{(-)} \in \mathcal{S}$ denote the intersection of two $S_2^{(-)}$:

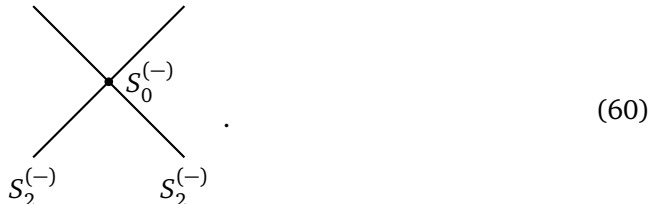

$$ (60) $$

Then we have

$$\sigma(S_0^{(-)}) = \pm D_0^{(\text{id})} \in \mathcal{S}(\mathfrak{T}), \qquad (61)$$

with different signs for the two vacua. The vacuum realizing the negative sign in the above equation is known as the **oblique confining** vacuum, while the one realizing the positive sign is known as the **standard confining** vacuum.

## 2.4 Symmetry TFT and the sandwich conjecture

The main concept that we will use to encode generalized charges of a symmetry $\mathcal{S}$ is that of the symmetry TFT (or SymTFT) associated to a symmetry $\mathcal{S}$. The purpose of this subsection is to introduce this concept to set up the stage for discussion of generalized charges in the subsequent sections. Roughly speaking, symmetry TFT is a $(d+1)$-dimensional topological quantum field theory (TQFT) $\mathfrak{Z}(\mathcal{S})$ that can be used to separate the $\mathcal{S}$ symmetry of a $d$-dimensional theory $\mathfrak{T}_\sigma$ from its dynamics [12], with the symmetry and dynamics living on two different $d$-dimensional boundary conditions $\mathfrak{B}_\mathcal{S}^{\text{sym}}$ and $\mathfrak{B}_{\mathfrak{T}_\sigma}^{\text{phys}}$ of $\mathfrak{Z}(\mathcal{S})$.

Before discussing this idea in more detail below, let us note a few points about the literature on this topic. First of all, our presentation is largely inspired by the recent seminal work [11] on this topic. The concept of SymTFTs is already being prominently used in the recent literature on the study of symmetries in high-energy physics [14, 22–24]. In condensed matter literature, this idea appeared under the nomenclature[4] "categorical symmetries" in [13] and has also appeared in other works, e.g. [25].

Given a symmetry described by a fusion $(d-1)$-category $\mathcal{S}$, we can associate to it the following objects:

---

**Definition 2.6: Symmetry TFT and symmetry boundary**

The **symmetry TFT** (or **SymTFT** in short) $\mathfrak{Z}(\mathcal{S})$ associated to a symmetry $\mathcal{S}$ is a $(d+1)$-dimensional TQFT fixed by the requirement that it admits a topological boundary condition $\mathfrak{B}_\mathcal{S}^{\text{sym}}$ such that the symmetry category $\mathcal{S}(\mathfrak{B}_\mathcal{S}^{\text{sym}})$ of the boundary $\mathfrak{B}_\mathcal{S}^{\text{sym}}$ coincides with $\mathcal{S}$

$$\mathcal{S}(\mathfrak{B}_\mathcal{S}^{\text{sym}}) = \mathcal{S}. \qquad (62)$$

That is, the topological defects living inside $\mathfrak{B}_\mathcal{S}^{\text{sym}}$, and unattached to topological defects

---

[4]The term "categorical symmetries" used in this reference does not refer to the notion of symmetry $\mathcal{S}$ (which is a fusion higher-category) as used in this paper. Instead, it more appropriately refers to the notion of generalized charges of the symmetry $\mathcal{S}$ discussed in subsequent sections.

living in the bulk of the TQFT $\mathfrak{Z}(\mathcal{S})$, form the fusion $(d-1)$-category $\mathcal{S}$.

The SymTFT $\mathfrak{Z}(\mathcal{S})$ may admit multiple topological boundary conditions whose symmetry category is $\mathcal{S}$. We fix such a topological boundary condition $\mathfrak{B}_{\mathcal{S}}^{\text{sym}}$ in what follows and refer to it as the **symmetry boundary** associated to the symmetry $\mathcal{S}$.

The fact that we are restricting to fusion rather than multi-fusion $(d-1)$-categories $\mathcal{S}$ reflects in the fact that the topological boundary $\mathfrak{B}_{\mathcal{S}}^{\text{sym}}$ is simple, i.e. the only genuine topological local operators on $\mathfrak{B}_{\mathcal{S}}^{\text{sym}}$ are multiples of the identity local operator. Let us note that the SymTFT $\mathfrak{Z}(\mathcal{S})$ is also simple.

Note that the topological defects on the symmetry boundary $\mathfrak{B}_{\mathcal{S}}^{\text{sym}}$ provide a physical realization of the abstract defects comprising a symmetry $\mathcal{S}$. As such, we will denote $p$-dimensional topological defects of $\mathfrak{B}_{\mathcal{S}}^{\text{sym}}$ by $S_p$.

The utility of the SymTFT is to separate the symmetry and dynamics of a theory $\mathfrak{T}_\sigma$, which is made possible by the sandwich construction described in [11], which we review below.

---

**Statement 2.1: Sandwich construction**

An $\mathcal{S}$-symmetric $d$-dimensional theory $\mathfrak{T}_\sigma$ can be expressed as an interval compactification of the SymTFT $\mathfrak{Z}(\mathcal{S})$, as shown in figure 4. The boundary conditions involved in the compactification are:

- **Symmetry Boundary** $\mathfrak{B}_{\mathcal{S}}^{\text{sym}}$: This is a topological boundary condition of $\mathfrak{Z}(\mathcal{S})$ discussed above, which will be displayed on the left.

- **Physical Boundary** $\mathfrak{B}_{\mathfrak{T}_\sigma}^{\text{phys}}$: This is a boundary condition that depends on the $\mathcal{S}$-symmetric theory $\mathfrak{T}_\sigma$, which will be displayed on the right.

---

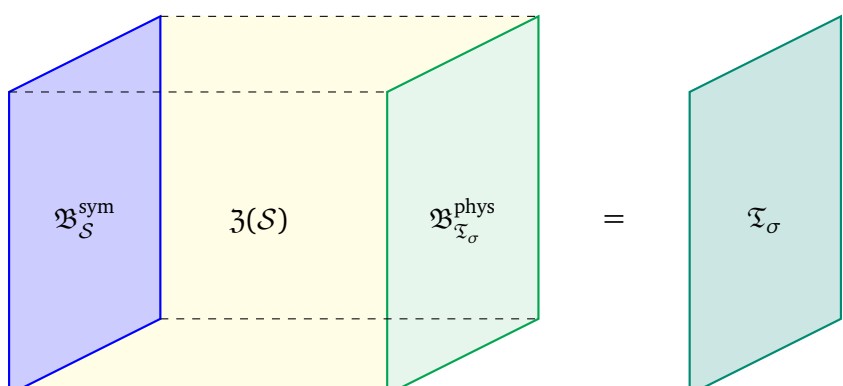

Figure 4: The Symmetry TFT (SymTFT) $\mathfrak{Z}(\mathcal{S})$ is a $(d+1)$-dimensional topological field theory associated to a symmetry $\mathcal{S}$ of $d$-dimensional theories. The SymTFT admits a topological boundary condition, namely the symmetry boundary condition $\mathfrak{B}_{\mathcal{S}}^{\text{sym}}$, whose symmetry category matches the fusion $(d-1)$-category associated to the symmetry $\mathcal{S}$. Given an $\mathcal{S}$-symmetric $d$-dimensional theory $\mathfrak{T}_\sigma$, there exists a corresponding boundary condition $\mathfrak{B}_{\mathfrak{T}_\sigma}^{\text{phys}}$ of the symmetry TFT $\mathfrak{Z}(\mathcal{S})$, which we refer to as the physical boundary condition. As shown in the figure, compactifying on an interval with the symmetry topological boundary condition $\mathfrak{B}_{\mathcal{S}}^{\text{sym}}$ on one side, and the physical (not necessarily topological) boundary condition $\mathfrak{B}_{\mathfrak{T}_\sigma}^{\text{phys}}$ on the other side, recovers the $\mathcal{S}$-symmetric theory $\mathfrak{T}_\sigma$.

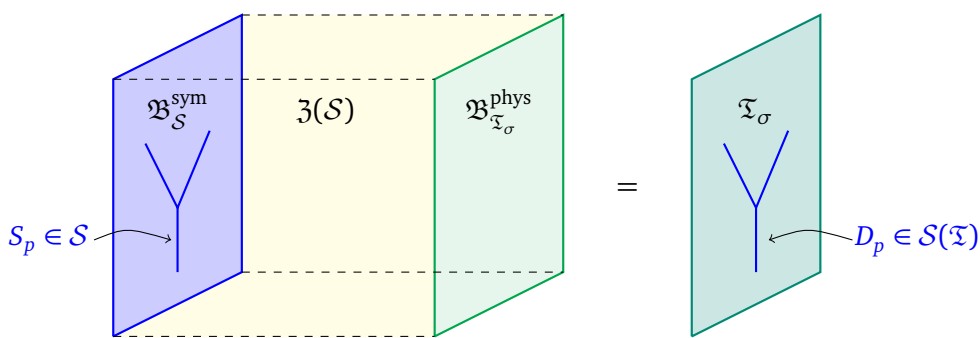

Figure 5: After the interval compactification, the topological defects $S_p \in \mathcal{S}$ (drawn as blue lines) of $\mathfrak{B}_{\mathcal{S}}^{\text{sym}}$ become topological defects $D_p \in \mathcal{S}(\mathfrak{T})$ of the theory $\mathfrak{T}$ that generate the $\mathcal{S}$ symmetry of $\mathfrak{T}_\sigma$.

Some comments are in order:

- If we have two different $\mathcal{S}$-symmetric theories

$$\mathfrak{T}_\sigma \neq \mathfrak{T}'_{\sigma'}, \tag{63}$$

  then the corresponding physical boundaries are also different

$$\mathfrak{B}_{\mathfrak{T}_\sigma}^{\text{phys}} \neq \mathfrak{B}_{\mathfrak{T}'_{\sigma'}}^{\text{phys}}. \tag{64}$$

  This is also true if the underlying theories are the same

$$\mathfrak{T} = \mathfrak{T}', \tag{65}$$

  but they have been made $\mathcal{S}$-symmetric in different ways, i.e.

$$\sigma \neq \sigma'. \tag{66}$$

- The boundary condition $\mathfrak{B}_{\mathfrak{T}_\sigma}^{\text{phys}}$ is topological if and only if the underlying theory $\mathfrak{T}$ is topological.

- The information about $\sigma$ is encoded in the interval compactification as follows. Pick a $p$-dimensional topological defect $S_p$ of the symmetry boundary $\mathfrak{B}_{\mathcal{S}}^{\text{sym}}$. After the interval compactification it is converted to a $p$-dimensional topological defect $D_p$ of $\mathfrak{T}$. This defines the functor $\sigma$ via

$$\sigma(S_p) = D_p \in \mathcal{S}(\mathfrak{T}). \tag{67}$$

  This is illustrated in figure 5.

Unlike the symmetry boundary $\mathfrak{B}_{\mathcal{S}}^{\text{sym}}$, a physical boundary $\mathfrak{B}^{\text{phys}}$ may not be simple, i.e. there may exist genuine topological local operators along $\mathfrak{B}^{\text{phys}}$ that are not multiples of identity. On the other hand, simple physical boundaries correspond to irreducible $\mathcal{S}$-symmetric theories.

---

**Definition 2.7: Irreducible $\mathcal{S}$-symmetric theory**

A $d$-dimensional $\mathcal{S}$-symmetric theory $\mathfrak{T}_\sigma$ is called **irreducible** if it cannot be expressed as a direct sum of two other $d$-dimensional $\mathcal{S}$-symmetric theories.
In other words, in an irreducible $\mathcal{S}$-symmetric theory, any two genuine topological local operators are related (upto multiplication by a $\mathbb{C}^\times$ element) by the action of $\mathcal{S}$.

---

---

**Statement 2.2: Irreducible $\mathcal{S}$-symmetric theories from SymTFT**

The physical boundary $\mathfrak{B}^{\text{phys}}_{\mathfrak{T}_\sigma}$ corresponding to an irreducible $\mathcal{S}$-symmetric theory $\mathfrak{T}_\sigma$ is simple, i.e. $\mathfrak{B}^{\text{phys}}_{\mathfrak{T}_\sigma}$ does not contain any genuine topological local operators other than multiples of the identity local operator.

---

## 2.5 Examples of symmetry TFTs

In this subsection we discuss SymTFTs for symmetries described by groups or higher-groups, with or without the presence of 't Hooft anomalies. We also discuss SymTFTs for (higher-)representation symmetries.

### 2.5.1 SymTFTs for non-anomalous higher-form symmetries

We start with the SymTFT for non-anomalous, invertible higher-form symmetries, $G^{(r)}$, $r = 0, \cdots, p-1$, which do not mix with each other, and the 0-form symmetry group $G^{(0)}$ is abelian. This is a special case of a $p$-group $\mathbb{G}^{(p)}$:

$$\mathbb{G}^{(p)} = \prod_{r=0}^{p-1} G^{(r)}. \tag{68}$$

Then the SymTFT $\mathfrak{Z}(\mathcal{S}_{\mathbb{G}^{(p)}})$ can be expressed as a BF-theory (or a generalized Dijkgraaf-Witten theory)[5] with action

$$S[\mathfrak{Z}(\mathcal{S}_{\mathbb{G}^{(p)}})] = \int_{M_{d+1}} \sum_{r=0}^{p-1} a_{r+1} \cup \delta b_{d-r-1}. \tag{69}$$

Here

- $a_{r+1}$ is a dynamical gauge field which is a $G^{(r)}$-valued $(r+1)$-cochain on $M_{d+1}$. For $r = 0$, i.e. for 0-form symmetry, this is a standard gauge field $a_1$, which is a 1-cochain.

- $b_{d-r-1}$ is another dynamical gauge field which is a $\widehat{G}^{(r)}$-valued $(d-r-1)$-cochain on $M_{d+1}$, where $\widehat{G}^{(r)}$ is the Pontryagin dual group of the group $G^{(r)}$. This is the dual gauge field involved in a BF-theory.

- The cup product between $a_{r+1}$ and $b_{d-r-1}$ involves the natural pairing $G^{(r)} \times \widehat{G}^{(r)} \to \mathbb{R}/\mathbb{Z}$.

For non-abelian $G^{(0)}$ or for an arbitrary $p$-group $\mathbb{G}^{(p)}$ in which component $r$-form symmetry groups mix with each other, we do not have a simple Lagrangian description like above, but the SymTFT can be described abstractly as the theory obtained by gauging a trivial theory. We discuss this below in section 2.5.2.

The symmetry boundary $\mathfrak{B}^{\text{sym}}_{\mathcal{S}_{\mathbb{G}^{(p)}}}$ for this symmetry is as follows:

- Dirichlet boundary conditions for the gauge fields $a_r$, $0 \leq r \leq p-1$, i.e. at the location of $\mathfrak{B}^{\text{sym}}_{\mathcal{S}_{\mathbb{G}^{(p)}}}$, we impose

$$a_{r+1}|_{\mathfrak{B}^{\text{sym}}_{\mathcal{S}_{\mathbb{G}^{(p)}}}} = A_{r+1}, \qquad \forall \, 1 \leq r \leq p-1, \tag{70}$$

where $A_{r+1}$ is a fixed background field for $r$-form symmetry on the boundary $\mathfrak{B}^{\text{sym}}_{\mathcal{S}_{\mathbb{G}^{(p)}}}$.

---

[5]The identification of Dijkgraaf-Witten theories as SymTFTs was discussed in [12,14].

- Neumann boundary conditions for the gauge fields $b_{d-r-1}$, $0 \le r \le p-1$, i.e. they are free to fluctuate on the boundary $\mathfrak{B}^{\text{sym}}_{\mathcal{S}_{\mathbb{G}^{(p)}}}$.

A further specialization occurs when the higher-form symmetry is cyclic (or a product of cyclic groups): $\mathbb{Z}_N$. For a $\mathbb{Z}_N^{(r)}$ $r$-form symmetry, the SymTFT can be written in terms of $U(1)$-valued forms $\alpha_{r+1}$ and $\beta_{d-r-1}$, with action

$$S[\mathfrak{Z}(\mathcal{S}_{\mathbb{Z}_N^{(r)}})] = \frac{i}{2\pi} N \int_{M_{d+1}} \alpha_{r+1} \wedge d\beta_{d-r-1} \,, \tag{71}$$

where the torsion is imposed by the equations of motion $N d\alpha_{r+1} = 0$ and $N d\beta_{d-r-1} = 0$. This is particularly familiar in applications in holography, gravity and string theory, where $\alpha, \beta$ are form-fields, with a BF-coupling.

### 2.5.2 SymTFTs for general non-anomalous higher-group symmetries

Consider now a general non-anomalous $p$-group symmetry $\mathbb{G}^{(p)}$

$$\mathcal{S} = \mathcal{S}_{\mathbb{G}^{(p)}} = (d-1)\text{-Vec}_{\mathbb{G}^{(p)}} \,. \tag{72}$$

To obtain the corresponding SymTFT $\mathfrak{Z}(\mathcal{S}_{\mathbb{G}^{(p)}})$, begin by considering the trivial $(d+1)$-dimensional theory $\mathfrak{Z}^{(\text{trivial})}$, whose symmetry category is the $d$-category $d$-Vec of $d$-vector spaces.[6] Different ways of making $\mathfrak{Z}^{(\text{trivial})}$ symmetric under $\mathbb{G}^{(p)}$ correspond to tensor functors

$$\sigma: \quad d\text{-Vec}_{\mathbb{G}^{(p)}} \to d\text{-Vec} \,. \tag{73}$$

There exists a trivial functor $\sigma = 0$ in this case which sends every $q$-dimensional defect $S_q \in d\text{-Vec}_{\mathbb{G}^{(p)}}$ to the identity $q$-dimensional defect $D_q^{(\text{id})}$ of $\mathfrak{Z}^{(\text{trivial})}$

$$\sigma(S_q) = D_q^{(\text{id})} \in d\text{-Vec} \,. \tag{74}$$

The corresponding $\mathbb{G}^{(p)}$-symmetric $(d+1)$-dimensional theory

$$\mathfrak{Z}_0^{(\text{trivial})} \,, \tag{75}$$

obtained by choosing $\sigma = 0$ is also referred to as the trivial $\mathbb{G}^{(p)}$-symmetric $(d+1)$-dimensional theory. Gauging the $\mathbb{G}^{(p)}$ symmetry of $\mathfrak{Z}_0^{(\text{trivial})}$ results in the SymTFT $\mathfrak{Z}(\mathcal{S}_{\mathbb{G}^{(p)}})$:

$$\mathfrak{Z}(\mathcal{S}_{\mathbb{G}^{(p)}}) = \mathfrak{Z}_0^{(\text{trivial})}/\mathbb{G}^{(p)} \,. \tag{76}$$

The symmetry boundary $\mathfrak{B}^{\text{sym}}_{\mathcal{S}_{\mathbb{G}^{(p)}}}$ can be physically identified as the **Dirichlet** boundary condition for the $\mathbb{G}^{(p)}$ gauge fields.

In the special case discussed above in section 2.5.1, the gauge field $a_{r+1}$ for each $r$ can be understood as arising from gauging the $G^{(r)}$ $r$-form symmetry of the $p$-group (68) in the trivial $(d+1)$-dimensional theory. Gauging all $r$-form symmetry groups gauges the full $p$-group (68).

---

[6]The trivial $(d+1)$-dimensional theory $\mathfrak{Z}^{(\text{trivial})}$ is also the SymTFT for the trivial symmetry $\mathcal{S} = (d-1)$-Vec of $d$-dimensional theories, but this fact is not very important in the discussion that follows.

### 2.5.3 SymTFTs for anomalous higher-form symmetries

We can also add an anomaly for the invertible higher-form or higher-group symmetries and characterize the SymTFT. We again start with the case of non-anomalous higher-form symmetries and abelian $G^{(0)}$.

Consider a $p$-group symmetry of the form discussed in (68), which decomposes into higher-form symmetries as

$$\mathbb{G}^{(p)} = \prod_{r=0}^{p-1} G^{(r)}, \tag{77}$$

and assume moreover that $G^{(0)}$ is an abelian group. There are various possible 't Hooft anomalies for such a symmetry, which are captured by tensor functors (73). We consider a special class of such functors, or in other words a special class of 't Hooft anomalies, which are described by elements

$$\omega \in H^{d+1}(B\mathbb{G}^{(p)}, \mathbb{C}^\times), \tag{78}$$

where

$$B\mathbb{G}^{(p)} = \prod_{r=0}^{p-1} B^{r+1} G^{(r)}, \tag{79}$$

is the classifying space of the $p$-group $\mathbb{G}^{(p)}$. See the appendix of [26] for background on these spaces. We label the fusion $(d-1)$-category describing such a $p$-group symmetry with a 't Hooft anomaly $\omega$ as

$$\mathcal{S} = \mathcal{S}_{\mathbb{G}^{(p)}}^\omega. \tag{80}$$

Then, like (69) we can provide a simple Lagrangian description of the SymTFT $\mathfrak{Z}(\mathcal{S}_{\mathbb{G}^{(p)}}^\omega)$ as a BF-theory (or a generalized Dijkgraaf-Witten theory) with a twist governed by $\omega$

$$S[\mathfrak{Z}(\mathcal{S}_{\mathbb{G}^{(p)}}^\omega)] = \int_{M_{d+1}} \sum_{r=0}^{p-1} a_{r+1} \cup \delta b_{d-r-1} + \mathcal{A}^* \omega(a_1, a_2, \cdots a_p), \tag{81}$$

where $\mathcal{A}^* \omega$, which is what is often referred to as the *twist*, is the pull-back to $M_{d+1}$ of $\omega$ under the map

$$\mathcal{A}: \quad M_{d+1} \to B\mathbb{G}^{(p)}, \tag{82}$$

induced by a choice of gauge fields $a_1, a_2, \cdots a_p$. As such, $\mathcal{A}^* \omega$ is a function of these gauge fields, which is explicitly shown in (81).

Another identification for $\mathcal{A}^* \omega$ is that

$$\int_{M_{d+1}} \mathcal{A}^* \omega(A_1, A_2, \cdots A_p), \tag{83}$$

is the effective action for the anomaly theory associated to $\mathbb{G}^{(p)}$-symmetry with 't Hooft anomaly $\omega$, which is a function of *background* gauge fields $A_{r+1}$ for $G^{(r)}$ $r$-form global symmetries.

Just like for the case without anomaly, the symmetry boundary $\mathfrak{B}_{\mathcal{S}_{\mathbb{G}^{(p)}}^\omega}^{\text{sym}}$ for such a special $p$-group involves

- Dirichlet boundary conditions for the gauge fields $a_{r+1}$, $0 \leq r \leq p-1$

$$a_{r+1}|_{\mathfrak{B}_{\mathcal{S}_{\mathbb{G}^{(p)}}^\omega}^{\text{sym}}} = A_{r+1}. \tag{84}$$

- Neumann boundary conditions for the gauge fields $b_{d-r-1}$, $0 \leq r \leq p-1$.

However, as we will discuss later, a key difference between the SymTFTs for anomalous and non-anomalous cases is that not all boundary conditions allowed for $\mathfrak{Z}(\mathcal{S}_{\mathbb{G}^{(p)}})$ are allowed for $\mathfrak{Z}(\mathcal{S}_{\mathbb{G}^{(p)}}^{\omega})$, as they can be obstructed by the presence of the twist term in the action. For example, if $\omega$ is non-trivial then one cannot impose boundary conditions in which all $b_{d-r-1}$ fields have Dirichlet b.c. and all $a_{r+1}$ fields have Neumann b.c.

### 2.5.4 SymTFTs for anomalous higher-group symmetries: General case

Consider now an arbitrary $p$-group $\mathbb{G}^{(p)}$. The 't Hooft anomalies are again parametrized by tensor functors (73) and a special class of 't Hooft anomalies is provided by elements $\omega$ in (78). We label the fusion $(d-1)$-category for $\mathbb{G}^{(p)}$ symmetry with 't Hooft anomaly $\sigma$ as

$$\mathcal{S} = \mathcal{S}_{\mathbb{G}^{(p)}}^{\sigma}. \tag{85}$$

Let us now describe the corresponding SymTFT $\mathfrak{Z}(\mathcal{S}_{\mathbb{G}^{(p)}}^{\sigma})$. For this purpose, consider again the trivial $(d+1)$-dimensional theory $\mathfrak{Z}^{(\text{trivial})}$ and consider making it $\mathbb{G}^{(p)}$-symmetric without any 't Hooft anomaly. As discussed earlier, these choices are parametrized precisely by the tensor functors of the form (73). So, consider the $\mathbb{G}^{(p)}$-symmetric $(d+1)$-dimensional theory

$$\mathfrak{Z}_{\sigma}^{(\text{trivial})}, \tag{86}$$

obtained by choosing the functor $\sigma$ describing the 't Hooft anomaly. Note the following:

- For non-trivial $\sigma$, the $\mathbb{G}^{(p)}$-symmetric theory $\mathfrak{Z}_{\sigma}^{(\text{trivial})}$ is non-trivial, even though the underlying theory $\mathfrak{Z}^{(\text{trivial})}$ is trivial. In other words, $\mathfrak{Z}_{\sigma}^{(\text{trivial})}$ is a non-trivial $(d+1)$-dimensional **SPT phase** for $\mathbb{G}^{(p)}$ $p$-group symmetry. One also refers to $\mathfrak{Z}_{\sigma}^{(\text{trivial})}$ as the **anomaly theory** associated to the 't Hooft anomaly $\sigma$ of $\mathbb{G}^{(p)}$ symmetry.

- The $\mathbb{G}^{(p)}$-symmetry of the $(d+1)$-dimensional theory $\mathfrak{Z}_{\sigma}^{(\text{trivial})}$ is non-anomalous, even though $\sigma$ captures a non-trivial 't Hooft anomaly for $\mathbb{G}^{(p)}$-symmetric theories in $d$ dimensions.

We can now gauge the $\mathbb{G}^{(p)}$ symmetry of $\mathfrak{Z}_{\sigma}^{(\text{trivial})}$, and this leads us to the required SymTFT

$$\mathfrak{Z}(\mathcal{S}_{\mathbb{G}^{(p)}}^{\sigma}) = \mathfrak{Z}_{\sigma}^{(\text{trivial})}/\mathbb{G}^{(p)}. \tag{87}$$

The symmetry boundary $\mathfrak{B}_{\mathcal{S}_{\mathbb{G}^{(p)}}^{\sigma}}^{\text{sym}}$ can again be physically recognized as the **Dirichlet** boundary condition for the $\mathbb{G}^{(p)}$ gauge fields.

### 2.5.5 SymTFTs for non-invertible higher-representation symmetries

We discussed a class of symmetries in example 2.2 whose associated fusion $(d-1)$-categories are

$$\mathcal{S} = (d-1)\text{-Rep}(\mathbb{G}^{(p)}). \tag{88}$$

The associated SymTFT for such a symmetry is

$$\mathfrak{Z}(\mathcal{S}) = \mathfrak{Z}(\mathcal{S}_{\mathbb{G}^{(p)}}). \tag{89}$$

That is the SymTFT for (higher-)representation symmetry (88) is the same as the SymTFT for the non-anomalous $p$-group symmetry $\mathbb{G}^{(p)}$.

We will see that this is actually a general statement: two symmetries related by gauging have the same SymTFTs.

## 2.6 Comparison: Symmetry TFTs and relative theories

The notions encountered in the study of symmetry TFTs are closely related to the notions encountered in the study of relative theories. In this subsection, our aim is to describe in which situations these two closely related setups differ from each other, in order to avoid any potential confusions. Let us begin with the definition of a relative theory:[7]

---

**Definition 2.8: Relative theory**

A **relative $d$-dimensional theory** is simply another name for a boundary condition of a non-trivial $(d+1)$-dimensional TQFT.

---

Thus, the physical boundary $\mathfrak{B}^{\text{phys}}_{\mathfrak{T}_\sigma}$ arising in the sandwich construction involving the SymTFT is a relative theory. However, it should be noted that not all relative theories are physical boundaries for some SymTFT. This happens whenever the TQFT associated to a relative theory does not admit a topological boundary condition, as in that case this TQFT cannot be a SymTFT for any symmetry. Let us provide two well-known examples of relative theories that are of this type:

---

**Example 2.5: Moore-Seiberg setup [28–30]**

A chiral rational CFT (RCFT) is a relative theory as it is a boundary condition of a 3d TQFT whose line defects form the modular tensor category (MTC) associated to the corresponding chiral algebra. However, in general such a 3d TQFT does not admit a topological boundary condition, and hence cannot be the SymTFT for any symmetry. An example is provided by the chiral part of Ising CFT, in which case the associated 3d TQFT is described by the Ising MTC.

---

**Example 2.6: 6d $\mathcal{N} = (2,0)$ theories**

6d $\mathcal{N} = (2,0)$ SCFTs are relative theories as they naturally arise as boundary conditions of Chern-Simons-type 7d TQFTs [31,32]. These 7d TQFTs do not always have topological boundary conditions. Examples are provided by 7d TQFTs associated to 6d $\mathcal{N} = (2,0)$ SCFTs corresponding to Lie algebras $\mathfrak{su}(p)$ for prime $p$. Again for the same reason as above, these 7d TQFTs cannot be SymTFTs for any symmetry.

---

However, when a relative theory can be converted into an absolute theory, the setup of relative theories coincides with that of SymTFTs. First, recall the definition of an absolute theory

---

**Definition 2.9: Absolute theory**

An **absolute $d$-dimensional theory** is a theory that is well-defined without having to be attached to any non-trivial higher-dimensional theory.

---

[7]The terminology was introduced in [27] and has been used since then in a variety of contexts which are unified by the definition that follows.

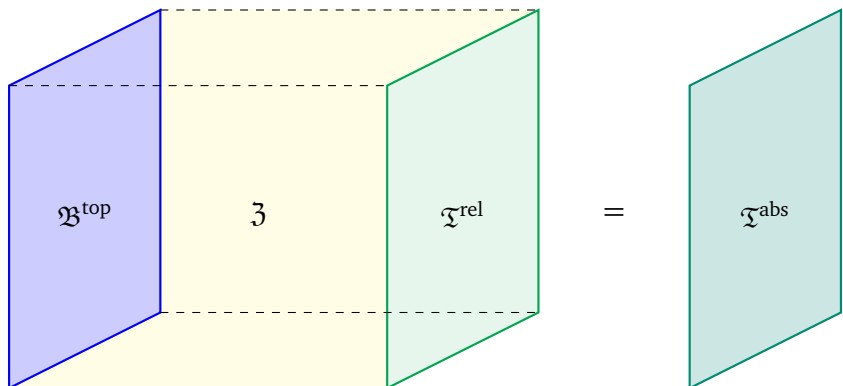

Figure 6: The sandwich construction converting a $d$-dimensional relative theory $\mathfrak{T}^{\text{rel}}$, which is a boundary condition of a $(d+1)$-dimensional TQFT $\mathfrak{Z}$, into an absolute theory $\mathfrak{T}^{\text{abs}}$, using a topological boundary condition $\mathfrak{B}^{\text{top}}$ of the TQFT $\mathfrak{Z}$.

One can convert a relative theory $\mathfrak{T}^{\text{rel}}$ into an absolute theory $\mathfrak{T}^{\text{abs}}$, if the $(d+1)$-dimensional TQFT $\mathfrak{Z}$ attached to the relative theory $\mathfrak{T}^{\text{rel}}$ admits a topological boundary condition $\mathfrak{B}^{\text{top}}$. This involves a sandwich construction similar to the one used in the context of SymTFT. See figure 6. Consequently, we can identify the two sandwich constructions as follows:

$$(\mathfrak{B}^{\text{sym}}_{\mathcal{S}}, \mathfrak{Z}(\mathcal{S}), \mathfrak{B}^{\text{phys}}_{\mathfrak{T}_\sigma}; \mathfrak{T}) = (\mathfrak{B}^{\text{top}}, \mathfrak{Z}, \mathfrak{T}^{\text{rel}}; \mathfrak{T}^{\text{abs}}), \tag{90}$$

with the symmetry $\mathcal{S}$ being identified as the symmetry category of the topological boundary

$$\mathcal{S} = \mathcal{S}(\mathfrak{B}^{\text{top}}). \tag{91}$$

## 2.7 Gauging and SymTFT

Since the symmetry $\mathcal{S}$ of a $d$-dimensional theory $\mathfrak{T}_\sigma$ can be separated out to the symmetry boundary $\mathfrak{B}^{\text{sym}}_{\mathcal{S}}$ of the SymTFT $\mathfrak{Z}(\mathcal{S})$, operations on the theory $\mathfrak{T}_\sigma$ related to the symmetry $\mathcal{S}$ are also separated out to operations on the symmetry boundary $\mathfrak{B}^{\text{sym}}_{\mathcal{S}}$. One such operation is that of gauging. In this subsection, we describe how it can be understood from the point of view of symmetry boundary $\mathfrak{B}^{\text{sym}}_{\mathcal{S}}$.

We want to consider the most general notion of gauging/condensation, which can be formalized into the existence of a topological interface between the original system and the system obtained after gauging. The physical idea behind this formalization is that we can provide Dirichlet boundary conditions to the gauge fields arising from the gauging procedure. The Dirichlet b.c. is then identified as the associated topological interface.

After the above physical motivation, let us describe the general procedure. Given an $\mathcal{S}$-symmetric theory $\mathfrak{T}_\sigma$, we can obtain an $\mathcal{S}'$-symmetric theory $\mathfrak{T}'_{\sigma'}$ if the following holds:

1. There exists a simple boundary condition $\mathfrak{B}^{\text{sym}}_{\mathcal{S}'}$ of the SymTFT $\mathfrak{Z}(\mathcal{S})$ such that the symmetry category of $\mathfrak{B}^{\text{sym}}_{\mathcal{S}'}$ is $\mathcal{S}'$

$$\mathcal{S}(\mathfrak{B}^{\text{sym}}_{\mathcal{S}'}) = \mathcal{S}'. \tag{92}$$

2. Moreover, there exists a topological $(d-1)$-dimensional interface

$$\mathcal{I}_{\mathcal{S},\mathcal{S}'}: \quad \mathfrak{B}^{\text{sym}}_{\mathcal{S}} \to \mathfrak{B}^{\text{sym}}_{\mathcal{S}'}, \tag{93}$$

from $\mathfrak{B}^{\text{sym}}_{\mathcal{S}}$ to $\mathfrak{B}^{\text{sym}}_{\mathcal{S}'}$, such that $\mathcal{I}_{\mathcal{S},\mathcal{S}'}$ is not attached to any $d$-dimensional topological defect of $\mathfrak{Z}(\mathcal{S})$. See figure 7. This means that the boundaries $\mathfrak{B}^{\text{sym}}_{\mathcal{S}'}$ and $\mathfrak{B}^{\text{sym}}_{\mathcal{S}}$ are related by condensation, meaning that they lie in the same **condensation component** of all topological boundary conditions of $\mathfrak{Z}(\mathcal{S})$.

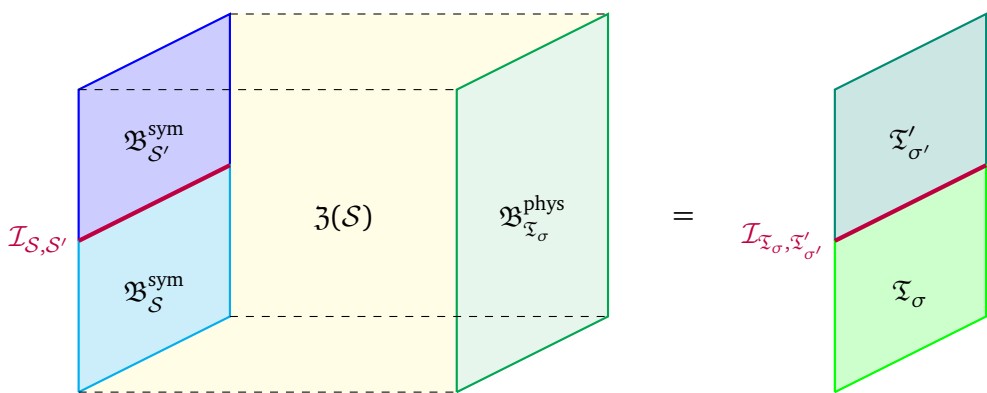

Figure 7: The interface $\mathcal{I}_{\mathcal{S},\mathcal{S}'}$ changes boundary $\mathfrak{B}_{\mathcal{S}}^{\mathrm{sym}}$ to boundary $\mathfrak{B}_{\mathcal{S}'}^{\mathrm{sym}}$. Using the two boundaries as symmetry boundaries in sandwich construction and keeping the physical boundary $\mathfrak{B}_{\mathfrak{T}_\sigma}^{\mathrm{phys}}$ fixed leads respectively to $\mathcal{S}$-symmetric theory $\mathfrak{T}_\sigma$ and $\mathcal{S}'$-symmetric theory $\mathfrak{T}'_{\sigma'}$. The interface $\mathcal{I}_{\mathcal{S},\mathcal{S}'}$ becomes a topological interface $\mathcal{I}_{\mathfrak{T}_\sigma,\mathfrak{T}'_{\sigma'}}$ from $\mathfrak{T}_\sigma$ to $\mathfrak{T}'_{\sigma'}$. In such a situation, the two topological boundary conditions $\mathfrak{B}_{\mathcal{S}}^{\mathrm{sym}}$ and $\mathfrak{B}_{\mathcal{S}'}^{\mathrm{sym}}$ are related by gauging, and equivalently the two theories $\mathfrak{T}_\sigma$ and $\mathfrak{T}'_{\sigma'}$ are related by gauging. Moreover, one also often says that the symmetries $\mathcal{S}$ and $\mathcal{S}'$ are related by gauging, or that they are dual symmetries.

Then $\mathfrak{T}'_{\sigma'}$ is the result of interval compactification of $\mathfrak{Z}(\mathcal{S})$ using the same physical boundary $\mathfrak{B}_{\mathfrak{T}_\sigma}^{\mathrm{phys}}$ but different symmetry boundary $\mathfrak{B}_{\mathcal{S}'}^{\mathrm{sym}}$. See figure 7. The compactification of $\mathcal{I}_{\mathcal{S},\mathcal{S}'}$ produces an $(\mathcal{S},\mathcal{S}')$-symmetric interface $\mathcal{I}_{\mathfrak{T}_\sigma,\mathfrak{T}'_{\sigma'}}$ from the $\mathcal{S}$-symmetric theory $\mathfrak{T}_\sigma$ to the $\mathcal{S}'$-symmetric theory $\mathfrak{T}'_{\sigma'}$. In such a situation, we say that the theories and symmetries are related by gauging:

---

**Definition 2.10: Theories related by gauging**

$\mathcal{S}'$-symmetric theory $\mathfrak{T}'_{\sigma'}$ and $\mathcal{S}$-symmetric theory $\mathfrak{T}_\sigma$ are related by gauging. That is, we can obtain $\mathfrak{T}'_{\sigma'}$ by gauging $\mathcal{S}$ symmetry of $\mathfrak{T}_\sigma$, and we can obtain $\mathfrak{T}_\sigma$ by gauging $\mathcal{S}'$ symmetry of $\mathfrak{T}'_{\sigma'}$.

---

**Definition 2.11: Symmetries related by gauging and Morita equivalence**

Symmetries $\mathcal{S}$ and $\mathcal{S}'$ are dual symmetries related to each other by gauging. That is, we can obtain $\mathcal{S}'$ by gauging $\mathcal{S}$, and vice versa.
In mathematical terminology, one also says that the fusion $(d-1)$ categories $\mathcal{S}$ and $\mathcal{S}'$ are **Morita equivalent** to each other.

---

Note that equivalence via gauging, i.e. Morita equivalence, implies that the associated SymTFT are same

$$\mathfrak{Z}(\mathcal{S}) = \mathfrak{Z}(\mathcal{S}'). \tag{94}$$

By performing gaugings of a symmetry $\mathcal{S}$, one can go to various different symmetries $\mathcal{S}'$. The full set of symmetries $\mathcal{S}'$ gauge-related to the symmetry $\mathcal{S}$ is parametrized by the symmetry categories of the boundary conditions lying in the condensation component of topological boundary conditions in which $\mathfrak{B}_{\mathcal{S}}^{\mathrm{sym}}$ lies.

---

### Example 2.7: Gauging of non-anomalous (higher-)group symmetries

As mentioned in example 2.5.5, the symmetries

$$\mathcal{S} = \mathcal{S}_{\mathbb{G}^{(p)}}, \quad \text{and} \quad \mathcal{S}' = (d-1)\text{-Rep}(\mathbb{G}^{(p)}), \tag{95}$$

are Morita equivalent, i.e. they are related by gauging. This has been discussed in detail in recent literature on symmetries [9, 10, 21].

The symmetry boundary conditions are as follows:

- For $\mathcal{S} = \mathcal{S}_{\mathbb{G}^{(p)}}$ the topological boundary conditions are **Dirichlet** for the $\mathbb{G}^{(p)}$ gauge fields.

- For $\mathcal{S}' = (d-1)\text{-Rep}(\mathbb{G}^{(p)})$ the topological boundary conditions are **Neumann** for the $\mathbb{G}^{(p)}$ gauge fields.

- The $(d-1)$-dimensional interface $\mathcal{I}_{\mathcal{S},\mathcal{S}'}$ can be physically understood as being constructed by imposing the Dirichlet boundary condition on the dynamical $\mathbb{G}^{(p)}$ gauge fields living on the Neumann boundary $\mathfrak{B}^{\text{sym}}_{\mathcal{S}'}$. This converts the Neumann boundary $\mathfrak{B}^{\text{sym}}_{\mathcal{S}'}$ to the Dirichlet boundary $\mathfrak{B}^{\text{sym}}_{\mathcal{S}}$.

In the special case when the $p$-group decomposes as a product of higher-form symmetry groups as in (68) with abelian $G^{(0)}$, the boundary condition $\mathfrak{B}^{\text{sym}}_{\mathcal{S}'}$ comprises of the Neumann boundary conditions for the gauge fields $a_r$, $0 \leq r \leq p-1$ and Dirichlet boundary conditions for the gauge fields $b_{d-r-1}$, $0 \leq r \leq p-1$ appearing in (69). Indeed, this claim can be easily verified explicitly. On the one hand, we have

$$\mathcal{S}' = \mathcal{S}_{\mathbb{G}^{(d-1)}_{\text{dual}}}, \tag{96}$$

associated to a non-anomalous $(d-1)$-group symmetry

$$\mathbb{G}^{(d-1)}_{\text{dual}} = \prod_{r=d-p-1}^{d-2} G^{(r)}_{\text{dual}}, \tag{97}$$

with the component $r$-form symmetry groups being

$$G^{(r)}_{\text{dual}} = \widehat{G}^{(d-r-2)}, \tag{98}$$

where the hat denotes the Pontryagin dual group. On the other hand, simply exchanging the Dirichlet with Neumann boundary conditions, interchanges the role of $a_r$ and $b_{d-r-2}$ in (69), thus identifying the resulting boundary condition as $\mathfrak{B}^{\text{sym}}_{\mathcal{S}_{\mathbb{G}^{(d-1)}_{\text{dual}}}}$ and

$$\mathfrak{Z}\left(\mathcal{S}_{\mathbb{G}^{(d-1)}_{\text{dual}}}\right) = \mathfrak{Z}\left(\mathcal{S}_{\mathbb{G}^{(p)}}\right). \tag{99}$$

One can also interchange Dirichlet and Neumann conditions for $(a_r, b_{d-r-2})$ for a few specific values of $r$, rather than all values of $r$ as done above. In this way, we obtain even more topological boundary conditions of $\mathfrak{Z}(\mathcal{S}_{\mathbb{G}^{(p)}})$. The symmetries captured by these boundary conditions are non-anomalous higher-group symmetries obtained from the $\mathbb{G}^{(p)}$ symmetry by gauging component $r$-form symmetry groups for those values of $r$ for which the Dirichlet-Neumann interchange was performed.

## 2.8 $\mathcal{S}$-symmetric TQFTs from SymTFT

In this subsection, we discuss an important physical application of SymTFTs. Given a UV $d$-dimensional theory, an important physical question is to understand all the possible infrared phases that the theory can flow to. If the UV theory carries symmetry $\mathcal{S}$, then the IR theory must also be $\mathcal{S}$-symmetric. As a simplification, we may restrict to understanding gapped/massive IR phases with symmetry $\mathcal{S}$, which are parametrized by possible $\mathcal{S}$-symmetric $d$-dimensional TQFTs. The classification of such TQFTs is thus of paramount importance and below we see that this classification can be rephrased as classification of topological boundary conditions of the SymTFT $\mathfrak{Z}(\mathcal{S})$. The latter problem is often much easier to tackle as one can bring in mathematical machinery that has been developed for understanding topological boundary conditions of TQFTs. We develop these lines of reasoning in more detail in [17, 18].

Given the above physical motivation, consider an irreducible $\mathcal{S}$-symmetric $d$-dimensional TQFT $\mathfrak{T}_\sigma$. According to statement 2.2, the physical boundary condition $\mathfrak{B}^{\text{phys}}_{\mathfrak{T}_\sigma}$ arising in its sandwich construction is simple. Moreover, since $\mathfrak{T}_\sigma$ is topological, the the physical boundary $\mathfrak{B}^{\text{phys}}_{\mathfrak{T}_\sigma}$ needs to also be topological. We are thus led to the following correspondence:

---

**Statement 2.3: Classification of $\mathcal{S}$-symmetric TQFTs**

Irreducible $\mathcal{S}$-symmetric $d$-dimensional TQFTs, for a fusion $(d-1)$-category $\mathcal{S}$, are in one-to-one correspondence with simple topological boundary conditions of the $(d+1)$-dimensional SymTFT $\mathfrak{Z}(\mathcal{S})$ associated to $\mathcal{S}$.

---

Let us denote by

$$\mathfrak{T}(\mathfrak{B}^{\text{sym}}_{\mathcal{S}'}) , \tag{100}$$

an irreducible $\mathcal{S}$-symmetric TQFT obtained by choosing the physical boundary $\mathfrak{B}^{\text{phys}}$ to be a simple topological boundary condition $\mathfrak{B}^{\text{sym}}_{\mathcal{S}'}$

$$\mathfrak{B}^{\text{phys}} = \mathfrak{B}^{\text{sym}}_{\mathcal{S}'} . \tag{101}$$

We have a few general comments:

- From the discussion of the previous subsection, we know that two such irreducible $\mathcal{S}$-symmetric TQFTs $\mathfrak{T}(\mathfrak{B}^{\text{sym}}_{\mathcal{S}'})$ and $\mathfrak{T}(\mathfrak{B}^{\text{sym}}_{\mathcal{S}''})$ are related by a gauging if the corresponding topological boundaries $\mathfrak{B}^{\text{sym}}_{\mathcal{S}'}$ and $\mathfrak{B}^{\text{sym}}_{\mathcal{S}''}$ are related by gauging.

  However, it should be noted that this gauging procedure does not involve the symmetry $\mathcal{S}$. In fact, by combining the symmetries descending from both the symmetry and physical boundaries, the $\mathcal{S}$-symmetric TQFT $\mathfrak{T}(\mathfrak{B}^{\text{sym}}_{\mathcal{S}'})$ actually has an enlarged

  $$\bar{\mathcal{S}}' \times \mathcal{S} , \tag{102}$$

  symmetry, where $\bar{\mathcal{S}}'$ denotes the opposite category of $\mathcal{S}'$, arising due to the fact that we need to invert the orientation of $\mathfrak{B}^{\text{sym}}_{\mathcal{S}'}$, when using it as a physical boundary rather than a symmetry boundary. The $\mathcal{S}$-symmetric TQFT $\mathfrak{T}(\mathfrak{B}^{\text{sym}}_{\mathcal{S}''})$ is obtained by gauging the $\bar{\mathcal{S}}'$ symmetry of $\mathfrak{T}(\mathfrak{B}^{\text{sym}}_{\mathcal{S}'})$. Similarly, the $\mathcal{S}$-symmetric TQFT $\mathfrak{T}(\mathfrak{B}^{\text{sym}}_{\mathcal{S}'})$ is obtained by gauging the $\bar{\mathcal{S}}''$ symmetry of $\mathfrak{T}(\mathfrak{B}^{\text{sym}}_{\mathcal{S}''})$.

- There is a canonical irreducible $\mathcal{S}$-symmetric TQFT for any symmetries $\mathcal{S}$, namely the TQFT

  $$\mathfrak{T}(\mathfrak{B}^{\text{sym}}_{\mathcal{S}}) , \tag{103}$$

obtained simply by choosing

$$\mathfrak{B}^{\text{phys}} = \mathfrak{B}^{\text{sym}}_{\mathcal{S}}. \tag{104}$$

Physically, $\mathfrak{T}(\mathfrak{B}^{\text{sym}}_{\mathcal{S}})$ describes an IR phase in which the symmetry $\mathcal{S}$ is completely **spontaneously broken**. See for instance the example 2.8 below where this notion of spontaneous breaking reduces to the well-known notion of spontaneous breaking of $G^{(0)}$ 0-form symmetry.

Let us observe the statement 2.3 at play in the following example:

---

**Example 2.8: $G^{(0)}$-symmetric 2d TQFTs**

It is well-known that irreducible 2d TQFTs with a non-anomalous $G^{(0)}$ 0-form symmetry are classified by:

1. A subgroup $H \subseteq G^{(0)}$ describing the symmetry that is not broken spontaneously in a particular vacuum $v$ of the TQFT.

2. An element

$$\beta \in H^2(H, \mathbb{C}^\times), \tag{105}$$

describing the SPT phase for the $H$ symmetry arising in the vacuum $v$.

Let us denote such a $G^{(0)}$-symmetric 2d TQFT as

$$\mathfrak{T}_{(H,\beta)}. \tag{106}$$

The structure of the full $G^{(0)}$-symmetric TQFT including the symmetry properties of the other vacua is completely determined in terms of the above two pieces of information.

On the other hand, the above two pieces of information also describe possible gaugings of a non-anomalous $G^{(0)}$ 0-form symmetry in 2d. $H$ is the subgroup being gauged and $\beta$ captures the discrete torsion for the gauging. Thus we have simple topological boundary conditions

$$\mathfrak{B}^{\text{sym}}_{(H,\beta)} \equiv \mathfrak{B}^{\text{sym}}_{\mathcal{S}_{G^{(0)}}}/(H,\beta), \tag{107}$$

of the SymTFT $\mathfrak{Z}(\mathcal{S}_{G^{(0)}})$ obtained by performing the corresponding gaugings of the $G^{(0)}$ symmetry of the symmetry boundary $\mathfrak{B}^{\text{sym}}_{\mathcal{S}_{G^{(0)}}}$.

These two facts are related as follows. The $G^{(0)}$-symmetric TQFT $\mathfrak{T}_{(H,\beta)}$ is obtained by performing the interval compactification of the SymTFT $\mathfrak{Z}(\mathcal{S}_{G^{(0)}})$ with $\mathfrak{B}^{\text{sym}}_{\mathcal{S}_{G^{(0)}}}$ as the symmetry boundary condition and

$$\mathfrak{B}^{\text{phys}} = \mathfrak{B}^{\text{sym}}_{(H,\beta)}, \tag{108}$$

as the physical boundary condition. That is, we have

$$\mathfrak{T}(\mathfrak{B}^{\text{sym}}_{(H,\beta)}) = \mathfrak{T}_{(H,\beta)}. \tag{109}$$

---

# 3 Generalized charges and the Drinfeld center

The previous section discussed aspects related to symmetries of a $d$-dimensional theory. In this section, we discuss how these symmetries act on operators of various dimensions in the theory.

Under the action of a symmetry, the operators combine together into multiplets, where each multiplet of operators can be associated a **generalized charge** for that symmetry. We further characterize generalized charges as follows:

---

**Definition 3.1: $q$-charges**

Generalized charges of $q$-dimensional operators are referred to as **$q$-charges**.

---

Note that not every $q$-charge needs to be realized in a theory having the symmetry. That is, given a particular $q$-charge, the theory may not contain a multiplet of $q$-dimensional operators transforming in that $q$-charge.

## 3.1 Generalized charges from SymTFT

In this subsection, we will see that generalized charges for a symmetry $\mathcal{S}$ can be neatly encoded in terms of the associated SymTFT $\mathfrak{Z}(\mathcal{S})$. See statement 3.2 for the main result.

**Sandwich construction of an uncharged operator.** Let us begin by considering a simple (possibly non-topological) genuine $q$-dimensional operator $\mathcal{O}_q$ in a $d$-dimensional $\mathcal{S}$-symmetric theory $\mathfrak{T}_\sigma$, and ask how it can be constructed via the sandwich construction. The simplest possibility, considered in figure 8, is that $\mathcal{O}_q$ lifts to a $q$-dimensional simple operator $\mathcal{M}_q$ living on the physical boundary $\mathfrak{B}_{\mathfrak{T}_\sigma}^{\text{phys}}$ that is not attached to any bulk operator of the SymTFT $\mathfrak{Z}(\mathcal{S})$. However, this means that $\mathcal{O}_q$ is left completely invariant by the action of the symmetry $\mathcal{S}$! This is because the symmetry $\mathcal{S}$ lives on the symmetry boundary $\mathfrak{B}_{\mathcal{S}}^{\text{sym}}$, and thus does not talk to the operator $\mathcal{M}_q$ living on the physical boundary $\mathfrak{B}_{\mathfrak{T}_\sigma}^{\text{phys}}$, which after interval compactification translates to the fact that the symmetry $\mathcal{S}$ of $\mathfrak{T}_\sigma$ does not act on the operator $\mathcal{O}_q$ of $\mathfrak{T}$.

**Sandwich construction of a charged operator.** Thus, if $\mathcal{O}_q$ carries a non-trivial $q$-charge under $\mathcal{S}$, its sandwich construction must be more complicated. The boundary operator $\mathcal{M}_q$ must be attached to a simple non-identity bulk topological operator $\boldsymbol{Q}_{q+1}$ of $\mathfrak{Z}(\mathcal{S})$, and the bulk operator $\boldsymbol{Q}_{q+1}$ must itself end on the symmetry boundary $\mathfrak{B}_{\mathcal{S}}^{\text{sym}}$ along a simple $q$-dimensional topological operator $\mathcal{E}_q$. See figure 9. Now the action of $\mathcal{S}$ symmetry on $\mathcal{O}_q$ is encoded in how the topological operators of $\mathfrak{B}_{\mathcal{S}}^{\text{sym}}$ interact with the ends $\mathcal{E}_q$ of $\boldsymbol{Q}_{q+1}$ along $\mathfrak{B}_{\mathcal{S}}^{\text{sym}}$. The latter information is purely a SymTFT information, encoded in how $\mathfrak{B}_{\mathcal{S}}^{\text{sym}}$ serves as a boundary condition of the TQFT $\mathfrak{Z}(\mathcal{S})$.

**Symmetry action generating a multiplet of charged operators.** The action of the topological operators $S_r$ of $\mathfrak{B}_{\mathcal{S}}^{\text{sym}}$ may convert an end $\mathcal{E}_q$ of $\boldsymbol{Q}_{q+1}$ to another end $\mathcal{E}'_q$ of $\boldsymbol{Q}_{q+1}$ along $\mathfrak{B}_{\mathcal{S}}^{\text{sym}}$

$$S_r: \qquad \mathcal{E}_q \to \mathcal{E}'_q. \tag{110}$$

Performing the sandwich construction using the end $\mathcal{E}'_q$ along $\mathfrak{B}_{\mathcal{S}}^{\text{sym}}$, while keeping the end $\mathcal{M}_q$ along $\mathfrak{B}_{\mathfrak{T}_\sigma}^{\text{phys}}$ fixed, leads to a different $q$-dimensional operator of $\mathfrak{T}$ that we label $\mathcal{O}'_q$. See figure 9. We learn that the symmetry $\mathcal{S}$ of $\mathfrak{T}_\sigma$ can act as

$$D_r = \sigma(S_r): \qquad \mathcal{O}_q \to \mathcal{O}'_q. \tag{111}$$

Physically, in such a situation, one would say that $\mathcal{O}_q$ and $\mathcal{O}'_q$ live in the same irreducible **multiplet** of $q$-dimensional operators under the action of symmetry $\mathcal{S}$.

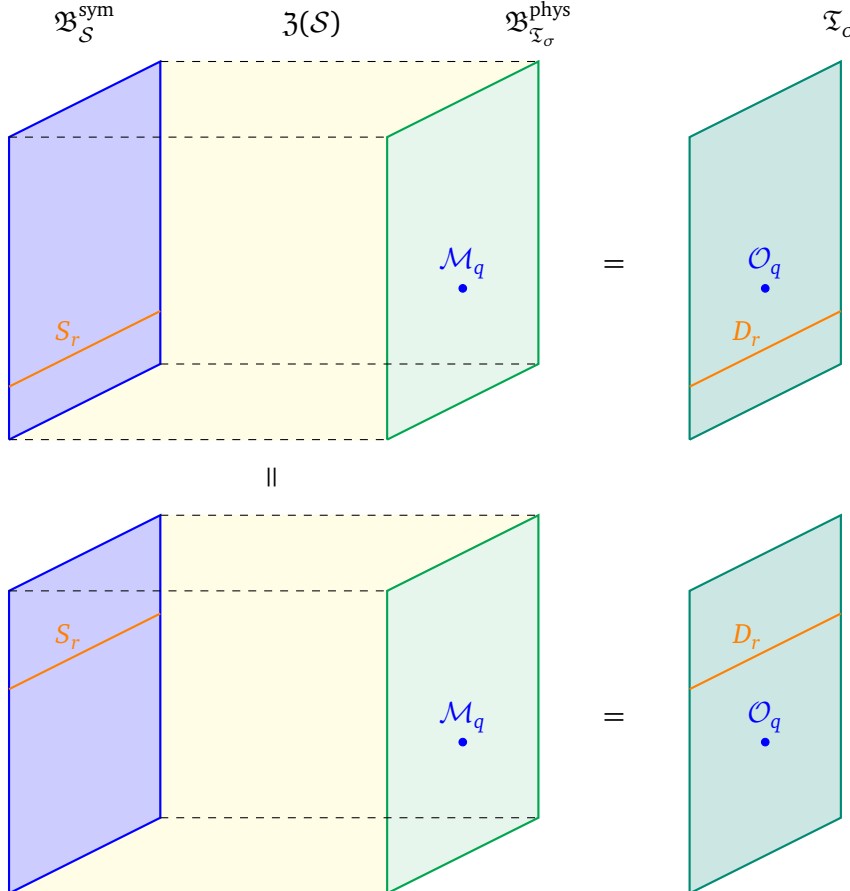

Figure 8: The sandwich construction of a $q$-dimensional operator $\mathcal{O}_q$ uncharged under $\mathcal{S}$ involves a $q$-dimensional operator $\mathcal{M}_q$ along the physical boundary $\mathfrak{B}^{\text{phys}}_{\mathfrak{T}_\sigma}$ that is not attached to any defects of the bulk SymTFT $\mathfrak{Z}(\mathcal{S})$. Since the symmetry is separated out to the other boundary, i.e. the symmetry boundary $\mathfrak{B}^{\text{sym}}_{\mathcal{S}}$, it does not act on $\mathcal{M}_q$, implying that $\mathcal{O}_q$ is uncharged. In the figure, $D_r := \sigma(S_r) \in \mathcal{S}(\mathfrak{T})$ is the $r$-dimensional topological defect of $\mathfrak{T}$ generating the symmetry corresponding to an $r$-dimensional topological defect $S_r \in \mathcal{S}$ of $\mathfrak{B}^{\text{sym}}_{\mathcal{S}}$.

**Twisted and untwisted sector operators in same multiplet.** In fact, if the symmetry $S_r \in \mathcal{S}$ is non-invertible, the action can be more complicated. The operator $\mathcal{E}'_q$ obtained by acting on the operator $\mathcal{E}_q$ may live at the end of a $(q+1)$-dimensional topological defect $S_{q+1}$ of $\mathfrak{B}^{\text{sym}}_{\mathcal{S}}$. See figure 10. The resulting operator $\mathcal{O}'_q$ of $\mathfrak{T}$ obtained after performing interval compactification of $Q_{q+1}$ with $\mathcal{M}_q$ as one end and $\mathcal{E}'_q$ as the other end, lives at the end of the $(q+1)$-dimensional topological defect $D_{q+1} = \sigma(S_{q+1})$ of $\mathfrak{T}$. In such a situation, one says that $\mathcal{O}'_q$ lives in the **twisted sector**[8] for the symmetry $S_{q+1}$. Thus the action of a non-invertible symmetry combines together untwisted and twisted sector operators in the same irreducible multiplet. More generally, a non-invertible symmetry mixes operators in different twisted sectors together.

**Multiplet as an operator on physical boundary.** Above we saw that two operators $\mathcal{O}_q$ and $\mathcal{O}'_q$ of $\mathfrak{T}$ obtained via the sandwich constructions involving the same operator $\mathcal{M}_q$ along $\mathfrak{B}^{\text{phys}}_{\mathfrak{T}_\sigma}$

---

[8]Note that such a twisted sector operator is in general a non-genuine operator living at the boundary of $D_{q+1}$, but could be a genuine operator if $D_{q+1}$ is the identity defect, i.e. if $D_{q+1} = D^{(\text{id})}_{q+1}$.

but different ends $\mathcal{E}_q$ and $\mathcal{E}'_q$ along $\mathfrak{B}^{\mathrm{sym}}_{\mathcal{S}}$ are in the same multiplet, if $\mathcal{E}_q$ and $\mathcal{E}'_q$ are related by the action of some symmetry $S_r \in \mathcal{S}$.

The key point in this discussion can be stated more simply. Beginning with an operator $\mathcal{O}_q$ of $\mathfrak{T}$ and studying its sandwich construction leads us to an operator $\mathcal{M}_q$ along $\mathfrak{B}^{\mathrm{phys}}_{\mathfrak{T}_\sigma}$ attached to a $(q+1)$-dimensional topological operator $\boldsymbol{Q}_{q+1}$ of $\mathfrak{Z}(\mathcal{S})$. Now this operator $\mathcal{M}_q$ can be used to construct a variety of operators $\mathcal{O}'_q$ of $\mathfrak{T}$ by performing sandwich construction with various ends $\mathcal{E}'_q$ of $\boldsymbol{Q}_{q+1}$ along $\mathfrak{B}^{\mathrm{sym}}_{\mathcal{S}}$. Thus using the existence of operator $\mathcal{O}_q$ of $\mathfrak{T}$, we have uncovered the existence of many other operators $\mathcal{O}'_q$ of $\mathfrak{T}$, where we only used information about the symmetry $\mathcal{S}$ in the process.

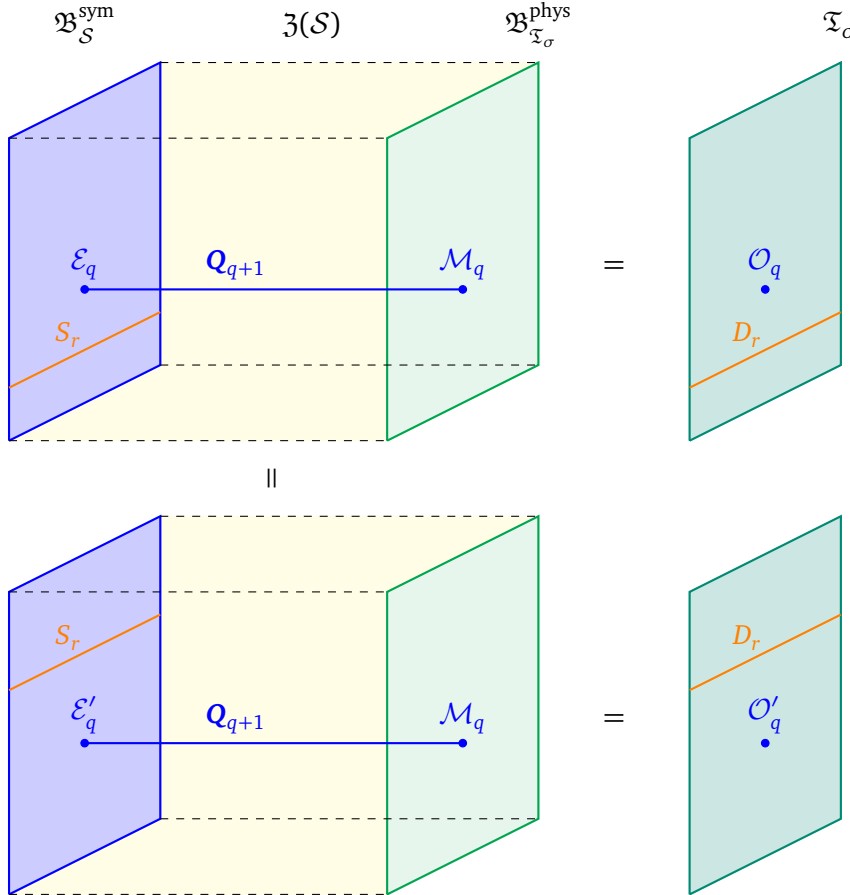

Figure 9: The sandwich construction of a $q$-dimensional operator $\mathcal{O}_q$ that is non-trivially charged under $\mathcal{S}$ involves a $q$-dimensional operator $\mathcal{M}_q$ along the physical boundary $\mathfrak{B}^{\mathrm{phys}}_{\mathfrak{T}_\sigma}$ that is attached to a non-trivial topological defect $\boldsymbol{Q}_{q+1}$ of the bulk SymTFT $\mathfrak{Z}(\mathcal{S})$. The bulk operator $\boldsymbol{Q}_{q+1}$ needs to subsequently end along a topological $q$-dimensional operator $\mathcal{E}_q$ along $\mathfrak{B}^{\mathrm{sym}}_{\mathcal{S}}$. As shown in the figure, an $r$-dimensional topological operator $S_r$ can act on $\mathcal{E}_q$ transforming into another $q$-dimensional topological end $\mathcal{E}'_q$ of $\boldsymbol{Q}_{q+1}$ along $\mathfrak{B}^{\mathrm{sym}}_{\mathcal{S}}$. This implies that, after the interval compactification, the $r$-dimensional topological operator $D_r := \sigma(S_r)$ of $\mathfrak{T}$ generating the symmetry $S_r$ acts on $\mathcal{O}_q$ transforming it into another $q$-dimensional operator $\mathcal{O}'_q$, which originates from the same operator $\mathcal{M}_q$ along $\mathfrak{B}^{\mathrm{phys}}_{\mathfrak{T}_\sigma}$, but the other end $\mathcal{E}'_q$ along $\mathfrak{B}^{\mathrm{sym}}_{\mathcal{S}}$. We say that the two operators $\mathcal{O}_q$ and $\mathcal{O}'_q$ live in the same irreducible multiplet under $\mathcal{S}$.

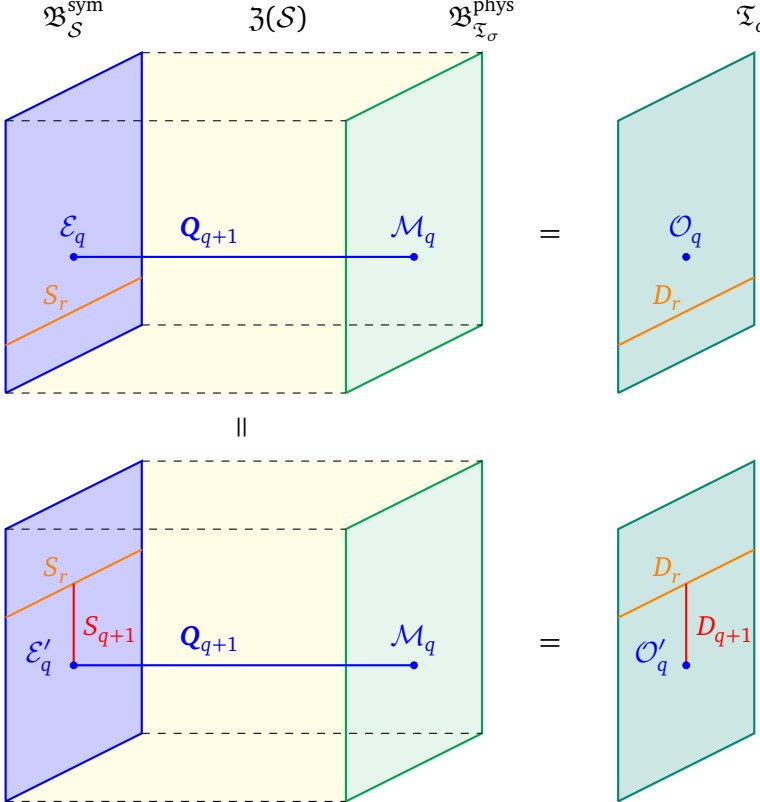

Figure 10: A non-invertible $r$-dimensional topological operator $S_r$ can act on $\mathcal{E}_q$ transforming it into another $q$-dimensional topological end $\mathcal{E}'_q$ of $\boldsymbol{Q}_{q+1}$ along $\mathfrak{B}^{\mathrm{sym}}_{\mathcal{S}}$, with the end $\mathcal{E}'_q$ having the property that it is attached to a $(q+1)$-dimensional topological defect $S_{q+1}$ of $\mathfrak{B}^{\mathrm{sym}}_{\mathcal{S}}$. This implies that the $q$-dimensional operator $\mathcal{O}'_q$ of $\mathfrak{T}$ lives in the same multiplet as the untwisted sector operator $\mathcal{O}_q$. The operator $\mathcal{O}'_q$ is in twisted sector for the symmetry $S_{q+1}$ and lives at the end of the corresponding topological defect $D_{q+1}$ of $\mathfrak{T}$ generating the symmetry $S_{q+1}$. Thus, in the presence of non-invertible symmetries, an irreducible multiplet of operators contains both untwisted and twisted sector operators.

Thus it is reasonable to refer to all operators $\mathcal{O}'_q$ obtainable from $\mathcal{O}_q$ as lying in the same multiplet as $\mathcal{O}_q$ under $\mathcal{S}$. From the above discussion we see that the whole multiplet can be deduced beginning from the operator $\mathcal{M}_q$ along $\mathfrak{B}^{\mathrm{phys}}_{\mathfrak{T}_\sigma}$. Thus, we are led to the following statement:

---

**Statement 3.1: Irreducible multiplets from the physical boundary**

An **irreducible multiplet** of simple $q$-dimensional operators under symmetry $\mathcal{S}$ of a theory $\mathfrak{T}_\sigma$ corresponds to a simple $q$-dimensional operator $\mathcal{M}_q$ along the associated physical boundary $\mathfrak{B}^{\mathrm{phys}}_{\mathfrak{T}_\sigma}$, which is allowed to be attached to a simple $(q+1)$-dimensional topological operator $\boldsymbol{Q}_{q+1}$ of the SymTFT $\mathfrak{Z}(\mathcal{S})$:

$$\underset{\boldsymbol{Q}_{q+1} \qquad \mathcal{M}_q}{\rule{4cm}{0.4pt}\bullet} \ . \tag{112}$$

---

A simple $q$-dimensional operator $\mathcal{O}_q$ of $\mathfrak{T}_\sigma$ in the multiplet $\mathcal{M}_q$ lives generally at the end of a simple topological $(q+1)$-dimensional operator $D_{q+1}$, and is extracted by choosing a simple $q$-dimensional operator $\mathcal{E}_q$ lying at the end of $\boldsymbol{Q}_{q+1}$ along the symmetry boundary $\mathfrak{B}^{\text{sym}}_{\mathcal{S}}$, which in general is attached to a simple $(q+1)$-dimensional topological operator $S_{q+1}$ of $\mathfrak{B}^{\text{sym}}_{\mathcal{S}}$.

$$
\begin{matrix}
\color{red}{S_{q+1}} & \color{blue}{\boldsymbol{Q}_{q+1}} & \color{blue}{\mathcal{M}_q} \\
\color{blue}{\mathcal{E}_q} \rule[-1ex]{0pt}{3ex} & \rule{6em}{0.4pt} & 
\end{matrix}
\quad = \quad
\begin{matrix}
\color{red}{D_{q+1}} \\
\color{blue}{\mathcal{O}_q}
\end{matrix}
\qquad (113)
$$

See the preceding discussion and figures 9 and 10 for more details.

**Generalized charge associated to a multiplet.** The action of $\mathcal{S}$ on simple $q$-dimensional operators of $\mathfrak{T}_\sigma$ living in an irreducible multiplet $\mathcal{M}_q$ is captured in how $\boldsymbol{Q}_{q+1}$ interacts with the topological symmetry boundary $\mathfrak{B}^{\text{sym}}_{\mathcal{S}}$. This is essentially the definition of the bulk operator $\boldsymbol{Q}_{q+1}$ in terms of the information of $\mathfrak{B}^{\text{sym}}_{\mathcal{S}}$, leading us to the following characterization of generalized charges in terms of SymTFT, which is the main statement of this paper:

> **Statement 3.2: Generalized charges as topological defects of SymTFT**
>
> Consider any $d$-dimensional QFT, whose symmetry is described by a fusion $(d-1)$-category $\mathcal{S}$. Then:
> **The generalized charges for the symmetry $\mathcal{S}$ are topological defects of the SymTFT $\mathfrak{Z}(\mathcal{S})$ associated to $\mathcal{S}$.**
> In particular, **$q$-charges are $(q+1)$-dimensional topological defects of $\mathfrak{Z}(\mathcal{S})$.** This includes $q$-charges in the range of $0 \leq q \leq d-2$. We do not study the general structure of $(d-1)$-charges in this paper.[9]
> More concretely, consider an irreducible multiplet of $q$-dimensional simple operators of $\mathfrak{T}_\sigma$ transforming under $\mathcal{S}$. The operators in this multiplet transform in the irreducible $q$-charge corresponding to a simple $(q+1)$-dimensional topological operator $\boldsymbol{Q}_{q+1}$ of $\mathfrak{Z}(\mathcal{S})$. The operator $\boldsymbol{Q}_{q+1}$ is attached to the simple $q$-dimensional topological operator $\mathcal{M}_q$ of $\mathfrak{B}^{\text{phys}}_{\mathfrak{T}_\sigma}$ corresponding to the irreducible multiplet under consideration.

It should be noted that in a particular $\mathcal{S}$-symmetric theory $\mathfrak{T}_\sigma$, there may not exist a multiplet of $q$-dimensional operators transforming under a $q$-charge $\boldsymbol{Q}_{q+1}$. In such a situation, although the SymTFT $\mathfrak{Z}(\mathcal{S})$ contains the corresponding $(q+1)$-dimensional topological operator $\boldsymbol{Q}_{q+1}$, it does not admit a $q$-dimensional end $\mathcal{M}_q$ along the physical boundary $\mathfrak{B}^{\text{phys}}_{\mathfrak{T}_\sigma}$.

In the above statement 3.2, we can include both genuine and non-genuine $(q+1)$-dimensional topological operators of the SymTFT $\mathfrak{Z}(\mathcal{S})$. Consequently, the higher-categorical structure of topological defects descends to generalized charges.

> **Statement 3.3: Generalized charges and Drinfeld center**
>
> $q$-charges (for $0 \leq q \leq d-2$) of a symmetry $\mathcal{S}$ combine to form the structure of a fusion $(d-1)$-category $\mathcal{Z}(\mathcal{S})$ of topological defects of codimension greater than or equal to 2

---

[9]The main reason for this is that in general we now need to account for ends of $d$ dimensional topological operators of $\mathfrak{Z}(\mathcal{S})$ on the symmetry boundary that change the symmetry boundary condition – these would correspond to $(d-1)$-charges of interfaces between an $\mathcal{S}$-symmetric theory $\mathfrak{T}$ and a gauged version $\mathfrak{T}/\mathcal{S}$ of it.

of the SymTFT $\mathfrak{Z}(\mathcal{S})$. This $(d-1)$-category $\mathcal{Z}(\mathcal{S})$ is known as the **Drinfeld center** of the fusion $(d-1)$-category $\mathcal{S}$.

The Drinfeld center $\mathcal{Z}(\mathcal{S})$ can be expressed in terms of the full symmetry category $\mathcal{S}\big(\mathfrak{Z}(\mathcal{S})\big)$ of the SymTFT $\mathfrak{Z}(\mathcal{S})$, which is a fusion $d$-category, as

$$\mathcal{Z}(\mathcal{S}) = \mathrm{End}_{\mathcal{S}\left(\mathfrak{Z}(\mathcal{S})\right)}\left(\mathbf{Q}_d^{(\mathrm{id})}\right), \tag{114}$$

where the right hand side denotes the endomorphism category, which is a fusion $(d-1)$-category, of the identity $d$-dimensional defect $\mathbf{Q}_d^{(\mathrm{id})}$ of $\mathfrak{Z}(\mathcal{S})$, or in other words the identity object of $\mathcal{S}\big(\mathfrak{Z}(\mathcal{S})\big)$.

**Invariance of generalized charges under gauging.** As discussed in the previous section, the SymTFT for two symmetries $\mathcal{S}$ and $\mathcal{S}'$ related by gauging are same

$$\mathfrak{Z}(\mathcal{S}) = \mathfrak{Z}(\mathcal{S}'). \tag{115}$$

This implies that the generalized charges for the two symmetries, which are topological defects of the SymTFTs, must also be the same

$$\mathcal{Z}(\mathcal{S}) = \mathcal{Z}(\mathcal{S}'). \tag{116}$$

In other words, gauging preserves the collection of generalized charges.

However, note that the corresponding symmetry boundaries are different

$$\mathfrak{B}_{\mathcal{S}}^{\mathrm{sym}} \neq \mathfrak{B}_{\mathcal{S}'}^{\mathrm{sym}}. \tag{117}$$

These boundaries carrying different collections of topological defects along them, which are respectively described by $\mathcal{S}$ and $\mathcal{S}'$. Consequently, the possible topological ends of a $(q+1)$-dimensional topological defect $\mathbf{Q}_{q+1}$ of the SymTFT are different along the two boundaries. Moreover, the action of topological defects living on these boundaries on the ends are different.

Thus, even though the generalized charges for $\mathcal{S}$ and $\mathcal{S}'$ form the same higher-category, the structure of a multiplet of operators transforming in a particular generalized charge, and the action of the symmetries on the multiplet, can be highly different for two cases $\mathcal{S}$ and $\mathcal{S}'$.

## 3.2 Computation of generalized charges

In this section we will develop how the topological defects of the SymTFT can be computed. We will motivate the definition of the Drinfeld center using a physical approach. The Drinfeld center of fusion 2-categories has been defined in [33, 34] and see [35] for study of Drinfeld center of fusion 3-categories.

### 3.2.1 Projections and Drinfeld center

As we discussed in the previous subsection, generalized charges of a symmetry $\mathcal{S}$ of $d$-dimensional theories form a (braided) fusion $(d-1)$ category $\mathcal{Z}(\mathcal{S})$ known as the Drinfeld center of the fusion $(d-1)$-category $\mathcal{S}$. The center $\mathcal{Z}(\mathcal{S})$ can be constructed from just the information of $\mathcal{S}$. In other words, the generalized charges of a symmetry $\mathcal{S}$ can be computed from just the information of the symmetry $\mathcal{S}$. Below we elucidate the mathematical construction of the center in physical terms.

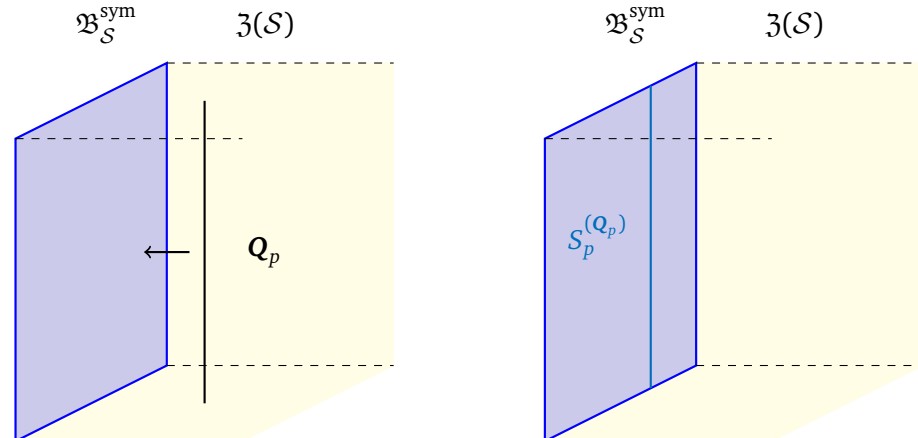

Figure 11: Projecting the bulk topological operator $Q_p$ parallel to the boundary gives rise to topological defect $S_p^{(Q_p)}$ on the $\mathfrak{B}_{\mathcal{S}}^{\text{sym}}$ boundary.

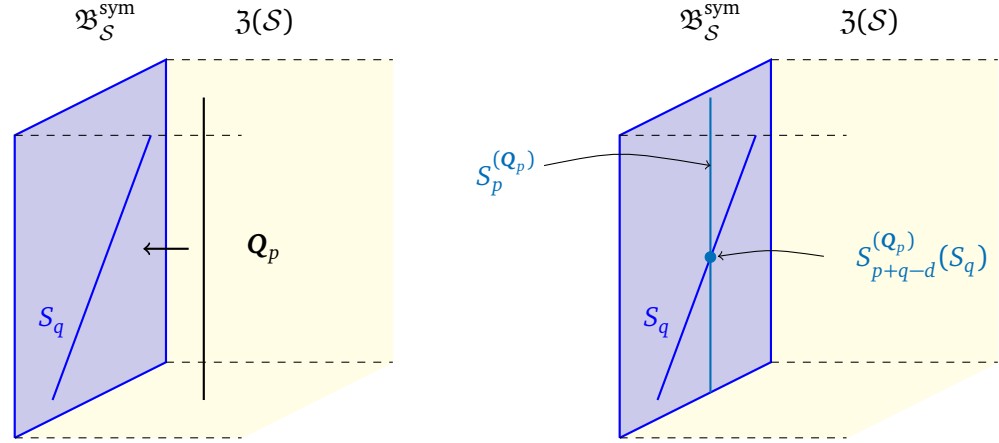

Figure 12: Projecting the bulk topological operator $Q_p$ parallel to the boundary in the presence of a $q$-dimensional topological defect $S_q$ on the boundary, which results in a junction operator $S_{p+q-d}^{(Q_p)}(S_q)$, which is sometimes referred to as the half-braiding.

**Projection.** The most basic information about a $p$-dimensional defect $Q_p \in \mathcal{Z}(\mathcal{S})$ is that of a $p$-dimensional defect $S_p^{(Q_p)} \in \mathcal{S}$. Physically, these two $p$-dimensional defects are related by a projection map as shown in figure 11. We take the topological defect $Q_p$ of the SymTFT $\mathfrak{Z}(\mathcal{S})$ parallel to the symmetry boundary $\mathfrak{B}_{\mathcal{S}}^{\text{sym}}$ and fuse it with $\mathfrak{B}_{\mathcal{S}}^{\text{sym}}$ to obtain the topological defect $S_p^{(Q_p)}$ of $\mathfrak{B}_{\mathcal{S}}^{\text{sym}}$. We refer to this process as projection of $Q_p$ onto $\mathfrak{B}_{\mathcal{S}}^{\text{sym}}$.

It should be noted that the projection does not preserve simplicity of operators. That is, even if $Q_p$ is a simple topological operator, its projection $S_p^{(Q_p)}$ may not be a simple topological operator. For example this happens when considering non-abelian group-like symmetries.

**Projections in the background of a boundary operator.** Consider a $q$-dimensional topological defect $S_q \in \mathcal{S}$ of $\mathfrak{B}_{\mathcal{S}}^{\text{sym}}$, where $p + q \geq d$ and consider performing the projection of $Q_p$ onto $\mathfrak{B}_{\mathcal{S}}^{\text{sym}}$ in the background of $S_q$ as shown in figure 12. The projection now provides a $(p + q - d)$-dimensional topological operator $S_{p+q-d}^{(Q_p)}(S_q)$ of $\mathfrak{B}_{\mathcal{S}}^{\text{sym}}$ living at the intersection of $S_q$ and $S_p^{(Q_p)}$.

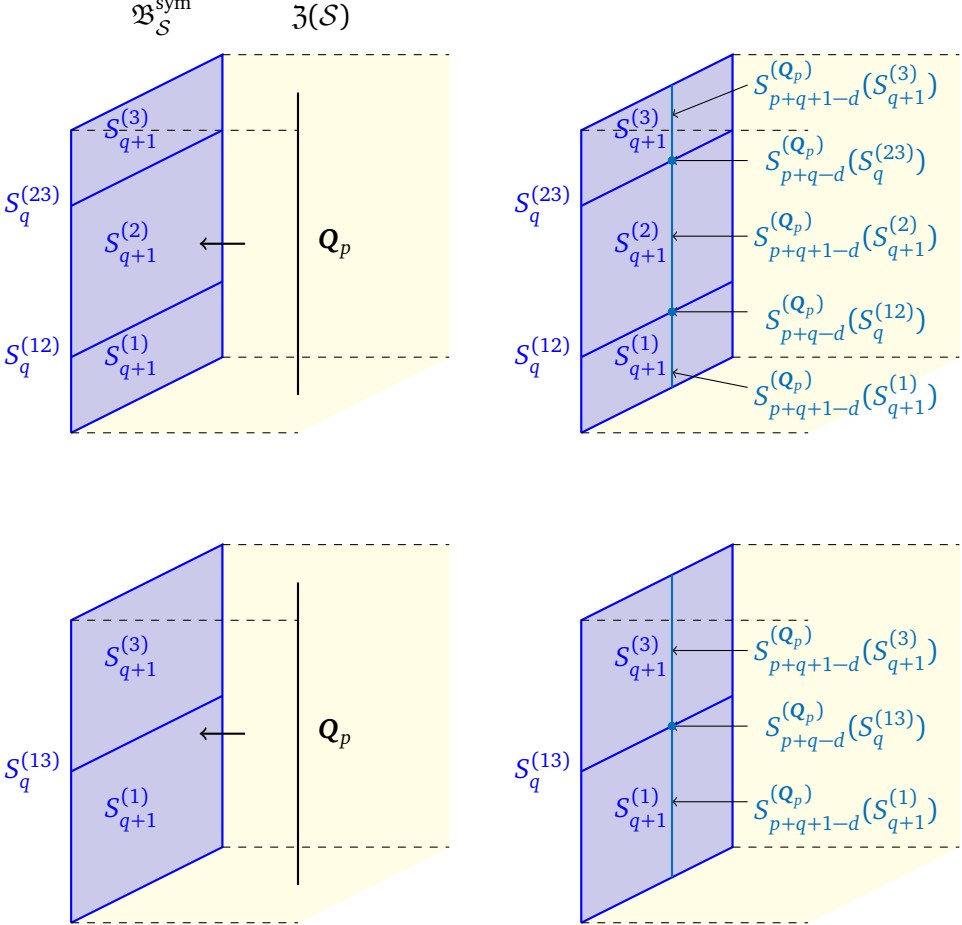

Figure 13: Consistency condition between projection and fusion: The top diagrams show three $q + 1$-dimensional topological defects on $\mathfrak{B}^{\mathrm{sym}}_{\mathcal{S}}$ with junctions $S_q^{(i\,i+1)}$. We can project the bulk defect (top right figure). Alternatively, one can fuse the two junctions $S_q^{(12)}$ and $S_q^{(23)}$ first and then project $Q_p$ onto the boundary – this is shown at the bottom. These operations need to commute. Let us emphasize that the defects $S_{q+1}^{(i)}$ do not fill the whole boundary $\mathfrak{B}^{\mathrm{sym}}_{\mathcal{S}}$, i.e. we have $q + 1 < d$.

**Consistency conditions on projections.**    So far we have associated to a defect $Q_p \in \mathcal{Z}(\mathcal{S})$ a collection of defects in $\mathcal{S}$ of various dimensions

$$\left\{ \left( S_p^{(Q_p)}, \, S_{p+q-d}^{(Q_p)}(S_q) \right) \middle| \, q \geq d - p, \, S_q \in \mathcal{S} \right\} . \tag{118}$$

The operator

$$S_{p+q-d}^{(Q_p)}(S_q), \tag{119}$$

is often referred to as the half-braiding. This collection of defects needs to satisfy various consistency conditions.

Consider topological defects

$$\begin{aligned} S_q^{(12)} : \quad & S_{q+1}^{(1)} \to S_{q+1}^{(2)} , \\ S_q^{(23)} : \quad & S_{q+1}^{(2)} \to S_{q+1}^{(3)} , \end{aligned} \tag{120}$$

of $\mathfrak{B}^{\mathrm{sym}}_{\mathcal{S}}$ with $q$ satisfying the condition that $p + q \geq d$. Projecting $Q_p$ onto this configuration of defects of $\mathfrak{B}^{\mathrm{sym}}_{\mathcal{S}}$ as in figure 13, we obtain $S_{p+q-d}^{(Q_p)}(S_q^{(12)})$ converting $S_{p+q+1-d}^{(Q_p)}(S_{q+1}^{(1)})$

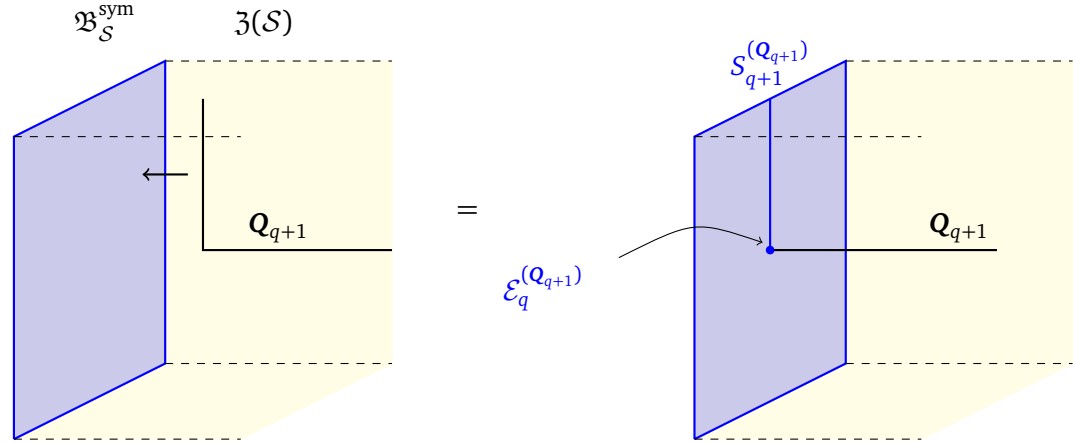

Figure 14: Projecting half of an L-shaped bulk topological operator $Q_{q+1}$ on the symmetry boundary $\mathfrak{B}_{\mathcal{S}}^{\text{sym}}$ produces a canonical topological end $\mathcal{E}_q^{(Q_{q+1})}$ of $Q_{q+1}$ along $\mathfrak{B}_{\mathcal{S}}^{\text{sym}}$, which is attached to the topological operator $S_{q+1}^{(Q_{q+1})}$ of $\mathfrak{B}_{\mathcal{S}}^{\text{sym}}$.

into $S_{p+q+1-d}^{(Q_p)}(S_{q+1}^{(2)})$ as it passes $S_q^{(12)}$, and $S_{p+q-d}^{(Q_p)}(S_q^{(23)})$ converting $S_{p+q+1-d}^{(Q_p)}(S_{q+1}^{(2)})$ into $S_{p+q+1-d}^{(Q_p)}(S_{q+1}^{(3)})$ as it passes $S_q^{(23)}$.

Alternatively, we can fuse $S_q^{(12)}$ and $S_q^{(23)}$ along $S_{q+1}^{(2)}$ to obtain a topological defect

$$S_q^{(13)} := S_q^{(12)} \otimes_{S_{q+1}^{(2)}} S_q^{(23)} : \quad S_{q+1}^{(1)} \to S_{q+1}^{(3)}. \tag{121}$$

Projecting $Q_p$ onto the fused configuration, we obtain $S_{p+q-d}^{(Q_p)}(S_q^{(13)})$ converting $S_{p+q+1-d}^{(Q_p)}(S_{q+1}^{(1)})$ into $S_{p+q+1-d}^{(Q_p)}(S_{q+1}^{(3)})$ as it passes $S_q^{(13)}$.

Clearly, the two options

1. first fusing and then projecting, or

2. first projecting and then fusing

should both lead to identical results. This imposes the consistency condition that the fusion of $S_{p+q-d}^{(Q_p)}(S_q^{(12)})$ and $S_{p+q-d}^{(Q_p)}(S_q^{(23)})$ along $S_{p+q+1-d}^{(Q_p)}(S_{q+1}^{(2)})$ should be equal to $S_{p+q-d}^{(Q_p)}(S_q^{(13)})$, or in equations

$$S_{p+q-d}^{(Q_p)}(S_q^{(12)}) \otimes_{S_{p+q+1-d}^{(Q_p)}(S_{q+1}^{(2)})} S_{p+q-d}^{(Q_p)}(S_q^{(23)}) \cong S_{p+q-d}^{(Q_p)}(S_q^{(13)}). \tag{122}$$

See figure 13.

**Collecting everything together.** A collection of defects (118) in $\mathcal{S}$ satisfying consistency conditions (122) is the mathematical definition of a defect $Q_p$ in the Drinfeld center $\mathcal{Z}(\mathcal{S})$ of $\mathcal{S}$. Above we have explained how this mathematical definition arises naturally by considering the physical process of projecting a defect of the SymTFT $\mathfrak{Z}(\mathcal{S})$ onto the physical boundary $\mathfrak{B}_{\mathcal{S}}^{\text{sym}}$ in a variety of ways.

### 3.2.2 Multiplet structure

The multiplet structure associated to a generalized charge $Q_{q+1}$ can be computed as follows. Consider an L-shaped version of a simple topological defect $Q_{q+1}$ of $\mathfrak{Z}(\mathcal{S})$ as shown in figure 14

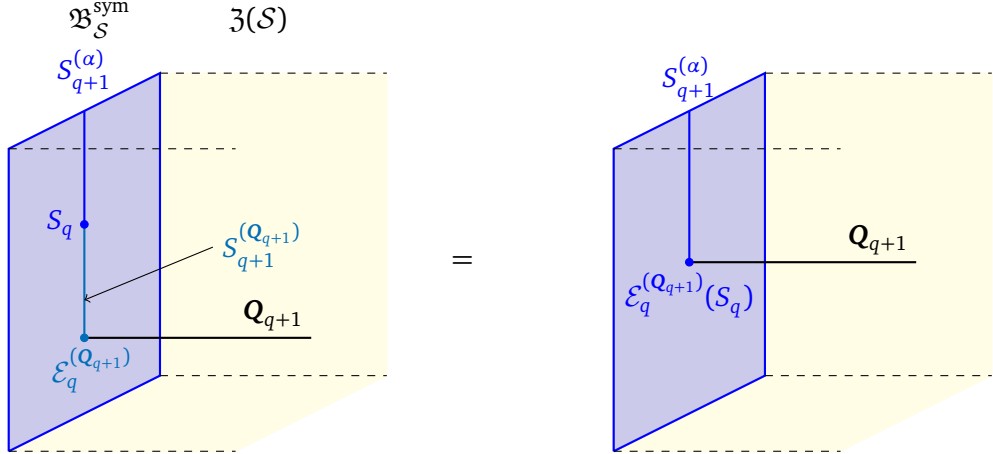

Figure 15: The end $\mathcal{E}_q^{(Q_{q+1})}(S_q)$ of $Q_{q+1}$ is obtained as the fusion of the defect $S_q$ with the canonical end $\mathcal{E}_q^{(Q_{q+1})}$.

and only project half of the L-shape onto $\mathfrak{B}_{\mathcal{S}}^{\mathrm{sym}}$. This provides a canonical simple end

$$\mathcal{E}_q^{(Q_{q+1})}, \tag{123}$$

of $Q_{q+1}$ along $\mathfrak{B}_{\mathcal{S}}^{\mathrm{sym}}$ which is attached to the $(q+1)$-dimensional topological defect $S_{q+1}^{(Q_{q+1})}$ of $\mathfrak{B}_{\mathcal{S}}^{\mathrm{sym}}$, namely the projection of $Q_{q+1}$ onto $\mathfrak{B}_{\mathcal{S}}^{\mathrm{sym}}$. See figure 14.

Now consider a simple $(q+1)$-dimensional topological defect $S_{q+1}^{(\alpha)}$ of $\mathfrak{B}_{\mathcal{S}}^{\mathrm{sym}}$. Then, possible simple ends of $Q_{q+1}$ along $\mathfrak{B}_{\mathcal{S}}^{\mathrm{sym}}$ that are attached to $S_{q+1}^{(\alpha)}$ are in one-to one correspondence with simple $q$-dimensional topological defects transitioning $S_{q+1}^{(Q_{q+1})}$ into $S_{q+1}^{(\alpha)}$.

In more detail, a simple $q$-dimensional topological defect

$$S_q : \quad S_{q+1}^{(Q_{q+1})} \to S_{q+1}^{(\alpha)}, \tag{124}$$

provides a simple topological end

$$\mathcal{E}_q^{(Q_{q+1})}(S_q), \tag{125}$$

of $Q_{q+1}$ along $\mathfrak{B}_{\mathcal{S}}^{\mathrm{sym}}$ that is attached to $S_{q+1}^{(\alpha)}$. This is obtained by fusing $\mathcal{E}_q^{(Q_{q+1})}$ with $S_q$ along $S_{q+1}^{(Q_{q+1})}$, i.e.

$$\mathcal{E}_q^{(Q_{q+1})}(S_q) = \mathcal{E}_q^{(Q_{q+1})} \otimes_{S_{q+1}^{(Q_{q+1})}} S_q, \tag{126}$$

as shown in figure 15.

### 3.2.3 Action of the symmetry

The canonical end $\mathcal{E}_q^{(Q_{q+1})}$ has the property that it is left invariant by the action of any topological defect $S_r \in \mathcal{S}$. This invariance is encoded in the equality of two defect configurations displayed in figure 16.

The action of $S_r$ on an arbitrary end $\mathcal{E}_q^{(Q_{q+1})}(S_q)$ of $Q_{q+1}$ is then computed by first resolving it into the canonical end $\mathcal{E}_q^{(Q_{q+1})}$ and $S_q$ as in (126), and then commuting $S_r$ across the canonical end $\mathcal{E}_q^{(Q_{q+1})}$. See figure 17. Now all the topological defects of $\mathfrak{B}_{\mathcal{S}}^{\mathrm{sym}}$ are on one side of the end

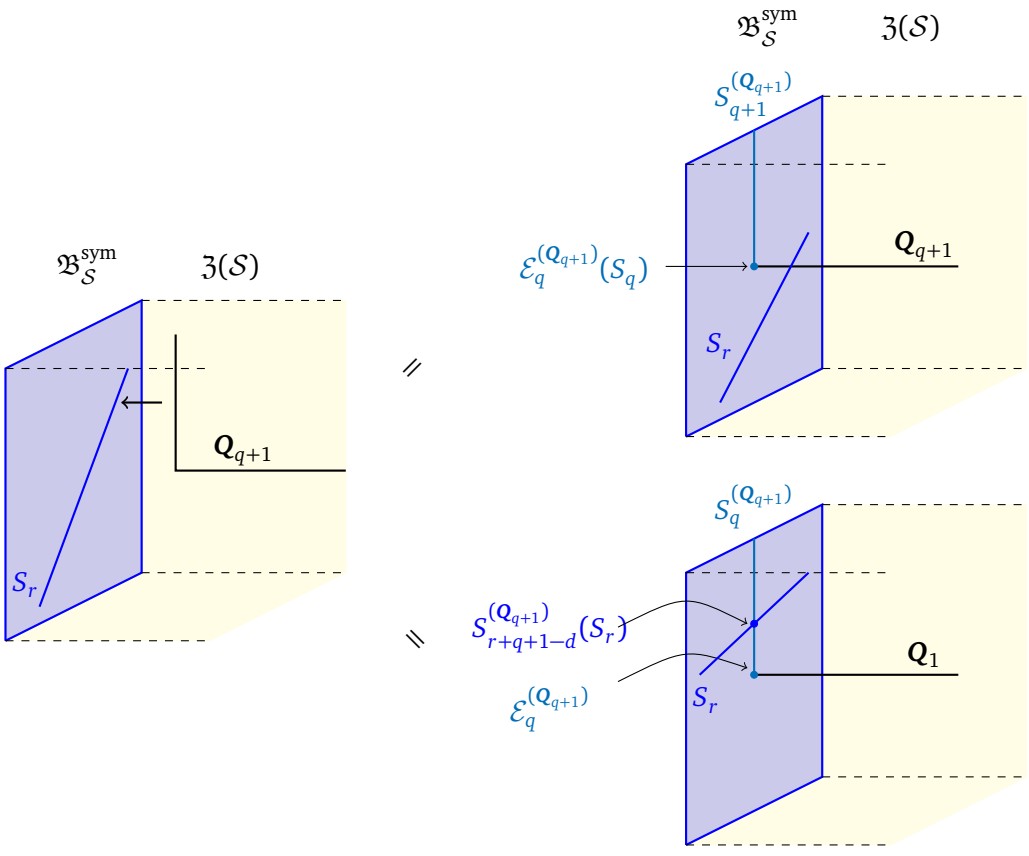

Figure 16: There are two ways of performing an L-dip in the presence of a boundary topological operator $S_r$. Equating the two ways implies that the half-braiding of $\boldsymbol{Q}_{q+1}$ with $S_r$ acts trivially on the canonical end $\mathcal{E}_q^{(\boldsymbol{Q}_{q+1})}$ of $\boldsymbol{Q}_{q+1}$.

$\mathcal{E}_q^{(\boldsymbol{Q}_{q+1})}$ and can be rearranged using the information of $\mathcal{S}$ as desired. After the rearrangement, we can perform a fusion with $\mathcal{E}_q^{(\boldsymbol{Q}_{q+1})}$ to obtain the resulting collection of ends.

After the interval compactification, this action of topological defects of $\mathfrak{B}_{\mathcal{S}}^{\text{sym}}$ on ends of $\boldsymbol{Q}_{q+1}$ descends to an action of topological defects generating symmetry $\mathcal{S}$ of theory $\mathfrak{T}_\sigma$ on $q$-dimensional operators living in a multiplet $\mathcal{M}_q$ transforming in $q$-charge $\boldsymbol{Q}_{q+1}$.

The above described action should be thought of as the simplest action of the symmetry $\mathcal{S}$ which generates all the possible actions that one might care about. For example, using the above action we can derive lasso and higher-lasso actions of symmetry $\mathcal{S}$, where the symmetry topological operators link the non-topological $q$-dimensional operators in various ways. We describe this concretely for $d = 2$ in section 4.3.

## 3.3 Examples of generalized charges

### 3.3.1 (Higher-)reps as charges: Group and higher-group symmetries

In Part I [1] of this series of papers and [3], generalized charges for group and higher-group symmetries were studied from a direct analysis, rather than relying on the technology provided by SymTFT. In this subsection, we recover this statement from the SymTFT point of view.

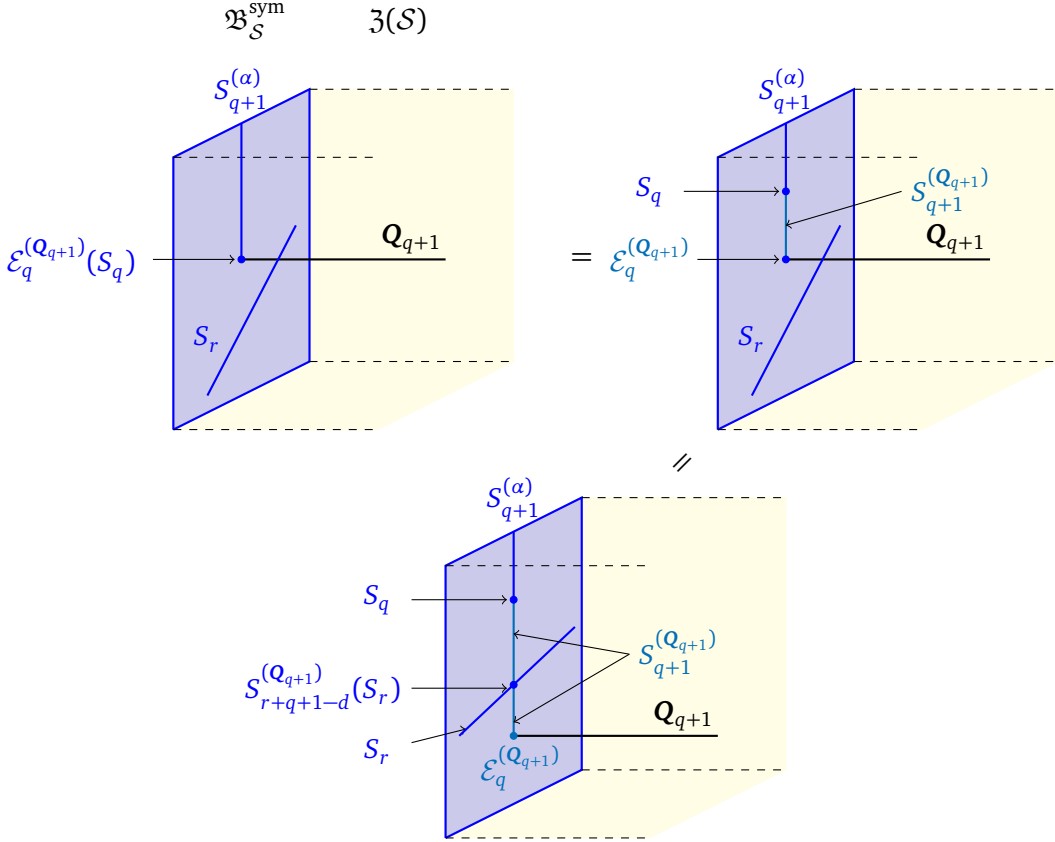

Figure 17: Action of a topological operator $S_r$ of $\mathfrak{B}^{\text{sym}}_{\mathcal{S}}$ on the end $\mathcal{E}_q^{(\boldsymbol{Q}_{q+1})}(S_q)$ of $\boldsymbol{Q}_{q+1}$. First we resolve the configuration using figure 15. We then use the property that $S_r$ acts on the canonical end $\mathcal{S}_1^{(\boldsymbol{Q}_{q+1})}$ by leaving it invariant, but producing an extra half-braiding.

The main statement of the paper [1] was

---

**Statement 3.4: Genuine generalized charges are higher-representations**

$q$-charges of genuine $q$-dimensional operators under a $p$-group symmetry $\mathbb{G}^{(p)}$ with a 't Hooft anomaly $[\omega]$ are $(q+1)$-representations of $\mathbb{G}^{(p)}$, for $0 \leq q \leq d-2$.

---

See the appendix B of [1] for a quick review on higher-representations of groups and higher-groups. We recover this statement in two ways: by performing Drinfeld center computation described in (3.2.1), and by using the theta defects discussed in [9, 21].

**Derivation using the Drinfeld center.** Let us say we are studying a $\mathbb{G}^{(p)}$ symmetry with arbitrary 't Hooft anomaly $\sigma$ with SymTFT $\mathfrak{Z}(\mathcal{S}^{\sigma}_{\mathbb{G}^{(p)}})$. Consider constructing a topological defect $\boldsymbol{Q}_{q+1}$ of $\mathfrak{Z}(\mathcal{S}^{\sigma}_{\mathbb{G}^{(p)}})$ whose projection onto the symmetry boundary $\mathfrak{B}^{\text{sym}}_{\mathcal{S}^{\sigma}_{\mathbb{G}^{(p)}}}$ is

$$S_{q+1}^{(\boldsymbol{Q}_{q+1})} = S_{q+1}^{(\mathfrak{T}_{q+1})}, \tag{127}$$

where $S_{q+1}^{(\mathfrak{T}_{q+1})}$ is a topological defect of $\mathfrak{B}^{\text{sym}}_{\mathcal{S}^{\sigma}_{\mathbb{G}^{(p)}}}$ obtained by inserting a decoupled copy of a $(q+1)$-dimensional TQFT $\mathfrak{T}_{q+1}$ inside the worldvolume of $\mathfrak{B}^{\text{sym}}_{\mathcal{S}^{\sigma}_{\mathbb{G}^{(p)}}}$.

Consider performing the projection of $Q_{q+1}$ on top of a topological operator

$$S_{d-r-1}^{(g)}, \qquad g \in G^{(r)}, \tag{128}$$

generating an $r$-form symmetry inside the $p$-group $\mathbb{G}^{(p)}$. As discussed in previous subsection, this provides an operator

$$S_{q-r}^{(Q_{q+1})}\left(S_{d-r-1}^{(g)}\right), \tag{129}$$

located at the intersection of $S_{d-r-1}^{(g)}$ and $S_{q+1}^{(\mathfrak{T}_{q+1})}$. But any such operator can be obtained by beginning with a topological operator $D_{q-r}$ inside $\mathfrak{T}_{q+1}$ and inserting both of them inside the worldvolume of $\mathfrak{B}_{\mathcal{S}_{\mathbb{G}^{(p)}}^{\sigma}}^{\text{sym}}$.

In other words, we obtain a map

$$G^{(r)} \to \left\{(q-r)\text{-dimensional topological operators of } \mathfrak{T}_{q+1}\right\}, \tag{130}$$

for all $r$-form symmetry groups. The consistency conditions (122) require the above map to describe a $(q+1)$-representation $\rho$ of the $p$-group $\mathbb{G}^{(p)}$.

Thus, using the Drinfeld center construction on $\mathcal{S}_{\mathbb{G}^{(p)}}^{\sigma}$, we have managed to construct a class of $(q+1)$-dimensional topological defects of the SymTFT $\mathfrak{Z}(\mathcal{S}_{\mathbb{G}^{(p)}}^{\sigma})$ that correspond to $(q+1)$-representations of the $p$-group $\mathbb{G}^{(p)}$. These are the generalized charges appearing in 3.4.

**Derivation using theta defects.** We can also easily recover statement 3.4 as a special case of the following more general construction of [9, 21]:

---

**Statement 3.5: Theta defects**

Any $(d+1)$-dimensional theory $\mathfrak{T}/\mathbb{G}^{(p)}$ that can be constructed by gauging a $\mathbb{G}^{(p)}$ symmetry of a $(d+1)$-dimensional theory $\mathfrak{T}$ admits a class of $(q+1)$-dimensional topological defects, known as **theta defects**, that correspond to $(q+1)$-representations of the $p$-group $\mathbb{G}^{(p)}$.

---

The idea behind theta defects is that inserting any $\mathbb{G}^{(p)}$-symmetric $(q+1)$-dimensional TQFT inside the spacetime occupied by $\mathfrak{T}$, and then performing the $\mathbb{G}^{(p)}$ gauging, produces a $(q+1)$-dimensional topological defect of the gauged theory $\mathfrak{T}/\mathbb{G}^{(p)}$. See figure 18. As explained in detail in section 3 of [21], $\mathbb{G}^{(p)}$-symmetric $(q+1)$-dimensional TQFTs, and hence $(q+1)$-dimensional theta defects, are parametrized by $(q+1)$-representations of $\mathbb{G}^{(p)}$.

Now, one can apply the statement 3.5 to the SymTFT $\mathfrak{Z}(\mathcal{S}_{\mathbb{G}^{(p)}}^{\sigma})$ of a $\mathbb{G}^{(p)}$ symmetry with arbitrary 't Hooft anomaly $\sigma$. This admits precisely a gauging construction (87)

$$\mathfrak{Z}(\mathcal{S}_{\mathbb{G}^{(p)}}^{\sigma}) = \mathfrak{Z}_{\sigma}^{\text{(trivial)}}/\mathbb{G}^{(p)}. \tag{131}$$

The theta-defects in this gauging provide a collection of $(q+1)$-dimensional topological defects of $\mathfrak{Z}(\mathcal{S}_{\mathbb{G}^{(p)}}^{\sigma})$ labeled by $(q+1)$-representations of $\mathbb{G}^{(p)}$. By the general statement 3.2, these topological defects of $\mathfrak{Z}(\mathcal{S}_{\mathbb{G}^{(p)}}^{\sigma})$ describe $q$-charges for symmetry $\mathcal{S}_{\mathbb{G}^{(p)}}^{\sigma}$.

**Genuine operators in the multiplet.** Let us describe the topological ends of $Q_{q+1}$ along $\mathfrak{B}_{\mathcal{S}_{\mathbb{G}^{(p)}}^{\sigma}}^{\text{sym}}$ that are not attached to any non-trivial topological defects of $\mathfrak{B}_{\mathcal{S}_{\mathbb{G}^{(p)}}^{\sigma}}^{\text{sym}}$. Such ends give rise to genuine operators in a multiplet transforming in the $q$-charge $Q_{q+1}$. Below, we will call such ends as genuine ends for brevity.

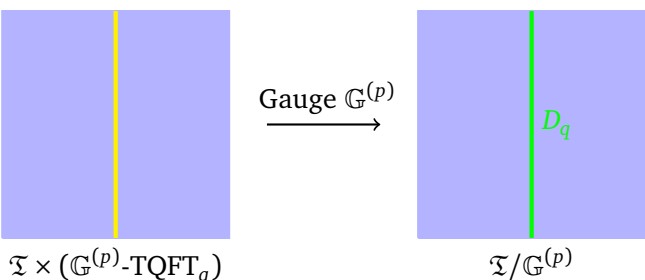

Figure 18: Theta defects: Consider a theory $\mathfrak{T}$ with $p$-group symmetry $\mathbb{G}^{(p)}$. We take the product with a $q$-dimensional $\mathbb{G}^{(p)}$-TQFT (shown in yellow), and gauge the diagonal $\mathbb{G}^{(p)}$. The TQFT then becomes a topological defect in the gauged theory $\mathfrak{T}/\mathbb{G}^{(p)}$: the theta-defect (shown in green).

The canonical end $\mathcal{E}_q^{(Q_{q+1})}$ of $Q_{q+1}$ is attached to $S_{q+1}^{(\mathfrak{T}_{q+1})}$. Thus possible genuine ends of $Q_{q+1}$ are in one-to-one correspondence with ends of $S_{q+1}^{(\mathfrak{T}_{q+1})}$ inside $\mathfrak{B}_{S_{\mathbb{G}^{(p)}}^\sigma}^{\mathrm{sym}}$, which in turn are in one-to-one correspondence with the topological boundary conditions of the TQFT $\mathfrak{T}_{q+1}$.

Note that $\mathfrak{T}_{q+1}$ has at least one topological boundary condition, and hence we have at least one genuine end for each topological operator $Q_{q+1}$ corresponding to a higher-representation $\rho$ of $G^{(p)}$. This justifies the remaining part of the statement 3.4 which claims that a multiplet of operators transforming in $q$-charge $Q_{q+1}$ contains genuine $q$-dimensional operators of a $S_{\mathbb{G}^{(p)}}^\sigma$-symmetric $d$-dimensional theory.

### 3.3.2 (Higher-)reps as charges: (Higher-)representation symmetries

As discussed in previous subsection a higher-representation type symmetry

$$S = (d-1)\text{-Rep}(\mathbb{G}^{(p)}),\tag{132}$$

can be obtained by gauging a non-anomalous $\mathbb{G}^{(p)}$ $p$-group symmetry. Thus the two symmetries have the same collection of generalized charges.

Moreover, above we constructed a class of generalized charges for a $\mathbb{G}^{(p)}$ $p$-group symmetry described by higher-representations of $\mathbb{G}^{(p)}$. Consequently, there is a class of generalized charges for $(d-1)\text{-Rep}(\mathbb{G}^{(p)})$ symmetry described by higher-representations of $\mathbb{G}^{(p)}$.

Consider such a $q$-charge $Q_{q+1}$ described by a $(q+1)$-representation $\rho$ of $\mathbb{G}^{(p)}$. Its projection onto the new symmetry boundary $\mathrm{B}_{(d-1)\text{-Rep}(\mathbb{G}^{(p)})}^{\mathrm{sym}}$ is

$$S_{q+1}^{(Q_{q+1})} = S_{q+1}^{(\rho)},\tag{133}$$

where $S_{q+1}^{(\rho)} \in (d-1)\text{-Rep}(\mathbb{G}^{(p)})$ is the topological operator on $\mathrm{B}_{(d-1)\text{-Rep}(\mathbb{G}^{(p)})}^{\mathrm{sym}}$ corresponding to $(q+1)$-representation $\rho$.

Thus the canonical end $\mathcal{E}_q^{(Q_{q+1})}$ is attached to $S_{q+1}^{(\rho)}$. This implies that the possible ends of $Q_{q+1}$ along $\mathrm{B}_{(d-1)\text{-Rep}(\mathbb{G}^{(p)})}^{\mathrm{sym}}$ that are attached to $S_{q+1}^{(\rho')}$ for some $(q+1)$-representation $\rho'$ are in one-to-one correspondence with intertwiners between $(q+1)$-representations $S_{q+1}^{(\rho)}$ and $S_{q+1}^{(\rho')}$, as these intertwiners describe the $q$-dimensional topological defects between $S_{q+1}^{(\rho)}$ and $S_{q+1}^{(\rho')}$.

In particular, the genuine ends of $\mathcal{E}_q^{(Q_{q+1})}$ are in one-to-one correspondence with the intertwiners between $(q+1)$-representation $\rho$ and the identity $(q+1)$-representation.

### 3.3.3 Twisted (higher-)reps as charges: Group symmetries

We can easily compute all the remaining generalized charges for a $G^{(0)}$ 0-form symmetry with a 't Hooft anomaly

$$[\omega] \in H^{d+1}(G^{(0)}, \mathbb{C}^{\times}). \tag{134}$$

In other words, we can finish the computation of the Drinfeld center

$$\mathcal{Z}\left(\mathcal{S}_{G^{(0)}}^{[\omega]}\right), \tag{135}$$

of the fusion $(d-1)$-category $\mathcal{S}_{G^{(0)}}^{[\omega]}$. In section 3.3.1, we encountered generalized charges corresponding to higher-representations of $G^{(0)}$. The remaining generalized charges correspond to twisted higher-representations of subgroups of $G^{(0)}$.

Consider a simple topological defect $\boldsymbol{Q}_{d-1}$ of $\mathfrak{Z}(\mathcal{S}_{G^{(0)}}^{[\omega]})$ whose projection $S_{d-1}^{(\boldsymbol{Q}_{d-1})}$ onto the symmetry boundary $\mathfrak{B}_{\mathcal{S}_{G^{(0)}}^{[\omega]}}^{\text{sym}}$ involves a $(d-1)$-dimensional topological defect in $\mathcal{S}_{G^{(0)}}^{[\omega]}$ lying in a non-trivial grade $g \in G^{(0)}$. In order for the projection of $\boldsymbol{Q}_{d-1}$ to be consistent in the background of a topological defect $S_{d-1}^{(g')}$ of $\mathfrak{B}_{\mathcal{S}_{G^{(0)}}^{[\omega]}}^{\text{sym}}$, the projection $S_{d-1}^{(\boldsymbol{Q}_{d-1})}$ must involve $(d-1)$-dimensional topological defects in $\mathcal{S}_{G^{(0)}}^{[\omega]}$ lying in all grades

$$g \in [g], \tag{136}$$

where $[g]$ is a non-trivial conjugacy class of $G^{(0)}$. We can thus express

$$S_{d-1}^{(\boldsymbol{Q}_{d-1})} = \bigoplus_{g \in [g]} S_{d-1}^{(\boldsymbol{Q}_{d-1}, g)}, \tag{137}$$

where $S_{d-1}^{(\boldsymbol{Q}_{d-1}, g)}$ is the piece of $S_{d-1}^{(\boldsymbol{Q}_{d-1})}$ lying in the grade $g$. The defect $S_{d-1}^{(\boldsymbol{Q}_{d-1}, g)}$ must be of the form

$$S_{d-1}^{(\boldsymbol{Q}_{d-1}, g)} = S_{d-1}^{(g)} \otimes S_{d-1}^{(\mathfrak{T}_{d-1})}, \tag{138}$$

where $S_{d-1}^{(\mathfrak{T}_{d-1})}$ is a topological defect of $\mathfrak{B}_{\mathcal{S}_{G^{(0)}}^{[\omega]}}^{\text{sym}}$ obtained by inserting a decoupled copy of a $(d-1)$-dimensional TQFT $\mathfrak{T}_{d-1}$ inside the worldvolume of $\mathfrak{B}_{\mathcal{S}_{G^{(0)}}^{[\omega]}}^{\text{sym}}$.

Now choose an element

$$h \in H_g, \tag{139}$$

where $H_g \subseteq G^{(0)}$ is the stabilizer subgroup of the element $g$, and perform a projection of $\boldsymbol{Q}_{q+1}$ on top of

$$S_{d-1}^{(h)}. \tag{140}$$

The resulting half-braiding $S_{d-2}^{\boldsymbol{Q}_{d-1}}(S_{d-1}^{(h)})$ descends to a half-braiding $S_{d-2}^{\boldsymbol{Q}_{d-1}, g}(S_{d-1}^{(h)})$ located at the intersection of $S_{d-1}^{(\boldsymbol{Q}_{d-1}, g)}$ and $S_{d-1}^{(h)}$. Following similar arguments as in section 3.3.1, we see that the half-braiding $S_{d-2}^{\boldsymbol{Q}_{d-1}, g}(S_{d-1}^{(h)})$ describes a map

$$H_g \rightarrow \{(d-2)\text{-dimensional topological operators of } \mathfrak{T}_{d-1}\}. \tag{141}$$

The consistency conditions (122) require the above map to describe a $[\omega_g]$-twisted $(d-1)$-representation $\boldsymbol{\rho}$ of $H_g$ where

$$[\omega_g] \in H^d(H_g, \mathbb{C}^{\times}), \tag{142}$$

is obtained as

$$\omega_g(h_1, h_2, \cdots, h_d) = \prod_{i=0}^{d} \omega^{s(i)}(h_1, \cdots, h_i, g, h_{i+1}, \cdots, h_d), \tag{143}$$

where $h_i \in H_g$, $s(i) = 1$ for even $i$ and $s(i) = -1$ for odd $i$.

Thus the possible simple $(q-1)$-dimensional topological defects of $\mathcal{Z}(\mathcal{S}_{G^{(0)}}^{[\omega]})$ are specified by two pieces of data:

1. A conjugacy class $[g] \subset G^{(0)}$.

2. An $\omega_g$-twisted irreducible $(d-1)$-representation $\boldsymbol{\rho}$ of the centralizer $H_g$ of an element $g \in [g]$.

We can thus express the full Drinfeld center as

$$\mathcal{Z}\left(\mathcal{S}_{G^{(0)}}^{[\omega]}\right) = \bigoplus_{[g]} (d-1)\text{-Rep}^{\omega_g}(H_g), \tag{144}$$

where the sum is over conjugacy classes of $G^{(0)}$, and $(d-1)\text{-Rep}^{\omega_g}(H_g)$ is the $(d-1)$-category of $\omega_g$-twisted $(d-1)$-representations of the centralizer $H_g$ of a representative $g$ in conjugacy class $[g]$. This is consistent in the case of 2-categories, where the center was discussed in [34, 36].

When there is no anomaly $\omega = 1$, then all $\omega_g = 1$, implying that we have standard $(d-1)$-representations of centralizers in the corresponding Drinfeld center:

$$\mathcal{Z}(\mathcal{S}_{G^{(0)}}) = \bigoplus_{[g]} (d-1)\text{-Rep}(H_g). \tag{145}$$

### 3.4 Comparison: Generalized charges and relative defects

Just like SymTFTs are related to relative theories (see section 2.6), generalized charges, which are topological defects of SymTFTs, are related to relative defects of relative theories. The notions of relative and absolute defects were introduced in [37] and will be reviewed below. We begin with the definition of a relative defect of a relative theory

---

**Definition 3.2: Relative defect of a relative theory**

A relative $\boldsymbol{q}$-dimensional defect $\mathcal{O}_q^{\text{rel}}$ of a relative $d$-dimensional theory $\mathfrak{T}^{\text{rel}}$ is a defect of $\mathfrak{T}^{\text{rel}}$ that is attached to a non-trivial $(q+1)$-dimensional topological defect $\boldsymbol{Q}_{q+1}$ of the $(d+1)$-dimensional TQFT $\mathfrak{Z}$ of which $\mathfrak{T}^{\text{rel}}$ is a boundary condition. See figure 19.

---

As discussed in section 2.6, a physical boundary $\mathfrak{B}_{\mathfrak{T}_\sigma}^{\text{phys}}$ of a SymTFT $\mathfrak{Z}(\mathcal{S})$ is an example of a relative $d$-dimensional theory. Then clearly a $q$-dimensional operator $\mathcal{M}_q$ living on the physical boundary $\mathfrak{B}_{\mathfrak{T}_\sigma}^{\text{phys}}$ attached to a non-trivial $(q+1)$-dimensional topological defect $\boldsymbol{Q}_{q+1}$ of the SymTFT $\mathfrak{Z}(\mathcal{S})$ is a relative defect of the relative theory $\mathfrak{B}_{\mathfrak{T}_\sigma}^{\text{phys}}$. In other words, a multiplet $\mathcal{M}_q$ of $q$-dimensional operators transforming non-trivially under the $\mathcal{S}$ symmetry of an absolute theory $\mathfrak{T}_\sigma$ corresponds to a relative defect of the relative theory $\mathfrak{B}_{\mathfrak{T}_\sigma}^{\text{phys}}$.

A similar notion is that of a relative defect of an absolute theory:

---

**Definition 3.3: Relative defect of an absolute theory**

A relative $\boldsymbol{q}$-dimensional defect $\mathcal{O}_q^{\text{rel}}$ of an absolute $d$-dimensional theory $\mathfrak{T}^{\text{abs}}$ is a defect

---

of $\mathfrak{T}^{\text{abs}}$ that is attached to a non-trivial $(q + 1)$-dimensional topological defect $D_{q+1}$ of $\mathfrak{T}^{\text{abs}}$. See figure 20.

On the other hand, an absolute defect is defined as:

---

**Definition 3.4: Absolute defect**

An **absolute defect** is a genuine defect not attached to any higher-dimensional defects.

---

We can have absolute defects in an absolute theory $\mathfrak{T}^{\text{abs}}$ or in a relative theory $\mathfrak{T}^{\text{rel}}$, but we will focus on absolute defects of absolute theories in what follows.

Now suppose that we are given a topological boundary condition $\mathfrak{B}^{\text{top}}$ of a TQFT $\mathfrak{Z}$ that converts a relative theory $\mathfrak{T}^{\text{rel}}$ into an absolute theory $\mathfrak{T}^{\text{abs}}$. It is then possible to convert a relative defect $\mathcal{O}_q^{\text{rel}}$ of the relative theory $\mathfrak{T}^{\text{rel}}$ into a relative or absolute defect $\mathcal{O}_q$ of the absolute theory $\mathfrak{T}^{\text{abs}}$. This can be done by performing an interval compactification shown in figure 21 after choosing a topological $q$-dimensional end $\mathcal{E}_q$ of $\boldsymbol{Q}_{q+1}$ along $\mathfrak{B}^{\text{top}}$. The end may

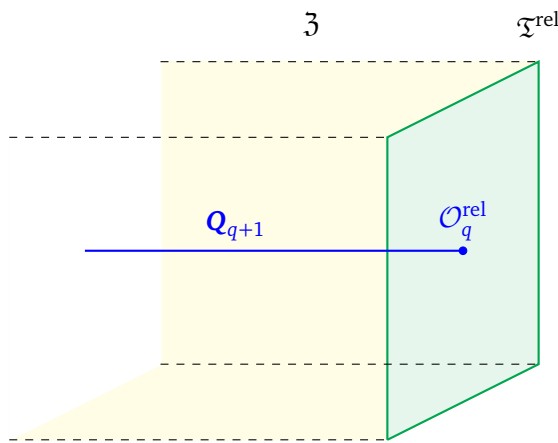

Figure 19: A $q$-dimensional relative defect $\mathcal{O}_q^{\text{rel}}$ of a $d$-dimensional relative theory $\mathfrak{T}^{\text{rel}}$ is attached to a non-trivial $(q + 1)$-dimensional topological defect $\boldsymbol{Q}_{q+1}$ of the $(d + 1)$-dimensional TQFT $\mathfrak{Z}$ associated to $\mathfrak{T}^{\text{rel}}$.

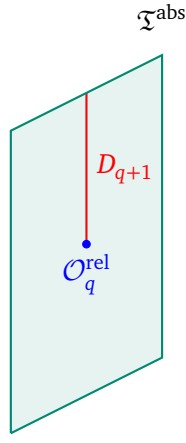

Figure 20: A $q$-dimensional relative defect $\mathcal{O}_q^{\text{rel}}$ of a $d$-dimensional absolute theory $\mathfrak{T}^{\text{abs}}$ is attached to a non-trivial $(q + 1)$-dimensional topological defect $D_{q+1}$ of $\mathfrak{T}^{\text{abs}}$.

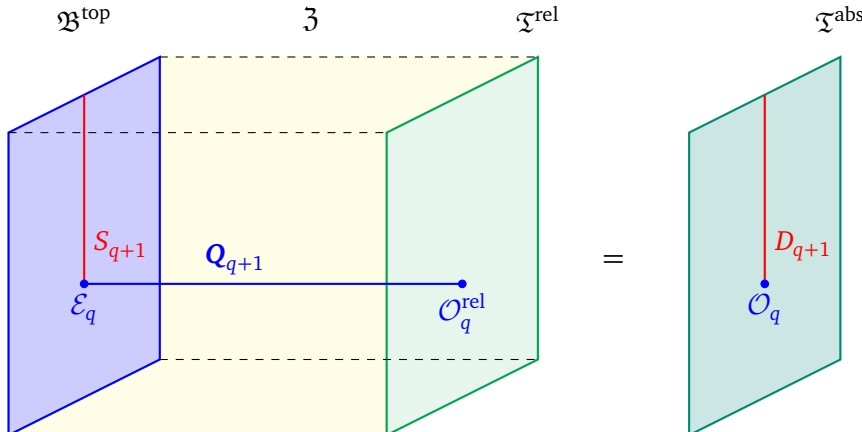

Figure 21: Conversion of a relative defect $\mathcal{O}_q^{\mathrm{rel}}$ of a relative theory $\mathfrak{T}^{\mathrm{rel}}$ into an absolute or a relative defect $\mathcal{O}_q$ of the absolute theory $\mathfrak{T}^{\mathrm{abs}}$.

be attached to a $(q+1)$-dimensional topological defect $S_{q+1}$ of $\mathfrak{B}^{\mathrm{top}}$. After the compactification, $S_{q+1}$ becomes a $(q+1)$-dimensional topological defect $D_{q+1}$ of $\mathfrak{T}^{\mathrm{abs}}$. If $D_{q+1}$ is trivial, then $\mathcal{O}_q$ is an absolute defect of the absolute theory $\mathfrak{T}^{\mathrm{abs}}$. On the other hand, if $D_{q+1}$ is non-trivial, then $\mathcal{O}_q$ is a relative defect of the absolute theory $\mathfrak{T}^{\mathrm{abs}}$.

Note that the above conversion procedure is the same as the procedure for converting a $q$-dimensional operator $\mathcal{M}_q$ along the physical boundary $\mathfrak{B}_{\mathfrak{T}_\sigma}^{\mathrm{phys}}$ associated to a multiplet of $q$-dimensional operators transforming in the $q$-charge $Q_{q+1}$ into operators $\mathcal{O}_q$ of the theory $\mathfrak{T}_\sigma$ comprising the multiplet associated to $\mathcal{M}_q$.

The following examples of relative defects of relative theories were studied in [37]:

---

**Example 3.1: Codimension-2 defects in 6d $\mathcal{N} = (2, 0)$ theories**

6d $\mathcal{N} = (2, 0)$ SCFTs are well-known examples of relative theories. In [37] an argument was put forward for the existence of relative codimension-2 defects in these theories. Such relative codimension-2 defects necessarily correspond to irregular punctures in Class S constructions, but not all irregular punctures come from relative codimension-2 defects.

A relative codimension-2 defect $\mathcal{M}_4^{\mathrm{rel}}$ of a 6d $\mathcal{N} = (2, 0)$ SCFT $\mathfrak{T}^{\mathrm{rel}}$ studied in [37] has the property that the codimension-2 topological defect $Q_5$ of the associated 7d TQFT $\mathfrak{Z}$ that $\mathcal{M}_4^{\mathrm{rel}}$ is attached to carries localized 2-form symmetries along its worldvolume. That is, there are 2-dimensional invertible topological operators $Q_2^{\mathrm{loc}}$ that live along the worldvolume of $Q_5$. This has the following consequence: as we convert the relative defect $\mathcal{M}_4^{\mathrm{rel}}$ into an absolute or relative codimension-2 defect $\mathcal{O}_4$ of an absolute 6d $\mathcal{N} = (2, 0)$ SCFT $\mathfrak{T}^{\mathrm{abs}}$, we find localized 1-form symmetries along the worldvolume of $\mathcal{O}_4$. Such localized 1-form symmetries (more precisely the defect group thereof) are essentially the source of the **trapped 1-form symmetries** carried by irregular punctures in Class S constructions discussed in [37].

---

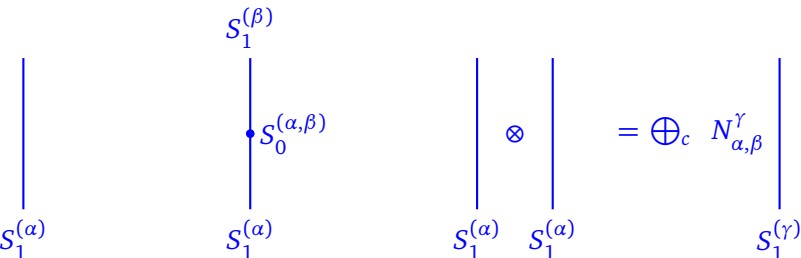

Figure 22: Topological lines $S_1$ and local operators $S_0$ of a symmetry $\mathcal{S}_{\mathfrak{T}}$ of a 2d theory $\mathfrak{T}$. The fusion of lines results in a sum of lines, with multiplicities given by the fusion coefficients $N^{\gamma}_{\alpha,\beta}$.

# 4  Generalized charges and SymTFT in $d = 2$

The general framework of the last two sections will now be put to work in concrete physical theories, starting in 2d, and working ourselves up in dimensions.

In 2d, a general (possibly non-invertible) symmetry $\mathcal{S}$ is described by a fusion 1-category. This is a very well-known statement in this spacetime dimension and predates the systematic studies of generalized global symmetries initiated by [2]. Some initial works advocating this viewpoint in 2d are [38–40]. See also [15] for a review of this approach using terminology similar to that used here, and [41–47] for recent works on categorical symmetries in 2d.

Much is well-known about the associated SymTFT $\mathfrak{Z}(\mathcal{S})$. The SymTFT is obtained by applying the Turaev-Viro-Barrett-Westbury construction [48,49] (see also [50,51]) and is often referred to as the **Turaev-Viro theory**[10] based on $\mathcal{S}$. The Drinfeld center for fusion categories have a long-standing history in mathematics, see [53,54] for a comprehensive discussion.

However the perspective of **0-charges of the symmetry $\mathcal{S}$ as topological line defects in the SymTFT $\mathfrak{Z}(\mathcal{S})$** is new and was recently initiated in [46]. We will shed new light onto applications of fusion category symmetries from this perspective in this very well-developed field in [17,18].

## 4.1  Symmetries and symmetric 2d theories

Consider a 2d theory $\mathfrak{T}$. Its symmetry category is a multi-tensor category $\mathcal{S}_{\mathfrak{T}}$. The objects of $\mathcal{S}_{\mathfrak{T}}$ are topological line defects of $\mathfrak{T}$, and morphisms of $\mathcal{S}_{\mathfrak{T}}$ are topological local operators between two topological line defects. See figure 22.

Abstracting out the above notion, a finite symmetry $\mathcal{S}$ of a 2d theory is described by a multi-fusion category $\mathcal{S}$. In this paper, we restrict to the case of a fusion category symmetry $\mathcal{S}$. We label objects of $\mathcal{S}$ by $S_1$ and morphisms by $S_0$, which are elements in the vector space of morphisms from $S_1$ to $S_1'$

$$S_0 \in \mathrm{Hom}(S_1, S_1'). \tag{146}$$

The fusion of two objects $S_1$ and $S_1'$ is denoted by

$$S_1 \otimes S_1'. \tag{147}$$

The fusion satisfies associativity

$$\left(S_1 \otimes S_1'\right) \otimes S_1'' \cong S_1 \otimes \left(S_1' \otimes S_1''\right), \tag{148}$$

---

[10]In the condensed matter literature, there is a corresponding construction, known as the Levin-Wen string-net construction [52], of a 2+1-dimensional lattice model using the information of a fusion category $\mathcal{S}$ which admits a gapped phase described by the 3d Turaev-Viro TQFT $\mathfrak{Z}(\mathcal{S})$.

and we are provided a canonical isomorphism, known as the associator,

$$S_0^{\text{ass}}\left(S_1, S_1', S_1''\right) \in \text{Hom}\left(\left(S_1 \otimes S_1'\right) \otimes S_1'', \, S_1 \otimes \left(S_1' \otimes S_1''\right)\right), \tag{149}$$

between these.

A fusion category has a finite set of isomorphism classes of simple objects, with the property that there are no morphisms between two simple objects lying in different isomorphism classes. We parametrize these isomorphism classes for $\mathcal{S}$ by $\alpha$ and pick a representative simple object

$$S_1^{(\alpha)}, \tag{150}$$

in each class. The fusion rules then specify non-negative integers $N_\gamma^{\alpha,\beta}$ via

$$S_1^{(\alpha)} \otimes S_1^{(\beta)} \cong \bigoplus_\gamma N_\gamma^{\alpha,\beta} S_1^{(\gamma)}. \tag{151}$$

A 2d theory $\mathfrak{T}$ is equipped with symmetry $\mathcal{S}$ if we choose a tensor functor as in definition 2.3

$$\sigma: \quad \mathcal{S} \rightarrow \mathcal{S}(\mathfrak{T}). \tag{152}$$

This means we choose a topological line defect

$$D_1 = \sigma(S_1), \tag{153}$$

of $\mathfrak{T}$ for every object $S_1$ of $\mathcal{S}$. We may choose the same topological line defect of $\mathfrak{T}$ for different objects of $\mathcal{S}$, i.e. we may have

$$\sigma(S_1) \cong \sigma(S_1') \in \mathcal{S}(\mathfrak{T}), \qquad S_1 \not\cong S_1' \in \mathcal{S}. \tag{154}$$

Similarly, we choose a topological local operator $D_0$ of $\mathfrak{T}$ for each morphism $S_0$ of $\mathcal{S}$

$$\sigma(S_0) = D_0: \quad D_1 \rightarrow D_1', \tag{155}$$

where

$$D_1 = \sigma(S_1), \qquad D_1' = \sigma(S_1'), \tag{156}$$

and

$$S_0: \quad S_1 \rightarrow S_1', \tag{157}$$

is a morphism in $\mathcal{S}$. Again, we may choose the same topological local operator of $\mathfrak{T}$ for different morphisms of $\mathcal{S}$. The map $\sigma$ needs to also respect the fusion rules and associativity of the fusion category $\mathcal{S}$. The pair

$$\mathfrak{T}_\sigma \equiv (\mathfrak{T}, \sigma), \tag{158}$$

describes an $\mathcal{S}$-symmetric 2d theory $\mathfrak{T}_\sigma$ whose underlying 2d theory $\mathfrak{T}$.

---

**Example 4.1: 0-form symmetries with anomaly in 2d**

Let us discuss the fusion category $\mathcal{S}_{G^{(0)}}^{[\omega]}$ associated to a finite $G^{(0)}$ 0-form symmetry in 2d with a 't Hooft anomaly

$$[\omega] \in H^3(G^{(0)}, \mathbb{C}^\times). \tag{159}$$

This fusion category is also denoted as

$$\mathcal{S}_{G^{(0)}}^{[\omega]} = \text{Vec}_{G^{(0)}}^{[\omega]}, \tag{160}$$

and recognized as the category of $G^{(0)}$-graded finite dimensional vector spaces with associator $[\omega]$. The isomorphism classes of simple objects of $\mathcal{S}_{G^{(0)}}^{[\omega]}$ are labeled by group

---

elements $g \in G^{(0)}$ and we pick representative simple objects

$$S_1^{(g)}, \tag{161}$$

in each class. The fusion rules are

$$S_1^{(g)} \otimes S_1^{(h)} \cong S_1^{(gh)}, \tag{162}$$

and let us choose (non-canonical) isomorphisms

$$S_0^{(g,h)}: \quad S_1^{(g)} \otimes S_1^{(h)} \to S_1^{(gh)}. \tag{163}$$

The group cohomology class $[\omega]$ captures the associator $S_0^{\mathrm{ass}}$ in (149). To see this, let us first construct two morphisms

$$S_0^{(g,h,k;L)}, \ S_0^{(g,h,k;R)} \in \mathrm{Hom}\Big( \big( S_1^{(g)} \otimes S_1^{(h)} \big) \otimes S_1^{(k)}, S_1^{(ghk)} \Big), \tag{164}$$

as[11]

$$S_0^{(g,h,k;L)} := S_0^{(gh,k)} \circ S_0^{(g,h)}, \tag{165}$$

and

$$S_0^{(g,h,k;R)} := S_0^{(g,hk)} \circ S_0^{(h,k)} \circ S_0^{\mathrm{ass}}\big( S_1^{(g)}, S_1^{(h)}, S_1^{(k)} \big). \tag{166}$$

Now, since $\mathrm{Hom}\Big( \big( S_1^{(g)} \otimes S_1^{(h)} \big) \otimes S_1^{(k)}, S_1^{(ghk)} \Big)$ is a one-dimensional vector space, we can relate the two morphisms $S_0^{(g,h,k;L)}$ and $S_0^{(g,h,k;R)}$ by a $\mathbb{C}^\times$ number, which is described by a representative $\omega$ of the class $[\omega]$ as follows

$$S_0^{(g,h,k;R)} = \omega(g,h,k) S_0^{(g,h,k;L)}. \tag{167}$$

The precise representative $\omega$ depends on the choices made for

$$S_0^{(g,h)} \in \mathrm{Hom}\Big( S_1^{(g)} \otimes S_1^{(h)}, S_1^{(gh)} \Big). \tag{168}$$

Since $\mathrm{Hom}\Big( S_1^{(g)} \otimes S_1^{(h)}, S_1^{(gh)} \Big)$ is a one-dimensional vector space, we can modify our choice by

$$S_0^{(g,h)} \to \beta(g,h) S_0^{(g,h)}, \tag{169}$$

for $\beta$ an arbitrary $\mathbb{C}^\times$ valued 2-cochain on $G^{(0)}$, which modifies $S_0^{(g,h,k;L)}$ and $S_0^{(g,h,k;R)}$ accordingly, such that the relation (167) is obeyed with the replacement

$$\omega \to \omega \times \delta\beta. \tag{170}$$

---

**Example 4.2: 2d SPT phases protected by 0-form symmetry**

Consider a non-anomalous finite $G^{(0)}$ 0-form symmetry. A 2d $G^{(0)}$-SPT phase is a $G^{(0)}$-symmetric theory $\mathfrak{T}_\sigma$ whose underlying theory $\mathfrak{T}$ is the trivial 2d theory, for which the symmetry category is

$$\mathcal{S}_{\mathfrak{T}} = \mathsf{Vec}, \tag{171}$$

---

[11]Here, and in all equations that follow, we drop tensor products with identity endomorphisms for brevity.

namely the category of finite dimensional vector spaces. An SPT phase is then described by a tensor functor

$$\sigma: \quad \mathcal{S}_{G^{(0)}} \to \mathsf{Vec}, \tag{172}$$

which is also known in the mathematics literature as a *fiber functor* from $\mathcal{S}_{G^{(0)}}$.

Since $[\omega] = 1$, we can pick morphisms $S_0^{(gg')}$ such that the representative $\omega = 1$ in (167). Now, the map $\sigma$ must be such that

$$D_1^{(g)} := \sigma\left(S_1^{(g)}\right) \cong D_1^{(\mathrm{id})}, \qquad \forall\, g \in G^{(0)}, \tag{173}$$

where $D_1^{(\mathrm{id})}$ is the identity line defect of $\mathfrak{T}$, or in other words the vector space $\mathbb{C}$ associated to complex numbers in Vec. That is, $D_1^{(g)}$ is a one-dimensional vector space in Vec. We choose some (non-canonical) isomorphisms

$$D_0^{(g)}: \quad D_1^{(g)} \to D_1^{(\mathrm{id})}. \tag{174}$$

Let

$$D_0^{(g,g')} := \sigma\left(S_0^{(g,g')}\right): \quad D_1^{(g)} \otimes D_1^{(g')} \to D_1^{(gg')}. \tag{175}$$

After applying the isomorphisms $D_0^{(g)}$ and their inverses, we can translate $D_0^{(g,g')}$ to a morphism in

$$\beta(g,g') \in \mathrm{Hom}\left(D_1^{(\mathrm{id})} \otimes D_1^{(\mathrm{id})}, D_1^{(\mathrm{id})}\right) = \mathrm{Hom}\left(D_1^{(\mathrm{id})}, D_1^{(\mathrm{id})}\right) = \mathbb{C}^\times, \tag{176}$$

specified by a $\mathbb{C}^\times$ valued 2-cochain $\beta$ on $G^{(0)}$. The equation (167) with $\omega = 1$ now translates to

$$\delta\beta = \omega = 1, \tag{177}$$

implying that $\beta$ must be a 2-cocycle. This 2-cocycle $\beta$ can be changed by exact 2-cocycles

$$\beta \to \beta \times \delta\gamma, \tag{178}$$

if the isomorphisms $D_0^{(g)}$ are modified as

$$D_0^{(g)} \to \gamma(g) D_0^{(g)}, \tag{179}$$

where $\gamma$ is an arbitrary $\mathbb{C}^\times$ valued 1-cochain on $G^{(0)}$.

We thus conclude that different fiber functors $\sigma$ correspond to elements

$$[\beta] \in H^2(G^{(0)}, \mathbb{C}^\times), \tag{180}$$

which classifies the 2d SPT phases protected by non-anomalous $G^{(0)}$ 0-form symmetry.

One may ask the same question for $G^{(0)}$ 0-form symmetry with non-trivial 't Hooft anomaly $[\omega]$. In this case, there are no SPT phases, because the equation (167)

$$\delta\beta = \omega, \tag{181}$$

is a contradiction to the fact that $[\omega] \neq 1$, and hence admits no solutions.

---

**Example 4.3: Representation symmetries**

Another interesting class of fusion categories are representation categories. That is, consider

$$\mathcal{S} = \mathsf{Rep}(G^{(0)}), \tag{182}$$

namely the category whose objects are finite dimensional representations of a finite group $G^{(0)}$ and morphisms are intertwiners between representations. The simple objects are thus irreducible representations. The fusion is tensor product of representations.

For an abelian group $G^{(0)}$, such a symmetry is invertible as we can identify the category as

$$\mathsf{Rep}(G^{(0)}) = \mathsf{Vec}_{\widehat{G}^{(0)}} = \mathcal{S}_{\widehat{G}^{(0)}}, \tag{183}$$

where $\widehat{G}^{(0)}$ is the group Pontryagin dual to $G^{(0)}$. For a non-abelian group, the symmetry is **non-invertible**, as a non-abelian group carries at least one irreducible representation of dimension bigger than one, which cannot be invertible as dimension is multiplicative under tensor product.

Such a symmetry is carried by a 2d theory $\mathfrak{T}_\sigma / G^{(0)}$ obtained by gauging a non-anomalous $G^{(0)}$-form symmetry of a 2d theory $\mathfrak{T}_\sigma$. For this reason, such a symmetry for non-abelian $G^{(0)}$ is referred to as a **non-intrinsic** non-invertible symmetry [22], given that the non-invertibility is not intrinsic since it can be removed by gauging $\mathsf{Rep}(G^{(0)})$ symmetry back to non-anomalous $G^{(0)}$ 0-form symmetry. See [15] for details on this gauging procedure.

---

**Example 4.4: SPT phases protected by $\mathsf{Rep}(G^{(0)})$ representation symmetry**

Consider a representation symmetry $\mathcal{S} = \mathsf{Rep}(G^{(0)})$. SPT phases protected by such a symmetry correspond to functors

$$\sigma: \quad \mathsf{Rep}(G^{(0)}) \to \mathsf{Vec}. \tag{184}$$

There is always such a functor for any $G^{(0)}$, including non-abelian $G^{(0)}$ for which the $\mathsf{Rep}(G^{(0)})$ symmetry is non-invertible. This is the forgetful functor that sends a representation to its underlying vector space and an intertwiner to the corresponding linear map between vector spaces. Thus, there is a canonical SPT phase for $\mathsf{Rep}(G^{(0)})$ symmetry, which we may refer to as the "trivial" SPT phase. This SPT phase can be obtained by gauging the $G^{(0)}$ symmetry of the 2d $G^{(0)}$-symmetric TQFT with $|G^{(0)}|$ vacua describing a phase in which $G^{(0)}$ symmetry is completely spontaneously broken.

It should be noted that there exist fusion categories which do not admit fiber functors, and hence the corresponding symmetries do not admit any SPT phase, not even a "trivial" SPT phase. We saw that $G^{(0)}$ 0-form symmetries with non-trivial 't Hooft anomalies provide examples of such symmetries. A non-invertible example is provided by Ising symmetry that is discussed in the next example. In other words, the trivial 2d theory can be made symmetric under a non-anomalous $G^{(0)}$ 0-form symmetry and under $\mathsf{Rep}(G^{(0)})$ representation symmetry, but not under an arbitrary categorical symmetry.

There may exist other SPT phases protected by $\mathsf{Rep}(G^{(0)})$ symmetry. We can describe a criteria for finding them. Choose a pair

$$(H, [\beta]), \tag{185}$$

where $H \subseteq G^{(0)}$ is a subgroup and

$$[\beta] \in H^2(H, \mathbb{C}^\times). \tag{186}$$

This pair describes an SPT phase for $\text{Rep}(G^{(0)})$ symmetry if and only if there is a single irreducible $[\beta]$-twisted representation[12] of $H$, upto isomorphism. In such a situation the category

$$\text{Rep}^{[\beta]}(H), \tag{187}$$

of $[\beta]$-twisted representations of $H$ can be identified with Vec and the action of $\text{Rep}(G^{(0)})$ on $\text{Rep}^{[\beta]}(H)$ captures the information of the corresponding fiber functor $\sigma$. Such an SPT phase can be obtained by gauging the $G^{(0)}$ symmetry of the 2d $G^{(0)}$-symmetric TQFT describing a phase in which $G^{(0)}$ symmetry is spontaneously broken to subgroup $H$ along with an SPT phase $[\beta]$ for $H$. See example 2.8 for more details on such $G^{(0)}$-symmetric 2d TQFTs.

---

**Example 4.5: Intrinsic non-invertible symmetries: Ising symmetries $\mathcal{S}_{\text{Ising}}^\pm$**

Let us introduce a couple of examples of **intrinsic** non-invertible symmetries, namely symmetries that cannot be obtained by gauging invertible symmetries. These symmetries correspond to two closely related fusion categories

$$\mathcal{S} = \mathcal{S}_{\text{Ising}}^\pm, \tag{188}$$

distinguished only by a sign in the associator. These fusion categories are referred to as Ising fusion categories, or Tambara-Yamagami categories based on the abelian group $\mathbb{Z}_2$. We would refer to the corresponding symmetries as **Ising symmetries**. The simple objects (upto isomorphisms) for both categories are

$$\text{Obj}(\mathcal{S}_{\text{Ising}}^\pm) = \left\{ S_1^{(\text{id})}, S_1^{(P)}, S_1^{(S)} \right\}. \tag{189}$$

The fusion rules can be taken to be

$$\begin{aligned}
S_1^{(P)} \otimes S_1^{(P)} &= S_1^{(\text{id})}, \\
S_1^{(P)} \otimes S_1^{(S)} &= S_1^{(S)}, \\
S_1^{(S)} \otimes S_1^{(P)} &= S_1^{(S)}, \\
S_1^{(S)} \otimes S_1^{(S)} &= S_1^{(\text{id})} \oplus S_1^{(P)},
\end{aligned} \tag{190}$$

with the non-trivial associators being

$$\begin{aligned}
S_0^{\text{ass}}\left( S_1^{(P)}, S_1^{(S)}, S_1^{(P)} \right) &= -1 \in \text{End}\left( S_1^{(S)} \right), \\
S_0^{\text{ass}}\left( S_1^{(S)}, S_1^{(P)}, S_1^{(S)} \right) &= (1, -1) \in \text{End}\left( S_1^{(\text{id})} \oplus S_1^{(P)} \right), \\
S_0^{\text{ass}}\left( S_1^{(S)}, S_1^{(S)}, S_1^{(S)} \right) &= \pm \frac{1}{\sqrt{2}} \begin{pmatrix} 1 & 1 \\ 1 & -1 \end{pmatrix} \in \text{End}\left( 2 S_1^{(S)} \right).
\end{aligned} \tag{191}$$

---

[12]A $[\beta]$-twisted representation is a representation of the $[\beta]$-twisted group algebra $\mathbb{C}[G^{(0)}]$ in which the algebra multiplication does not quite follow group rules, but is twisted by $\beta$: $v_{g_1} \cdot v_{g_2} = \beta(g_1, g_2) v_{g_1 g_2}$.

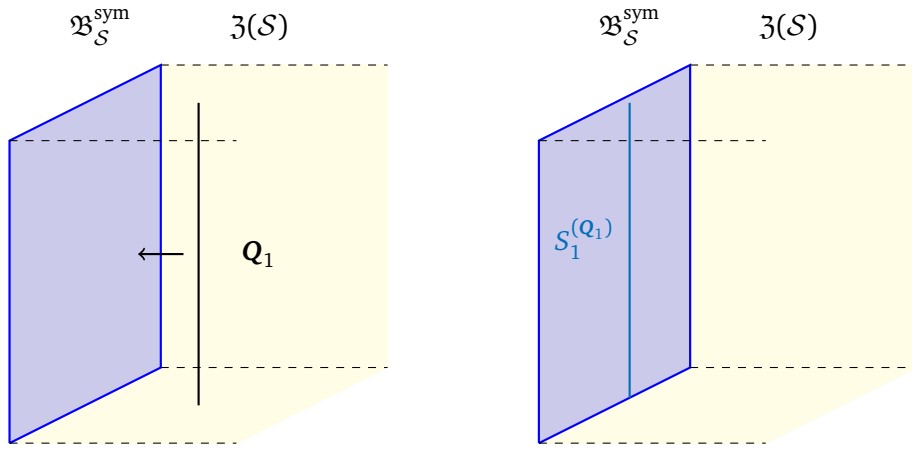

Figure 23: Projecting the bulk topological lines $Q_1$ parallel to the boundary.

The sign in the final associator $S_0^{\text{ass}}\left(S_1^{(S)}, S_1^{(S)}, S_1^{(S)}\right)$ differentiates the two Ising fusion categories $\mathcal{S}_{\text{Ising}}^+$ and $\mathcal{S}_{\text{Ising}}^-$. All other associators are trivial, i.e. equal to identity endomorphisms.

## 4.2 SymTFT topological lines and the Drinfeld center

According to the general discussion of section 3.1, the 0-charges for a symmetry $\mathcal{S}$ in 2d, namely different ways in which a symmetry $\mathcal{S}$ can act on local operators in 2d, are parametrized by the topological line defects of the associated Turaev-Viro theory $\mathfrak{Z}(\mathcal{S})$, which is the SymTFT associated to $\mathcal{S}$. The topological line defects of $\mathfrak{Z}(\mathcal{S})$ form a modular tensor category $\mathcal{Z}(\mathcal{S})$ that can be recognized as what is known mathematically as the **Drinfeld center** of $\mathcal{S}$. This can be seen by studying the interactions of topological line operators of $\mathfrak{Z}(\mathcal{S})$ with a topological boundary condition $\mathfrak{B}_{\mathcal{S}}^{\text{sym}}$ of $\mathfrak{Z}(\mathcal{S})$ whose symmetry category is $\mathcal{S}$, i.e. the topological line operators on $\mathfrak{B}_{\mathcal{S}}^{\text{sym}}$ form the fusion category $\mathcal{S}$. In the language of SymTFT, we refer to $\mathfrak{B}_{\mathcal{S}}^{\text{sym}}$ as the symmetry boundary condition for the sandwich construction of $\mathcal{S}$-symmetric 2d theories.

Following the general analysis of section 4.2, let us study the interactions of topological line operators of $\mathfrak{Z}(\mathcal{S})$ with the boundary $\mathfrak{B}_{\mathcal{S}}^{\text{sym}}$ (see also [51]). Consider a topological line operator $Q_1$ of $\mathfrak{Z}(\mathcal{S})$ and push it to the symmetry boundary $\mathfrak{B}_{\mathcal{S}}^{\text{sym}}$, as shown in figure 23. In this way, we obtain a topological line operator $S_1^{(Q_1)}$ of $\mathfrak{B}_{\mathcal{S}}^{\text{sym}}$, which is an object of $\mathcal{S}$

$$Q_1 \to S_1^{(Q_1)}. \tag{192}$$

Note that we are working with general non-simple objects and topological lines here. Also, note, that not all symmetry generators will be simply projections to the boundary of $Q_1$ lines (e.g. we will see later that the $S_3$ symmetry lines do not arise as a projection).

Now, let us perform the above projection in the presence of a topological line operator $S_1$ of $\mathfrak{B}_{\mathcal{S}}^{\text{sym}}$. As shown in figure 24, this procedure produces a topological local operator $S_0^{(Q_1)}(S_1)$ on $\mathfrak{B}_{\mathcal{S}}^{\text{sym}}$

$$S_0^{(Q_1)}(S_1): \quad S_1 \otimes S_1^{(Q_1)} \to S_1^{(Q_1)} \otimes S_1, \tag{193}$$

which mathematically is a morphism in the fusion category $\mathcal{S}$, sometimes referred to as a **half-braiding**. Let us collect $S_0^{(Q_1)}(S_1)$ for all $S_1$, and refer to this collection as $S_0^{(Q_1)}$.

Now, instead of performing the projection of $Q_1$ in the presence of a single topological line of $\mathfrak{B}_{\mathcal{S}}^{\text{sym}}$, consider performing the projection in the presence of two topological lines $S_1$ and $S_1'$

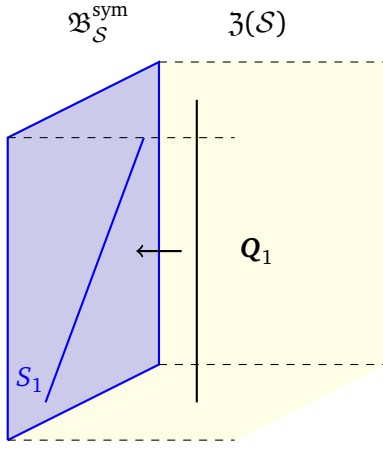 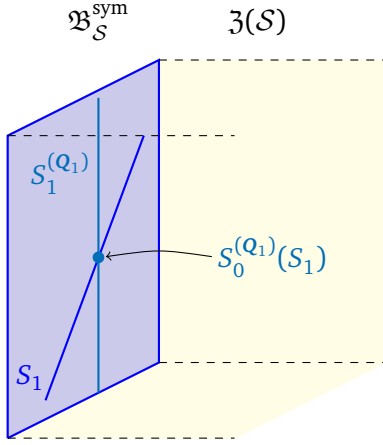

Figure 24: Projecting the bulk topological lines $Q_1$ parallel to the boundary in the presence of a line $S_1$ on the boundary, which results in a junction operator $S_0^{(Q_1)}(S_1)$, which is sometimes referred to as the half-braiding.

of $\mathfrak{B}_{\mathcal{S}}^{\text{sym}}$. As shown in figure 25, we can now either project $Q_1$ first and then fuse $S_1, S_1'$, or first fuse $S_1, S_1'$ and then project $Q_1$. These two processes must yield identical results, leading to a consistency condition on the pair

$$\left( S_1^{(Q_1)}, S_0^{(Q_1)} \right), \tag{194}$$

which can be expressed as

$$S_0^{\text{ass}}\left( S_1^{(Q_1)}, S_1, S_1' \right) \circ \underline{S_0^{(Q_1)}(S_1)} \circ \left( S_0^{\text{ass}} \right)^{-1} \left( S_1, S_1^{(Q_1)}, S_1' \right) \circ \underline{S_0^{(Q_1)}\left( S_1' \right)} \circ S_0^{\text{ass}}\left( S_1, S_1', S_1^{(Q_1)} \right)$$
$$= \underline{S_0^{(Q_1)}\left( S_1 \otimes S_1' \right)}. \tag{195}$$

An object $S_1^{(Q_1)}$ of $\mathcal{S}$ along with a collection $S_0^{(Q_1)}$ of morphisms of type (193), satisfying the condition (195) is precisely the mathematical definition for an object $Q_1$ of the Drinfeld center $\mathcal{Z}(\mathcal{S})$ of $\mathcal{S}$

$$Q_1 = \left( S_1^{(Q_1)}, S_0^{(Q_1)} \right). \tag{196}$$

Thus we have justified the following statement:

---

**Statement 4.1: Topological lines of SymTFT from Drinfeld center**

Consider a symmetry of 2d theories described by a fusion category $\mathcal{S}$. Topological line operators of the associated SymTFT $\mathfrak{Z}(\mathcal{S})$, which is a 3d TQFT, correspond to objects of the Drinfeld center category $\mathcal{Z}(\mathcal{S})$ of the fusion category $\mathcal{S}$.

---

Now consider a topological local operator $Q_0$ between two topological line operators $Q_1$ and $Q_1'$ of the SymTFT $\mathfrak{Z}(\mathcal{S})$

$$Q_0: \quad Q_1 \to Q_1'. \tag{197}$$

This local operator corresponds to a morphism in the Drinfeld center $\mathcal{Z}(\mathcal{S})$. To see this, project the configuration

$$(Q_1, Q_1', Q_0), \tag{198}$$

to the symmetry boundary $\mathfrak{B}_{\mathcal{S}}^{\text{sym}}$ as shown in 26. This projects to a configuration

$$(Q_1, Q_1', Q_0) \to \left( S_1^{(Q_1)}, S_1^{(Q_1')}, S_0^{(Q_0)} \right), \tag{199}$$

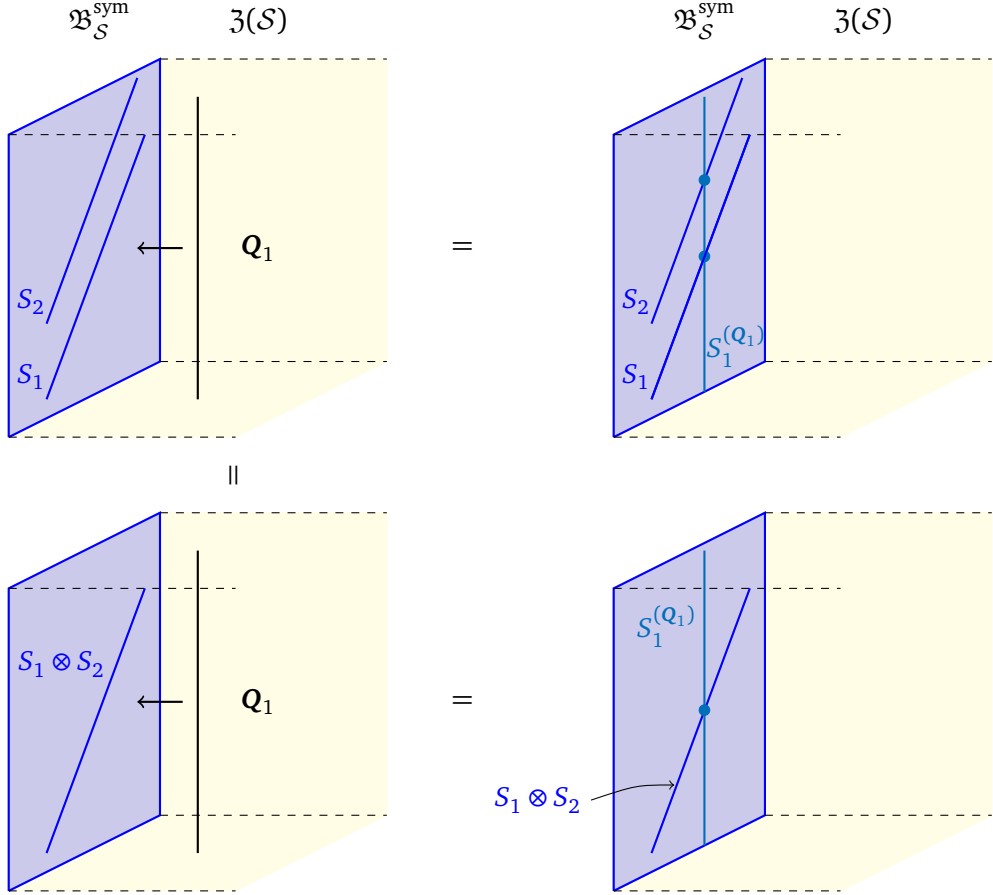

Figure 25: Projection of the bulk topological lines $\boldsymbol{Q}_1$ has to commute with the process of fusion of boundary topological lines.

where $S_0^{(\boldsymbol{Q}_0)}$ is a topological local operator on $\mathfrak{B}_\mathcal{S}^{\text{sym}}$ between topological lines $S_1^{(\boldsymbol{Q}_1)}$, $S_1^{(\boldsymbol{Q}_1')}$ of $\mathfrak{B}_\mathcal{S}^{\text{sym}}$, described by a morphism of $\mathcal{S}$

$$S_0^{(\boldsymbol{Q}_0)}: \quad S_1^{(\boldsymbol{Q}_1)} \to S_1^{(\boldsymbol{Q}_1')}. \tag{200}$$

Now consider projecting the configuration (198) in the presence of a boundary line $S_1 \in \mathcal{S}$. There are two possible projections as shown in figure 27 including now lines on the symmetry boundary, which should yield identical results. This means that the morphism $S_0^{(\boldsymbol{Q}_0)}$ has to satisfy the following consistency condition with the half-braidings

$$S_0^{(\boldsymbol{Q}_1')}(S_1) \circ S_0^{(\boldsymbol{Q}_0)} = S_0^{(\boldsymbol{Q}_0)} \circ S_0^{(\boldsymbol{Q}_1)}(S_1). \tag{201}$$

A morphism (200) satisfying the condition (201) is precisely the mathematical definition for a morphism of the Drinfeld center $\mathcal{Z}(\mathcal{S})$ of $\mathcal{S}$. Thus we have justified the following statement:

---

**Statement 4.2: Category of topological lines of SymTFT is Drinfeld center**

Consider a symmetry of 2d theories described by a fusion category $\mathcal{S}$. The category formed by topological line and local operators of the associated SymTFT $\mathfrak{Z}(\mathcal{S})$ is the Drinfeld center $\mathcal{Z}(\mathcal{S})$ of the fusion category $\mathcal{S}$.

---

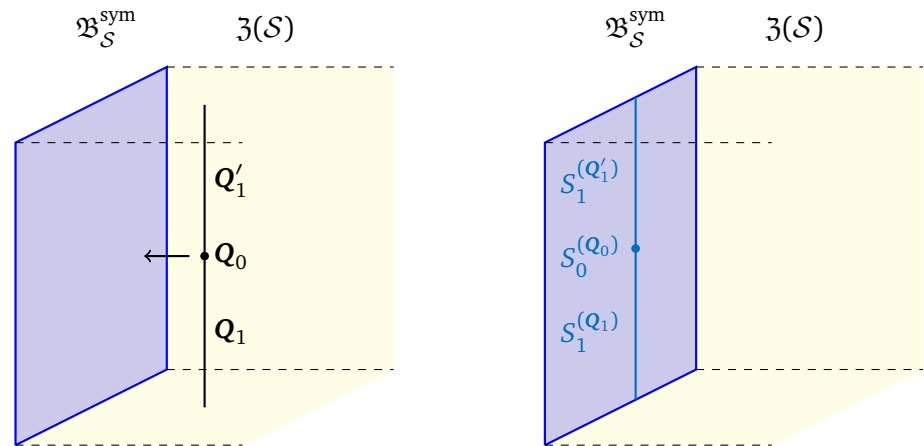

Figure 26: Projecting the bulk topological lines $Q_1$, $Q_1'$ with junction $Q_0$ parallel to the boundary.

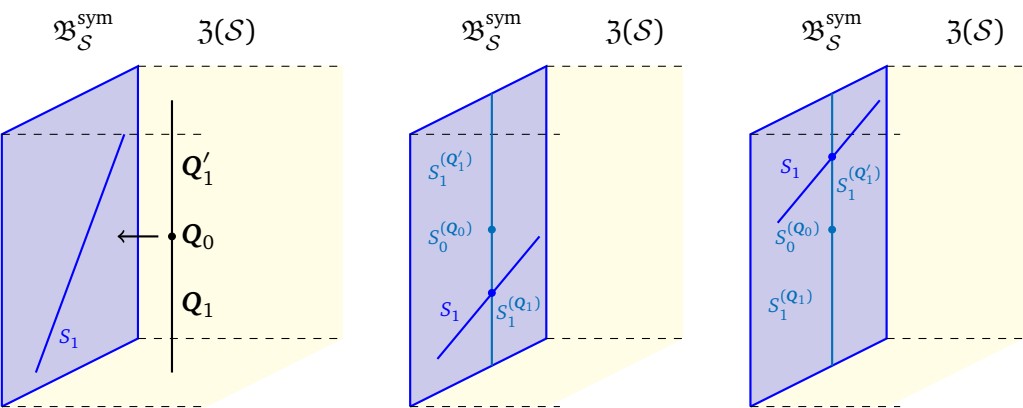

Figure 27: Projecting the bulk topological lines $Q_1$, $Q_1'$ with junction $Q_0$ parallel to the boundary in the presence of a line $S_1$ on the boundary. The equality of the diagrams results in the consistency condition (201).

### 4.3 Action of topological operators on charges

In order to understand how a topological line operator $Q_1 \in \mathcal{Z}(\mathcal{S})$ encodes a 0-charge for an $\mathcal{S}$ symmetry, we need to understand the topological local operators arising at the end of $Q_1$ along $\mathfrak{B}_{\mathcal{S}}^{\mathrm{sym}}$ and how these topological local operators interact with the topological line operators of $\mathfrak{B}_{\mathcal{S}}^{\mathrm{sym}}$.

Denote by

$$V^{(Q_1)}(S_1), \tag{202}$$

the vector space of topological local operators arising at the end of bulk line $Q_1$ along $\mathfrak{B}_{\mathcal{S}}^{\mathrm{sym}}$, attached to a boundary line $S_1 \in \mathcal{S}$. See figure 28. A topological local operator

$$S_0 \in \mathrm{Hom}(S_1, S_1'), \tag{203}$$

on $\mathfrak{B}_{\mathcal{S}}^{\mathrm{sym}}$ acts as a linear map

$$L(S_0): \quad V^{(Q_1)}(S_1) \to V^{(Q_1)}(S_1'), \tag{204}$$

by performing a fusion operation as shown in figure 29.

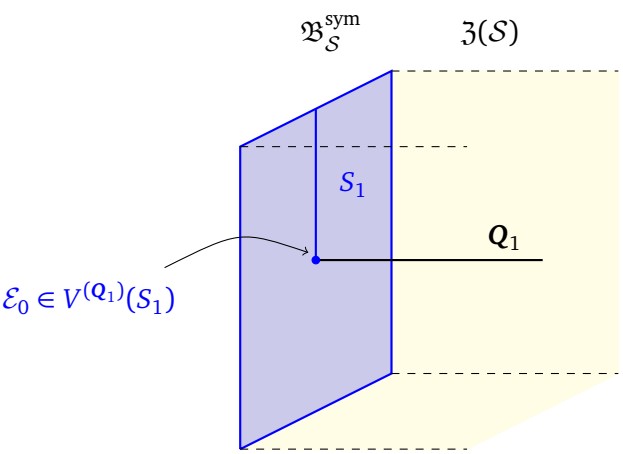

Figure 28: $V^{\boldsymbol{Q}_1}(S_1)$ is the vector space at the junction between the bulk line $\boldsymbol{Q}_1$ and the topological defect $S_1$ on the boundary $\mathfrak{B}_{\mathcal{S}}^{\text{sym}}$.

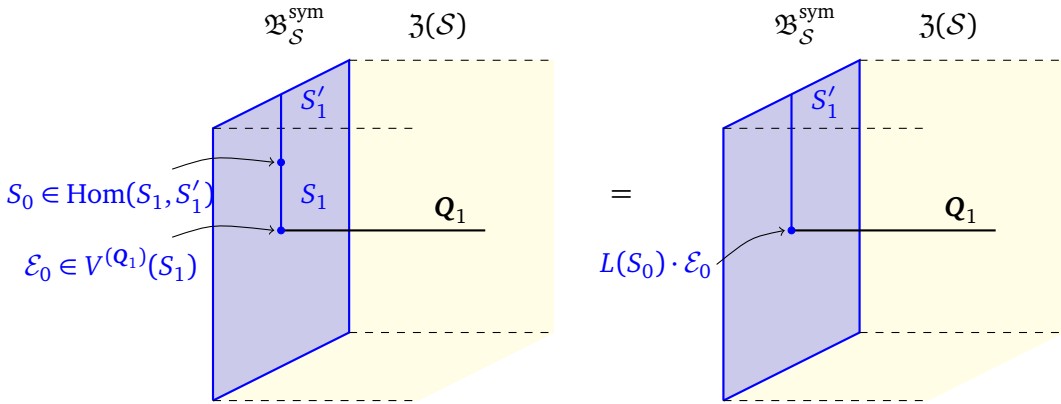

Figure 29: A topological line $\boldsymbol{Q}_1$ in the SymTFT ends on the symmetry boundary, with junction $\mathcal{E}_0$ with the line $S_1$. A topological local operator $S_0 \in \text{Hom}(S_1, S'_1)$ can be fused with $\mathcal{E}_0$ to create a junction $L(S_0) \cdot \mathcal{E}_0$ between $\boldsymbol{Q}_1$ and $S'_1$.

Now note that there is a canonical local operator

$$\mathcal{E}_0^{(\boldsymbol{Q}_1)} \in V^{(\boldsymbol{Q}_1)}\left(S_1^{(\boldsymbol{Q}_1)}\right), \tag{205}$$

obtained by dipping half of an L-shaped $\boldsymbol{Q}_1$ bulk line into $\mathfrak{B}_{\mathcal{S}}^{\text{sym}}$ as shown in figure 30. Recall that $S_1^{(\boldsymbol{Q}_1)}$ is the projection of $\boldsymbol{Q}_1$ onto the symmetry boundary (parallel to the boundary). This operator commutes with the half-braiding (193) of $S_1$ as argued in figure 31, which means that the action of all half-braidings on $\mathcal{E}_0^{(\boldsymbol{Q}_1)}$ is trivial.

Moreover, this canonical operator allows us to express any operator $\mathcal{E}_0 \in V^{(\boldsymbol{Q}_1)}(S_1)$ as

$$\mathcal{E}_0 = L(S_0^{(\mathcal{E}_0)}) \cdot \mathcal{E}_0^{(\boldsymbol{Q}_1)}, \quad \text{for some} \quad S_0^{(\mathcal{E}_0)} \in \text{Hom}\left(S_1^{(\boldsymbol{Q}_1)}, S_1\right). \tag{206}$$

See figure 32 for an explanation. Thus, the operator $\mathcal{E}_0^{(\boldsymbol{Q}_1)}$ provides a natural isomorphism

$$V^{(\boldsymbol{Q}_1)}(S_1) \cong \text{Hom}\left(S_1^{(\boldsymbol{Q}_1)}, S_1\right). \tag{207}$$

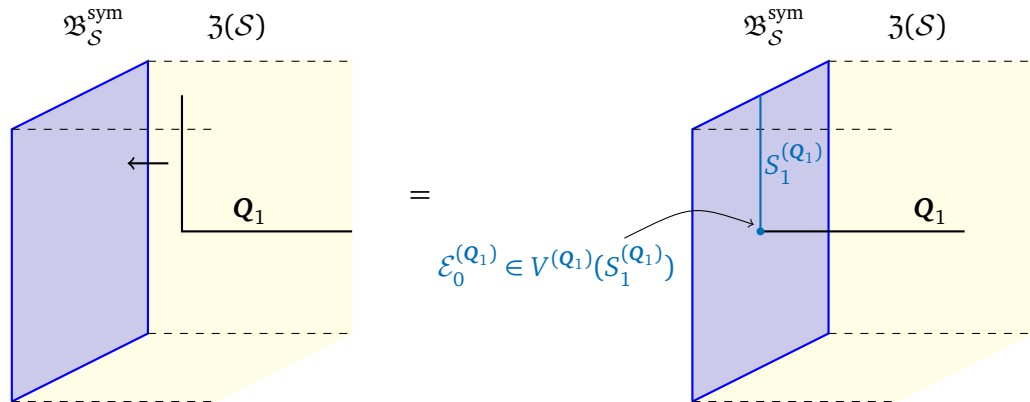

Figure 30: The L-dip. Projecting half of an L-shaped SymTFT topological line operator $Q_1$ onto the symmetry boundary $\mathfrak{B}_S^{\mathrm{sym}}$, results in a canonical end $\mathcal{E}_0^{(Q_1)}$ of $Q_1$ along $\mathfrak{B}_S^{\mathrm{sym}}$ attached to a topological line defect $S_1^{(Q_1)}$ of $\mathfrak{B}_S^{\mathrm{sym}}$.

**Action of symmetry on 0-charges.** We can now describe various types of actions of $S$ on an end local operator

$$\mathcal{E}_0 \in V^{(Q_1)}(S_1), \tag{208}$$

for some $S_1$. All such actions can be deduced from the following simple action. Consider a line $S_1' \in S$ which does not intersect $S_1$ and move it past $\mathcal{E}_0$. The result of this move can be represented as shown in figure 33, where we have the end local operator $\mathcal{E}_0^{(Q_1)}$ emitting $S_1^{(Q_1)}$ which is converted into $S_1$ by a morphism

$$S_0^{(\mathcal{E}_0, S_1')} := S_0^{(\mathcal{E}_0)} \circ S_0^{(Q_1)}(S_1') \in \mathrm{Hom}\left(S_1' \otimes S_1^{(Q_1)}, S_1 \otimes S_1'\right), \tag{209}$$

where

$$S_0^{(\mathcal{E}_0)} \in \mathrm{Hom}(S_1^{(Q_1)}, S_1), \tag{210}$$

is obtained from $\mathcal{E}_0$ by applying the isomorphism (207).

Any other action can be understood in terms of the above action. For example, consider a line $S_1'$ linking $\mathcal{E}_0$, such that as it intersects $S_1$, it converts $S_1$ into $S_1''$ via a local operator

$$S_0 \in \mathrm{Hom}(S_1'^* \otimes S_1, \, S_1'' \otimes S_1'^*). \tag{211}$$

This is also known as the lasso action [41, 46]. As shown in figure 34, by applying the above-discussed simple action, we can replace this configuration by a configuration involving end operator $\mathcal{E}_0^{(Q_1)}$ emitting $S_1^{(Q_1)}$ which is converted into $S_1''$ by a morphism

$$S_0^{\mathrm{ev}}(S_1') \circ S_0^{\mathrm{ass}}(S_1'', S_1'^*, S_1') \circ S_0 \circ (S_0^{\mathrm{ass}})^{-1}(S_1'^*, S_1, S_1') \circ S_0^{(\mathcal{E}_0, S_1')} \circ S_0^{\mathrm{ass}}\left(S_1'^*, S_1', S_1^{(Q_1)}\right) \circ S_0^{\mathrm{coev}}(S_1'^*), \tag{212}$$

in $\mathrm{Hom}(S_1^{(Q_1)}, S_1'')$, where $S_0^{\mathrm{ev}}$ and $S_0^{\mathrm{coev}}$ are evaluation and co-evaluation maps respectively.

Ultimately, all of the information regarding the action is encoded in the bulk topological line operator $Q_1$ that we begin with. We have thus justified the following statement:

---

**Statement 4.3: 0-charges of a symmetry in 2d**

0-charges of a symmetry $S$ of 2d theories correspond to topological line operators of the associated SymTFT $\mathfrak{Z}(S)$.

---

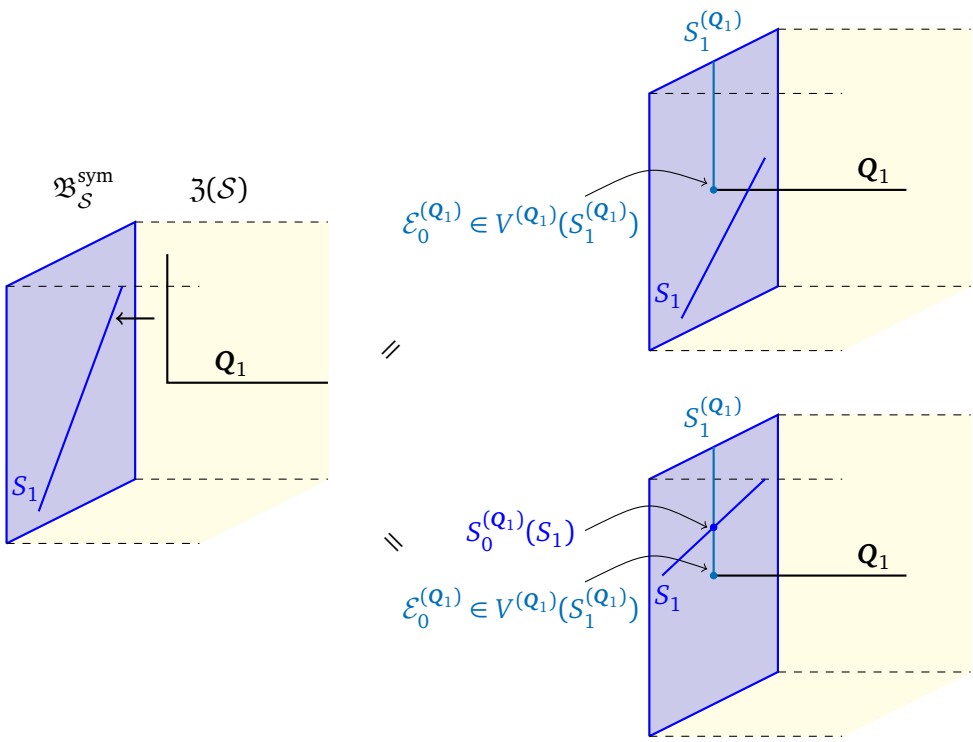

Figure 31: There are two ways of performing an L-dip in the presence of a boundary topological line $S_1$. Equating the two ways implies that the half-braiding of $\boldsymbol{Q}_1$ with $S_1$ acts trivially on the canonical end of $\boldsymbol{Q}_1$.

## 4.4 Examples of 0-charges

We now provide a large class of examples of 0-charges and the action of symmetries on these: starting with (not necessarily abelian) group-like 0-form symmetries, including the specialization to abelian groups and including anomalies. We also discuss the representation-category symmetries. Finally, we study intrinsic non-invertible symmetries given by the Ising category.

### 4.4.1 Invertible symmetries: General case

Consider a $G^{(0)}$ 0-form symmetry in 2d with a 't Hooft anomaly

$$\omega \in H^3(G^{(0)}, \mathbb{C}^\times), \tag{213}$$

discussed in detail in example 4.1. Let us compute the Drinfeld center $\mathcal{Z}(\mathcal{S}^\omega_{G^{(0)}})$, or in other words the possible 0-charges.

**Untwisted 0-charges.** First of all, let us consider untwisted 0-charges, i.e. those 0-charges characterizing multiplets containing only local operators in untwisted sector for $G^{(0)}$ 0-form symmetry. For such a 0-charge $\boldsymbol{Q}_1$, we have the projection onto $\mathfrak{B}^{\text{sym}}_{\mathcal{S}^\omega_{G^{(0)}}}$

$$S_1^{(\boldsymbol{Q}_1)} \cong n S_1^{(\text{id})}, \quad n > 0 \in \mathbb{Z}, \tag{214}$$

where $S_1^{(\text{id})}$ is the identity line on $\mathfrak{B}^{\text{sym}}_{\mathcal{S}^\omega_{G^{(0)}}}$. The local operator $S_0^{(\boldsymbol{Q}_1)}\left(S_1^{(g)}\right)$ can be identified with an endomorphism of $n S_1^{(g)}$, or equivalently as an endomorphism of a vector space

$$V_{\boldsymbol{Q}_1} \cong \mathbb{C}^n. \tag{215}$$

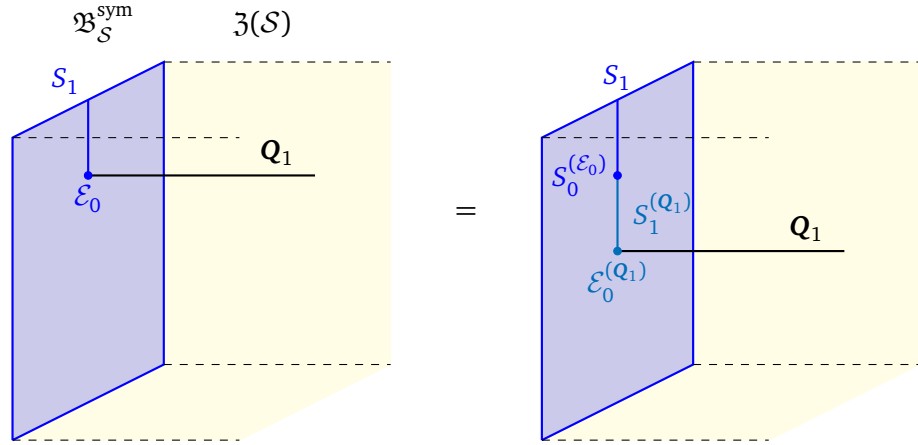

Figure 32: Mapping from any $\mathcal{E}_0$ operator to the canonical operator $\mathcal{E}_0^{(Q_1)}$ with a morphism $S_0^{(\mathcal{E}_0)} \in \mathrm{Hom}(S_1^{(Q_1)}, S_1)$. This is done just by partially projecting $Q_1$ below $\mathcal{E}_0$.

The condition (195) becomes

$$S_0^{(Q_1)}\left(S_1^{(g)}\right) S_0^{(Q_1)}\left(S_1^{(g')}\right) = S_0^{(Q_1)}\left(S_1^{(gg')}\right), \tag{216}$$

converting $V_{Q_1}$ into a **representation** of the 0-form symmetry group $G^{(0)}$. Thus, **untwisted 0-charges are characterized by representations of $G^{(0)}$**.

**Twisted 0-charges.** Now consider an irreducible twisted 0-charge $Q_1$, i.e. a 0-charge characterizing multiplets containing local operators in twisted sector for $G^{(0)}$ 0-form symmetry. For morphisms $S_0^{(Q_1)}$ to exist, we must have

$$S_1^{(Q_1)} \cong n \bigoplus_{g \in [g]} S_1^{(g)}, \quad n > 0 \in \mathbb{Z}, \tag{217}$$

where $[g]$ is a conjugacy class of $G^{(0)}$. The half-braidings can now be identified as endomorphisms of a vector space

$$V_{Q_1} = \bigoplus_{g \in [g]} V_{Q_1}^{(g)}, \tag{218}$$

where each $V_{Q_1}^{(g)} \cong \mathbb{C}^n$ admits a closed action of

$$S_0^{(Q_1)}\left(S_1^{(h)}\right), \quad h \in H_g \subseteq G^{(0)}, \tag{219}$$

where $H_g$ is the centralizer subgroup of the element $g \in G^{(0)}$. The condition (195) becomes

$$\omega_g(h, h') S_0^{(Q_1)}\left(S_1^{(h)}\right) S_0^{(Q_1)}\left(S_1^{(h')}\right) = S_0^{(Q_1)}\left(S_1^{(hh')}\right), \tag{220}$$

for all $h, h' \in H_g$, where

$$\omega_g(h, h') := \frac{\omega(g, h, h') \omega(h, h', g)}{\omega(h, g, h')} \in \mathbb{C}^\times, \tag{221}$$

is a 2-cocycle on $H_g$, referred to as being obtained by taking the **slant product** of $\omega$ with $g$ (see e.g. [55]). The equation (220) means that $V_{Q_1}^{(g)}$ forms an $\omega_g$-twisted representation of $H_g$. In fact, the whole action of $G^{(0)}$ on the space $V_{Q_1}$ is induced from this $\omega_g$-twisted representation of $H_g$. Thus, irreducible **twisted 0-charges** are characterized by following two pieces of data:

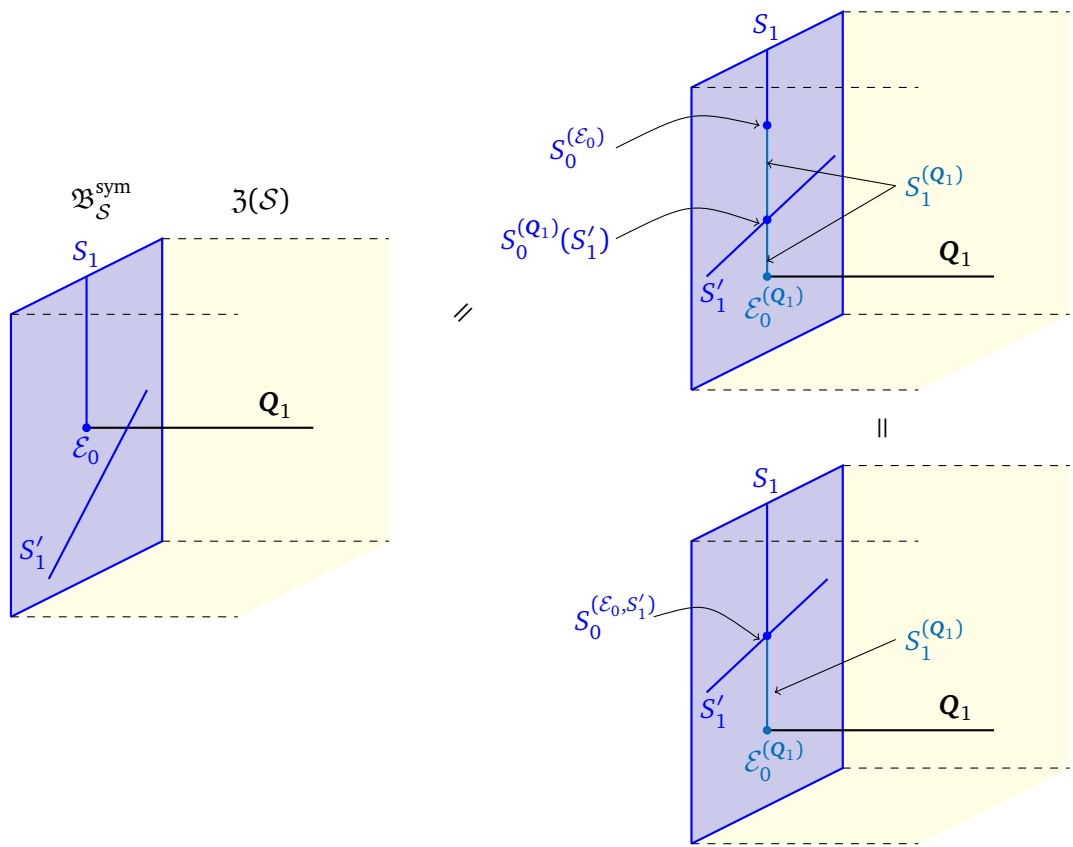

Figure 33: Key action of the symmetry $\mathcal{S}$ on the local operators $\mathcal{E}_0$ at the ends of topological lines $\boldsymbol{Q}_1$ of the SymTFT. This is the 2d version of figure 17, carrying out the fusion at the end explicitly.

1. A conjugacy class $[g] \subset G^{(0)}$.

2. An $\omega_g$-twisted irreducible representation $\boldsymbol{R}$ of the centralizer $H_g$ of an element $g \in [g]$, where $\omega_g$ is specified in (221).

We denote an irreducible 0-charge specified by the above two data as

$$\boldsymbol{Q}_1^{([g],\boldsymbol{R})}. \tag{222}$$

**Full Drinfeld center.** We can express the category of 0-charges as

$$\mathcal{Z}(\mathcal{S}_{G^{(0)}}^{\omega}) = \bigoplus_{[g]} \mathsf{Rep}^{\omega_g}(H_g), \tag{223}$$

where the sum is over conjugacy classes of $G^{(0)}$, and $\mathsf{Rep}^{\omega_g}(H_g)$ is the category of $\omega_g$-twisted representations of the centralizer $H_g$ of a representative $g$ in conjugacy class $[g]$. The morphisms are intertwiners of these representations. When there is no anomaly $\omega = 1$, then all $\omega_g = 1$, implying that we have standard representations of centralizers in (223):

$$\mathcal{Z}(\mathcal{S}_{G^{(0)}}) = \bigoplus_{[g]} \mathsf{Rep}(H_g). \tag{224}$$

**Action of symmetry on 0-charges.** Consider an irreducible 0-charge $\boldsymbol{Q}_1^{([g],\boldsymbol{R})}$. A multiplet $\mathcal{M}_0$ of local operators in a $\mathcal{S}_{G^{(0)}}^{\omega}$-symmetric 2d theory $\mathfrak{T}_\sigma$ transforming in the 0-charge $\boldsymbol{Q}_1^{([g],\boldsymbol{R})}$

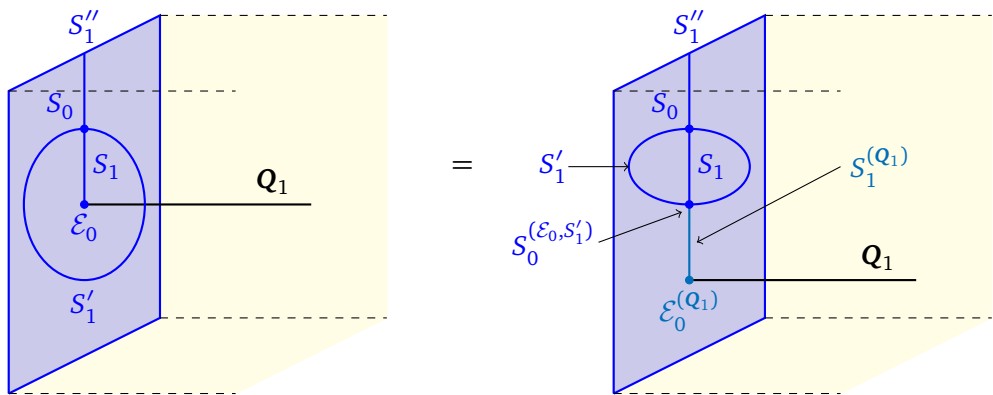

Figure 34: Lasso action follows from the general action in figure 33.

comprises of local operators in $g$-twisted sectors for all $g \in [g]$. The vector space formed by $g$-twisted sector operators in the multiplet $\mathcal{M}_0$ can be identified with the vector space $V_{Q_1}^{(g)}$ appearing in (218). In total, combining the different twisted sector operators together, the local operators living in the multiplet $\mathcal{M}_0$ can be identified with the vector space $V_{Q_1}$ appearing in (218). The action of $G^{(0)}$ on these local operators is identified with the action of $G^{(0)}$ on $V_{Q_1}^{(g)}$.

### 4.4.2 Invertible symmetries: Abelian case

For abelian $G^{(0)}$, we can decompose the Drinfeld center as

$$\mathcal{Z}\left(\mathcal{S}_{G^{(0)}}^{\omega}\right) = \mathrm{Vec}_{G^{(0)}} \boxtimes \mathrm{Rep}\left(G^{(0)}\right). \tag{225}$$

That is, the simple topological line operators of the SymTFT $\mathfrak{Z}\left(\mathcal{S}_{G^{(0)}}^{\omega}\right)$ can be labeled as

$$Q_1^{(g,R)}, \tag{226}$$

where $g \in G^{(0)}$ and $R$ is a one-dimensional representation of $G^{(0)}$ twisted by $\omega_g$. Such twisted representations are in one-to-one correspondence with standard linear representations, as one can go from one twisted representation to another twisted representation by tensoring with a linear representation. We can in fact identify the line operators $Q_1^{(g,R)}$ in terms of the Lagrangian formulation (81), which takes the following form in this case

$$S[\mathfrak{Z}(\mathcal{S}_{G^{(0)}}^{\omega})] = \int_{M_3} a_1 \cup \delta b_1 + \mathcal{A}^* \omega(a_1). \tag{227}$$

The identification is as follows:

- The lines $Q_1^{(\mathrm{id},R)}$ for various linear representations $R$ of $G^{(0)}$ can be identified with Wilson lines $W_R(a_1)$ in the representation $R$ for the $G^{(0)}$-valued gauge field $a_1$

$$Q_1^{(\mathrm{id},R)} = W_R(a_1). \tag{228}$$

- Similarly, we can try to construct Wilson lines using the $\widehat{G}^{(0)}$-valued gauge field $b_1$, which requires a choice of a representation of $\widehat{G}^{(0)}$, or equivalently a choice of an element $g \in G^{(0)}$: $\widehat{W}_g(b_1)$. However, such a Wilson line is not gauge invariant under bulk gauge

transformations. This is an effect of the twist term $\mathcal{A}^*\omega(a_1)$. Under bulk gauge transformations, the line $\widehat{W}_g(b_1)$ picks up a term which is a function of $a_1$. To cancel this gauge variation, we have to dress the Wilson line $\widehat{W}_g(b_1)$ with another gauge non-invariant Wilson line for $a_1$ transforming in a $\omega_g$-twisted representation $\boldsymbol{R}$ of $G^{(0)}$. The combined Wilson line for $b_1$ and $a_1$ gauge fields is gauge invariant,

$$\boldsymbol{Q}_1^{(g,R)} = \widehat{W}_g(b_1)W_{\boldsymbol{R}^{\omega_g}}(a_1),\tag{229}$$

and in this way we obtain the lines $\boldsymbol{Q}_1^{(g,R)}$ for general $g$. This alternate construction does not use information of the symmetry boundary $\mathfrak{B}^{\text{sym}}_{\mathcal{S}^\omega_{G^{(0)}}}$.

The different Drinfeld centers (225) for different $\omega$ but same abelian $G^{(0)}$ are distinguished by the spins of the lines $\boldsymbol{Q}_1^{(g,R)}$, which is computed as

$$\theta(\boldsymbol{Q}_1^{(g,R)}) = \boldsymbol{R}(g) \in U(1),\tag{230}$$

where $\boldsymbol{R}(g)$ is the phase by which $g$ acts in the one-dimensional twisted representation $\boldsymbol{R}$.

### 4.4.3 Invertible symmetries: Some special cases

Let us discuss the 0-charges for a few special cases of invertible symmetries.

1. $\underline{\mathbb{Z}_2 \textbf{ without Anomaly}}$: Consider a non-anomalous

$$G^{(0)} = \mathbb{Z}_2 = \langle \text{id}, b\rangle,\qquad b^2 = \text{id},\tag{231}$$

0-form symmetry, where id is the identity element and $b$ is the generator. The representations will be denoted by $\boldsymbol{R} = \pm$, corresponding to the trivial $+$ and non-trivial $-$ representation of $\mathbb{Z}_2$. In this case, the SymTFT is also known as the **toric code** model. We have four irreducible 0-charges:

$$\begin{aligned}
\boldsymbol{Q}_1^{(\text{id},+)} &= \{\text{Untwisted sector operator with } b \text{ acting trivially}\},\\
\boldsymbol{Q}_1^{(\text{id},-)} &= \{\text{Untwisted sector operator with } b \text{ acting by a minus sign}\},\\
\boldsymbol{Q}_1^{(b,+)} &= \{\text{Twisted sector operator with } b \text{ acting trivially}\},\\
\boldsymbol{Q}_1^{(b,-)} &= \{\text{Twisted sector operator with } b \text{ acting by a minus sign}\}.
\end{aligned}\tag{232}$$

2. $\underline{\mathbb{Z}_2 \textbf{ with Anomaly}}$: Since

$$H^3(\mathbb{Z}_2, \mathbb{C}^\times) = \mathbb{Z}_2,\tag{233}$$

a $\mathbb{Z}_2$ 0-form symmetry can have a unique non-trivial anomaly $\omega$. Let us study 0-charges for a $\mathbb{Z}_2$ 0-form symmetry with the non-trivial anomaly. In this case, the SymTFT is also known as the **double semion** model. We have four irreducible 0-charges:

$$\begin{aligned}
\boldsymbol{Q}_1^{(\text{id},+)} &= \{\text{Untwisted sector operator with } b \text{ acting trivially}\},\\
\boldsymbol{Q}_1^{(\text{id},-)} &= \{\text{Untwisted sector operator with } b \text{ acting by a minus sign}\},\\
\boldsymbol{Q}_1^{(b,i)} &= \{\text{Twisted sector operator with } b \text{ acting by } i\},\\
\boldsymbol{Q}_1^{(b,-i)} &= \{\text{Twisted sector operator with } b \text{ acting by } -i\}.
\end{aligned}\tag{234}$$

The action of $b$ is $\pm i$ for twisted sector operators because that is how $b$ acts in irreducible $\omega_b$-twisted representations. To see this, choose a representative of $\omega$ such that the only non-trivial evaluation is

$$\omega(b,b,b) = -1,\tag{235}$$

from which we compute that the only non-trivial evaluation of the slant product $\omega_b$ is

$$\omega_b(b,b) = -1 \,. \tag{236}$$

Consequently the square of the action of $b$ in a $\omega_b$-twisted representation is $-1$, implying that $b$ acts in the two irreducible $\omega_b$-twisted representations as $\pm i$ respectively.

3. $\underline{S_3 \text{ without Anomaly}}$: For abelian 0-form symmetries, all irreducible multiplets contain a single operator. However, for all non-abelian 0-form symmetries, one has irreducible multiplets containing multiple operators, and this happens already in the untwisted sector. The simplest of non-abelian finite groups is $G^{(0)} = S_3$, namely the permutation group of three elements, which we parameterize as

$$G^{(0)} = S_3 = \{\mathrm{id}, a, a^2, b, ab, a^2 b\} \,, \tag{237}$$

with relationships

$$a^3 = b^2 = \mathrm{id} \,, \qquad ba = a^2 b \,. \tag{238}$$

Let us consider non-anomalous $G^{(0)} = S_3$ 0-form symmetry. The irreducible representations will be denoted by $+$ (trivial), $-$ (sign), $E$ (2d representation). The untwisted irreducible 0-charges are

$$
\begin{aligned}
Q_1^{(\mathrm{id},+)} &= \{\text{Untwisted sector operator with } (a,b) \text{ acting trivially}\} \,, \\
Q_1^{(\mathrm{id},-)} &= \{\text{Untwisted sector operator with } (a,b) \text{ acting as } (1,-1)\} \,, \\
Q_1^{(\mathrm{id},E)} &= \Big\{\text{Two-dimensional multiplet } (\mathcal{O}_0^{(E,1)}, \mathcal{O}_0^{(E,2)}) \text{ of untwisted sector operators;} \\
&\qquad \text{with } b \text{ acting as the exchange } \mathcal{O}_0^{(E,1)} \longleftrightarrow \mathcal{O}_0^{(E,2)} \\
&\qquad \text{and } a \text{ acting diagonally as } \mathcal{O}_0^{(E,1)} \longrightarrow \omega \mathcal{O}_0^{(E,1)}, \quad \mathcal{O}_0^{(E,2)} \longrightarrow \omega^2 \mathcal{O}_0^{(E,2)}\Big\} \,,
\end{aligned}
\tag{239}
$$

where $\omega$ is a third root of unity.

In addition to the above, there are two different types of twisted sector charges as there are two non-trivial conjugacy classes $[a]$ and $[b]$, whose representatives can be chosen to be $a$ and $b$ respectively. The corresponding stabilizers are

$$H_a = \{\mathrm{id}, a, a^2\} = \mathbb{Z}_3 \,, \qquad H_b = \{\mathrm{id}, b\} = \mathbb{Z}_2 \,. \tag{240}$$

Let us discuss first the twisted sector operators associated to the conjugacy class $[a]$. An irreducible multiplet involving such twisted sector operators is two-dimensional, with one of the operators $\mathcal{O}_0^{(a)}$ being in $a$-twisted sector and the other operator $\mathcal{O}_0^{(a^2)}$ being in $a^2$-twisted sector. On top of that, we can specify the action of $S_3$ on the multiplet by specifying the action of $H_a$ on the operator $\mathcal{O}_0^{(a)}$. We have three irreducible 0-charges of this type

$$
\begin{aligned}
Q_1^{([a],\mathrm{id})} &= \Big\{a \text{ acting as } \mathcal{O}_0^{(a)} \longrightarrow \mathcal{O}_0^{(a)}\Big\} \,, \\
Q_1^{([a],\omega)} &= \Big\{a \text{ acting as } \mathcal{O}_0^{(a)} \longrightarrow \omega \mathcal{O}_0^{(a)}\Big\} \,, \\
Q_1^{([a],\omega^2)} &= \Big\{a \text{ acting as } \mathcal{O}_0^{(a)} \longrightarrow \omega^2 \mathcal{O}_0^{(a)}\Big\} \,.
\end{aligned}
\tag{241}
$$

Let us now discuss the twisted sector operators associated to conjugacy class $[b]$. An irreducible multiplet involving such twisted sector operators is three-dimensional, with operators $\mathcal{O}_0^{(b)}, \mathcal{O}_0^{(ab)}, \mathcal{O}_0^{(a^2 b)}$ being respectively in $b$-twisted, $ab$-twisted and $a^2 b$-twisted

sectors. On top of that, we can specify the action of $S_3$ on the multiplet by specifying the action of $H_b$ on the operator $\mathcal{O}_0^{(b)}$. We have two irreducible 0-charges of this type

$$
\begin{aligned}
Q_1^{([b],+)} &= \left\{ b \text{ acting as } \mathcal{O}_0^{(b)} \longrightarrow \mathcal{O}_0^{(b)} \right\}, \\
Q_1^{([b],-)} &= \left\{ b \text{ acting as } \mathcal{O}_0^{(b)} \longrightarrow -\mathcal{O}_0^{(b)} \right\}.
\end{aligned}
\tag{242}
$$

4. $\mathbb{Z}_4$ **without Anomaly vs.** $\mathbb{Z}_2 \times \mathbb{Z}_2$ **with Mixed Anomaly**: Let us now discuss two cases of invertible symmetries that have the same set of 0-charges. On the one hand, we have a non-anomalous

$$
G^{(0)} = \mathbb{Z}_4 = \{\text{id}, a, a^2, a^3\},
\tag{243}
$$

0-form symmetry, for which there are 16 irreducible 0-charges

$$
Q_1^{(a^p, i^q)}, \qquad 0 \le p, q \le 3,
\tag{244}
$$

describing $a^p$-twisted sector operators on which $a$ acts as $i^q \in U(1)$. The fusion rules of the corresponding 16 topological line operators of the SymTFT $\mathfrak{Z}(\mathcal{S}_{\mathbb{Z}_4})$ follow group multiplication of $\mathbb{Z}_4 \times \mathbb{Z}_4$ and their spins are

$$
\theta\left(Q_1^{(a^p, i^q)}\right) = i^{pq}.
\tag{245}
$$

On the other hand, consider

$$
G^{(0)} = \mathbb{Z}_2 \times \mathbb{Z}_2 = \{\text{id}, a, b, ab\},
\tag{246}
$$

with a mixed 't Hooft anomaly between the two $\mathbb{Z}_2$ factors associated to anomaly theory

$$
A_1 \cup B_1,
\tag{247}
$$

where $A_1$ and $B_1$ are background fields for the two $\mathbb{Z}_2$ factors. An explicit group-cocycle $\omega$ realizing the anomaly is such that the only non-trivial evaluations are

$$
\omega(x, y, z) = -1, \qquad \forall \, x \in \{a, ab\}; \; y, z \in \{b, ab\}.
\tag{248}
$$

From this one computes the following 16 irreducible 0-charges

$$
\begin{aligned}
Q_1^{(\text{id}, s, s')} &= \{\text{Untwisted sector operator with } (a, b) \text{ acting as } (s, s')\}, \\
Q_1^{(b, s, s')} &= \{b\text{-twisted sector operator with } (a, b) \text{ acting as } (s, s')\}, \\
Q_1^{(a, s, is')} &= \{b\text{-twisted sector operator with } (a, b) \text{ acting as } (s, is')\}, \\
Q_1^{(ab, s, is')} &= \{ab\text{-twisted sector operator with } (a, b) \text{ acting as } (s, is')\},
\end{aligned}
\tag{249}
$$

where $s, s' \in \{+1, -1\}$. The fusions of the corresponding 16 topological line operators of the SymTFT $\mathfrak{Z}(\mathcal{S}_{\mathbb{Z}_2 \times \mathbb{Z}_2}^{\omega})$ again follow $\mathbb{Z}_4 \times \mathbb{Z}_4$ group multiplication and their spins are

$$
\theta\left(Q_1^{(\text{id}, s, s')}\right) = 1, \qquad \theta\left(Q_1^{(b, s, s')}\right) = s', \qquad \theta\left(Q_1^{(a, s, s')}\right) = s, \qquad \theta\left(Q_1^{(ab, s, s')}\right) = iss'.
\tag{250}
$$

The fusion rules and spins for line operators of $\mathfrak{Z}(\mathcal{S}_{\mathbb{Z}_4})$ and $\mathfrak{Z}(\mathcal{S}_{\mathbb{Z}_2 \times \mathbb{Z}_2}^{\omega})$ match, implying that the two Drinfeld centers match

$$
\mathcal{Z}(\mathcal{S}_{\mathbb{Z}_4}) = \mathcal{Z}(\mathcal{S}_{\mathbb{Z}_2 \times \mathbb{Z}_2}^{\omega}),
\tag{251}
$$

and since we are in 3d the two SymTFTs also match

$$
\mathfrak{Z}(\mathcal{S}_{\mathbb{Z}_4}) = \mathfrak{Z}(\mathcal{S}_{\mathbb{Z}_2 \times \mathbb{Z}_2}^{\omega}).
\tag{252}
$$

In fact $\mathfrak{B}_{\mathcal{S}_{\mathbb{Z}_4}}^{\text{sym}}$ and $\mathfrak{B}_{\mathcal{S}_{\mathbb{Z}_2 \times \mathbb{Z}_2}^{\omega}}^{\text{sym}}$ are two different topological boundary conditions of this SymTFT, which can be related by gauging non-anomalous $\mathbb{Z}_2$ subgroups of these symmetries.

#### 4.4.4 Non-invertible representation symmetries

The simplest examples of non-invertible symmetries in 2d are described by fusion categories of the form

$$\mathcal{S} = \text{Rep}(G^{(0)}), \tag{253}$$

for a finite non-abelian group $G^{(0)}$. These symmetries are obtained by gauging non-anomalous $G^{(0)}$ 0-form symmetry. Consequently the 0-charges for (253) symmetry is the same as that for non-anomalous $G^{(0)}$ 0-form symmetry, which we computed in section 4.4.1

$$\mathcal{Z}(\text{Rep}(G^{(0)})) = \mathcal{Z}(\mathcal{S}_{G^{(0)}}) = \bigoplus_{[g]} \text{Rep}(H_g), \tag{254}$$

where the sum is over conjugacy classes of $G^{(0)}$, and $\text{Rep}(H_g)$ is the category of linear representations of the centralizer $H_g \subseteq G^{(0)}$ of a representative $g$ in conjugacy class $[g]$.

However, the untwisted 0-charges for $\mathcal{S} = \mathcal{S}_{G^{(0)}}$ become twisted 0-charges for $\mathcal{S} = \text{Rep}(G^{(0)})$ and vice versa. For example, an irreducible 0-charge

$$\mathbf{Q}_1^{(\text{id},\mathbf{R})} \in \text{Rep}(H_{\text{id}}) = \text{Rep}(G^{(0)}) \subset \mathcal{Z}(\mathcal{S}_{G^{(0)}}), \tag{255}$$

for $\mathcal{S} = \mathcal{S}_{G^{(0)}}$ describes untwisted sector operators transforming in irreducible representation $\mathbf{R}$ of $G^{(0)}$. However, for $\mathcal{S} = \text{Rep}(G^{(0)})$, the same 0-charge $\mathbf{Q}_1^{(\text{id},\mathbf{R})}$ describes operators in **twisted sector** for the symmetry

$$S_1^{(\mathbf{R})} \in \mathcal{S} = \text{Rep}(G^{(0)}), \tag{256}$$

corresponding to irreducible representation $\mathbf{R}$.

#### 4.4.5 Intrinsic non-invertible symmetries: Ising symmetries

One can also equally study 0-charges of **intrinsic** non-invertible symmetries, like Ising symmetries described in example 4.5. Using this one can directly compute the Drinfeld centers of these fusion categories. These were also computed via a different method in [56]. Most of the computation is similar for the two Ising categories $\mathcal{S}_{\text{Ising}}^+$ and $\mathcal{S}_{\text{Ising}}^-$. While making statements that are valid for both of these categories, we will denote them by $\mathcal{S}_{\text{Ising}}^s$, where $s$ is a sign that can take any value $s \in \{+,-\}$.

First consider simple objects of $\mathcal{Z}(\mathcal{S}_{\text{Ising}}^s)$ that project to the identity object of $\mathcal{S}_{\text{Ising}}^s$. Let the half-braiding (193) of such an object with $S_1^{(P)}$ and $S_1^{(S)}$ be $m_P$ and $m_S$ respectively. Then, we find that $m_P$ and $m_S$ need to satisfy the following consistency conditions, coming from the fusion rules in $\mathcal{S}_{\text{Ising}}^s$

$$m_P^2 = 1, \qquad m_P m_S = m_S, \qquad m_S^2 = m_P = 1, \tag{257}$$

which imply that we have two solutions, both of which have $m_P = 1$, but are differentiated by $m_S = \pm 1$. We label the two simple objects of $\mathcal{Z}(\mathcal{S}_{\text{Ising}}^s)$ as

$$\mathbf{Q}_1^{(\text{id},+,\pm)} = \left\{ \text{Untwisted sector operator with } \left(S_1^{(P)}, S_1^{(S)}\right) \text{ acting as } (1,\pm 1) \right\}, \tag{258}$$

respectively. Note that $\mathbf{Q}_1^{(\text{id},+,+)}$ is the identity object of $\mathcal{Z}(\mathcal{S}_{\text{Ising}}^s)$.

Second, consider the simple objects of $\mathcal{Z}(\mathcal{S}_{\text{Ising}}^s)$ that project to the object $S_1^{(P)}$. Again, let $m_P$ and $m_S$ be half-braidings, but now they have to satisfy the following consistency conditions

$$m_P^2 = 1, \qquad -m_P m_S = m_S, \qquad m_S^2 = m_P = -1. \tag{259}$$

Again we have two solutions, both of which have $m_P = -1$, but are differentiated by $m_S = \pm i$. We label the two simple objects of $\mathcal{Z}(\mathcal{S}^s_{\text{Ising}})$ as

$$Q_1^{(P,-,\pm i)} = \left\{ S_1^{(P)}\text{-twisted sector operator with } \left( S_1^{(P)}, S_1^{(S)} \right) \text{ acting as } (-1, \pm i) \right\}, \qquad (260)$$

respectively.

Third, consider the simple objects of $\mathcal{Z}(\mathcal{S}^s_{\text{Ising}})$ that project to the object $S_1^{(S)}$. Let $m_P$ be the half-braiding with $P$. There are two half-braidings with $S$, as such a half-braiding is an endomorphism of $S_1^{(\text{id})} \oplus S_1^{(P)}$. Let $(m_{S,1}, m_{S,P})$ be the half-braiding endomorphism. Now, from $(S_1^{(P)})^2 = S_1^{(\text{id})}$, we obtain the consistency condition

$$m_P^2 = -1, \qquad (261)$$

where the minus sign comes from the associator $S_0^{\text{ass}}\left( S_1^{(P)}, S_1^{(S)}, S_1^{(P)} \right)$ in (195). From the fusion rule $S_1^{(P)} \otimes S_1^{(S)} = S_1^{(S)}$, we obtain the consistency condition

$$m_P m_{S,1} = -m_{S,P}. \qquad (262)$$

From the fusion rule $S_1^{(S)} \otimes S_1^{(S)} \cong S_1^{(\text{id})} \oplus S_1^{(P)}$, we obtain the consistency condition

$$\begin{pmatrix} 1 & 0 \\ 0 & m_P \end{pmatrix} = \frac{s}{2\sqrt{2}} \begin{pmatrix} 1 & 1 \\ 1 & -1 \end{pmatrix} \begin{pmatrix} m_{S,1} & 0 \\ 0 & m_{S,P} \end{pmatrix} \begin{pmatrix} 1 & 1 \\ 1 & -1 \end{pmatrix} \begin{pmatrix} m_{S,1} & 0 \\ 0 & m_{S,P} \end{pmatrix} \begin{pmatrix} 1 & 1 \\ 1 & -1 \end{pmatrix}, \qquad (263)$$

where on the left side we have half-braiding with $S_1^{(\text{id})} \oplus S_1^{(P)}$, while on the right hand side we have three associators and two half-braidings with $S_1^{(S)}$. The above matrix equation gives rise to one new condition which can simple be stated as

$$m_{S,1}^2 = \frac{s}{\sqrt{2}}(1 + m_P). \qquad (264)$$

Now we obtain a total of four solutions:

$$\begin{aligned}
m_P &= i, & m_{S,1} &= \sqrt{s}\, e^{\frac{2\pi i}{16}}, \\
m_P &= i, & m_{S,1} &= \sqrt{s}\, e^{\frac{18\pi i}{16}}, \\
m_P &= -i, & m_{S,1} &= \sqrt{s}\, e^{\frac{-2\pi i}{16}}, \\
m_P &= -i, & m_{S,1} &= \sqrt{s}\, e^{\frac{14\pi i}{16}},
\end{aligned} \qquad (265)$$

where we have chosen $\sqrt{s} = 1, i$ respectively for $s = 1, -1$. We label these four simple objects of $\mathcal{Z}(\mathcal{S}^+_{\text{Ising}})$ as

$$\begin{aligned}
Q_1^{(S,i,1/16)} = \Big\{ &S_1^{(S)}\text{-twisted sector operator with } S_1^{(P)} \text{ acting as } i \\
&\text{and } S_1^{(S)} \text{ acting as } \left( e^{\frac{2\pi i}{16}}, -ie^{\frac{2\pi i}{16}} \right) \in \text{End}\left( S_1^{(\text{id})} \oplus S_1^{(P)} \right) \Big\}, \\
Q_1^{(S,i,-7/16)} = \Big\{ &S_1^{(S)}\text{-twisted sector operator with } S_1^{(P)} \text{ acting as } i \\
&\text{and } S_1^{(S)} \text{ acting as } \left( e^{\frac{-14\pi i}{16}}, -ie^{\frac{-14\pi i}{16}} \right) \in \text{End}\left( S_1^{(\text{id})} \oplus S_1^{(P)} \right) \Big\}, \\
Q_1^{(S,-i,-1/16)} = \Big\{ &S_1^{(S)}\text{-twisted sector operator with } S_1^{(P)} \text{ acting as } -i \\
&\text{and } S_1^{(S)} \text{ acting as } \left( e^{\frac{-2\pi i}{16}}, ie^{\frac{-2\pi i}{16}} \right) \in \text{End}\left( S_1^{(\text{id})} \oplus S_1^{(P)} \right) \Big\}, \\
Q_1^{(S,-i,7/16)} = \Big\{ &S_1^{(S)}\text{-twisted sector operator with } S_1^{(P)} \text{ acting as } -i \\
&\text{and } S_1^{(S)} \text{ acting as } \left( e^{\frac{14\pi i}{16}}, ie^{\frac{14\pi i}{16}} \right) \in \text{End}\left( S_1^{(\text{id})} \oplus S_1^{(P)} \right) \Big\},
\end{aligned} \qquad (266)$$

$$
\begin{array}{c}
\Big| \quad \mathcal{O}_0^{(S,i,1/16)} \!\!-\!\!\!-\!\!\! S_1^{(S)} \quad = \quad \mathcal{O}_0^{(S,i,1/16)} \Big| \!-\!\! S_1^{(S)} \\[4pt]
S_1^{(S)} \qquad\qquad\qquad\qquad\qquad S_1^{(S)} \qquad\qquad S_1^{(S)}
\end{array}
$$

$$
= \quad e^{2\pi i/16} \; \mathcal{O}_0^{(S,i,1/16)} \underset{S_1^{(S)}}{\Big\lfloor}\!\!-\!\! S_1^{(S)} \;+\; (-i)e^{2\pi i/16} \; \underset{S_1^{(S)}}{\overset{S_1^{(S)}}{\mathcal{O}_0^{(S,i,1/16)}\Big| S_1^{(P)}}}\!\!-\!\! S_1^{(S)}
$$

Figure 35: Action of $S_1^{(S)}$ on an $S_1^{(S)}$-twisted sector operator $\mathcal{O}_0^{(S,i,1/16)}$ transforming in 0-charge $\mathbf{Q}_1^{(S,i,1/16)}$.

and the corresponding four simple objects of $\mathcal{Z}(\mathcal{S}_{\text{Ising}}^-)$ as

$$
\begin{aligned}
\mathbf{Q}_1^{(S,i,5/16)} &= \Big\{ S_1^{(S)}\text{-twisted sector operator with } S_1^{(P)} \text{ acting as } i \\
&\quad \text{and } S_1^{(S)} \text{ acting as } \Big( e^{\frac{10\pi i}{16}}, -i e^{\frac{10\pi i}{16}} \Big) \in \text{End}\Big( S_1^{(\text{id})} \oplus S_1^{(P)} \Big) \Big\}, \\
\mathbf{Q}_1^{(S,i,-3/16)} &= \Big\{ S_1^{(S)}\text{-twisted sector operator with } S_1^{(P)} \text{ acting as } i \\
&\quad \text{and } S_1^{(S)} \text{ acting as } \Big( e^{\frac{-6\pi i}{16}}, -i e^{\frac{-6\pi i}{16}} \Big) \in \text{End}\Big( S_1^{(\text{id})} \oplus S_1^{(P)} \Big) \Big\}, \\
\mathbf{Q}_1^{(S,-i,3/16)} &= \Big\{ S_1^{(S)}\text{-twisted sector operator with } S_1^{(P)} \text{ acting as } -i \\
&\quad \text{and } S_1^{(S)} \text{ acting as } \Big( e^{\frac{6\pi i}{16}}, i e^{\frac{6\pi i}{16}} \Big) \in \text{End}\Big( S_1^{(\text{id})} \oplus S_1^{(P)} \Big) \Big\}, \\
\mathbf{Q}_1^{(S,-i,-5/16)} &= \Big\{ S_1^{(S)}\text{-twisted sector operator with } S_1^{(P)} \text{ acting as } -i \\
&\quad \text{and } S_1^{(S)} \text{ acting as } \Big( e^{\frac{-10\pi i}{16}}, i e^{\frac{-10\pi i}{16}} \Big) \in \text{End}\Big( S_1^{(\text{id})} \oplus S_1^{(P)} \Big) \Big\}.
\end{aligned}
\tag{267}
$$

Notice that $S_1^{(S)}$ acts on such an $S_1^{(S)}$-twisted sector operator by two $U(1)$ valued phase factors. This statement may seem confusing and so we clarify its meaning in figure 35.

Finally, consider simple objects of $\mathcal{Z}(\mathcal{S}_{\text{Ising}}^s)$ that project to the object $S_1^{(\text{id})} \oplus S_1^{(P)}$. The half-braiding of such an object with $S_1^{(P)}$ is an endomorphism of $S_1^{(\text{id})} \oplus S_1^{(P)}$, which we denote as $(m_1, m_P)$. The half-braiding with $S_1^{(S)}$ is an endomorphism of $2S_1^{(S)}$, which we denote as the matrix

$$
\begin{pmatrix} m_{1,1} & m_{1,P} \\ m_{P,1} & m_{P,P} \end{pmatrix},
\tag{268}
$$

where $m_{1,1}$ describes sub half-braiding

$$
S_1^{(S)} \otimes S_1^{(\text{id})} \to S_1^{(\text{id})} \otimes S_1^{(S)},
\tag{269}
$$

$m_{1,P}$ describes sub half-braiding

$$
S_1^{(S)} \otimes S_1^{(\text{id})} \to S_1^{(P)} \otimes S_1^{(S)},
\tag{270}
$$

$m_{P,1}$ describes sub half-braiding

$$
S_1^{(S)} \otimes S_1^{(P)} \to S_1^{(\text{id})} \otimes S_1^{(S)},
\tag{271}
$$

and $m_{P,P}$ describes sub half-braiding

$$S_1^{(S)} \otimes S_1^{(P)} \to S_1^{(P)} \otimes S_1^{(S)}. \tag{272}$$

From $(S_1^{(P)})^2 = S_1^{(\text{id})}$, we obtain the consistency conditions

$$m_1^2 = m_P^2 = 1. \tag{273}$$

From $S_1^{(S)} \otimes S_1^{(P)} = S_1^{(S)}$, we obtain the consistency conditions

$$
\begin{aligned}
m_{1,1} m_1 &= m_{1,1}, \\
-m_{1,P} m_1 &= m_{1,P}, \\
m_{P,1} m_P &= m_{P,1}, \\
-m_{P,P} m_P &= m_{P,P}.
\end{aligned} \tag{274}
$$

From $S_1^{(S)} \otimes S_1^{(S)} = S_1^{(\text{id})} \oplus S_1^{(P)}$, we obtain the following consistency conditions

$$
\begin{aligned}
m_{1,1} m_{1,1} + m_{1,P} m_{P,1} &= 1, \\
m_{P,1} m_{1,P} - m_{P,P} m_{P,P} &= 1, \\
m_{1,1} m_{1,P} + m_{1,P} m_{P,P} &= 0, \\
m_{1,1} m_{P,1} - m_{P,P} m_{P,1} &= 0, \\
m_{1,1} m_{1,1} - m_{1,P} m_{P,1} &= m_1, \\
m_{P,1} m_{1,P} + m_{P,P} m_{P,P} &= m_P, \\
m_{1,1} m_{1,P} - m_{1,P} m_{P,P} &= 0, \\
m_{1,1} m_{P,1} + m_{P,P} m_{P,1} &= 0.
\end{aligned} \tag{275}
$$

A solution of these equations which is not a direct sum of previous solutions is

$$m_{1,1} = m_{P,P} = 0, \qquad m_{1,P} = m_{P,1} = 1, \qquad -m_1 = m_P = 1. \tag{276}$$

This simple object of $\mathcal{Z}(\mathcal{S}_{\text{Ising}}^s)$ corresponds to an irreducible 0-charge describing a multiplet composed of an untwisted sector local operator $\mathcal{O}_0^{(\text{id})}$ and an $S_1^{(P)}$-twisted sector local operator $\mathcal{O}_0^{(P)}$. We denote the object by

$$
\begin{aligned}
Q_1^{(\text{id}\oplus P, \mp, \text{ex})} = \Big\{ & S_1^{(P)} \text{ acting as } \left( \mathcal{O}_0^{(\text{id})}, \mathcal{O}_0^{(P)} \right) \longrightarrow \left( -\mathcal{O}_0^{(\text{id})}, \mathcal{O}_0^{(P)} \right) \\
& \text{and } S_1^{(S)} \text{ acting as the exchange } \mathcal{O}_0^{(\text{id})} \longleftrightarrow \mathcal{O}_0^{(P)} \Big\}.
\end{aligned} \tag{277}
$$

Note that including this simple object, we reproduce the total quantum dimension 16 for $\mathcal{Z}(\mathcal{S}_{\text{Ising}}^s)$. Thus, we have found all isomorphism classes of simple objects in $\mathcal{Z}(\mathcal{S})$.

---

**Example 4.6: 0-charges in the Ising CFT**

All of the above 0-charges for the Ising symmetry $\mathcal{S}_{\text{Ising}}^+$ discussed above are realized by conformal primary operators in the 2d Ising CFT, which admits the Ising symmetry.

See [41] for a table of these operators. The 0-charges for these operators are

$$\begin{aligned}
\boldsymbol{Q}_1^{(\text{id},+,-)} &= \left\{ \text{Energy operator } \epsilon \right\}, \\
\boldsymbol{Q}_1^{(P,-,i)} &= \left\{ \text{Operator } \psi \text{ with spin-1/2} \right\}, \\
\boldsymbol{Q}_1^{(P,-,-i)} &= \left\{ \text{Operator } \widetilde{\psi} \text{ with spin-1/2} \right\}, \\
\boldsymbol{Q}_1^{(S,i,1/16)} &= \left\{ \text{Operator } s \text{ with spin } e^{2\pi i/16} \right\}, \\
\boldsymbol{Q}_1^{(S,i,-7/16)} &= \left\{ \text{Operator } \Lambda \text{ with spin } e^{-14\pi i/16} \right\}, \\
\boldsymbol{Q}_1^{(S,-i,-1/16)} &= \left\{ \text{Operator } \widetilde{s} \text{ with spin } e^{-2\pi i/16} \right\}, \\
\boldsymbol{Q}_1^{(S,-i,7/16)} &= \left\{ \text{Operator } \widetilde{\Lambda} \text{ with spin } e^{14\pi i/16} \right\}, \\
\boldsymbol{Q}_1^{(\text{id}\oplus P,\mp,\text{ex})} &= \left\{ \text{Multiplet } (\sigma, \mu) \text{ of spin/order operator } \sigma \right. \\
&\qquad \left. \text{and disorder operator } \mu \right\}.
\end{aligned} \tag{278}$$

There is an interesting consequence for the operator spectrum that we can deduce from the study of 0-charges:

---

**Statement 4.4: Order operator exists iff disorder operator exists**

Consider an Ising-symmetric 2d QFT $\mathfrak{T}_\sigma$ admitting an untwisted sector local operator $\mathcal{O}_0^{(\sigma)}$ charged non-trivially under the $\mathbb{Z}_2$ subsymmetry of Ising symmetry generated by $S_1^{(P)}$. From an analysis of the 0-charges, we see that such an operator must transform in the 0-charge $\boldsymbol{Q}_1^{(\text{id}\oplus P,\mp,\text{ex})}$, and hence must arise in a multiplet $(\mathcal{O}_\sigma, \mathcal{O}_\mu)$ of two local operators, where the other operator $\mathcal{O}_\mu$ lies in the twisted sector for the $\mathbb{Z}_2$ subsymmetry, and is uncharged under the $\mathbb{Z}_2$ subsymmetry.

---

## 4.5 Symmetry boundaries and gauging

In this subsection, our aim is to discuss general methods using which one can determine different topological boundary conditions of a 3d SymTFT $\mathfrak{Z}(\mathcal{S})$. Such boundary conditions can be used as symmetry boundary conditions for the same SymTFT $\mathfrak{Z}(\mathcal{S})$. We discuss two different methods:

1. The first method involves beginning with the symmetry boundary $\mathfrak{B}_{\mathcal{S}}^{\text{sym}}$ and gauging the $\mathcal{S}$ symmetry of it. Different gaugings lead to different topological boundary conditions of $\mathfrak{Z}(\mathcal{S})$. As we will discuss later, mathematically, this involves determining indecomposable module categories of $\mathcal{S}$, or equivalently special types of algebras in the fusion category $\mathcal{S}$.

2. The second method uses the fact that a topological boundary condition $\mathfrak{B}^{\text{top}}$ of $\mathfrak{Z}(\mathcal{S})$ is specified by specifying which topological line operators of $\mathfrak{Z}(\mathcal{S})$ can end on $\mathfrak{B}^{\text{top}}$. Thus, this method relies only on the knowledge of the center $\mathcal{Z}(\mathcal{S})$ but not the fusion category $\mathcal{S}$ itself. As we will discuss later, mathematically, this involves determining Lagrangian algebras in the Drinfeld center $\mathcal{Z}(\mathcal{S})$.

Apriori, the second method might seem more general than the first, as the first method can only produce topological boundary conditions of $\mathfrak{Z}(\mathcal{S})$ that may be obtained by gauging the

$$\mathfrak{B}^{\text{sym}}_{\mathcal{S}} \qquad \mathfrak{B}^{\text{sym}}_{\mathcal{S}} \qquad \mathfrak{B}^{\text{sym}}_{\mathcal{S}'} \quad = \quad \mathfrak{B}^{\text{sym}}_{\mathcal{S}} \qquad \mathfrak{B}^{\text{sym}}_{\mathcal{S}'}$$

$$S_1 \in \mathcal{S} \qquad I_1 \in \mathcal{M}_{\mathcal{S},\mathcal{S}'} \qquad\qquad S_1 \otimes I_1 \in \mathcal{M}_{\mathcal{S},\mathcal{S}'}$$

Figure 36: A topological line $S_1$ on $\mathfrak{B}^{\text{sym}}_{\mathcal{S}}$ can act on an interface $I_1$ between $\mathfrak{B}^{\text{sym}}_{\mathcal{S}}$ and $\mathfrak{B}^{\text{sym}}_{\mathcal{S}'}$ to give rise to another interface between the two boundaries.

boundary $\mathfrak{B}^{\text{sym}}_{\mathcal{S}}$. However, for 3d SymTFTs, it is a general fact that all topological boundary conditions are related by gaugings. So the above two methods are equivalent.

Let us discuss the two methods in detail below.

### 4.5.1 Method 1: Topological boundary conditions by gauging

We are interested in gauging the $\mathcal{S}$ symmetry of $\mathfrak{B}^{\text{sym}}_{\mathcal{S}}$ to produce another topological boundary condition $\mathfrak{B}^{\text{sym}}_{\mathcal{S}'}$. This problem was discussed for a general two-dimensional system in the language used in this paper in [15]. Here we briefly review the key concepts involved and refer the reader to [15] for more details, where the reader can also find more mathematical references.

We can approach this problem from the point of view of the fact that if $\mathfrak{B}^{\text{sym}}_{\mathcal{S}}$ and $\mathfrak{B}^{\text{sym}}_{\mathcal{S}'}$ are related by gauging, then there exist topological interfaces between them. The collection of all such topological interfaces forms an indecomposable module category $\mathcal{M}_{\mathcal{S},\mathcal{S}'}$ of the fusion category $\mathcal{S}$. Let us quickly justify this statement. The fact that the topological interfaces form a 1-category $\mathcal{M}_{\mathcal{S},\mathcal{S}'}$ is straightforward: different interfaces describe objects of the category and interface changing local operators describe morphisms of the category. Moreover, a topological line operator of $\mathfrak{B}^{\text{sym}}_{\mathcal{S}}$ may act on such an interface to produce another interface. This action converts the category $\mathcal{M}_{\mathcal{S},\mathcal{S}'}$ of topological interfaces into a (left-)module category for the fusion category $\mathcal{S}$ describing topological line operators of $\mathfrak{B}^{\text{sym}}_{\mathcal{S}}$. See figure 36.

The topological line operators of $\mathfrak{B}^{\text{sym}}_{\mathcal{S}'}$, which are valued in the fusion category $\mathcal{S}'$, act in a similar fashion, thus converting $\mathcal{M}_{\mathcal{S},\mathcal{S}'}$ into a right module category for $\mathcal{S}'$. An equivalent way of stating this is that $\mathcal{S}'$ describes endofunctors of the category $\mathcal{M}_{\mathcal{S},\mathcal{S}'}$ that commute with the action of $\mathcal{S}$ on $\mathcal{M}_{\mathcal{S},\mathcal{S}'}$. This provides a way of deriving the symmetry $\mathcal{S}'$ from the knowledge of the indecomposable module category $\mathcal{M}_{\mathcal{S},\mathcal{S}'}$ of $\mathcal{S}$.

One can also directly perform the gauging. The idea is that the gauged boundary $\mathfrak{B}^{\text{sym}}_{\mathcal{S}'}$ is obtained from $\mathfrak{B}^{\text{sym}}_{\mathcal{S}}$ by inserting a fine-enough trivalent mesh of topological operators of $\mathfrak{B}^{\text{sym}}_{\mathcal{S}}$. We require that the size and shape of the mesh is irrelevant, i.e. correlation functions of the mesh do not depend on the size and shape. Then, we can make the mesh arbitrarily dense and think of $\mathfrak{B}^{\text{sym}}_{\mathcal{S}}$ with infinitely dense mesh as a new topological boundary condition $\mathfrak{B}^{\text{sym}}_{\mathcal{S}'}$. See figure 37.

Such a mesh can be produced by choosing an algebra

$$A = (A_1, A_0^{\text{p}}, A_0^{\text{cp}}), \tag{279}$$

in the fusion category $\mathcal{S}$, where $A_1$ is a (possibly non-simple) line defect in $\mathcal{S}$ and

$$\begin{aligned} A_0^{\text{p}}: \quad & A_1 \otimes A_1 \to A_1, \\ A_0^{\text{cp}}: \quad & A_1 \to A_1 \otimes A_1, \end{aligned} \tag{280}$$

are local operators providing trivalent junctions for the line $A_1$. These junction operators are required to satisfy the conditions shown in figure 38, which ensures that changing the size and shape of a mesh built using algebra $A$ does not impact correlation functions of the mesh.

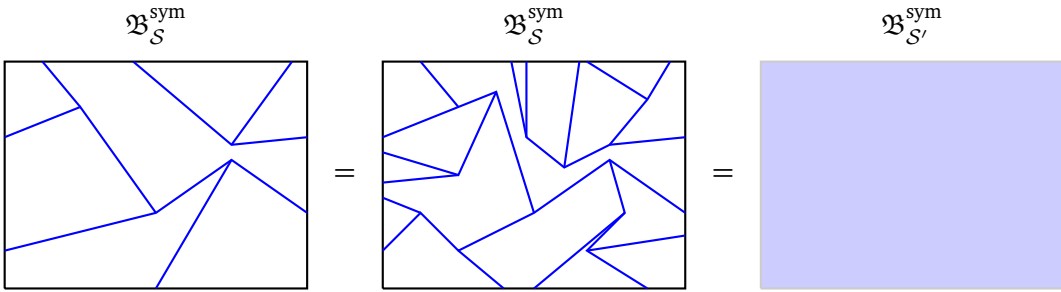

Figure 37: If the correlation function of a fine-enough mesh of topological lines of $\mathfrak{B}_{\mathcal{S}}^{\mathrm{sym}}$ is independent of how dense the mesh is, then $\mathfrak{B}_{\mathcal{S}}^{\mathrm{sym}}$ with mesh can be understood as a new boundary $\mathfrak{B}_{\mathcal{S}'}^{\mathrm{sym}}$. The intuitive idea is that $\mathfrak{B}_{\mathcal{S}'}^{\mathrm{sym}}$ is obtained by taking an infinitely dense mesh on $\mathfrak{B}_{\mathcal{S}}^{\mathrm{sym}}$, but by the above property that is the same as having a fine-enough but finitely dense mesh on $\mathfrak{B}_{\mathcal{S}}^{\mathrm{sym}}$.

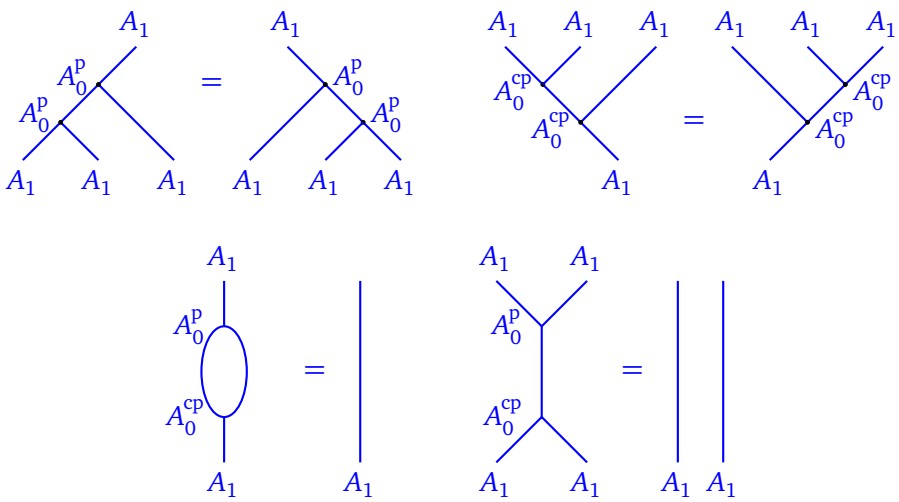

Figure 38: Consistency conditions on the algebra $A = (A_1, A_0^{\mathrm{p}}, A_0^{\mathrm{cp}})$.

Now, we have two different descriptions of gauging: one in terms of topological interfaces $\mathcal{M}_{\mathcal{S},\mathcal{S}'}$, and the other in terms of a mesh $A$ of topological operators of $\mathfrak{B}_{\mathcal{S}}^{\mathrm{sym}}$. These two descriptions are equivalent. A topological interface can be described in terms of the algebra $A$ as a (possibly non-simple) topological line $M_1 \in \mathcal{S}$ of $\mathfrak{B}_{\mathcal{S}}^{\mathrm{sym}}$ where the mesh can end. There are two types of ends, described by local operators $M_0^{\mathrm{p}}, M_0^{\mathrm{cp}} \in \mathcal{S}$

$$
\begin{aligned}
M_0^{\mathrm{p}}: &\quad M_1 \otimes A_1 \to M_1, \\
M_0^{\mathrm{cp}}: &\quad M_1 \to M_1 \otimes A_1.
\end{aligned}
\tag{281}
$$

See figure 39.

Different ways in which the mesh can end must be equivalent, which means that the local operators $M_0^{\mathrm{p}}, M_0^{\mathrm{cp}}$ satisfy conditions shown in the figure 39. Now, a topological interface in the category $\mathcal{M}_{\mathcal{S},\mathcal{S}'}$ is obtained from the information

$$
M = (M_1, M_0^{\mathrm{p}}, M_0^{\mathrm{cp}}),
\tag{282}
$$

by taking the limit of infinitely dense mesh in the presence of $M$ as shown in figure 40.

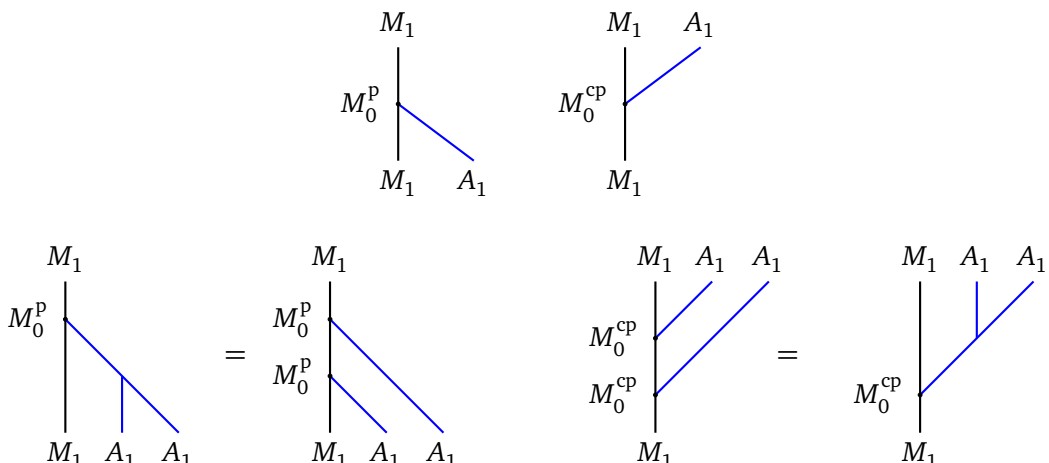

Figure 39: Two types of ends of the algebra object $A_1$ on the topological line defect $M_1$, and the consistency conditions satisfied by these ends.

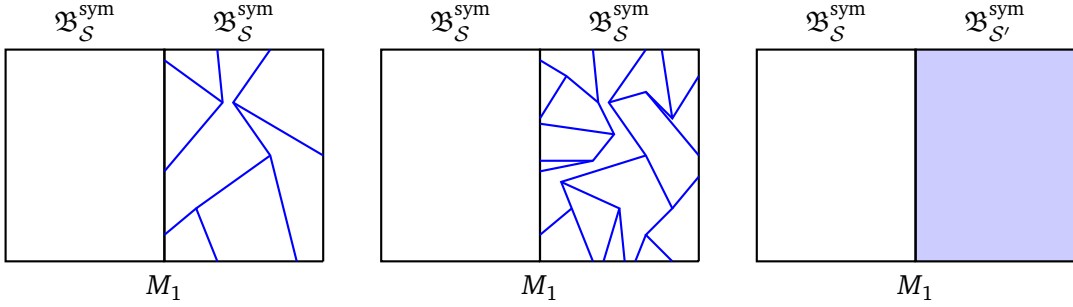

Figure 40: Placing a mesh of $A$ in the presence of a line operator $M_1$ participating in a right module $M$ of algebra $A$).

Mathematically, $M$ is called a right module of the algebra $A$ in the fusion category $\mathcal{S}$. Thus, the left module category $\mathcal{M}_{\mathcal{S},\mathcal{S}'}$ of the fusion category $\mathcal{S}$ is obtained as the category of right $A$-modules $\mathsf{Mod}_A(\mathcal{S})$ for an algebra $A$ in the fusion category $\mathcal{S}$

$$\mathcal{M}_{\mathcal{S},\mathcal{S}'} = \mathsf{Mod}_A(\mathcal{S}). \tag{283}$$

Similarly, the line operators on the topological boundary $\mathfrak{B}^{\mathrm{sym}}_{\mathcal{S}'}$ can be understood as line operators of $\mathfrak{B}^{\mathrm{sym}}_{\mathcal{S}}$ on which the $A$ mesh can end from both sides. Let $B_1 \in \mathcal{S}$ be such a (possibly non-simple) topological line operator. Then we need topological operators providing ends of $A$ on $B_1$

$$
\begin{aligned}
B_0^{\mathrm{r,p}} &: & B_1 \otimes A_1 &\to B_1 , \\
B_0^{\mathrm{r,cp}} &: & B_1 &\to B_1 \otimes A_1 , \\
B_0^{\mathrm{l,p}} &: & A_1 \otimes B_1 &\to B_1 , \\
B_0^{\mathrm{l,cp}} &: & B_1 &\to A_1 \otimes B_1 ,
\end{aligned}
\tag{284}
$$

which satisfy the module conditions for $A$ from both left and right sides of $B_1$, along with extra conditions shown in figure 41. Together these conditions allow taking the limit of infinitely dense mesh.

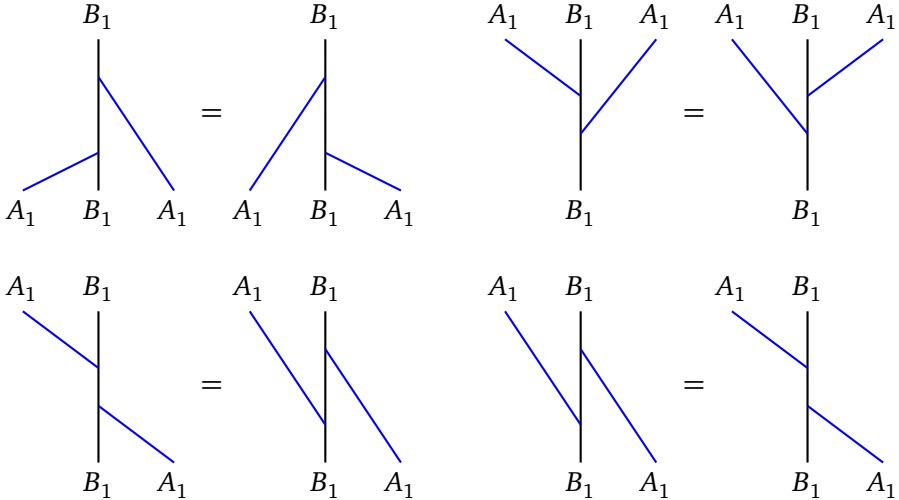

Figure 41: Extra conditions converting a left and right module into a bimodule for the algebra $A$. We omit the labels for the morphisms, but any $A_1$ line ending from the left (right) gives rise to $B_0^{l(r)}$, etc.

Mathematically, the collection

$$B = (B_1, B_0^{\mathrm{r,p}}, B_0^{\mathrm{r,cp}}, B_0^{\mathrm{l,p}}, B_0^{\mathrm{l,cp}}), \tag{285}$$

is known as a bimodule for the algebra $A$ and thus we can identify

$$\mathcal{S}' = \mathrm{Bimod}_A(\mathcal{S}), \tag{286}$$

where the right hand side denotes the category of $A$-bimodules in the fusion category $\mathcal{S}$.

Let us discuss some examples of gaugings via this method.

**Example 4.7: Gaugings of invertible symmetries**

Consider a $G^{(0)}$ 0-form symmetry with a 't Hooft anomaly $[\omega] \in H^3(G^{(0)}, \mathbb{C}^\times)$. Its possible gaugings are described by following two pieces of data:

1. A subgroup $H \subseteq G^{(0)}$ that we would like to gauge. This is only possible if the 't Hooft anomaly restricted to $H$ is trivial, i.e.

$$[\omega]|_H = 0 \in H^3(H, \mathbb{C}^\times). \tag{287}$$

2. A discrete torsion $[\beta]$ for the gauging, which is an equivalence class of 2-cochains on $H$ valued in $\mathbb{C}^\times$ whose representatives satisfy

$$\delta\beta = \omega|_H, \tag{288}$$

where $\omega$ is a fixed representative 3-cocycle of the class $[\omega]$. The equivalence relation between such 2-cochains is

$$\beta \sim \beta + \delta\gamma, \tag{289}$$

for 1-cochains $\gamma$ on $H$ valued in $\mathbb{C}^\times$.

We can obtain this classification of possible gaugings of the symmetry $\mathcal{S}^{\omega}_{G^{(0)}}$ by computing both module categories of $\mathcal{S}^{\omega}_{G^{(0)}}$, and by computing the possible algebras in $\mathcal{S}^{\omega}_{G^{(0)}}$. Below, let us roughly sketch how these two computations proceed.

First of all, consider an indecomposable module category $\mathcal{M}$ of $\mathcal{S}^{\omega}_{G^{(0)}}$. The group $G^{(0)}$ acts on simple line operators in $\mathcal{M}$. Pick a simple line $M_1 \in \mathcal{M}$. Let $H$ be the subgroup of $G^{(0)}$ which leaves $M_1$ invariant. In the language of [1], $H$ is a 0-form symmetry of line $M_1$ induced from the bulk $G^{(0)}$ 0-form symmetry. In order to derive the restriction (287), let us look more closely at the action of this induced 0-form symmetry. Let $M_0^{(h)}$ for $h \in H$ be local operators implementing the action of $H$ on the line $M_1$. These local operators in general have the fusion

$$M_0^{(h)} M_0^{(h')} = \beta(h, h') M_0^{(hh')} \,, \tag{290}$$

for a 2-cochain $\beta$ on $H$ valued in $\mathbb{C}^{\times}$. The associativity of the action of $H$ induced symmetry leads to equation (288), which implies (287). We can modify

$$M_0^{(h)} \to \gamma(h) M_0^{(h)} \,, \tag{291}$$

for a 1-cochain $\gamma$ on $H$ valued in $\mathbb{C}^{\times}$, which changes $\beta$ according to the equivalence relation (289), but does not modify (288). Thus, we have recovered the classification of possible gaugings of $\mathcal{S}^{\omega}_{G^{(0)}}$ from the point of view of its module categories. Before moving on, let us note that for the case without any 't Hooft anomaly $[\omega] = 0$, the equivalence class $[\beta] \in H^2(H, \mathbb{C}^{\times})$ describes a 't Hooft anomaly of the induced $H$ 0-form symmetry on the line $M_1$.

Now, let us discuss algebras $A$ of $\mathcal{S}^{\omega}_{G^{(0)}}$ for such gaugings. For gauging $H \subseteq G^{(0)}$, the line operator comprising the algebra is

$$A_1 = \bigoplus_{h \in H} S_1^{(h)} \,. \tag{292}$$

The algebra product $A_0^{\mathrm{p}}$ involves morphisms

$$S_1^{(h)} \otimes S_1^{(h')} \to S_1^{(hh')} \,, \tag{293}$$

which we take to be $\beta(h, h') \in \mathrm{End}(S_1^{(hh')}) = \mathbb{C}^{\times}$. The associativity of $A_0^{\mathrm{p}}$ shown in figure 38 implies the condition (288), which further implies (287).

### Example 4.8: Gaugings of representation symmetries

Consider a symmetry associated to a representation category of a finite group

$$\mathcal{S} = \mathrm{Rep}(G^{(0)}) \,. \tag{294}$$

This is obtained as dual symmetry after gauging a non-anomalous $G^{(0)}$ 0-form symmetry in 2d. As such, its possible gaugings should coincide with possible gaugings of $\mathcal{S}_{G^{(0)}}$ symmetry discussed in the previous example, where we saw that such gaugings are classified by following two pieces of data

1. A subgroup $H \subseteq G^{(0)}$.

2. An element $[\beta] \in H^2(H, \mathbb{C}^{\times})$.

Indeed, it is easy to construct module categories for $\text{Rep}(G^{(0)})$ corresponding to the above two pieces of data. We begin with the category

$$\text{Rep}^{[\beta]}(H), \tag{295}$$

of projective representations of $H$ in the class $[\beta]$. A linear representation $R_H$ of $H$ acts on such a projective representation $R'$ by tensor product

$$R' \to R_H \otimes R', \tag{296}$$

such that the resulting projective representation $R_H \otimes R'$ is also in the class $[\beta]$. We can thus act a representation $R$ of $G^{(0)}$ on a projective representation $R'$ of $H$ by decomposing $R$ into a representation $R_H$ of the subgroup $H$ and then acting $R_H$ on $R'$ as in (296).

In general, these gaugings of $\text{Rep}(G^{(0)})$ are generalized gaugings of non-invertible symmetries. For example, the gauging corresponding to module category Vec associated to $H = 1$ and $[\beta] = 0$ converts $\text{Rep}(G^{(0)})$ symmetry into dual non-anomalous $G^{(0)}$ 0-form symmetry. For non-abelian $G^{(0)}$, $\text{Rep}(G^{(0)})$ is a non-invertible symmetry and the above gauging is necessarily a generalized non-group-like gauging of non-invertible $\text{Rep}(G^{(0)})$ symmetry.

---

**Example 4.9: Gaugings of Ising symmetries**

For an Ising symmetry $\mathcal{S}^s_{\text{Ising}}$, there is a unique topological boundary condition $\mathfrak{B}^{\text{sym}}_{\mathcal{S}^s_{\text{Ising}}}$ of the SymTFT $\mathfrak{Z}(\mathcal{S}^s_{\text{Ising}})$ upto isomorphisms/dualities. One can gauge the $\mathbb{Z}_2$ subsymmetry of Ising symmetry generated by $S_1^{(P)}$, but the resulting topological boundary condition of $\mathfrak{Z}(\mathcal{S}^s_{\text{Ising}})$ is isomorphic to $\mathfrak{B}^{\text{sym}}_{\mathcal{S}^s_{\text{Ising}}}$ and carries a dual symmetry that is again the same Ising symmetry $\mathcal{S}^s_{\text{Ising}}$. See [57] for a classification of module categories of Ising fusion categories.

---

### 4.5.2 Method 2: Topological boundary conditions from the SymTFT

Another method for characterizing a topological boundary condition $\mathfrak{B}^{\text{top}}$ of a 3d TQFT $\mathfrak{Z}$ is in terms of the bulk topological line operators that can end on the boundary $\mathfrak{B}^{\text{top}}$. This method does not involve knowledge of another topological boundary condition of $\mathfrak{Z}$ to produce $\mathfrak{B}^{\text{top}}$.

Let $A_1^{(\alpha)}$ for various $\alpha$ be the simple line operators of $\mathfrak{Z}$ that can end on $\mathfrak{B}^{\text{top}}$, and let $V_\alpha$ be the vector space of topological local operators lying at the end of $A_1^{(\alpha)}$ along $\mathfrak{B}^{\text{top}}$. From this information, we can define a line operator $A_1$ of $\mathfrak{Z}$ as

$$A_1 = \bigoplus_\alpha V_\alpha A_1^{(\alpha)}. \tag{297}$$

As discussed in figure 42, the fusion of these local operators lying in $V_\alpha$ provide a product operation $A_0^{\text{p}}$ on the line $A_1$. As the fusion is associative, $(A_1, A_0^{\text{p}})$ defines an algebra $A$ in the category $\mathcal{Z}$ of topological line operators of the 3d TQFT $\mathfrak{Z}$. There is further structure on the algebra $A$. We can first braid the ends and then fuse them, and this should be the same as fusing them without braiding, meaning that the algebra $A$ needs to be commutative. See figure 43. Finally, there is a technical condition requiring that $A$ must be a maximal algebra satisfying the above conditions. Imposing all these requirements, we can recognize $A$ as a Lagrangian algebra in $\mathcal{Z}$, leading to the following statement

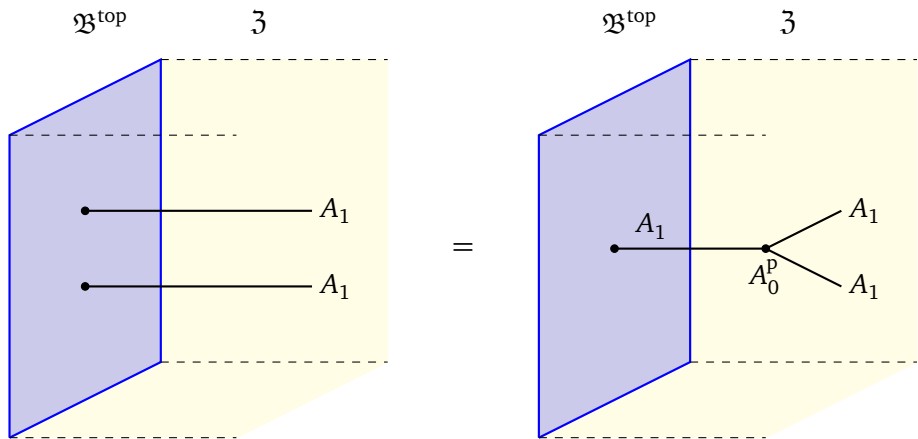

Figure 42: Fusion product on the $A$-lines.

Topological boundary conditions of a simple 3d TQFT $\mathfrak{Z}$, modulo stacking with invertible 2d TQFTs, correspond to **Lagrangian algebras** of the modular fusion category $\mathcal{Z}$ formed by topological line defects of $\mathfrak{Z}$.

An important condition that is extremely helpful in identifying the various possible Lagrangian algebras is that the line $A_1$ comprising a Lagrangian algebra $A$ has the property that

$$\dim(A_1) = \sqrt{\dim(\mathcal{Z})}, \tag{298}$$

where $\dim(A_1)$ is the quantum dimension of the line $A_1$ and $\dim(\mathcal{Z})$ is the quantum dimension of the category $\mathcal{Z}$ which is defined to be

$$\dim(\mathcal{Z}) = \sum_a \left(\dim Q_1^{(a)}\right)^2, \tag{299}$$

where $Q_1^{(a)}$ are different simple lines of $\mathcal{Z}$. Now consider any topological boundary $\mathfrak{B}_{\mathcal{S}}^{\text{sym}}$ of $\mathfrak{Z}$. Let $\mathcal{S}$ be the fusion category describing symmetry category of $\mathfrak{B}_{\mathcal{S}}^{\text{sym}}$, which allows us to express the bulk 3d TQFT as the SymTFT for $\mathcal{S}$ symmetry

$$\mathfrak{Z} = \mathfrak{Z}(\mathcal{S}), \tag{300}$$

and the category of bulk topological lines as the Drinfeld center for $\mathcal{S}$

$$\mathcal{Z} = \mathcal{Z}(\mathcal{S}). \tag{301}$$

Then, using the general fact that

$$\sqrt{\dim\big(\mathcal{Z}(\mathcal{S})\big)} = \dim(\mathcal{S}), \tag{302}$$

we can rewrite the condition (298) for a Lagrangian algebra $A$ associated to an arbitrary topological boundary condition $\mathfrak{B}^{\text{top}}$ of the SymTFT $\mathfrak{Z}(\mathcal{S})$ for $\mathcal{S}$ symmetry as

$$\dim(A_1) = \dim(\mathcal{S}). \tag{303}$$

Let us discuss some computations of topological boundary conditions via this method.

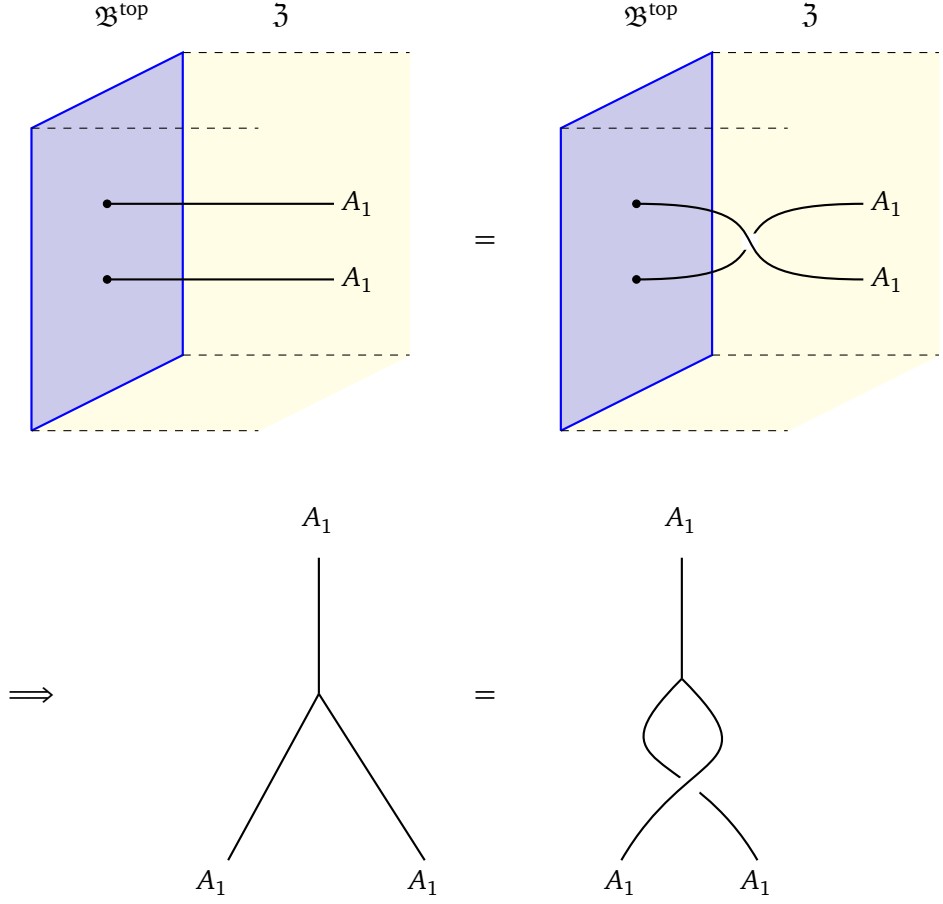

Figure 43: Braiding the ends of $A_1$ and then fusing them should be the same as simply fusing them. This implies commutativity of the algebra $A_1$.

---

**Example 4.10: Topological boundary conditions of 3d DW gauge theories**

Consider a 3d gauge theory for a finite gauge group $G^{(0)}$ without any DW twist. This theory is the SymTFT $\mathfrak{Z}(\mathcal{S}_{G^{(0)}})$ for a non-anomalous $G^{(0)}$ 0-form symmetry of 2d theories. As discussed earlier, the simple line operators of this 3d TQFT can be labeled as

$$Q_1^{([g],\mathbf{R})}, \tag{304}$$

where $[g]$ is a conjugacy class of $G^{(0)}$ and $\mathbf{R}$ is an irreducible representation of the centralizer $H_g \subseteq G^{(0)}$ of a representative $g \in G^{(0)}$ of the conjugacy class $[g]$. The quantum dimension of such a line operator is

$$\dim\left(Q_1^{([g],\mathbf{R})}\right) = \text{size}([g]) \times \dim(\mathbf{R}), \tag{305}$$

where $\text{size}([g])$ is the number of elements of $G^{(0)}$ in the conjugacy class $[g]$ and $\dim(\mathbf{R})$ is the dimension of the vector space underlying the representation $\mathbf{R}$.

The symmetry boundary condition $\mathfrak{B}^{\text{sym}}_{\mathcal{S}_{G^{(0)}}}$ of $\mathfrak{Z}(\mathcal{S}_{G^{(0)}})$ realizing non-anomalous $G^{(0)}$ 0-form symmetry is obtained by choosing a Lagrangian algebra $A^D$ whose underlying line

operator is

$$A_1^D = \bigoplus_R Q_1^{([\mathrm{id}],\boldsymbol{R})}, \tag{306}$$

where $[\mathrm{id}]$ is the conjugacy class containing only the identity element id of $G^{(0)}$, whose centralizer is $H_{\mathrm{id}} = G^{(0)}$, and the sum is over all irreducible representations $\boldsymbol{R}$ of $G^{(0)}$. It is straightforward to see that the condition (303) is satisfied, as we have

$$\dim(A_1^D) = \dim(\mathcal{S}_{G^{(0)}}) = \left| G^{(0)} \right|, \tag{307}$$

where $\left| G^{(0)} \right|$ is the order of the group $G^{(0)}$.

On the other hand, the symmetry boundary condition $\mathrm{B}_{\mathsf{Rep}(G^{(0)})}^{\mathrm{sym}}$ of $\mathfrak{Z}(\mathcal{S}_{G^{(0)}})$ realizing

$$\mathcal{S} = \mathsf{Rep}(G^{(0)}), \tag{308}$$

symmetry is obtained by choosing a Lagrangian algebra $A^N$ whose underlying line operator is

$$A_1^N = \bigoplus_{[g]} Q_1^{([g],\mathrm{id})}, \tag{309}$$

where the sum is over all conjugacy classes of $G^{(0)}$ and id is the identity representation of each centralizer $H_g$. Again the condition (303) is satisfied, as we have

$$\dim(A_1^N) = \dim(\mathsf{Rep}(G^{(0)})) = \left| G^{(0)} \right|. \tag{310}$$

Other topological boundary conditions of $\mathfrak{Z}(\mathcal{S}_{G^{(0)}})$ involve both non-trivial conjugacy classes and non-trivial representations of centralizers. See [58] for the full classification, which also discusses the case of non-trivial DW twist $[\omega] \in H^3(G^{(0)}, \mathbb{C}^\times)$.

---

**Example 4.11: Topological boundary conditions and defect group**

It is illustrative though to study all topological boundary conditions for abelian $G^{(0)}$ without any DW twist. In this case it is well-known that the topological line operators of the bulk DW theory $\mathfrak{Z}(\mathcal{S}_{G^{(0)}})$ are parametrized by the **defect group** (the terminology was introduced in [59]) which is the product of the group and its Pontryagin dual

$$\mathcal{L} = G^{(0)} \times \widehat{G}^{(0)}. \tag{311}$$

The lines can be labeled as

$$Q_1^{(g,\widehat{g})}, \qquad g \in G^{(0)}, \qquad \widehat{g} \in \widehat{G}^{(0)}. \tag{312}$$

There is a **pairing** on $\mathcal{L}$

$$\eta : \quad \mathcal{L} \times \mathcal{L} \to U(1), \tag{313}$$

which is given as

$$\eta\big((g,\widehat{g}),(g',\widehat{g}')\big) = \frac{\widehat{g}(g')}{\widehat{g}'(g)}, \tag{314}$$

where

$$\widehat{g}(g') \in U(1), \tag{315}$$

is the action of $g'$ in one-dimensional representation labeled by $\widehat{g}$ of $G^{(0)}$.

The various topological boundary conditions are obtained by choosing a **Lagrangian subgroup** $\Lambda$ (also called **polarization**) of the defect group $\mathcal{L}$, which is a maximal subgroup of $\mathcal{L}$ on which the pairing $\eta$ trivializes

$$\eta|_{\Lambda} = 1 \,. \tag{316}$$

We can describe $\Lambda$ also as

$$\Lambda = (H, \theta) \,, \tag{317}$$

where

$$H \subseteq G^{(0)} \,, \tag{318}$$

is a subgroup and

$$\theta : \quad H \to \widehat{H} \,, \tag{319}$$

is a homomorphism. The subgroup $H$ is obtained as

$$H = \pi_{G^0}(\Lambda) \,, \tag{320}$$

where

$$\pi_{G^0} : \mathcal{L} = G^{(0)} \times \widehat{G}^{(0)} \to G^{(0)} \,, \tag{321}$$

is the projection from $\mathcal{L}$ to its $G^{(0)}$ factor. For the homomorphism $\theta$, pick an element

$$(h, \widehat{g}) \in \Lambda \,, \qquad h \in H \,, \qquad \widehat{g} \in \widehat{G}^{(0)} \,, \tag{322}$$

and specify

$$\theta(h) = \widehat{i}_H(\widehat{g}) \,, \tag{323}$$

where

$$\widehat{i}_H : \quad \widehat{G}^{(0)} \to \widehat{H} \,, \tag{324}$$

is the Pontryagin dual surjective map to the injective map

$$i_H : \quad H \to G^{(0)} \,, \tag{325}$$

corresponding to the choice of subgroup $H$ of $G^{(0)}$. The homomorphism $\theta$ is well-defined and does not depend on the choice of the element $(h, \widehat{g}) \in \Lambda$. Moreover, the fact that $\eta$ trivializes on $\theta$ translates to a condition

$$\theta(h)(h')\theta(h')(h) = 1 \in U(1) \,, \tag{326}$$

which means that $\theta$ describes an element

$$[\beta] \in H^2(H, U(1)) \,. \tag{327}$$

The two are related as follows [9]:

$$\theta(h)(h') = \beta(h, h')\beta(h^{-1}, h'^{-1}) \,. \tag{328}$$

Let us label the topological boundary condition of $\mathfrak{Z}(\mathcal{S}_{G^{(0)}})$ obtained from the polarization $\Lambda$ as $\mathfrak{B}^{\Lambda}$. The boundary condition

$$\mathfrak{B}^{\mathrm{sym}}_{\mathcal{S}_{G^{(0)}}} = \mathfrak{B}^{\Lambda_D} \,, \tag{329}$$

recovering the non-anomalous $G^{(0)}$ 0-form symmetry is obtained from polarization

$$\Lambda_D = \{Q_1^{(\mathrm{id},\widehat{g})}|\widehat{g} \in \widehat{G}^{(0)}\}\,. \tag{330}$$

An arbitrary $\mathfrak{B}^\Lambda$ is obtained by gauging of $G^{(0)}$ symmetry of $\mathfrak{B}^{\Lambda_D}$. The subgroup $H$ being gauged is precisely the one described in (320) and the discrete torsion $[\beta]$ involved in the gauging is described in (327).

---

**Example 4.12: Topological boundary conditions of 3d doubled Ising TQFT**

Consider a doubled Ising TQFT in 3d, which is the SymTFT for an Ising symmetry in 2d. Its topological line operators were discussed in section 4.4.5. We claimed earlier that there is a single topological boundary condition, which is the symmetry boundary $\mathfrak{B}^{\mathrm{sym}}_{\mathcal{S}^s_{\mathrm{Ising}}}$.

The line operator underlying the corresponding Lagrangian algebra $A^{\mathrm{Ising}}$ can be deduced from the Drinfeld center description of the bulk topological lines. From the discussion in section 4.4.5, we know that the only bulk lines that can end on $\mathfrak{B}^{\mathrm{sym}}_{\mathcal{S}^s_{\mathrm{Ising}}}$ are $Q_1^{(\mathrm{id},+,+)}$, $Q_1^{(\mathrm{id},+,-)}$ and $Q_1^{(\mathrm{id}\oplus P,\mp,\mathrm{ex})}$, and there is only a one-dimensional space of topological local operators that can lie at the end of each such bulk topological line along the boundary $\mathfrak{B}^{\mathrm{sym}}_{\mathcal{S}^s_{\mathrm{Ising}}}$. This means that the topological line operator underlying $A^{\mathrm{Ising}}$ must be

$$A_1^{\mathrm{Ising}} = Q_1^{(\mathrm{id},+,+)} \oplus Q_1^{(\mathrm{id},+,-)} \oplus Q_1^{(\mathrm{id}\oplus P,\mp,\mathrm{ex})}\,. \tag{331}$$

We can easily check that the condition (303) on the quantum dimensions is satisfied.

---

# 5 Generalized charges and SymTFT in $d=3$

In 3d, a general (possibly non-invertible) symmetry $\mathcal{S}$ is described by a fusion 2-category. The study of such 2-categorical (and higher-categorical) symmetries in non-topological quantum field theories was initiated in [8], with many subsequent followups exploring and developing the structure further [9, 10, 21, 60–64].

The associated SymTFT $\mathfrak{Z}(\mathcal{S})$ has been studied for arbitrary fusion 2-categories by [65, 66] which we could refer to as the **Douglas-Reutter-Walker TFT**, while some special cases have been studied long ago [67, 68] going under the name of the **Crane-Yetter** construction.[13]

We will discuss 1-charges and 0-charges of such symmetries in this section. The Drinfeld centers of fusion 2-categories have been studied by [36, 70–73].

## 5.1 Fusion 2-Categorical Symmetries

Consider a symmetry $\mathcal{S}$ in 3d given by a fusion 2-category – for a mathematical background see [20, 65]. The topological operators have now an extra layer compared to the fusion category case:

- Objects, which are topological surface operators: $S_2^{(\alpha)}$.

---

[13]In the condensed matter literature, the corresponding constructions of lattice models flowing to these 4d TFTs was provided by Walker-Wang [69] for the Crane-Yetter case.

- 1-Morphisms, which are topological line operators. These are $S_1^{(\alpha,\beta)}$ lines in $\text{Hom}(S_2^{(\alpha)}, S_2^{(\beta)})$. These form a 1-category.

- 2-Morphisms, which are topological local operators, which form a vector space, i.e. $S_0^{(a,b)} \in \text{Hom}(S_1^{(a)}, S_1^{(b)})$.

In each dimension of topological defects we have a fusion product

$$S_p^{(\alpha)} \otimes S_p^{(\beta)} = \bigoplus_\gamma N_\gamma^{\alpha,\beta} S_p^{(\gamma)}. \tag{332}$$

And given two lines

$$\begin{aligned} S_1^{(12)}: \quad & S_2^{(1)} \to S_2^{(2)}, \\ S_1^{(23)}: \quad & S_2^{(2)} \to S_2^{(3)}, \end{aligned} \tag{333}$$

we have a composition operation, which can be understood as fusion along the surface $S_2^{(2)}$, to produce a new line

$$S_1^{(12)} \otimes_{S_2^{(2)}} S_1^{(23)}: \quad S_2^{(1)} \to S_2^{(3)}. \tag{334}$$

There are various coherence relations for whose details we refer the reader to [65].

A recent classification result [72] states that any fusion 2-category is gauge/Morita equivalent to a fusion 2-category of the form

$$2-\text{Vec}_{G^{(0)}}^\omega \boxtimes \text{Mod}(\mathcal{B}), \tag{335}$$

where $2-\text{Vec}_{G^{(0)}}^\omega$ describes a $G^{(0)}$ 0-form symmetry with anomaly $\omega$, and $\text{Mod}(\mathcal{B})$ is the fusion 2-category formed by module categories of a non-degenerate braided fusion category $\mathcal{B}$. As a corollary, since Drinfeld center is Morita equivalent, we learn that all Drinfeld centers of fusion 2-categories are of the form

$$\mathcal{Z}(2-\text{Vec}_{G^{(0)}}^\omega) \boxtimes \mathcal{Z}(\text{Mod}(\mathcal{B})), \tag{336}$$

but $\mathcal{Z}(\text{Mod}(\mathcal{B}))$ is trivial as $\mathcal{B}$ is non-degenerate. In conclusion, the Drinfeld center of any fusion 2-category is of the form

$$\mathcal{Z}(2-\text{Vec}_{G^{(0)}}^\omega), \tag{337}$$

which we described already in section 3.3.3.

A few examples of such symmetries are discussed below.

---

**Example 5.1: 0-form symmetries with anomaly in 3d: $2\text{-Vec}_{G^{(0)}}^\omega$**

Consider a finite 0-form symmetry group $G^{(0)}$ with a 't Hooft anomaly

$$\omega \in H^4(G^{(0)}, \mathbb{C}^\times), \tag{338}$$

in 3d. The associated fusion 2-category is

$$\mathcal{S}_{G^{(0)}}^\omega = 2\text{-Vec}_{G^{(0)}}^\omega. \tag{339}$$

The distinct simple topological surfaces are

$$S_2^{(g)}, \qquad g \in G^{(0)}, \tag{340}$$

with fusion

$$S_2^{(g)} \otimes S_2^{(h)} \cong S_2^{(gh)}, \qquad g, h \in G^{(0)}. \tag{341}$$

The only lines are along each surface $S_2^{(g)}$, i.e. 1-endomorphisms of $S_2^{(g)}$. There is a single simple line

$$S_1^{(g)}, \tag{342}$$

upto isomorphism on $S_2^{(g)}$, which is the identity line on $S_2^{(g)}$.

The anomaly $\omega$ describes coherence relations generalizing the associator involved in the 2d case discussed in example 4.1.

For these fusion categories, we argued that the center is given by (144), which was discussed also in [36].

---

**Example 5.2: 2-representation and 1-form symmetries: 2-$\mathsf{Rep}_{G^{(0)}}$**

Gauging non-anomalous $G^{(0)}$ 0-form symmetry in 3d, leads to

$$\mathcal{S} = 2\text{-}\mathsf{Rep}(G^{(0)}), \tag{343}$$

symmetry. If $G^{(0)}$ is non-abelian, then this is the fusion 2-category associated to a non-anomalous

$$G^{(1)} = \widehat{G}^{(0)}, \tag{344}$$

1-form symmetry, i.e.

$$2\text{-}\mathsf{Rep}(G^{(0)}) = \mathcal{S}_{G^{(1)}} = 2\text{-}\mathsf{Vec}_{G^{(1)}}. \tag{345}$$

If $G^{(0)}$ is non-abelian, we can understand it as the fusion 2-category associated to a non-invertible $\mathsf{Rep}(G^{(0)})$ 1-form symmetry.

The symmetry 2-$\mathsf{Rep}(G^{(0)})$ always involves non-invertible simple surfaces. The full list of distinct simple surfaces is provided by irreducible 2-representations of $G^{(0)}$, which are characterized by following two pieces of data:

1. A subgroup $H \subseteq G^{(0)}$.

2. An element $\beta \in H^2(H, \mathbb{C}^\times)$.

See [9, 10] for more details.

These surface operators can be understood as being obtained by gauging $G^{(1)}$ or $\mathsf{Rep}(G^{(0)})$ 1-form symmetry on an identity surface in 3d spacetime, and are thus referred to as condensation defects.

Unlike 2-$\mathsf{Vec}_{G^{(0)}}^\omega$, the symmetry 2-$\mathsf{Rep}_{G^{(0)}}$ involves line operators between distinct simple surfaces. These lines correspond to intertwiners between the two irreducible 2-representations corresponding to the two simple surfaces.

---

**Example 5.3: General non-invertible 1-form symmetries**

In the previous example, we saw the appearance of a $\mathsf{Rep}(G^{(0)})$ 1-form symmetry, which is non-invertible for non-abelian $G^{(0)}$. A general non-invertible symmetry in 3d is associated to a braided fusion category $\mathcal{B}$ describing the line operators generating the symmetry. The associated fusion 2-category describing such a symmetry is

$$\mathcal{S}(\mathcal{B}) = \mathsf{Mod}(\mathcal{B}), \tag{346}$$

in which surfaces correspond to module categories of $\mathcal{B}$. All of these surfaces can be obtained by condensing the lines in $\mathcal{B}$. Indeed, module categories describe the possible gaugings of lines along a two-dimensional surface. See section 4.5.1 for more details.

---

**Example 5.4: 2-groups and their 2-representations**

Above we have discussed the fusion 2-categories associated to 0-form and 1-form symmetry groups in 3d. Generalizing these, if we have a $\mathbb{G}^{(2)}$ 2-group symmetry in 3d with 't Hooft anomaly

$$\omega \in H^4(B\mathbb{G}^{(2)}, \mathbb{C}^\times), \tag{347}$$

then the corresponding fusion 2-category is

$$\mathcal{S}^\omega_{\mathbb{G}^{(2)}} = 2\text{-Vec}^\omega_{\mathbb{G}^{(2)}}. \tag{348}$$

A general simple surface in this 2-category is obtained by gauging appropriate subgroups of 1-form symmetries on the surfaces generating 0-form symmetries.

Similarly, gauging a non-anomalous $\mathbb{G}^{(2)}$ 2-group symmetry, we obtain a symmetry

$$\mathcal{S} = 2\text{-Rep}(\mathbb{G}^{(2)}), \tag{349}$$

in which surfaces are described by 2-representations of the 2-group $\mathbb{G}^{(2)}$. The key difference from the case of $2\text{-Rep}(G^{(0)})$ is that now not every surface is a condensation surface obtained by gauging line operators! Examples of such type were discussed at length in [9].

---

## 5.2 Generalized charges for $\mathbb{Z}_2$ 0-form symmetry

Consider non-anomalous $\mathbb{Z}_2$ 0-form symmetry in 3d, for which the associated fusion 2-category is

$$\mathcal{S}_{\mathbb{Z}_2^{(0)}} = 2\text{-Vec}_{\mathbb{Z}_2}, \tag{350}$$

whose simple objects up to isomorphisms are

$$\text{Obj}(\mathcal{S}_{\mathbb{Z}_2^{(0)}}) = \left\{ S_2^{(\text{id})}, S_2^{(V)} \right\}, \tag{351}$$

where $S_2^{(V)}$ is the topological surface operator generating the $\mathbb{Z}_2$ 0-form symmetry of the 3d symmetry boundary $\mathfrak{B}^{\text{sym}}_{2\text{-Vec}_{\mathbb{Z}_2}}$ of the associated 4d SymTFT $\mathsf{Z}(2\text{-Vec}_{\mathbb{Z}_2})$.

The generalized charges for this symmetry are described by the Drinfeld center

$$\mathcal{Z}(2\text{-Vec}_{\mathbb{Z}_2}) = 2\text{-Vec}_{\mathbb{Z}_2} \boxtimes 2\text{-Rep}(\mathbb{Z}_2). \tag{352}$$

Its simple objects upto isomorphisms are

$$\text{Obj}\left(\mathcal{Z}(2\text{-Vec}_{\mathbb{Z}_2})\right) = \left\{ \mathbf{Q}_2^{(\text{id})}, \mathbf{Q}_2^{(\mathbb{Z}_2)}, \mathbf{Q}_2^{(V)}, \mathbf{Q}_2^{(V\mathbb{Z}_2)} \right\}, \tag{353}$$

where $\mathbf{Q}_2^{(\mathbb{Z}_2)}$ generates the $2\text{-Rep}(\mathbb{Z}_2)$ factor and $\mathbf{Q}_2^{(V)}$ generates the $2\text{-Vec}_{\mathbb{Z}_2}$ factor. These objects describe different 1-charges, whose physics is discussed below.

**Projection onto the symmetry boundary $B^{\text{sym}}_{2\text{-Vec}_{\mathbb{Z}_2}}$.** The above simple topological lines of $Z(2\text{-Vec}_{\mathbb{Z}_2})$ project onto the symmetry boundary $B^{\text{sym}}_{2\text{-Vec}_{\mathbb{Z}_2}}$ carrying the $\mathbb{Z}_2$ 0-form symmetry as

$$
\begin{aligned}
Q_2^{(\text{id})} &\to S_2^{(Q_2^{(\text{id})})} \cong S_2^{(\text{id})}, & Q_2^{(\mathbb{Z}_2)} &\to S_2^{(Q_2^{(\mathbb{Z}_2)})} \cong 2S_2^{(\text{id})}, \\
Q_2^{(V)} &\to S_2^{(Q_2^{(V)})} \cong S_2^{(V)}, & Q_2^{(V\mathbb{Z}_2)} &\to S_2^{(Q_2^{(V\mathbb{Z}_2)})} \cong 2S_2^{(V)}.
\end{aligned}
\tag{354}
$$

Note that both topological surface operators $S_2^{(\text{id})}$ and $S_2^{(V)}$ on the symmetry boundary $B^{\text{sym}}_{2\text{-Vec}_{\mathbb{Z}_2}}$ are straightforwardly recovered via such a projection of bulk surfaces. This fact does not hold quite so straightforwardly for a non-abelian $G^{(0)}$ 0-form symmetry, simply because surfaces in a 4d SymTFT must have commutative fusion rules, while the surfaces on the boundary generating $G^{(0)}$ 0-form symmetry have non-commutative fusion rules.

### 5.2.1 1-charges

**1-charge $Q_2^{(\text{id})}$.** This is the trivial 1-charge and corresponds to the identity topological surface operator of the 4d SymTFT $Z(2\text{-Vec}_{\mathbb{Z}_2})$. Its ends along the 3d symmetry boundary $B^{\text{sym}}_{2\text{-Vec}_{\mathbb{Z}_2}}$ correspond to topological line defects of $B^{\text{sym}}_{2\text{-Vec}_{\mathbb{Z}_2}}$. In this case, the only topological line defect is the identity line defect, so the only possible end is

$$
\mathcal{E}_1^{(\text{id})} = S_1^{(\text{id})} \in \mathcal{S}_{\mathbb{Z}_2^{(0)}}.
\tag{355}
$$

Thus a multiplet $\mathcal{M}_1^{(\text{id})}$ of line operators of a $\mathbb{Z}_2$ 0-form symmetric 3d theory $\mathfrak{T}_\sigma$ transforming in the 1-charge $Q_2^{(\text{id})}$ comprises of a single simple genuine line operator

$$
\mathcal{O}_1^{(\text{id})},
\tag{356}
$$

on which the $\mathbb{Z}_2$ 0-form symmetry acts trivially.

**1-charge $Q_2^{(\mathbb{Z}_2)}$.** The defect $Q_2^{(\mathbb{Z}_2)}$ ends on the boundary $B^{\text{sym}}_{2\text{-Vec}_{\mathbb{Z}_2}}$, with the ends given in terms of two choices of lines

$$
\mathcal{E}_1^{(\mathbb{Z}_2)(\pm)}.
\tag{357}
$$

This follows from the fact that the projection of $Q_2^{(\mathbb{Z}_2)}$ is $2S_2^{(\text{id})}$ as discussed in (354), and $2S_2^{(\text{id})}$ has two simple 1-morphisms to the trivial surface $S_2^{(\text{id})}$.

Correspondingly, a multiplet $\mathcal{M}_1^{(\mathbb{Z}_2)}$ of line operators of a $\mathbb{Z}_2$ 0-form symmetric 3d theory $\mathfrak{T}_\sigma$ transforming in the 1-charge $Q_2^{(\mathbb{Z}_2)}$ comprises of two simple *genuine* line operators

$$
\mathcal{O}_1^{(\mathbb{Z}_2)(\pm)}.
\tag{358}
$$

The action of $\mathbb{Z}_2$ 0-form exchanges the two simple genuine line operators

$$
\mathbb{Z}_2: \quad \mathcal{O}_1^{(\mathbb{Z}_2)(+)} \longleftrightarrow \mathcal{O}_1^{(\mathbb{Z}_2)(-)}.
\tag{359}
$$

There are no ends of the surface operator $Q_2^{(\mathbb{Z}_2)}$ along $B^{\text{sym}}_{2\text{-Vec}_{\mathbb{Z}_2}}$ that are attached to non-trivial topological surface defects of $B^{\text{sym}}_{2\text{-Vec}_{\mathbb{Z}_2}}$. This is because its projection $2S_2^{(\text{id})}$ admits no 1-morphisms to $S_2^{(V)}$. Thus the multiplet $\mathcal{M}_1^{(\mathbb{Z}_2)}$ does not contain any twisted sector line operators.

**1-charge $Q_2^{(V)}$.** There is only one end

$$\mathcal{E}_1^{(V)}, \tag{360}$$

of the surface operator $Q_2^{(V)}$ along the symmetry boundary $B_{2\text{-Vec}_{\mathbb{Z}_2}}^{\text{sym}}$ for $\mathbb{Z}_2$ 0-form symmetry. Additionally, the topological line operator $\mathcal{E}_1^{(V)}$ is attached to the non-trivial topological surface operator $S_2^{(V)}$ of $B_{2\text{-Vec}_{\mathbb{Z}_2}}^{\text{sym}}$.

Thus a multiplet $\mathcal{M}_1^{(V)}$ of line operators of a $\mathbb{Z}_2$ 0-form symmetric 3d theory $\mathfrak{T}_\sigma$ transforming in the 1-charge $Q_2^{(V)}$ comprises of a single simple line operator

$$\mathcal{O}_1^{(V)}, \tag{361}$$

which lies in the twisted sector for the generator $S_2^{(V)}$ of the $\mathbb{Z}_2$ 0-form symmetry. In particular, $\mathcal{O}_1^{(V)}$ lies at the end of the topological surface operator

$$D_2^{(V)} := \sigma\left(S_2^{(V)}\right), \tag{362}$$

of the underlying theory $\mathfrak{T}$ generating the $\mathbb{Z}_2$ 0-form symmetry that converts the 3d theory $\mathfrak{T}$ into a $\mathbb{Z}_2$ 0-form symmetric 3d theory $\mathfrak{T}_\sigma$, determined by a functor

$$\sigma: \quad \mathcal{S}_{\mathbb{Z}_2^{(0)}} \to \mathcal{S}_{\mathfrak{T}}, \tag{363}$$

where $\mathcal{S}_{\mathfrak{T}}$ is the symmetry 2-category of $\mathfrak{T}$. If $D_2^{(V)}$ is a non-trivial topological surface operator, then $\mathcal{O}_1^{(V)}$ is a non-genuine line operator living at the end of the surface operator $D_2^{(V)}$. On the other hand, it is possible for $D_2^{(V)}$ to be the trivial topological surface operator $D_2^{(\text{id})}$ (i.e. when the $\mathbb{Z}_2$ 0-form symmetry is non-faithful), in which case $\mathcal{O}_1^{(V)}$ is a genuine line operator.

The action of $\mathbb{Z}_2$ on $\mathcal{O}_1^{(V)}$ is trivial.

**1-charge $Q_2^{(V\mathbb{Z}_2)}$.** This is a combination of the above two 1-charges. There are only two possible ends

$$\mathcal{E}_1^{(V\mathbb{Z}_2)(\pm)}, \tag{364}$$

of the surface operator $Q_2^{(V\mathbb{Z}_2)}$ of $Z(2\text{-Vec}_{\mathbb{Z}_2})$ along $B_{2\text{-Vec}_{\mathbb{Z}_2}}^{\text{sym}}$. Both of the topological line operators $\mathcal{E}_1^{(V\mathbb{Z}_2)(\pm)}$ are additionally attached to the non-trivial topological surface operator $S_2^{(V)}$ of $B_{2\text{-Vec}_{\mathbb{Z}_2}}^{\text{sym}}$.

Correspondingly, a multiplet $\mathcal{M}_1^{(V\mathbb{Z}_2)}$ of line operators of a $\mathbb{Z}_2$ 0-form symmetric 3d theory $\mathfrak{T}_\sigma$ transforming in the 1-charge $Q_2^{(V\mathbb{Z}_2)}$ comprises of two simple line operators

$$\mathcal{O}_1^{(V\mathbb{Z}_2)(\pm)}, \tag{365}$$

which lie in the twisted sector for the generator $S_2^{(V)}$ of the $\mathbb{Z}_2$ 0-form symmetry. These operators are either non-genuine or genuine depending on whether or not the $\mathbb{Z}_2$ 0-form symmetry of $\mathfrak{T}_\sigma$ is realized faithfully.

The action of $\mathbb{Z}_2$ 0-form exchanges the two simple twisted sector line operators

$$\mathbb{Z}_2: \quad \mathcal{O}_1^{(V\mathbb{Z}_2)(+)} \longleftrightarrow \mathcal{O}_1^{(V\mathbb{Z}_2)(-)}. \tag{366}$$

### 5.2.2 0-charges

0-charges correspond to 1-morphisms in the Drinfeld center 2-category (352). First of all, the simple 1-endomorphisms (upto isomorphisms) of various simple objects are

$$
\begin{aligned}
\text{End}\left(\boldsymbol{Q}_2^{(\text{id})}\right) &= \left\{\boldsymbol{Q}_1^{(\text{id})}, \boldsymbol{Q}_1^{(-)}\right\} \cong \text{Rep}(\mathbb{Z}_2), \\
\text{End}\left(\boldsymbol{Q}_2^{(\mathbb{Z}_2)}\right) &= \left\{\boldsymbol{Q}_1^{(\mathbb{Z}_2;\text{id})}, \boldsymbol{Q}_1^{(\mathbb{Z}_2;-)}\right\} \cong \text{Vec}(\mathbb{Z}_2), \\
\text{End}\left(\boldsymbol{Q}_2^{(V)}\right) &= \left\{\boldsymbol{Q}_1^{(V;\text{id})}, \boldsymbol{Q}_1^{(V;-)}\right\} \cong \text{Rep}(\mathbb{Z}_2), \\
\text{End}\left(\boldsymbol{Q}_2^{(V\mathbb{Z}_2)}\right) &= \left\{\boldsymbol{Q}_1^{(V\mathbb{Z}_2;\text{id})}, \boldsymbol{Q}_1^{(V\mathbb{Z}_2;-)}\right\} \cong \text{Vec}(\mathbb{Z}_2).
\end{aligned}
\tag{367}
$$

The endomorphisms of the identity defect $\boldsymbol{Q}_2^{(\text{id})}$ should be thought of as genuine line defects of the Drinfeld center.

Then we also have simple 1-morphisms (upto isomorphisms) between different simple objects, which are

$$
\begin{aligned}
\text{Hom}\left(\boldsymbol{Q}_2^{(\text{id})}, \boldsymbol{Q}_2^{(\mathbb{Z}_2)}\right) &= \left\{\boldsymbol{Q}_1^{(\text{id},\mathbb{Z}_2)}\right\} \cong \text{Vec}, \\
\text{Hom}\left(\boldsymbol{Q}_2^{(V)}, \boldsymbol{Q}_2^{(V\mathbb{Z}_2)}\right) &= \left\{\boldsymbol{Q}_1^{(V,V\mathbb{Z}_2)}\right\} \cong \text{Vec}.
\end{aligned}
\tag{368}
$$

Below, we discuss the physics of the corresponding 0-charges.

**Projections.** The projections of these lines onto the symmetry boundary $\text{B}_{2\text{-Vec}_{\mathbb{Z}_2}}^{\text{sym}}$ are

$$
\begin{aligned}
S_1^{(\boldsymbol{Q}_1^{(\text{id})})} &\cong S_1^{(\boldsymbol{Q}_1^{(-)})} \cong S_1^{(\text{id})}, \\
S_1^{(\boldsymbol{Q}_1^{(V;\text{id})})} &\cong S_1^{(\boldsymbol{Q}_1^{(V;-)})} \cong S_1^{(V;\text{id})},
\end{aligned}
\tag{369}
$$

where $S_1^{(V;\text{id})}$ is the identity line on $S_2^{(V)}$, and

$$
\begin{aligned}
S_1^{(\boldsymbol{Q}_1^{(\mathbb{Z}_2;\text{id})})} &\cong \begin{pmatrix} S_1^{(\text{id})} & 0 \\ 0 & S_1^{(\text{id})} \end{pmatrix} \in \text{End}(2S_2^{(\text{id})}), \\
S_1^{(\boldsymbol{Q}_1^{(\mathbb{Z}_2;-)})} &\cong \begin{pmatrix} 0 & S_1^{(\text{id})} \\ S_1^{(\text{id})} & 0 \end{pmatrix} \in \text{End}(2S_2^{(\text{id})}), \\
S_1^{(\boldsymbol{Q}_1^{(V\mathbb{Z}_2;\text{id})})} &\cong \begin{pmatrix} S_1^{(V;\text{id})} & 0 \\ 0 & S_1^{(V;\text{id})} \end{pmatrix} \in \text{End}(2S_2^{(V)}), \\
S_1^{(\boldsymbol{Q}_1^{(V\mathbb{Z}_2;-)})} &\cong \begin{pmatrix} 0 & S_1^{(V;\text{id})} \\ S_1^{(V;\text{id})} & 0 \end{pmatrix} \in \text{End}(2S_2^{(V)}), \\
S_1^{(\boldsymbol{Q}_1^{(\text{id},\mathbb{Z}_2)})} &\cong \begin{pmatrix} S_1^{(\text{id})} & S_1^{(\text{id})} \end{pmatrix} \in \text{Hom}(S_2^{(\text{id})}, 2S_2^{(\text{id})}), \\
S_1^{(\boldsymbol{Q}_1^{(V,V\mathbb{Z}_2)})} &\cong \begin{pmatrix} S_1^{(V;\text{id})} & S_1^{(V;\text{id})} \end{pmatrix} \in \text{Hom}(S_2^{(V)}, 2S_2^{(V)}).
\end{aligned}
\tag{370}
$$

**0-charge $\boldsymbol{Q}_1^{(\text{id})}$.** The line operator $\boldsymbol{Q}_1^{(\text{id})}$ is the identity line operator of the 4d SymTFT $Z(2\text{-Vec}_{\mathbb{Z}_2})$. It can end on $\text{B}_{2\text{-Vec}_{\mathbb{Z}_2}}^{\text{sym}}$ in a topological local operator of $\text{B}_{2\text{-Vec}_{\mathbb{Z}_2}}^{\text{sym}}$. The only topological local operator on $\text{B}_{2\text{-Vec}_{\mathbb{Z}_2}}^{\text{sym}}$ is the identity operator, so the only possible end is

$$
\mathcal{E}_0^{(\text{id})} = S_0^{(\text{id})} \in \mathcal{S}_{\mathbb{Z}_2^{(0)}}.
\tag{371}
$$

Thus a multiplet $\mathcal{M}_0^{(\mathrm{id})}$ of local operators of a $\mathbb{Z}_2$ 0-form symmetric 3d theory $\mathfrak{T}_\sigma$ transforming in the 0-charge $Q_1^{(\mathrm{id})}$ comprises of a one-dimensional space of local operators generated by an operator

$$\mathcal{O}_0^{(\mathrm{id})}: \quad \mathcal{O}_1^{(\mathrm{id})(1)} \to \mathcal{O}_1^{(\mathrm{id})(2)}, \tag{372}$$

between two line operators $\mathcal{O}_1^{(\mathrm{id})(1)}$ and $\mathcal{O}_1^{(\mathrm{id})(2)}$ of $\mathfrak{T}_\sigma$ transforming in the 1-charge $Q_2^{(\mathrm{id})}$. The $\mathbb{Z}_2$ 0-form symmetry acts trivially on $\mathcal{O}_0^{(\mathrm{id})}$.

The identity line operator $D_1^{(\mathrm{id})}$ of $\mathfrak{T}$ transforms in the 1-charge $Q_2^{(\mathrm{id})}$. So we can choose

$$\mathcal{O}_1^{(\mathrm{id})(1)} = \mathcal{O}_1^{(\mathrm{id})(2)} = D_1^{(\mathrm{id})}. \tag{373}$$

In this case an operator $\mathcal{O}_0^{(\mathrm{id})}$ is a genuine local operator of the theory $\mathfrak{T}$, which transforms trivially under $\mathbb{Z}_2$ 0-form symmetry.

**0-charge $Q_1^{(-)}$.** This is a genuine topological line defect of $Z(2\text{-Vec}_{\mathbb{Z}_2})$ that projects to the trivial line in $B_{2\text{-Vec}_{\mathbb{Z}_2}}^{\mathrm{sym}}$ and has a single possible topological end

$$\mathcal{E}_0^{(-)}. \tag{374}$$

Thus a multiplet $\mathcal{M}_0^{(-)}$ of local operators of a $\mathbb{Z}_2$ 0-form symmetric 3d theory $\mathfrak{T}_\sigma$ transforming in the 0-charge $Q_1^{(-)}$ comprises of a one-dimensional space of local operators generated by an operator

$$\mathcal{O}_0^{(-)}: \quad \mathcal{O}_1^{(\mathrm{id})(1)} \to \mathcal{O}_1^{(\mathrm{id})(2)}, \tag{375}$$

between two line operators $\mathcal{O}_1^{(\mathrm{id})(1)}$ and $\mathcal{O}_1^{(\mathrm{id})(2)}$ of $\mathfrak{T}_\sigma$ transforming in the 1-charge $Q_2^{(\mathrm{id})}$. The $\mathbb{Z}_2$ 0-form symmetry acts on $\mathcal{O}_0^{(-)}$ as

$$\mathbb{Z}_2: \quad \mathcal{O}_0^{(-)} \to -\mathcal{O}_0^{(-)}. \tag{376}$$

If we choose

$$\mathcal{O}_1^{(\mathrm{id})(1)} = \mathcal{O}_1^{(\mathrm{id})(2)} = D_1^{(\mathrm{id})}, \tag{377}$$

then $\mathcal{O}_0^{(-)}$ is a genuine local operator of the theory $\mathfrak{T}$ which transforms in the non-trivial one-dimensional irreducible representation of the $\mathbb{Z}_2$ 0-form symmetry group.

**0-charge $Q_1^{(\mathbb{Z}_2;\mathrm{id})}$.** This is the identity topological line defect in the topological surface defect $Q_2^{(\mathbb{Z}_2)}$ of the 4d SymTFT $Z(2\text{-Vec}_{\mathbb{Z}_2})$. It has two possible topological ends

$$\mathcal{E}_0^{(\mathbb{Z}_2;\mathrm{id})(++)}: \quad \mathcal{E}_1^{(\mathbb{Z}_2)(+)} \to \mathcal{E}_1^{(\mathbb{Z}_2)(+)}, \tag{378}$$

and

$$\mathcal{E}_0^{(\mathbb{Z}_2;\mathrm{id})(--)}: \quad \mathcal{E}_1^{(\mathbb{Z}_2)(-)} \to \mathcal{E}_1^{(\mathbb{Z}_2)(-)}, \tag{379}$$

along $B_{2\text{-Vec}_{\mathbb{Z}_2}}^{\mathrm{sym}}$, where as discussed earlier $\mathcal{E}_1^{(\mathbb{Z}_2)(\pm)}$ are ends of $Q_2^{(\mathbb{Z}_2)}$.

Thus a multiplet $\mathcal{M}_0^{(\mathbb{Z}_2;\mathrm{id})}$ of local operators of a $\mathbb{Z}_2$ 0-form symmetric 3d theory $\mathfrak{T}_\sigma$ transforming in the 0-charge $Q_1^{(\mathbb{Z}_2;\mathrm{id})}$ comprises of a two-dimensional space of local operators generated by operators

$$\begin{aligned}
\mathcal{O}_0^{(\mathbb{Z}_2;\mathrm{id})(++)}: \quad \mathcal{O}_1^{(\mathbb{Z}_2)(+)(1)} &\to \mathcal{O}_1^{(\mathbb{Z}_2)(+)(2)}, \\
\mathcal{O}_0^{(\mathbb{Z}_2;\mathrm{id})(--)}: \quad \mathcal{O}_1^{(\mathbb{Z}_2)(-)(1)} &\to \mathcal{O}_1^{(\mathbb{Z}_2)(-)(2)},
\end{aligned} \tag{380}$$

where $\mathcal{O}_1^{(\mathbb{Z}_2)(\pm)(1)}$ and $\mathcal{O}_1^{(\mathbb{Z}_2)(\pm)(2)}$ are two multiplets of line operators (which can be same or different) in $\mathfrak{T}_\sigma$ transforming in the 1-charge $Q_2^{(\mathbb{Z}_2)}$. The $\mathbb{Z}_2$ 0-form symmetry acts as

$$\mathcal{O}_0^{(\mathbb{Z}_2;\mathrm{id})(++)} \longleftrightarrow \mathcal{O}_0^{(\mathbb{Z}_2;\mathrm{id})(--)}. \tag{381}$$

**0-charge $Q_1^{(\mathbb{Z}_2;-)}$.** This is a non-identity topological line defect along the topological surface defect $Q_2^{(\mathbb{Z}_2)}$ of the 4d SymTFT $Z(2\text{-Vec}_{\mathbb{Z}_2})$. It has two possible topological ends

$$\mathcal{E}_0^{(\mathbb{Z}_2;\mathrm{id})(+-)}: \quad \mathcal{E}_1^{(\mathbb{Z}_2)(+)} \to \mathcal{E}_1^{(\mathbb{Z}_2)(-)}, \tag{382}$$

and

$$\mathcal{E}_0^{(\mathbb{Z}_2;\mathrm{id})(-+)}: \quad \mathcal{E}_1^{(\mathbb{Z}_2)(-)} \to \mathcal{E}_1^{(\mathbb{Z}_2)(+)}, \tag{383}$$

along $B_{2\text{-Vec}_{\mathbb{Z}_2}}^{\mathrm{sym}}$, where as discussed earlier $\mathcal{E}_1^{(\mathbb{Z}_2)(\pm)}$ are ends of $Q_2^{(\mathbb{Z}_2)}$.

Thus a multiplet $\mathcal{M}_0^{(\mathbb{Z}_2;-)}$ of local operators of a $\mathbb{Z}_2$ 0-form symmetric 3d theory $\mathfrak{T}_\sigma$ transforming in the 0-charge $Q_1^{(\mathbb{Z}_2;-)}$ comprises of a two-dimensional space of local operators generated by operators

$$\begin{aligned} \mathcal{O}_0^{(\mathbb{Z}_2;\mathrm{id})(+-)}: \quad \mathcal{O}_1^{(\mathbb{Z}_2)(+)(1)} &\to \mathcal{O}_1^{(\mathbb{Z}_2)(-)(2)}, \\ \mathcal{O}_0^{(\mathbb{Z}_2;\mathrm{id})(-+)}: \quad \mathcal{O}_1^{(\mathbb{Z}_2)(-)(1)} &\to \mathcal{O}_1^{(\mathbb{Z}_2)(+)(2)}, \end{aligned} \tag{384}$$

where $\mathcal{O}_1^{(\mathbb{Z}_2)(\pm)(1)}$ and $\mathcal{O}_1^{(\mathbb{Z}_2)(\pm)(2)}$ are two multiplets of line operators (which can be same or different) in $\mathfrak{T}_\sigma$ transforming in the 1-charge $Q_2^{(\mathbb{Z}_2)}$. The $\mathbb{Z}_2$ 0-form symmetry acts as

$$\mathcal{O}_0^{(\mathbb{Z}_2;\mathrm{id})(+-)} \longleftrightarrow \mathcal{O}_0^{(\mathbb{Z}_2;\mathrm{id})(-+)}. \tag{385}$$

**0-charges $Q_1^{(V;\mathrm{id})}$, $Q_1^{(V;-)}$, $Q_1^{(V\mathbb{Z}_2;\mathrm{id})}$ and $Q_1^{(V\mathbb{Z}_2;-)}$.** The 0-charges $Q_1^{(V;\mathrm{id})}$ and $Q_1^{(V;-)}$ are analogous to the 0-charges $Q_1^{(\mathrm{id})}$ and $Q_1^{(-)}$ respectively, with the only difference being that the local operators transforming under the 0-charges $Q_1^{(V;\mathrm{id})}$ and $Q_1^{(V;-)}$ are additionally attached to the topological surface operator $D_2^{(V)}$ generating the $\mathbb{Z}_2$ 0-form symmetry of $\mathfrak{T}_\sigma$.

In a similar fashion, the 0-charges $Q_1^{(V\mathbb{Z}_2;\mathrm{id})}$ and $Q_1^{(V\mathbb{Z}_2;-)}$ are analogous to the 0-charges $Q_1^{(\mathbb{Z}_2;\mathrm{id})}$ and $Q_1^{(\mathbb{Z}_2;-)}$, respectively.

**0-charge $Q_1^{(\mathrm{id},\mathbb{Z}_2)}$.** This 0-charge corresponds to a topological line defect lying at the end of the surface defect $Q_2^{(\mathbb{Z}_2)}$ of the SymTFT $Z(2\text{-Vec}_{\mathbb{Z}_2})$, see (368). It has two possible topological ends

$$\mathcal{E}_0^{(\mathrm{id},\mathbb{Z}_2)(\pm)}: \quad \mathcal{E}_1^{(\mathrm{id})} \to \mathcal{E}_1^{(\mathbb{Z}_2)(\pm)}, \tag{386}$$

along $B_{2\text{-Vec}_{\mathbb{Z}_2}}^{\mathrm{sym}}$, where as discussed earlier $\mathcal{E}_1^{(\mathbb{Z}_2)(\pm)}$ are ends of $Q_2^{(\mathbb{Z}_2)}$ and $\mathcal{E}_1^{(\mathrm{id})}$ is the identity line on $B_{2\text{-Vec}_{\mathbb{Z}_2}}^{\mathrm{sym}}$.

Thus a multiplet $\mathcal{M}_0^{(\mathrm{id},\mathbb{Z}_2)}$ of local operators of a $\mathbb{Z}_2$ 0-form symmetric 3d theory $\mathfrak{T}_\sigma$ transforming in the 0-charge $Q_1^{(\mathrm{id},\mathbb{Z}_2)}$ comprises of a two-dimensional space of local operators generated by operators

$$\mathcal{O}_0^{(\mathrm{id},\mathbb{Z}_2)(\pm)}: \quad \mathcal{O}_1^{(\mathrm{id})} \to \mathcal{O}_1^{(\mathbb{Z}_2)(\pm)}, \tag{387}$$

where $\mathcal{O}_1^{(\mathbb{Z}_2)(\pm)}$ are two line operators of $\mathfrak{T}_\sigma$ in a multiplet transforming in the 1-charge $Q_2^{(\mathbb{Z}_2)}$ and $\mathcal{O}_1^{(\mathrm{id})}$ is a line operator of $\mathfrak{T}_\sigma$ on which $\mathbb{Z}_2$ acts trivially. The $\mathbb{Z}_2$ 0-form symmetry acts as

$$\mathcal{O}_0^{(\mathrm{id},\mathbb{Z}_2)(+)} \longleftrightarrow \mathcal{O}_0^{(\mathrm{id},\mathbb{Z}_2)(-)}. \tag{388}$$

**0-charge $Q_1^{(V,V\mathbb{Z}_2)}$.** This is analogous to the 0-charge $Q_1^{(\mathrm{id},\mathbb{Z}_2)}$, with the only difference being that the local operators transforming under the 0-charge $Q_1^{(V,V\mathbb{Z}_2)}$ are additionally attached to the topological surface operator $D_2^{(V)}$ generating the $\mathbb{Z}_2$ 0-form symmetry of $\mathfrak{T}_\sigma$.

Table 1: Summary of 1-charges for $\mathbb{Z}_2$ magnetic 0-form symmetry of 3d pure $SO(3)$ gauge theory: the first column indicates the 1-charge $\boldsymbol{Q}_2^{(a)}$, the second column indicates some line operators in the gauge theory realizing the 1-charge, and the last column specifies whether this is an untwisted or twisted sector line operator for the $\mathbb{Z}_2$ 0-form symmetry. The 0-charges are discussed in the text.

| $\boldsymbol{Q}_2^{(a)}$ | Gauge Theory Realization $\mathcal{O}_1^{(a)}$ | Sector |
|---|---|---|
| $\boldsymbol{Q}_2^{(\text{id})}$ | $D_1^{(\text{id})},\ W_1^{\text{adj}}$ | untwisted |
| $\boldsymbol{Q}_2^{(\mathbb{Z}_2)}$ | $2D_1^{(\text{id})}, 2W_1^{\text{adj}}$ | untwisted |
| $\boldsymbol{Q}_2^{(V)}$ | $W_1^{\text{fund}}$ | twisted |
| $\boldsymbol{Q}_2^{(V\mathbb{Z}_2)}$ | $2W_1^{\text{fund}}$ | twisted |

### 5.2.3 Realization in 3d pure $SO(3)$ gauge theory

Consider 3d pure gauge theory with $SO(3)$ gauge group. This theory has a $\mathbb{Z}_2$ 0-form symmetry, known as **magnetic 0-form symmetry**, associated to the fact that

$$\pi_1(SO(3)) = \mathbb{Z}_2 . \tag{389}$$

The topological codimension-1 operator generating this 0-form symmetry would be referred to as

$$D_2^{(V)} . \tag{390}$$

Let us discuss the realization of the above-discussed 1-charges and 0-charges in this theory.

**1-charge $\boldsymbol{Q}_2^{(\text{id})}$.** This is always realized trivially in any theory with $\mathbb{Z}_2$ 0-form symmetry by the identity line operator $D_1^{(\text{id})}$.

In the particular $SO(3)$ gauge theory under study, there also exist non-topological operators realizing this 1-charge. An example is provided by the Wilson line operator $W_1^{\text{adj}}$ in adjoint representation of $SO(3)$.

**1-charge $\boldsymbol{Q}_2^{(\mathbb{Z}_2)}$.** This can also always be realized trivially in any theory with $\mathbb{Z}_2$ 0-form symmetry. We simply take two copies $D_1^{(\text{id})(1)}$ and $D_1^{(\text{id})(2)}$ of the identity line

$$D_1^{(\mathbb{Z}_2)} := D_1^{(\text{id})(1)} \oplus D_1^{(\text{id})(2)} = 2D_1^{(\text{id})} . \tag{391}$$

Although $D_1^{(\mathbb{Z}_2)}$ is a non-simple line operator, it can be converted into a simple $\mathbb{Z}_2$-symmetric line operator by imposing a $\mathbb{Z}_2$ action that exchanges the two copies of $D_1^{(\text{id})}$

$$\mathbb{Z}_2 : \quad D_1^{(\text{id})(1)} \longleftrightarrow D_1^{(\text{id})(2)} . \tag{392}$$

The above action implies that this $\mathbb{Z}_2$-symmetric line operator $D_1^{(\mathbb{Z}_2)}$ transforms in the 1-charge $\boldsymbol{Q}_2^{(\mathbb{Z}_2)}$.

Non-topological line operators transforming in 1-charge $\boldsymbol{Q}_2^{(\mathbb{Z}_2)}$ can also be obtained similarly starting from a non-topological line operator $\mathcal{O}_1^{(\text{id})}$ transforming in trivial 1-charge $\boldsymbol{Q}_2^{(\text{id})}$. In the case of $SO(3)$ gauge theory, let us take it to be

$$\mathcal{O}_1^{(\text{id})} = W_1^{\text{adj}} . \tag{393}$$

Then take two copies $\mathcal{O}_1^{(\mathrm{id})(1)}$ and $\mathcal{O}_1^{(\mathrm{id})(2)}$ of $\mathcal{O}_1^{(\mathrm{id})}$

$$\mathcal{O}_1^{(\mathbb{Z}_2)} := \mathcal{O}_1^{(\mathrm{id})(1)} \oplus \mathcal{O}_1^{(\mathrm{id})(2)} = 2\mathcal{O}_1^{(\mathrm{id})}, \tag{394}$$

and impose $\mathbb{Z}_2$ action

$$\mathbb{Z}_2: \quad \mathcal{O}_1^{(\mathrm{id})(1)} \longleftrightarrow \mathcal{O}_1^{(\mathrm{id})(2)}. \tag{395}$$

This converts $\mathcal{O}_1^{(\mathbb{Z}_2)}$ into a simple $\mathbb{Z}_2$-symmetric line operator of the $SO(3)$ gauge theory, transforming in the 1-charge $\mathbf{Q}_2^{(\mathbb{Z}_2)}$.

We do not know of line operators in $SO(3)$ gauge theory that transform in 1-charge $\mathbf{Q}_2^{(\mathbb{Z}_2)}$ and do not descend in the above way from line operators transforming in the trivial 1-charge $\mathbf{Q}_2^{(\mathrm{id})}$. However, we will exhibit an example of line operators of this type in the case of 3d pure $SU(3)$ gauge theory discussed below.

**1-charge $\mathbf{Q}_2^{(V)}$.** It is not possible to realize this 1-charge in every $\mathbb{Z}_2$ 0-form symmetric 3d theory. For example, consider the 3d BF theory with action

$$S = \int_{M_3} a_2 \cup \delta b_0, \tag{396}$$

where $a_2$ and $b_0$ are $\mathbb{Z}_2$ valued 2-cocycle and 0-cocycle gauge fields respectively. In this case, the Wilson surface operator

$$\exp\left(\int i\pi a_2\right), \tag{397}$$

generates a $\mathbb{Z}_2$ 0-form symmetry, but there are no operators in the twisted sector for this symmetry, as it would be in contradiction with the existence of the 2-form symmetry in this theory generated by the Wilson point-like operator

$$\exp(i\pi b_0). \tag{398}$$

This is because if the Wilson surface operator can end, then cannot be charged non-trivially under a 2-form symmetry, but in this theory it is charged under the 2-form symmetry generated by the above Wilson point operator.

However, in the case of $SO(3)$ gauge theory under study, there do exist line operators transforming in the 1-charge $\mathbf{Q}_2^{(V)}$. An example is provided by the Wilson line operator $W_1^{\mathrm{fund}}$ in fundamental representation of the gauge algebra $\mathfrak{su}(2)$, i.e.

$$\mathcal{O}_1^{(V)} = W_1^{\mathrm{fund}}. \tag{399}$$

This operator is attached to the generator $D_2^{(V)}$ of the $\mathbb{Z}_2$ 0-form symmetry, and hence lies in the twisted sector. The $\mathbb{Z}_2$ symmetry acts trivially on $W_1^{\mathrm{fund}}$.

**1-charge $\mathbf{Q}_2^{(V\mathbb{Z}_2)}$.** This 1-charge can always be realized in a $\mathbb{Z}_2$ 0-form symmetric 3d theory $\mathfrak{T}_\sigma$ if the theory $\mathfrak{T}_\sigma$ admits a line operator $\mathcal{O}_1^{(V)}$ transforming in the 1-charge $\mathbf{Q}_2^{(V)}$. We take two copies $\mathcal{O}_1^{(V)(1)}$ and $\mathcal{O}_1^{(V)(2)}$ of $\mathcal{O}_1^{(V)}$

$$\mathcal{O}_1^{(V\mathbb{Z}_2)} := \mathcal{O}_1^{(V)(1)} \oplus \mathcal{O}_1^{(V)(2)} = 2\mathcal{O}_1^{(V)}, \tag{400}$$

and impose $\mathbb{Z}_2$ action

$$\mathbb{Z}_2: \quad \mathcal{O}_1^{(V)(1)} \longleftrightarrow \mathcal{O}_1^{(V)(2)}. \tag{401}$$

This converts $\mathcal{O}_1^{(V\mathbb{Z}_2)}$ into a simple $\mathbb{Z}_2$-symmetric line operator of $\mathfrak{T}_\sigma$ transforming in the 1-charge $\mathbf{Q}_2^{(V\mathbb{Z}_2)}$.

In the case of 3d $SO(3)$ gauge theory, we can pick

$$\mathcal{O}_1^{(V)} = W_1^{\text{fund}}, \tag{402}$$

and construct the corresponding operator $\mathcal{O}_1^{(V\mathbb{Z}_2)}$ as above.

We do not know of line operators in $SO(3)$ gauge theory that transform in 1-charge $\mathbf{Q}_2^{(V\mathbb{Z}_2)}$ and do not descend in the above way from line operators transforming in the 1-charge $\mathbf{Q}_2^{(V)}$. However, we will exhibit an example of line operators of this type in the case of 3d pure $SU(3)$ gauge theory discussed below. A summary of all 1-charges can be found in table 1.

**0-charge $\mathbf{Q}_1^{(\text{id})}$.** Let us now discuss 0-charges. The 0-charge $\mathbf{Q}_1^{(\text{id})}$ is always realized trivially in any theory with $\mathbb{Z}_2$ 0-form symmetry by the identity local operator $D_0^{(\text{id})}$.

In the particular $SO(3)$ gauge theory under study, there also exist non-topological operators realizing this 0-charge. An example is provided by the genuine (i.e. gauge invariant) local operator

$$\mathcal{O}_0^{(\text{id})} = \text{Tr}\, F, \tag{403}$$

where $F$ is the field strength. The magnetic $\mathbb{Z}_2$ 0-form symmetry acts trivially on it.

This theory also contains non-genuine operators transforming in the 0-charge $\mathbf{Q}_1^{(\text{id})}$. An example is provided by the field strength $F$, which is a local operator arising at the end of the adjoint Wilson line operator $W_1^{\text{adj}}$, because the field strength transforms in the adjoint representation of the gauge group. The magnetic $\mathbb{Z}_2$ 0-form symmetry acts trivially on this non-genuine local operator.

**0-charge $\mathbf{Q}_1^{(-)}$.** This is not always realized in an arbitrary $\mathbb{Z}_2$ 0-form symmetric 3d theory $\mathfrak{T}_\sigma$. An example is provided by trivial 3d theory made $\mathbb{Z}_2$ 0-form symmetric in the trivial way. This theory does not carry a local operator charged non-trivially under the $\mathbb{Z}_2$ 0-form symmetry.

However, in the $SO(3)$ gauge theory under study, there exist magnetic monopole operators that carry this 0-charge. Such a monopole operator has the property that we have

$$\int_{S^2} w_2 = 1 \in \mathbb{Z}/2\mathbb{Z}, \tag{404}$$

on a small sphere $S^2$ surrounding the monopole operator. Here $w_2$ is second Stiefel-Whitney class for $SO(3)$ gauge bundles. It being non-trivial means that the monopole configuration associated to such a monopole operator cannot be lifted to a monopole configuration for the double cover $SU(2)$ of $SO(3)$.

**0-charges $\mathbf{Q}_1^{(\mathbb{Z}_2;\text{id})}$ and $\mathbf{Q}_1^{(\mathbb{Z}_2;-)}$.** This 0-charge can always be realized in an arbitrary $\mathbb{Z}_2$ 0-form symmetric 3d theory $\mathfrak{T}_\sigma$. Recall from (391) that any such theory contains a line operator $D_1^{(\mathbb{Z}_2)}$ comprised of two copies $D_1^{(\text{id})(1)}$ and $D_1^{(\text{id})(2)}$ of the identity line operator $D_1^{(\text{id})}$. We have topological local operators

$$\begin{aligned}
D_0^{(\text{id})(11)}: \quad & D_1^{(\text{id})(1)} \to D_1^{(\text{id})(1)}, \\
D_0^{(\text{id})(22)}: \quad & D_1^{(\text{id})(2)} \to D_1^{(\text{id})(2)}, \\
D_0^{(\text{id})(12)}: \quad & D_1^{(\text{id})(1)} \to D_1^{(\text{id})(2)}, \\
D_0^{(\text{id})(21)}: \quad & D_1^{(\text{id})(2)} \to D_1^{(\text{id})(1)},
\end{aligned} \tag{405}$$

which are identity local operators going between the two copies of the identity line. The $\mathbb{Z}_2$ 0-form symmetry action on $D_1^{(\mathbb{Z}_2)}$ exchanges these local operators as

$$
\begin{aligned}
D_0^{(\text{id})(11)} &\longleftrightarrow D_0^{(\text{id})(22)}, \\
D_0^{(\text{id})(12)} &\longleftrightarrow D_0^{(\text{id})(21)},
\end{aligned}
\tag{406}
$$

and thus the $D_0^{(\text{id})(11)} \oplus D_0^{(\text{id})(22)}$ and $D_0^{(\text{id})(12)} \oplus D_0^{(\text{id})(21)}$ realize the 0-charges $\boldsymbol{Q}_1^{(\mathbb{Z}_2;\text{id})}$ and $\boldsymbol{Q}_1^{(\mathbb{Z}_2;-)}$ respectively.

In fact, more generally consider an operator $\mathcal{O}_0^{(\text{id})}$ of $\mathfrak{T}_\sigma$ transforming in the 0-charge $\boldsymbol{Q}_1^{(\text{id})}$ transitioning between two line operators $\mathcal{O}_1^{(\text{id})(\alpha)}$ and $\mathcal{O}_1^{(\text{id})(\beta)}$ both transforming in the trivial 1-charge $\boldsymbol{Q}_2^{(\text{id})}$. In the same fashion as above, using four copies of $\mathcal{O}_0^{(\text{id})}$ transitioning between two copies of $\mathcal{O}_1^{(\text{id})(\alpha)}$ and $\mathcal{O}_1^{(\text{id})(\beta)}$, we can construct multiplets of local operators transforming in the 0-charges $\boldsymbol{Q}_1^{(\mathbb{Z}_2;\text{id})}$ and $\boldsymbol{Q}_1^{(\mathbb{Z}_2;-)}$.

**0-charge $\boldsymbol{Q}_1^{(V;\text{id})}$.** If $\mathfrak{T}_\sigma$ contains a line operator $\mathcal{O}_1^{(V)}$ transforming in 1-charge $\boldsymbol{Q}_2^{(V)}$, then it definitely contains local operators $\mathcal{O}_0^{(V;\text{id})}$ transforming in the 0-charge $\boldsymbol{Q}_1^{(V;\text{id})}$. An example is provided by the identity local operator along the line $\mathcal{O}_1^{(V)}$. In general, we can take any genuine local operator $\mathcal{O}_0^{(\text{id})}$ of $\mathfrak{T}_\sigma$ transforming in the 0-charge $\boldsymbol{Q}_1^{(\text{id})}$ and take its OPE with the line $\mathcal{O}_1^{(V)}$ to obtain non-identity local operators on $\mathcal{O}_1^{(V)}$ transforming in 0-charge $\boldsymbol{Q}_1^{(V;\text{id})}$. Similarly, taking OPE of non-genuine local operators transforming in $\boldsymbol{Q}_1^{(\text{id})}$ transitioning between two line operators transforming in $\boldsymbol{Q}_2^{(\text{id})}$ with the line $\mathcal{O}_1^{(V)}$ produces non-genuine local operators transforming in $\boldsymbol{Q}_1^{(V;\text{id})}$ transitioning between two different line operators transforming in $\boldsymbol{Q}_2^{(V)}$.

In the $SO(3)$ gauge theory under study, we have local operators comprised of field strength between two different Wilson line operators in half-integral spin representations of $\mathfrak{su}(2)$. Such Wilson lines transform in 1-charge $\boldsymbol{Q}_2^{(V)}$ and the local operators transform in 0-charge $\boldsymbol{Q}_1^{(V)}$.

**0-charge $\boldsymbol{Q}_1^{(V;-)}$.** If $\mathfrak{T}_\sigma$ contains a local operator $\mathcal{O}_0^{(-)}$ transforming in 0-charge $\boldsymbol{Q}_1^{(-)}$ and a line operator $\mathcal{O}_1^{(V)}$ transforming in 1-charge $\boldsymbol{Q}_2^{(V)}$, then it contains local operators transforming in 0-charge $\boldsymbol{Q}_1^{(V;-)}$. Examples are produced by taking OPE of $\mathcal{O}_0^{(-)}$ and $\mathcal{O}_1^{(V)}$.

For the $SO(3)$ gauge theory under discussion, we can take $\mathcal{O}_0^{(-)}$ to be a monopole operator for $SO(3)$ whose associated monopole configuration cannot be lifted to an $SU(2)$ monopole configuration, and $\mathcal{O}_1^{(V)}$ to be the Wilson line $W_1^{\text{fund}}$. Taking the OPE we obtain $SO(3)$ monopole operators living along $W_1^{\text{fund}}$, on which $\mathbb{Z}_2$ 0-form symmetry acts non-trivially.

**0-charge $\boldsymbol{Q}_1^{(\text{id},\mathbb{Z}_2)}$.** Any $\mathbb{Z}_2$ 0-form symmetric theory $\mathfrak{T}_\sigma$ contains a local operator transforming in such a 0-charge. Simply take two copies $D_1^{(\text{id})(1)} D_1^{(\text{id})(2)}$ of identity line operator with $\mathbb{Z}_2$ 0-form symmetry acting by exchanging the two copies. As discussed earlier, this gives rise to a $\mathbb{Z}_2$-symmetric line $D_1^{(\mathbb{Z}_2)}$ transforming in 1-charge $\boldsymbol{Q}_2^{(\mathbb{Z}_2)}$. Then take two copies of identity local operators

$$
D_0^{(\text{id})(i)}: \quad D_1^{(\text{id})} \to D_1^{(\text{id})(i)}, \qquad i \in \{1,2\},
\tag{407}
$$

with $\mathbb{Z}_2$ action

$$
D_0^{(\text{id})(1)} \longleftrightarrow D_0^{(\text{id})(2)}.
\tag{408}
$$

The multiplet $D_0^{(\text{id})(1)} \oplus D_0^{(\text{id})(2)}$ transforms in 0-charge $\boldsymbol{Q}_1^{(\text{id},\mathbb{Z}_2)}$.

Table 2: Summary of 1-charges for $\mathbb{Z}_2$ charge conjugation 0-form symmetry of 3d pure $SU(3)$ gauge theory: the first column indicates the 1-charge $Q_2^{(a)}$, the second column indicates some line operators in the gauge theory realizing the 1-charge, and the last column specifies whether this is an untwisted or twisted sector line operator for the $\mathbb{Z}_2$ 0-form symmetry. The 0-charges are discussed in the text.

| $Q_2^{(a)}$ | Gauge Theory Realization $\mathcal{O}_1^{(a)}$ | Sector |
|---|---|---|
| $Q_1^{(\mathrm{id})}$ | $W_1^{R_{(n,n)}}$ | untwisted |
| $Q_1^{(\mathbb{Z}_2)}$ | $W_1^{R_{(n,m)}} \oplus W_1^{R_{(m,n)}}, \quad n \neq m$ | untwisted |
| $Q_1^{(V)}$ | Vortex line for $\widetilde{SU}(3)V_1$ | twisted |
| $Q_2^{(V\mathbb{Z}_2)}$ | Vortex lines for $\widetilde{PSU}(3)V_1^{(\omega)} \oplus V_1^{(\omega^2)}$ | twisted |

Similarly, given any local operator $\mathcal{O}_0^{(\mathrm{id})}$ transforming in 0-charge $Q_1^{(\mathrm{id})}$ transitioning between two line operators $\mathcal{O}_1^{(\mathrm{id})(\alpha)}$ and $\mathcal{O}_1^{(\mathrm{id})(\beta)}$ transforming in 1-charge $Q_2^{(\mathrm{id})}$, we can construct a multiplet of local operators transforming in 0-charge $Q_1^{(\mathrm{id},\mathbb{Z}_2)}$ by taking two copies of $\mathcal{O}_0^{(\mathrm{id})}$, that transitions between line operators $\mathcal{O}_1^{(\mathrm{id})(\alpha)}$ and $\mathcal{O}_1^{(\mathbb{Z}_2)(\beta)}$ where $\mathcal{O}_1^{(\mathbb{Z}_2)(\beta)}$ is obtained from $\mathcal{O}_1^{(\mathrm{id})(\beta)}$ as in (394).

**0-charge $Q_1^{(V,V\mathbb{Z}_2)}$.** Given any local operator $\mathcal{O}_0^{(V)}$ transforming in 0-charge $Q_1^{(V;\mathrm{id})}$ transitioning between two line operators $\mathcal{O}_1^{(V)(\alpha)}$ and $\mathcal{O}_1^{(V)(\beta)}$ transforming in 1-charge $Q_2^{(V)}$, we can construct a multiplet of local operators transforming in 0-charge $Q_1^{(V,V\mathbb{Z}_2)}$ by taking two copies of $\mathcal{O}_0^{(V)}$, that transitions between line operators $\mathcal{O}_1^{(V)(\alpha)}$ and $\mathcal{O}_1^{(V\mathbb{Z}_2)(\beta)}$ where $\mathcal{O}_1^{(V\mathbb{Z}_2)(\beta)}$ is obtained from $\mathcal{O}_1^{(V)(\beta)}$ as in (400).

### 5.2.4 Realization in 3d pure $SU(3)$ gauge theory

In the previous subsection we discussed all kinds of 1-charges and 0-charges for $\mathbb{Z}_2$ 0-form symmetry, as realized in 3d pure $SO(3)$ gauge theory with the $\mathbb{Z}_2$ 0-form symmetry being the magnetic one. As we saw, most of these charges can be realized non-trivially in the $SO(3)$ gauge theory, but some charges (like the 1-charges $Q_2^{(\mathbb{Z}_2)}$ and $Q_2^{(\mathbb{Z}_2 V)}$) can only be realized trivially using operators transforming in other charges. In this subsection, we study another gauge theory in 3d with $\mathbb{Z}_2$ 0-form symmetry which provides non-trivial realizations of all charges.

The theory that we study in this subsection is pure gauge theory in 3d with gauge group $SU(3)$, and the

$$\mathbb{Z}_2^{(0)} = \text{charge conjugation symmetry}, \tag{409}$$

descending from the $\mathbb{Z}_2$ outer automorphism of the $\mathfrak{su}(3)$ gauge algebra. The topological codimension-1 operator generating this 0-form symmetry would again be referred to as $D_2^{(V)}$.

**1-charge $Q_2^{(\mathrm{id})}$.** This is realized by any Wilson line corresponding to an irreducible representation of $SU(3)$ whose highest weight has same Dynkin coefficients $(n, n)$ for the two nodes

$$\mathcal{O}_1^{(\mathrm{id})} = W_1^{R_{(n,n)}}. \tag{410}$$

As the outer-automorphism exchanges the two Dynkin coefficients, such a representation, and hence the corresponding Wilson line, is left invariant under the action of outer-automorphism.

An example is provided by the adjoint Wilson line whose highest weight has Dynkin coefficients $(1,1)$.

**1-charge $Q_2^{(\mathbb{Z}_2)}$.** This is realized by any Wilson line $W_1^{R_{(m,n)}}$ corresponding to an irreducible representation of $SU(3)$ whose highest weight has different Dynkin coefficients $(m,n)$ for the two nodes for $m \neq n$. Under the action of $\mathbb{Z}_2$ outer-automorphism, such a Wilson line is exchanged with the Wilson line is exchanged with a Wilson line $W_1^{R_{(n,m)}}$ corresponding to an irreducible representation of $SU(3)$ whose highest weight has Dynkin coefficients $(n,m)$. An example is provided by the fundamental Wilson line whose highest weight has Dynkin coefficients $(1,0)$, which is exchanged with the anti-fundamental Wilson line whose highest weight has Dynkin coefficients $(0,1)$.

**1-charge $Q_2^{(V)}$.** This is realized by a Gukov-Witten/vortex line operator $V_1$ having the property that

$$\int_{S^1} w_1 = 1 \in \mathbb{Z}/2\mathbb{Z}, \tag{411}$$

on a small circle linking the vortex line $V_1$. Here $w_1$ is a characteristic class for bundles of the disconnected group

$$\widetilde{SU}(3) = SU(3) \rtimes \mathbb{Z}_2, \tag{412}$$

constructed using the outer-automorphism action of $\mathbb{Z}_2$ on $SU(3)$. An $\widetilde{SU}(3)$ bundle with non-trivial class $w_1$ does not restrict to an $SU(3)$ bundle. The fact that we have a non-trivial $w_1$ on a circle around the line $V_1$ means that $V_1$ arises at the end of the topological surface operator $D_2^{(V)}$ generating the $\mathbb{Z}_2$ outer-automorphism symmetry.

**1-charge $Q_2^{(V\mathbb{Z}_2)}$.** This is realized by vortex line operators $V_1^{(\omega)}$ and $V_1^{(\omega^2)}$, around which we have $\widetilde{PSU}(3)$ bundles where

$$\widetilde{PSU}(3) = PSU(3) \rtimes \mathbb{Z}_2 = \frac{SU(3)}{\mathbb{Z}_3} \rtimes \mathbb{Z}_2. \tag{413}$$

These line operators have the property that

$$\int_{S^1} w_1 = 1 \in \mathbb{Z}/2\mathbb{Z}, \tag{414}$$

on a small circle linking these lines. Here the characteristic class $w_1$ captures the obstruction for being able to restrict a $\widetilde{PSU}(3)$ bundle to a $PSU(3)$ bundle. As a consequence, both these vortex lines are attached to $D_2^{(V)}$.

Moreover, consider a small disk $D^2$ intersecting $V_1^{(\omega)}$ at a point. Then, we have

$$\int_{D^2} w_2 = 1 \in \mathbb{Z}/3\mathbb{Z}, \tag{415}$$

where the characteristic class $w_2$ captures the obstruction for being able to lift a

$$\widetilde{PSU}(3) = \widetilde{SU}(3)/\mathbb{Z}_3, \tag{416}$$

bundle to a $\widetilde{SU}(3)$ bundle. Similarly, for $V_1^{(\omega^2)}$, we instead have

$$\int_{D^2} w_2 = 2 \in \mathbb{Z}/3\mathbb{Z}. \tag{417}$$

Now since $\mathbb{Z}_2$ outer-automorphism acts non-trivially on the $\mathbb{Z}_3$ center, the $\mathbb{Z}_2$ 0-form symmetry exchanges the two vortex lines

$$\mathbb{Z}_2: \quad V_1^{(\omega)} \longleftrightarrow V_1^{(\omega^2)}. \tag{418}$$

**0-charge $Q_1^{(\mathbf{id})}$.** This is realized by any monopole operator for which the co-character

$$\phi: \quad U(1) \to SU(3), \tag{419}$$

has equal winding numbers along the two $U(1)_i$ comprising the maximal torus $U(1)_1 \times U(1)_2 \subset SU(3)$. The winding numbers are exchanged by the $\mathbb{Z}_2$ outer-automorphism.

**0-charge $Q_1^{(-)}$.** This is realized by the genuine local operator

$$\mathcal{O}_0^{(\mathrm{Tr}\,F)}, \tag{420}$$

constructed using the gauge singlet

$$\mathrm{Tr}\,F, \tag{421}$$

comprised from the field strength $F$. The action of the outer-automorphism on $\mathfrak{su}(3)$ implies that the above operator picks up a sign under the action of this $\mathbb{Z}_2$

$$\mathbb{Z}_2: \quad \mathcal{O}_0^{(\mathrm{Tr}\,F)} \to -\mathcal{O}_0^{(\mathrm{Tr}\,F)}. \tag{422}$$

**0-charge $Q_1^{(\mathbb{Z}_2;\mathbf{id})}$.** Local operators carrying this 0-charge are realized by taking OPE of monopole operators having equal winding numbers with the Wilson lines having unequal Dynkin coefficients.

**0-charge $Q_1^{(\mathbb{Z}_2;-)}$.** Local operators carrying this 0-charge are realized by non-genuine operators transitioning between Wilson lines $W_1^{R_{(n,m)}}$ and $W_1^{R_{(m,n)}}$ for $m \neq n$ composed out of field strength.

**0-charge $Q_1^{(V;\mathbf{id})}$.** Local operators carrying this 0-charge are realized by taking OPE of monopole operators with equal winding numbers with the vortex line operator $V_1$ for $\widetilde{SU}(3)$ discussed above.

**0-charge $Q_1^{(V;-)}$.** Local operators carrying this 0-charge are realized by taking OPE of genuine local operator $\mathcal{O}_0^{(\mathrm{Tr}\,F)}$ with the vortex line operator $V_1$ for $\widetilde{SU}(3)$ discussed above.

**0-charge $Q_1^{(V\mathbb{Z}_2;\mathbf{id})}$.** Local operators carrying such charges are realized by taking OPE of monopole operators having equal winding numbers with the vortex line operators $V_1^{(\omega)}$ and $V_1^{(\omega^2)}$ for $\widetilde{PSU}(3)$ discussed above.

**0-charge $Q_1^{(V\mathbb{Z}_2;-)}$.** Local operators carrying such charges are realized by non-genuine monopole operators transitioning between the vortex line operators $V_1^{(\omega)}$ and $V_1^{(\omega^2)}$ for $\widetilde{PSU}(3)$ discussed above. Such monopole operators have the property that on a small sphere $S^2$ linking them, we have

$$\int_{S^2} w_2 \neq 0 \in \mathbb{Z}/3\mathbb{Z}. \tag{423}$$

## 5.3 $\mathbb{Z}_2$ 1-form symmetry

Consider now a non-anomalous $\mathbb{Z}_2$ 1-form symmetry in 3d, for which the associated fusion 2-category is

$$\mathcal{S}_{\mathbb{Z}_2^{(1)}} = 2\text{-Rep}(\mathbb{Z}_2), \tag{424}$$

whose simple objects up to isomorphisms are

$$\text{Obj}(\mathcal{S}_{\mathbb{Z}_2}) = \left\{ S_2^{(\text{id})}, S_2^{(\mathbb{Z}_2)} \right\}, \tag{425}$$

with fusion

$$S_2^{(\mathbb{Z}_2)} \otimes S_2^{(\mathbb{Z}_2)} \cong 2 S_2^{(\mathbb{Z}_2)}, \tag{426}$$

simple 1-endomorphisms (upto isomorphisms) are

$$\begin{aligned} \text{End}(S_2^{(\text{id})}) &= \left\{ S_1^{(\text{id})}, S_1^{(-)} \right\}, \\ \text{End}(S_2^{(\mathbb{Z}_2)}) &= \left\{ S_1^{(\mathbb{Z}_2;\text{id})}, S_1^{(\mathbb{Z}_2;-)} \right\}. \end{aligned} \tag{427}$$

$S_1^{(-)}$ is the generator of the $\mathbb{Z}_2$ 1-form symmetry and $S_1^{(\mathbb{Z}_2;-)}$ generates a 0-form symmetry localized along the surface $S_2^{(\mathbb{Z}_2)}$. There are also simple 1-morphisms between the two simple objects, which up to isomorphisms are

$$\begin{aligned} \text{Hom}(S_2^{(\text{id})}, S_2^{(\mathbb{Z}_2)}) &= \left\{ S_1^{(\text{id},\mathbb{Z}_2)} \right\}, \\ \text{Hom}(S_2^{(\mathbb{Z}_2)}, S_2^{(\text{id})}) &= \left\{ S_1^{(\mathbb{Z}_2,\text{id})} \right\}. \end{aligned} \tag{428}$$

Since this symmetry can be obtained by a gauging a non-anomalous $\mathbb{Z}_2$ 0-form symmetry in 3d, the associated SymTFTs are same, i.e.

$$Z(2\text{-Vec}_{\mathbb{Z}_2}) = \mathfrak{Z}(2\text{-Rep}(\mathbb{Z}_2)), \tag{429}$$

but the symmetry boundaries are different

$$\mathsf{B}^{\text{sym}}_{2\text{-Vec}_{\mathbb{Z}_2}} \neq \mathfrak{B}^{\text{sym}}_{2\text{-Rep}(\mathbb{Z}_2)}. \tag{430}$$

$S_2^{(\mathbb{Z}_2)}$ is a topological surface operator on the 3d symmetry boundary $\mathsf{B}^{\text{sym}}_{2\text{-Rep}(\mathbb{Z}_2)}$ obtained by gauging the $\mathbb{Z}_2$ 1-form symmetry of $\mathsf{B}^{\text{sym}}_{2\text{-Rep}(\mathbb{Z}_2)}$ along a two-dimensional worldvolume inside the three-dimensional worldvolume occupied by $\mathsf{B}^{\text{sym}}_{2\text{-Rep}(\mathbb{Z}_2)}$. This surface operator is also known as the **condensation surface defect** for the $\mathbb{Z}_2$ 1-form symmetry.

Since the SymTFTs are same, their topological operators, in other words the Drinfeld centers are the same

$$\mathcal{Z}(2\text{-Vec}_{\mathbb{Z}_2}) = \mathcal{Z}(2\text{-Rep}(\mathbb{Z}_2)). \tag{431}$$

Physically this means that the sets of 1-charges and 0-charges are the same for both symmetries. However, since the symmetry boundaries are different, the ends of topological operators along the boundaries are different, which leads to different multiplet structure for operators transforming in these charges. Let us begin with a discussion of 1-charges.

**Projections.** The projections of bulk surfaces to the symmetry boundary $\mathsf{B}^{\text{sym}}_{2\text{-Rep}(\mathbb{Z}_2)}$ are

$$S_2^{(Q_2^{(\text{id})})} \cong S_2^{(Q_2^{(V)})} \cong S_2^{(\text{id})}, \qquad S_2^{(Q_2^{(\mathbb{Z}_2)})} \cong S_2^{(Q_2^{(V\mathbb{Z}_2)})} \cong S_2^{(\mathbb{Z}_2)}. \tag{432}$$

### 5.3.1 1-charges

**1-charge $Q_2^{(\text{id})}$.** Since this corresponds to the identity topological surface operator of the 4d SymTFT $Z(2\text{-Rep}(\mathbb{Z}_2))$, its ends along the 3d symmetry boundary $B^{\text{sym}}_{2\text{-Rep}(\mathbb{Z}_2)}$ correspond to topological line defects of $B^{\text{sym}}_{2\text{-Rep}(\mathbb{Z}_2)}$ that lie at the end of some (trivial or non-trivial) topological surface operator $S_2 \in 2\text{-Rep}(\mathbb{Z}_2)$. Thus, there are three possible simple ends

$$\mathcal{E}_1^{(\text{id})(+)} := S_1^{(\text{id})}, \qquad \mathcal{E}_1^{(\text{id})(-)} := S_1^{(-)}, \qquad \mathcal{E}_1^{(\text{id})(\mathbb{Z}_2)} := S_1^{(\text{id},\mathbb{Z}_2)}, \tag{433}$$

where $\mathcal{E}_1^{(\text{id})(+)}$ and $\mathcal{E}_1^{(\text{id})(-)}$ are not attached to any topological surfaces of $B^{\text{sym}}_{2\text{-Rep}(\mathbb{Z}_2)}$, while $\mathcal{E}_1^{(\text{id})(\mathbb{Z}_2)}$ is attached to the topological surface $S_2^{(\mathbb{Z}_2)}$ of $B^{\text{sym}}_{2\text{-Rep}(\mathbb{Z}_2)}$.

Consequently, a multiplet $\mathcal{M}_1^{(\text{id})}$ of line operators of a $\mathbb{Z}_2$ 1-form symmetric 3d theory $\mathfrak{T}_\sigma$ transforming in the 1-charge $Q_2^{(\text{id})}$ comprises of three line operators

$$\mathcal{O}_1^{(\text{id})(+)}, \qquad \mathcal{O}_1^{(\text{id})(-)}, \qquad \mathcal{O}_1^{(\text{id})(\mathbb{Z}_2)}. \tag{434}$$

Here $\mathcal{O}_1^{(\text{id})(\pm)}$ are genuine line operators of $\mathfrak{T}$ which are not charged under the 1-form symmetry, but are permuted into each other by fusion with the topological line operator

$$D_1^{(-)} := \sigma(S_1^{(-)}), \tag{435}$$

generating the $\mathbb{Z}_2$ 1-form symmetry of $\mathfrak{T}_\sigma$. That is, we have

$$\begin{aligned} D_1^{(-)} \otimes \mathcal{O}_1^{(\text{id})(+)} &= \mathcal{O}_1^{(\text{id})(-)}, \\ D_1^{(-)} \otimes \mathcal{O}_1^{(\text{id})(-)} &= \mathcal{O}_1^{(\text{id})(+)}. \end{aligned} \tag{436}$$

On the other hand, $\mathcal{O}_1^{(\text{id})(\mathbb{Z}_2)}$ is a line operator living at the end of the topological surface defect

$$D_2^{(\mathbb{Z}_2)} := \sigma\left(S_2^{(\mathbb{Z}_2)}\right), \tag{437}$$

generating the $S_2^{(\mathbb{Z}_2)}$ symmetry of $\mathfrak{T}_\sigma$, and is left invariant by fusion with the topological line operator

$$D_1^{(\mathbb{Z}_2;-)} := \sigma\left(S_1^{(\mathbb{Z}_2;-)}\right), \tag{438}$$

generating the localized $\mathbb{Z}_2$ 0-form symmetry of $D_2^{(\mathbb{Z}_2)}$. That is, we have the fusion rule

$$\mathcal{O}_1^{(\text{id})(\mathbb{Z}_2)} \otimes_{D_2^{(\mathbb{Z}_2)}} D_1^{(\mathbb{Z}_2;-)} = \mathcal{O}_1^{(\text{id})(\mathbb{Z}_2)}. \tag{439}$$

It should be noted that, if the 1-form symmetry is *not faithful* and is generated by the identity line defect $D_1^{(\text{id})}$ of $\mathfrak{T}$, i.e. if we have

$$D_1^{(-)} \cong D_1^{(\text{id})}, \tag{440}$$

then

$$D_2^{(\mathbb{Z}_2)} \cong 2D_2^{(\text{id})}, \tag{441}$$

namely two copies of the identity surface defect $D_2^{(\text{id})}$ of $\mathfrak{T}$. In this case, we also have

$$\mathcal{O}_1^{(\text{id})(+)} \cong \mathcal{O}_1^{(\text{id})(-)}, \tag{442}$$

and

$$\mathcal{O}_1^{(\mathrm{id})(\mathbb{Z}_2)} \cong 2\mathcal{O}_1^{(\mathrm{id})(+)} . \tag{443}$$

However, if $D_1^{(-)}$ acts faithfully and is distinct from the identity line

$$D_1^{(-)} \not\cong D_1^{(\mathrm{id})} , \tag{444}$$

then $D_2^{(\mathbb{Z}_2)}$ is a non-trivial topological surface operator of $\mathfrak{T}$ that cannot be expressed in terms of the identity surface operator. In this case, we have

$$\mathcal{O}_1^{(\mathrm{id})(+)} \not\cong \mathcal{O}_1^{(\mathrm{id})(-)} , \tag{445}$$

and

$$\mathcal{O}_1^{(\mathrm{id})(\mathbb{Z}_2)} \not\cong 2\mathcal{O}_1^{(\mathrm{id})(+)} . \tag{446}$$

We can compose the line operator $\mathcal{O}_1^{(\mathrm{id})(\mathbb{Z}_2)}$ with the topological line operator

$$D_1^{(\mathbb{Z}_2,\mathrm{id})} = \sigma\left(S_1^{(\mathbb{Z}_2,\mathrm{id})}\right) , \tag{447}$$

ending the topological surface $D_2^{(\mathbb{Z}_2)}$ to obtain a sum of the other two lines in the multiplet $\mathcal{M}_1^{(\mathrm{id})}$

$$\mathcal{O}_1^{(\mathrm{id})(\mathbb{Z}_2)} \otimes_{D_2^{(\mathbb{Z}_2)}} D_1^{(\mathbb{Z}_2,\mathrm{id})} \cong \mathcal{O}_1^{(\mathrm{id})(+)} \oplus \mathcal{O}_1^{(\mathrm{id})(-)} . \tag{448}$$

In other words, $\mathcal{O}_1^{(\mathrm{id})(\mathbb{Z}_2)}$ is a relative line defect in the absolute theory $\mathfrak{T}$ attached to the condensation surface defect $D_2^{(\mathbb{Z}_2)}$. Moreover, it can be converted into an absolute line defect

$$\mathcal{O}_1^{(\mathrm{id})(+)} \oplus \mathcal{O}_1^{(\mathrm{id})(-)} , \tag{449}$$

of the absolute theory $\mathfrak{T}$. According to the discussion of section 5.4 on condensation twisted charges of [1], this absolute line defect should realize $\mathbb{Z}_2$ 1-form symmetry of $\mathfrak{T}_\sigma$ as an induced 0-form symmetry on its worldvolume. Indeed this is the case, as we have

$$\left(\mathcal{O}_1^{(\mathrm{id})(+)} \oplus \mathcal{O}_1^{(\mathrm{id})(-)}\right) \otimes D_1^{(-)} \cong \mathcal{O}_1^{(\mathrm{id})(+)} \oplus \mathcal{O}_1^{(\mathrm{id})(-)} . \tag{450}$$

**1-charge $Q_2^{(\mathbb{Z}_2)}$.** This surface operator of $Z(2\text{-Rep}(\mathbb{Z}_2))$ can end along $\mathrm{B}_{2\text{-Rep}(\mathbb{Z}_2)}^{\mathrm{sym}}$ in three types of ends

$$\mathcal{E}_1^{(\mathbb{Z}_2)(\mathrm{id})}, \qquad \mathcal{E}_1^{(\mathbb{Z}_2)(\mathbb{Z}_2)(\pm)} , \tag{451}$$

where $\mathcal{E}_1^{(\mathbb{Z}_2)(\mathrm{id})}$ is not attached to any topological surfaces of $\mathrm{B}_{2\text{-Rep}(\mathbb{Z}_2)}^{\mathrm{sym}}$, while $\mathcal{E}_1^{(\mathbb{Z}_2)(\mathbb{Z}_2)(\pm)}$ are both attached to the topological surface $S_2^{(\mathbb{Z}_2)}$ of $\mathrm{B}_{2\text{-Rep}(\mathbb{Z}_2)}^{\mathrm{sym}}$. This corresponds to the fact that the projection $S_2^{(\mathbb{Z}_2)}$ of $Q_2^{(\mathbb{Z}_2)}$ admits one simple line operator to $S_2^{(\mathrm{id})}$ and two simple line operators back to $S_2^{(\mathbb{Z}_2)}$.

Moreover we have the fusion rule

$$\mathcal{E}_1^{(\mathbb{Z}_2)(\mathrm{id})} \otimes S_1^{(-)} \cong \mathcal{E}_1^{(\mathbb{Z}_2)(\mathrm{id})} . \tag{452}$$

Since the $\mathbb{Z}_2$ 1-form symmetry of $\mathrm{B}_{2\text{-Rep}(\mathbb{Z}_2)}^{\mathrm{sym}}$ descends to an induced $\mathbb{Z}_2$ 0-form symmetry on $\mathcal{E}_1^{(\mathbb{Z}_2)(\mathrm{id})}$, we can convert $\mathcal{E}_1^{(\mathbb{Z}_2)(\mathrm{id})}$ into a line operator attached to the condensation surface defect $S_2^{(\mathbb{Z}_2)}$. The latter line operator can be taken to be either of the other two ends $\mathcal{E}_1^{(\mathbb{Z}_2)(\mathbb{Z}_2)}$, i.e. we have the fusion rules

$$\mathcal{E}_1^{(\mathbb{Z}_2)(\mathbb{Z}_2)(\pm)} \otimes_{S_2^{(\mathbb{Z}_2)}} S_1^{(\mathbb{Z}_2,\mathrm{id})} \cong \mathcal{E}_1^{(\mathbb{Z}_2)(\mathrm{id})} . \tag{453}$$

On the other hand, the localized $\mathbb{Z}_2$ 0-form symmetry of $S_2^{(\mathbb{Z}_2)}$ generated by $S_1^{(\mathbb{Z}_2;-)}$ exchanges the other two ends

$$\mathcal{E}_1^{(\mathbb{Z}_2)(\mathbb{Z}_2)(+)} \otimes_{S_2^{(\mathbb{Z}_2)}} S_1^{(\mathbb{Z}_2;-)} \cong \mathcal{E}_1^{(\mathbb{Z}_2)(\mathbb{Z}_2)(-)} \,. \tag{454}$$

In other words, $\mathcal{E}_1^{(\mathbb{Z}_2)(\mathbb{Z}_2)(\pm)}$ form a non-trivial 2-representation (or equivalently 1-charge) under localized $\mathbb{Z}_2$ 0-form symmetry of $S_2^{(\mathbb{Z}_2)}$.

Consequently, a multiplet $\mathcal{M}_1^{(\mathbb{Z}_2)}$ of line operators of a $\mathbb{Z}_2$ 1-form symmetric 3d theory $\mathfrak{T}_\sigma$ transforming in the 1-charge $Q_2^{(\mathbb{Z}_2)}$ comprises of three line operators

$$\mathcal{O}_1^{(\mathbb{Z}_2)(\text{id})} \,, \qquad \mathcal{O}_1^{(\mathbb{Z}_2)(\mathbb{Z}_2)(\pm)} \,. \tag{455}$$

Here $\mathcal{O}_1^{(\mathbb{Z}_2)(\text{id})}$ is a genuine line operator of $\mathfrak{T}$ which, as a consequence of the fusion rule (452), is left invariant under fusion by $D_1^{(-)}$

$$\mathcal{O}_1^{(\mathbb{Z}_2)(\text{id})} \otimes D_1^{(-)} \cong \mathcal{O}_1^{(\mathbb{Z}_2)(\text{id})} \,, \tag{456}$$

and hence carries induced $\mathbb{Z}_2$ 0-form symmetry. Note that unlike the case of the multiplet $\mathcal{M}_1^{(\text{id})}$, this fusion rule holds irrespective of whether the $\mathbb{Z}_2$ 1-form symmetry is realized faithfully on $\mathfrak{T}$ or not.

The fusion rules (453) on $B_{2\text{-Rep}(\mathbb{Z}_2)}^{\text{sym}}$ descends to the following fusion rules in $\mathfrak{T}$

$$\mathcal{O}_1^{(\mathbb{Z}_2)(\mathbb{Z}_2)(\pm)} \otimes_{D_2^{(\mathbb{Z}_2)}} D_1^{(\mathbb{Z}_2,\text{id})} \cong \mathcal{O}_1^{(\mathbb{Z}_2)(\text{id})} \,. \tag{457}$$

Again we can interpret $\mathcal{O}_1^{(\mathbb{Z}_2)(\text{id})}$ as an absolute defect of the absolute theory $\mathfrak{T}$ arising from the relative defects $\mathcal{O}_1^{(\mathbb{Z}_2)(\mathbb{Z}_2)(\pm)}$ of the absolute theory $\mathfrak{T}$ attached to the condensation defect $D_2^{(\mathbb{Z}_2)}$. In this case, the absolute defect $\mathcal{O}_1^{(\mathbb{Z}_2)(\text{id})}$ is simple, unlike the case with multiplet $\mathcal{M}_1^{(\text{id})}$.

We also have fusion rule

$$\mathcal{O}_1^{(\mathbb{Z}_2)(\mathbb{Z}_2)(+)} \otimes_{D_2^{(\mathbb{Z}_2)}} D_1^{(\mathbb{Z}_2;-)} \cong \mathcal{O}_1^{(\mathbb{Z}_2)(\mathbb{Z}_2)(-)} \,. \tag{458}$$

**1-charge $Q_2^{(V)}$.** The possible ends of $Q_2^{(V)}$ along the 3d symmetry boundary $B_{2\text{-Rep}(\mathbb{Z}_2)}^{\text{sym}}$ are

$$\mathcal{E}_1^{(V)(+)} \,, \qquad \mathcal{E}_1^{(V)(-)} \,, \qquad \mathcal{E}_1^{(V)(\mathbb{Z}_2)} \,, \tag{459}$$

where $\mathcal{E}_1^{(V)(\pm)}$ are not attached to any topological surfaces of $B_{2\text{-Rep}(\mathbb{Z}_2)}^{\text{sym}}$, while $\mathcal{E}_1^{(V)(\mathbb{Z}_2)}$ is attached to the topological surface $S_2^{(\mathbb{Z}_2)}$ of $B_{2\text{-Rep}(\mathbb{Z}_2)}^{\text{sym}}$. Moreover, we have the fusion rule

$$\mathcal{E}_1^{(V)(+)} \otimes S_1^{(-)} \cong \mathcal{E}_1^{(V)(-)} \,, \tag{460}$$

and the fusion rules

$$\begin{aligned}
\mathcal{E}_1^{(V)(\mathbb{Z}_2)} \otimes_{S_2^{(\mathbb{Z}_2)}} S_1^{(\mathbb{Z}_2,\text{id})} &\cong \mathcal{E}_1^{(V)(+)} \oplus \mathcal{E}_1^{(V)(-)} \,, \\
\mathcal{E}_1^{(V)(\mathbb{Z}_2)} \otimes_{S_2^{(\mathbb{Z}_2)}} S_1^{(\mathbb{Z}_2;-)} &\cong \mathcal{E}_1^{(V)(\mathbb{Z}_2)} \,.
\end{aligned} \tag{461}$$

Consequently, a multiplet $\mathcal{M}_1^{(V)}$ of line operators of a $\mathbb{Z}_2$ 1-form symmetric 3d theory $\mathfrak{T}_\sigma$ transforming in the 1-charge $Q_2^{(V)}$ comprises of three line operators

$$\mathcal{O}_1^{(V)(+)} \,, \qquad \mathcal{O}_1^{(V)(-)} \,, \qquad \mathcal{O}_1^{(V)(\mathbb{Z}_2)} \,. \tag{462}$$

Here $\mathcal{O}_1^{(V)(\pm)}$ are genuine line operators of $\mathfrak{T}$ interchanged by fusion with $D_1^{(-)}$

$$\mathcal{O}_1^{(V)(+)} \otimes D_1^{(-)} \cong \mathcal{O}_1^{(V)(-)}. \tag{463}$$

Both these line operators $\mathcal{O}_1^{(V)(\pm)}$ carry a non-trivial charge under $\mathbb{Z}_2$ 1-form symmetry. On the other hand, $\mathcal{O}_1^{(V)(\mathbb{Z}_2)}$ is a line attached to the topological surface $D_2^{(\mathbb{Z}_2)}$ which is related to $\mathcal{O}_1^{(V)(\pm)}$ via

$$\mathcal{O}_1^{(V)(\mathbb{Z}_2)} \otimes_{D_2^{(\mathbb{Z}_2)}} D_1^{(\mathbb{Z}_2,\mathrm{id})} \cong \mathcal{O}_1^{(V)(+)} \oplus \mathcal{O}_1^{(V)(-)}. \tag{464}$$

Finally, the $\mathbb{Z}_2$ 0-form localized symmetry of $D_2^{(\mathbb{Z}_2)}$ descends to an induced 0-form symmetry of $\mathcal{O}_1^{(V)(\mathbb{Z}_2)}$

$$\mathcal{O}_1^{(V)(\mathbb{Z}_2)} \otimes_{D_2^{(\mathbb{Z}_2)}} D_1^{(\mathbb{Z}_2;-)} \cong \mathcal{O}_1^{(V)(\mathbb{Z}_2)}. \tag{465}$$

**1-charge $Q_2^{(V\mathbb{Z}_2)}$.** The possible ends of $Q_2^{(V\mathbb{Z}_2)}$ along the 3d symmetry boundary $\mathrm{B}_{\text{2-Rep}(\mathbb{Z}_2)}^{\mathrm{sym}}$ are

$$\mathcal{E}_1^{(V\mathbb{Z}_2)(\mathrm{id})}, \qquad \mathcal{E}_1^{(V\mathbb{Z}_2)(\mathbb{Z}_2)(\pm)}, \tag{466}$$

where $\mathcal{E}_1^{(V\mathbb{Z}_2)(\mathrm{id})}$ is not attached to any topological surface of $\mathrm{B}_{\text{2-Rep}(\mathbb{Z}_2)}^{\mathrm{sym}}$, while $\mathcal{E}_1^{(V\mathbb{Z}_2)(\mathbb{Z}_2)(\pm)}$ are attached to the topological surface $S_2^{(\mathbb{Z}_2)}$ of $\mathrm{B}_{\text{2-Rep}(\mathbb{Z}_2)}^{\mathrm{sym}}$. Moreover, we have the fusion rules

$$\begin{aligned}
\mathcal{E}_1^{(V\mathbb{Z}_2)(\mathrm{id})} \otimes S_1^{(-)} &\cong \mathcal{E}_1^{(V\mathbb{Z}_2)(\mathrm{id})}, \\
\mathcal{E}_1^{(V\mathbb{Z}_2)(\mathbb{Z}_2)(\pm)} \otimes_{S_2^{(\mathbb{Z}_2)}} S_1^{(\mathbb{Z}_2,\mathrm{id})} &\cong \mathcal{E}_1^{(V\mathbb{Z}_2)(\mathrm{id})}, \\
\mathcal{E}_1^{(V\mathbb{Z}_2)(\mathbb{Z}_2)(+)} \otimes_{S_2^{(\mathbb{Z}_2)}} S_1^{(\mathbb{Z}_2;-)} &\cong \mathcal{E}_1^{(V\mathbb{Z}_2)(\mathbb{Z}_2)(-)}.
\end{aligned} \tag{467}$$

Consequently, a multiplet $\mathcal{M}_1^{(V\mathbb{Z}_2)}$ of line operators of a $\mathbb{Z}_2$ 1-form symmetric 3d theory $\mathfrak{T}_\sigma$ transforming in the 1-charge $Q_2^{(V\mathbb{Z}_2)}$ comprises of three line operators

$$\mathcal{O}_1^{(V\mathbb{Z}_2)(\mathrm{id})}, \qquad \mathcal{O}_1^{(V\mathbb{Z}_2)(\mathbb{Z}_2)(\pm)}. \tag{468}$$

Here $\mathcal{O}_1^{(V\mathbb{Z}_2)(\mathrm{id})}$ is a genuine line operator of $\mathfrak{T}$ kept invariant by the fusion with $D_1^{(-)}$

$$\mathcal{O}_1^{(V\mathbb{Z}_2)(\mathrm{id})} \otimes D_1^{(-)} \cong \mathcal{O}_1^{(V\mathbb{Z}_2)(\mathrm{id})}. \tag{469}$$

Moreover, this line operator $\mathcal{O}_1^{(V\mathbb{Z}_2)(\mathrm{id})}$ carries a non-trivial charge under $\mathbb{Z}_2$ 1-form symmetry. On the other hand, $\mathcal{O}_1^{(V\mathbb{Z}_2)(\mathbb{Z}_2)(\pm)}$ are lines attached to the topological surface $D_2^{(\mathbb{Z}_2)}$ which are related to $\mathcal{O}_1^{(V\mathbb{Z}_2)(\mathrm{id})}$ via

$$\mathcal{O}_1^{(V\mathbb{Z}_2)(\mathbb{Z}_2)(\pm)} \otimes_{D_2^{(\mathbb{Z}_2)}} D_1^{(\mathbb{Z}_2,\mathrm{id})} \cong \mathcal{O}_1^{(V\mathbb{Z}_2)(\mathrm{id})}. \tag{470}$$

These lines are permuted by fusion with localized $\mathbb{Z}_2$ 0-form symmetry of $D_2^{(\mathbb{Z}_2)}$

$$\mathcal{O}_1^{(V\mathbb{Z}_2)(\mathbb{Z}_2)(+)} \otimes_{D_2^{(\mathbb{Z}_2)}} D_1^{(\mathbb{Z}_2;-)} \cong \mathcal{O}_1^{(V\mathbb{Z}_2)(\mathbb{Z}_2)(-)}. \tag{471}$$

### 5.3.2 0-charges

**0-charge $Q_1^{(\text{id})}$.** This is the identity line operator of the 4d SymTFT $Z(2\text{-Rep}(\mathbb{Z}_2))$. As we are viewing it as a line operator living on identity surface operator $Q_2^{(\text{id})}$ of $Z(2\text{-Rep}(\mathbb{Z}_2))$, we have to describe the ends of $Q_1^{(\text{id})}$ along ends of $Q_2^{(\text{id})}$, possibly attached to other topological line operators of $B_{2\text{-Rep}(\mathbb{Z}_2)}^{\text{sym}}$. Thus, in total $Q_1^{(\text{id})}$ has the following ends along $B_{2\text{-Rep}(\mathbb{Z}_2)}^{\text{sym}}$:

$$
\begin{array}{ccc}
\mathcal{E}_0^{(\text{id})(++)}, & \mathcal{E}_0^{(\text{id})(--)}, & \mathcal{E}_0^{(\text{id})(\mathbb{Z}_2)(+)}, \\
\mathcal{E}_0^{(\text{id})(+-)}, & \mathcal{E}_0^{(\text{id})(-+)}, & \mathcal{E}_0^{(\text{id})(\mathbb{Z}_2)(-)}.
\end{array}
\tag{472}
$$

The ends $\mathcal{E}_0^{(\text{id})(++)}$, $\mathcal{E}_0^{(\text{id})(--)}$ and $\mathcal{E}_0^{(\text{id})(\mathbb{Z}_2)(+)}$ can respectively be identified with identity local operators of the ends (433) of $Q_2^{(\text{id})}$. On the other hand, the end $\mathcal{E}_0^{(\text{id})(+-)}$ transitions $\mathcal{E}_1^{(\text{id})(+)}$ into $\mathcal{E}_1^{(\text{id})(-)}$, and is additionally attached to the topological line operator $S_1^{(-)}$ of $B_{2\text{-Rep}(\mathbb{Z}_2)}^{\text{sym}}$. Similarly, the end $\mathcal{E}_0^{(\text{id})(-+)}$ transitions $\mathcal{E}_1^{(\text{id})(-)}$ into $\mathcal{E}_1^{(\text{id})(+)}$, and is additionally attached to the topological line operator $S_1^{(-)}$ of $B_{2\text{-Rep}(\mathbb{Z}_2)}^{\text{sym}}$. Finally, the end $\mathcal{E}_0^{(\text{id})(\mathbb{Z}_2)(-)}$ lives along $\mathcal{E}_1^{(\text{id})(\mathbb{Z}_2)}$, and is additionally attached to the topological line operator $S_1^{(\mathbb{Z}_2;-)}$ of $S_2^{(\mathbb{Z}_2)}$.

Consequently, a multiplet $\mathcal{M}_0^{(\text{id})}$ of local operators of a $\mathbb{Z}_2$ 1-form symmetric 3d theory $\mathfrak{T}_\sigma$ transforming in the 0-charge $Q_1^{(\text{id})}$ comprises of six local operators

$$
\begin{array}{ccc}
\mathcal{O}_0^{(\text{id})(++)}, & \mathcal{O}_0^{(\text{id})(--)}, & \mathcal{O}_0^{(\text{id})(\mathbb{Z}_2)(+)}, \\
\mathcal{O}_0^{(\text{id})(+-)}, & \mathcal{O}_0^{(\text{id})(-+)}, & \mathcal{O}_0^{(\text{id})(\mathbb{Z}_2)(-)},
\end{array}
\tag{473}
$$

where $\mathcal{O}_0^{(\text{id})(++)}$ lives between two genuine line operators $\mathcal{O}_1^{(\text{id})(+)(1)}$ and $\mathcal{O}_1^{(\text{id})(+)(2)}$ living in multiplets

$$
\begin{aligned}
\mathcal{M}_1^{(\text{id})(1)} &= \left\{ \mathcal{O}_1^{(\text{id})(+)(1)}, \mathcal{O}_1^{(\text{id})(-)(1)}, \mathcal{O}_1^{(\text{id})(\mathbb{Z}_2)(1)} \right\}, \\
\mathcal{M}_1^{(\text{id})(2)} &= \left\{ \mathcal{O}_1^{(\text{id})(+)(2)}, \mathcal{O}_1^{(\text{id})(-)(2)}, \mathcal{O}_1^{(\text{id})(\mathbb{Z}_2)(2)} \right\},
\end{aligned}
\tag{474}
$$

both transforming in the same 1-charge $Q_2^{(\text{id})}$. Similarly, $\mathcal{O}_0^{(\text{id})(--)}$ lives between the genuine line operators $\mathcal{O}_1^{(\text{id})(-)(1)}$ and $\mathcal{O}_1^{(\text{id})(-)(2)}$, $\mathcal{O}_0^{(\text{id})(\mathbb{Z}_2)(+)}$ lives between the $S_2^{(\mathbb{Z}_2)}$-twisted sector line operators $\mathcal{O}_1^{(\text{id})(\mathbb{Z}_2)(1)}$ and $\mathcal{O}_1^{(\text{id})(\mathbb{Z}_2)(2)}$. These operators $\mathcal{O}_0^{(\text{id})(++)}$, $\mathcal{O}_0^{(\text{id})(--)}$ and $\mathcal{O}_0^{(\text{id})(\mathbb{Z}_2)(+)}$ are not attached to topological line operators participating in the 2-Rep$(\mathbb{Z}_2)$ symmetry of $\mathfrak{T}_\sigma$. On the other hand, the local operator $\mathcal{O}_0^{(\text{id})(+-)}$ lives between the genuine line operators $\mathcal{O}_1^{(\text{id})(+)(1)}$ and $\mathcal{O}_1^{(\text{id})(-)(2)}$, and is attached additionally to topological line operator $D_1^{(-)}$. Similarly, the local operator $\mathcal{O}_0^{(\text{id})(-+)}$ lives between the genuine line operators $\mathcal{O}_1^{(\text{id})(-)(1)}$ and $\mathcal{O}_1^{(\text{id})(+)(2)}$, and is attached additionally to topological line operator $D_1^{(-)}$. Finally, the local operator $\mathcal{O}_0^{(\text{id})(\mathbb{Z}_2)(-)}$ lives between the $D_2^{(\mathbb{Z}_2)}$-twisted sector line operators $\mathcal{O}_1^{(\text{id})(\mathbb{Z}_2)(1)}$ and $\mathcal{O}_1^{(\text{id})(\mathbb{Z}_2)(2)}$, and is attached additionally to topological line operator

$$
D_1^{(\mathbb{Z}_2;-)} := \sigma\left( S_1^{(\mathbb{Z}_2;-)} \right),
\tag{475}
$$

on the surface $D_2^{(\mathbb{Z}_2)}$ attached to $\mathcal{O}_1^{(\text{id})(\mathbb{Z}_2)(1)}$ and $\mathcal{O}_1^{(\text{id})(\mathbb{Z}_2)(2)}$. This is summarized as follows (now understood in the theory $\mathfrak{T}_\sigma$)

$$
\tag{476}
$$

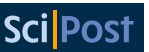

The local operators $\mathcal{O}_0^{(\text{id})(\mathbb{Z}_2)(\pm)}$ are related to the local operators $\mathcal{O}_0^{(\text{id})(++)}$ and $\mathcal{O}_0^{(\text{id})(--)}$ via sandwich constructions involving the condensation surface $D_2^{(\mathbb{Z}_2)}$ inside the 3d theory $\mathfrak{T}$ as shown here

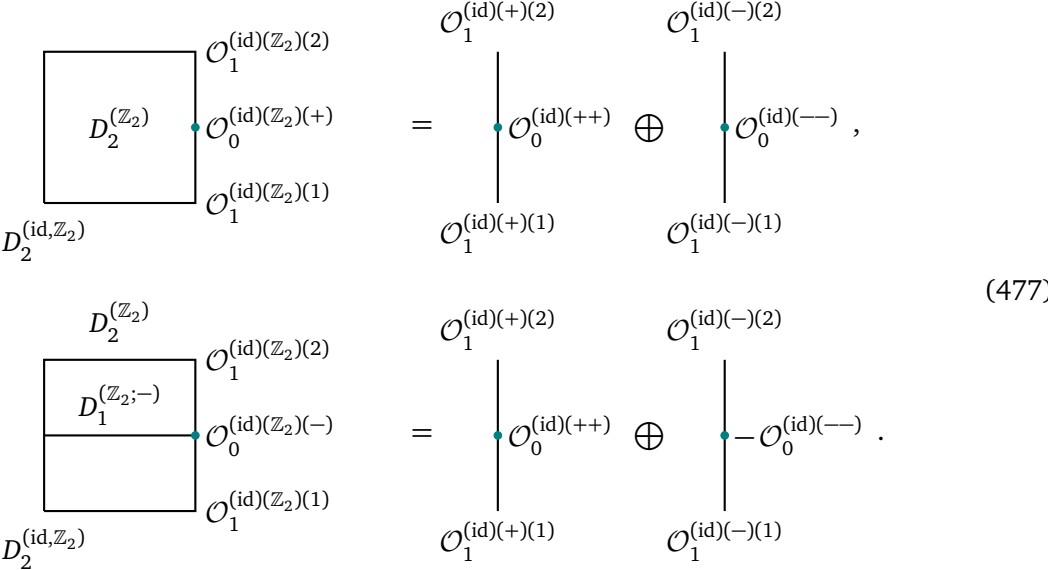

$$(477)$$

A key property of the local operator $\mathcal{O}_0^{(\text{id})(\mathbb{Z}_2)(+)}$ is that it commutes with the action of $\mathbb{Z}_2$ 0-form symmetries of $\mathcal{O}_1^{(\text{id})(\mathbb{Z}_2)(1)}$ and $\mathcal{O}_1^{(\text{id})(\mathbb{Z}_2)(2)}$ induced from the localized $\mathbb{Z}_2$ 0-form symmetry of $D_2^{(\mathbb{Z}_2)}$

$$(478)$$

In other words, the operator $\mathcal{O}_0^{(\text{id})(\mathbb{Z}_2)(+)}$ is uncharged under the induced $\mathbb{Z}_2$ 0-form symmetry.

**0-charge $Q_1^{(-)}$.** This line operator of the 4d SymTFT $Z(2\text{-Rep}(\mathbb{Z}_2))$ has the following ends along $B_{2\text{-Rep}(\mathbb{Z}_2)}^{\text{sym}}$

$$
\begin{aligned}
\mathcal{E}_0^{(-)(++)}, && \mathcal{E}_0^{(-)(--)}, && \mathcal{E}_0^{(-)(\mathbb{Z}_2)(+)}, \\
\mathcal{E}_0^{(-)(+-)}, && \mathcal{E}_0^{(-)(-+)}, && \mathcal{E}_0^{(-)(\mathbb{Z}_2)(-)}.
\end{aligned}
\tag{479}
$$

The ends $\mathcal{E}_0^{(-)(++)}$ and $\mathcal{E}_0^{(-)(--)}$ live respectively along $\mathcal{E}_1^{(\text{id})(+)}$ and $\mathcal{E}_1^{(\text{id})(-)}$, and are additionally attached to topological line operator $S_1^{(-)}$ of $B_{2\text{-Rep}(\mathbb{Z}_2)}^{\text{sym}}$. The end $\mathcal{E}_0^{(-)(\mathbb{Z}_2)(+)}$ lives along $\mathcal{E}_1^{(\text{id})(\mathbb{Z}_2)}$, and is not additionally attached to $S_1^{(\mathbb{Z}_2;-)}$. The end $\mathcal{E}_0^{(-)(\mathbb{Z}_2)(-)}$ lives along $\mathcal{E}_1^{(\text{id})(\mathbb{Z}_2)}$, and is additionally attached to $S_1^{(\mathbb{Z}_2;-)}$. On the other hand, the end $\mathcal{E}_0^{(-)(+-)}$ transitions $\mathcal{E}_1^{(\text{id})(+)}$ into $\mathcal{E}_1^{(\text{id})(-)}$, and is additionally not attached to the topological line operator $S_1^{(-)}$ of $B_{2\text{-Rep}(\mathbb{Z}_2)}^{\text{sym}}$. Similarly, the end $\mathcal{E}_0^{(-)(-+)}$ transitions $\mathcal{E}_1^{(\text{id})(-)}$ into $\mathcal{E}_1^{(\text{id})(+)}$, and is additionally not attached to the topological line operator $S_1^{(-)}$ of $B_{2\text{-Rep}(\mathbb{Z}_2)}^{\text{sym}}$. Finally, the end $\mathcal{E}_0^{(-)(\mathbb{Z}_2)(+)}$ is non-trivially

charged under the $\mathbb{Z}_2$ 0-form symmetry of $\mathcal{E}_1^{(\mathrm{id})(\mathbb{Z}_2)}$ induced from the localized 0-form symmetry of $S_2^{(\mathbb{Z}_2)}$.

Consequently, a multiplet $\mathcal{M}_0^{(-)}$ of local operators of a $\mathbb{Z}_2$ 1-form symmetric 3d theory $\mathfrak{T}_\sigma$ transforming in the 0-charge $\boldsymbol{Q}_1^{(-)}$ comprises of six local operators

$$
\begin{aligned}
&\mathcal{O}_0^{(-)(++)}, && \mathcal{O}_0^{(-)(--)}, && \mathcal{O}_0^{(-)(\mathbb{Z}_2)(+)}, \\
&\mathcal{O}_0^{(-)(+-)}, && \mathcal{O}_0^{(-)(-+)}, && \mathcal{O}_0^{(-)(\mathbb{Z}_2)(-)},
\end{aligned}
\tag{480}
$$

where $\mathcal{O}_0^{(-)(++)}$ lives between two genuine line operators $\mathcal{O}_1^{(\mathrm{id})(+)(1)}$ and $\mathcal{O}_1^{(\mathrm{id})(+)(2)}$ living in multiplets

$$
\begin{aligned}
\mathcal{M}_1^{(\mathrm{id})(1)} &= \left\{ \mathcal{O}_1^{(\mathrm{id})(+)(1)}, \mathcal{O}_1^{(\mathrm{id})(-)(1)}, \mathcal{O}_1^{(\mathrm{id})(\mathbb{Z}_2)(1)} \right\}, \\
\mathcal{M}_1^{(\mathrm{id})(2)} &= \left\{ \mathcal{O}_1^{(\mathrm{id})(+)(2)}, \mathcal{O}_1^{(\mathrm{id})(-)(2)}, \mathcal{O}_1^{(\mathrm{id})(\mathbb{Z}_2)(2)} \right\},
\end{aligned}
\tag{481}
$$

both transforming in the same 1-charge $\boldsymbol{Q}_2^{(\mathrm{id})}$, and is additionally attached to the topological line operator $D_1^{(S)}$. Similarly, $\mathcal{O}_0^{(-)(--)}$ lives between the genuine line operators $\mathcal{O}_1^{(\mathrm{id})(-)(1)}$ and $\mathcal{O}_1^{(\mathrm{id})(-)(2)}$, and is additionally attached to topological line operator $D_1^{(S)}$. $\mathcal{O}_0^{(-)(\mathbb{Z}_2)(+)}$ lives between the $S_2^{(\mathbb{Z}_2)}$-twisted sector line operators $\mathcal{O}_1^{(\mathrm{id})(\mathbb{Z}_2)(1)}$ and $\mathcal{O}_1^{(\mathrm{id})(\mathbb{Z}_2)(2)}$, and is not additionally attached to $D_1^{(\mathbb{Z}_2;-)}$. On the other hand, the local operator $\mathcal{O}_0^{(-)(+-)}$ lives between the genuine line operators $\mathcal{O}_1^{(\mathrm{id})(+)(1)}$ and $\mathcal{O}_1^{(\mathrm{id})(-)(2)}$, and is not attached additionally to topological line operator $D_1^{(-)}$. Similarly, the local operator $\mathcal{O}_0^{(-)(-+)}$ lives between the genuine line operators $\mathcal{O}_1^{(\mathrm{id})(-)(1)}$ and $\mathcal{O}_1^{(\mathrm{id})(+)(2)}$, and is not attached additionally to topological line operator $D_1^{(-)}$. Finally, the local operator $\mathcal{O}_0^{(-)(\mathbb{Z}_2)(-)}$ lives between the $S_2^{(\mathbb{Z}_2)}$-twisted sector line operators $\mathcal{O}_1^{(\mathrm{id})(\mathbb{Z}_2)(1)}$ and $\mathcal{O}_1^{(\mathrm{id})(\mathbb{Z}_2)(2)}$, and is attached additionally to $D_1^{(\mathbb{Z}_2;-)}$.

The local operators $\mathcal{O}_0^{(-)(\mathbb{Z}_2)(\pm)}$ are related to the local operators $\mathcal{O}_0^{(-)(+-)}$ and $\mathcal{O}_0^{(-)(-+)}$ via sandwich constructions involving condensation surface $D_2^{(\mathbb{Z}_2)}$ inside the 3d theory $\mathfrak{T}$.

A key property of the local operator $\mathcal{O}_0^{(-)(\mathbb{Z}_2)(+)}$ is that it anti-commutes with the action of $\mathbb{Z}_2$ 0-form symmetries of $\mathcal{O}_1^{(\mathrm{id})(\mathbb{Z}_2)(1)}$ and $\mathcal{O}_1^{(\mathrm{id})(\mathbb{Z}_2)(2)}$ induced from the localized $\mathbb{Z}_2$ 0-form symmetry of $D_2^{(\mathbb{Z}_2)}$ as shown here:

$$
\tag{482}
$$

**0-charge $\boldsymbol{Q}_1^{(\mathbb{Z}_2;\mathrm{id})}$.** This line operator of $Z(2\text{-Rep}(\mathbb{Z}_2))$ has six ends along $B_{2\text{-Rep}(\mathbb{Z}_2)}^{\mathrm{sym}}$

$$
\begin{aligned}
&\mathcal{E}_0^{(\mathbb{Z}_2;\mathrm{id})(\mathrm{id})(+)}, && \mathcal{E}_0^{(\mathbb{Z}_2;\mathrm{id})(\mathbb{Z}_2)(++)}, && \mathcal{E}_0^{(\mathbb{Z}_2;\mathrm{id})(\mathbb{Z}_2)(--)}, \\
&\mathcal{E}_0^{(\mathbb{Z}_2;\mathrm{id})(\mathrm{id})(-)}, && \mathcal{E}_0^{(\mathbb{Z}_2;\mathrm{id})(\mathbb{Z}_2)(+-)}, && \mathcal{E}_0^{(\mathbb{Z}_2;\mathrm{id})(\mathbb{Z}_2)(-+)},
\end{aligned}
\tag{483}
$$

where $\mathcal{E}_0^{(\mathbb{Z}_2;\mathrm{id})(\mathrm{id})(\pm)}$ live along the end $\mathcal{E}_1^{(\mathbb{Z}_2)(\mathrm{id})}$ of $\boldsymbol{Q}_2^{(\mathbb{Z}_2)}$, $\mathcal{E}_0^{(\mathbb{Z}_2;\mathrm{id})(\mathbb{Z}_2)(ss')}$ transitions between ends $\mathcal{E}_1^{(\mathbb{Z}_2)(\mathbb{Z}_2)(s)}$ and $\mathcal{E}_1^{(\mathbb{Z}_2)(\mathbb{Z}_2)(s')}$ of $\boldsymbol{Q}_2^{(\mathbb{Z}_2)}$, for $s, s' \in \{+, -\}$. Additionally, $\mathcal{E}_0^{(\mathbb{Z}_2;\mathrm{id})(\mathrm{id})(+)}$, $\mathcal{E}_0^{(\mathbb{Z}_2;\mathrm{id})(\mathbb{Z}_2)(++)}$ and $\mathcal{E}_0^{(\mathbb{Z}_2;\mathrm{id})(\mathbb{Z}_2)(--)}$ are not attached to any non-trivial line operators of $B_{2\text{-Rep}(\mathbb{Z}_2)}^{\mathrm{sym}}$. On the

other hand, $\mathcal{E}_0^{(\mathbb{Z}_2;\text{id})(\text{id})(-)}$ is attached to $S_1^{(-)}$, and $\mathcal{E}_0^{(\mathbb{Z}_2;\text{id})(\mathbb{Z}_2)(+-)}$ and $\mathcal{E}_0^{(\mathbb{Z}_2;\text{id})(\mathbb{Z}_2)(-+)}$ are attached to $S_1^{(\mathbb{Z}_2;-)}$. The operator $\mathcal{E}_0^{(\mathbb{Z}_2;\text{id})(\text{id})(+)}$ is uncharged under the induced $\mathbb{Z}_2$ 0-form symmetry of $\mathcal{E}_1^{(\mathbb{Z}_2)(\text{id})}$. The ends $\mathcal{E}_0^{(\mathbb{Z}_2;\text{id})(\mathbb{Z}_2)(ss')}$ can be converted into the end $\mathcal{E}_0^{(\mathbb{Z}_2;\text{id})(\text{id})(+)}$ by sandwich construction inside $\mathsf{B}_{2\text{-Rep}(\mathbb{Z}_2)}^{\text{sym}}$ involving condensation defect $S_2^{(\mathbb{Z}_2)}$.

Correspondingly, a multiplet $\mathcal{M}_0^{(\mathbb{Z}_2;\text{id})}$ of local operators of a $\mathbb{Z}_2$ 1-form symmetric 3d theory $\mathfrak{T}_\sigma$ transforming in the 0-charge $Q_1^{(\mathbb{Z}_2;\text{id})}$ comprises of local operators

$$\begin{array}{ccc} \mathcal{O}_0^{(\mathbb{Z}_2;\text{id})(\text{id})(++)}, & \mathcal{O}_0^{(\mathbb{Z}_2;\text{id})(\mathbb{Z}_2)(++)}, & \mathcal{O}_0^{(\mathbb{Z}_2;\text{id})(\mathbb{Z}_2)(--)}, \\ \mathcal{O}_0^{(\mathbb{Z}_2;\text{id})(\text{id})(-)}, & \mathcal{O}_0^{(\mathbb{Z}_2;\text{id})(\mathbb{Z}_2)(+-)}, & \mathcal{O}_0^{(\mathbb{Z}_2;\text{id})(\mathbb{Z}_2)(-+)}, \end{array} \tag{484}$$

where $\mathcal{O}_0^{(\mathbb{Z}_2;\text{id})(\text{id})(\pm)}$ live between two genuine line operators $\mathcal{O}_1^{(\mathbb{Z}_2)(\text{id})(1)}$ and $\mathcal{O}_1^{(\mathbb{Z}_2)(\text{id})(2)}$ living in multiplets

$$\begin{aligned} \mathcal{M}_1^{(\mathbb{Z}_2)(1)} &= \left\{ \mathcal{O}_1^{(\mathbb{Z}_2)(\text{id})(1)}, \mathcal{O}_1^{(\mathbb{Z}_2)(\mathbb{Z}_2)(+)(1)}, \mathcal{O}_1^{(\mathbb{Z}_2)(\mathbb{Z}_2)(-)(1)} \right\}, \\ \mathcal{M}_1^{(\mathbb{Z}_2)(2)} &= \left\{ \mathcal{O}_1^{(\mathbb{Z}_2)(\text{id})(2)}, \mathcal{O}_1^{(\mathbb{Z}_2)(\mathbb{Z}_2)(+)(2)}, \mathcal{O}_1^{(\mathbb{Z}_2)(\mathbb{Z}_2)(-)(2)} \right\}, \end{aligned} \tag{485}$$

both transforming in the same 1-charge $Q_2^{(\mathbb{Z}_2)}$, and $\mathcal{O}_0^{(\mathbb{Z}_2;\text{id})(\mathbb{Z}_2)(ss')}$ transition between the $S_2^{(\mathbb{Z}_2)}$-twisted sector lines operators $\mathcal{O}_1^{(\mathbb{Z}_2)(\mathbb{Z}_2)(s)(1)}$ and $\mathcal{O}_1^{(\mathbb{Z}_2)(\mathbb{Z}_2)(s')(2)}$ in the above described multiplets. The operators $\mathcal{O}_0^{(\mathbb{Z}_2;\text{id})(\text{id})(+)}$, $\mathcal{O}_0^{(\mathbb{Z}_2;\text{id})(\mathbb{Z}_2)(++)}$ and $\mathcal{O}_0^{(\mathbb{Z}_2;\text{id})(\mathbb{Z}_2)(--)}$ are not attached to any non-identity line operators. The operator $\mathcal{O}_0^{(\mathbb{Z}_2;\text{id})(\text{id})(-)}$ is attached to $D_1^{(-)}$, and the operators $\mathcal{O}_0^{(\mathbb{Z}_2;\text{id})(\mathbb{Z}_2)(+-)}$ and $\mathcal{O}_0^{(\mathbb{Z}_2;\text{id})(\mathbb{Z}_2)(-+)}$ are attached to $D_1^{(\mathbb{Z}_2;-)}$.

The local operators $\mathcal{O}_0^{(\mathbb{Z}_2;\text{id})(\mathbb{Z}_2)(ss')}$ can be converted into the operator $\mathcal{O}_0^{(\mathbb{Z}_2;\text{id})(\text{id})(+)}$ by sandwich construction involving condensation surface $D_2^{(\mathbb{Z}_2)}$ inside the 3d theory $\mathfrak{T}$.

A key property of the local operator $\mathcal{O}_0^{(\mathbb{Z}_2;\text{id})(\text{id})(+)}$ is that it commutes with the action of $\mathbb{Z}_2$ 0-form symmetries induced on lines $\mathcal{O}_1^{(\mathbb{Z}_2)(\text{id})(1)}$ and $\mathcal{O}_1^{(\mathbb{Z}_2)(\text{id})(2)}$ from the $\mathbb{Z}_2$ 1-form symmetry of $\mathfrak{T}_\sigma$:

$$\tag{486}$$

**0-charge $Q_1^{(\mathbb{Z}_2;-)}$.** This line operator of $Z(2\text{-Rep}(\mathbb{Z}_2))$ has six ends along $\mathsf{B}_{2\text{-Rep}(\mathbb{Z}_2)}^{\text{sym}}$

$$\begin{array}{ccc} \mathcal{E}_0^{(\mathbb{Z}_2;-)(\text{id})(+)}, & \mathcal{E}_0^{(\mathbb{Z}_2;-)(\mathbb{Z}_2)(++)}, & \mathcal{E}_0^{(\mathbb{Z}_2;-)(\mathbb{Z}_2)(--)}, \\ \mathcal{E}_0^{(\mathbb{Z}_2;-)(\text{id})(-)}, & \mathcal{E}_0^{(\mathbb{Z}_2;-)(\mathbb{Z}_2)(+-)}, & \mathcal{E}_0^{(\mathbb{Z}_2;-)(\mathbb{Z}_2)(-+)}, \end{array} \tag{487}$$

where $\mathcal{E}_0^{(\mathbb{Z}_2;-)(\text{id})(\pm)}$ live along the end $\mathcal{E}_1^{(\mathbb{Z}_2)(\text{id})}$ of $Q_2^{(\mathbb{Z}_2)}$, $\mathcal{E}_0^{(\mathbb{Z}_2;-)(\mathbb{Z}_2)(ss')}$ transitions between ends $\mathcal{E}_1^{(\mathbb{Z}_2)(\mathbb{Z}_2)(s)}$ and $\mathcal{E}_1^{(\mathbb{Z}_2)(\mathbb{Z}_2)(s')}$ of $Q_2^{(\mathbb{Z}_2)}$, for $s, s' \in \{+, -\}$. Additionally, $\mathcal{E}_0^{(\mathbb{Z}_2;-)(\text{id})(+)}$, $\mathcal{E}_0^{(\mathbb{Z}_2;-)(\mathbb{Z}_2)(+-)}$ and $\mathcal{E}_0^{(\mathbb{Z}_2;-)(\mathbb{Z}_2)(-+)}$ are not attached to any non-trivial line operators of $\mathsf{B}_{2\text{-Rep}(\mathbb{Z}_2)}^{\text{sym}}$. On the other hand, $\mathcal{E}_0^{(\mathbb{Z}_2;-)(\text{id})(-)}$ is attached to $S_1^{(-)}$, and $\mathcal{E}_0^{(\mathbb{Z}_2;-)(\mathbb{Z}_2)(++)}$ and $\mathcal{E}_0^{(\mathbb{Z}_2;-)(\mathbb{Z}_2)(--)}$ are attached to $S_1^{(\mathbb{Z}_2;-)}$. The operator $\mathcal{E}_0^{(\mathbb{Z}_2;-)(\text{id})(+)}$ is charged non-trivially under the induced $\mathbb{Z}_2$ 0-form symmetry of $\mathcal{E}_1^{(\mathbb{Z}_2)(\text{id})}$. The ends $\mathcal{E}_0^{(\mathbb{Z}_2;-)(\mathbb{Z}_2)(ss')}$ can be converted into the end $\mathcal{E}_0^{(\mathbb{Z}_2;-)(\text{id})(+)}$ by sandwich construction inside $\mathsf{B}_{2\text{-Rep}(\mathbb{Z}_2)}^{\text{sym}}$ involving condensation defect $S_2^{(\mathbb{Z}_2)}$.

Correspondingly, a multiplet $\mathcal{M}_0^{(\mathbb{Z}_2;-)}$ of local operators of a $\mathbb{Z}_2$ 1-form symmetric 3d theory $\mathfrak{T}_\sigma$ transforming in the 0-charge $Q_1^{(\mathbb{Z}_2;-)}$ comprises of local operators

$$
\begin{array}{ccc}
\mathcal{O}_0^{(\mathbb{Z}_2;-)(\mathrm{id})(+)}, & \mathcal{O}_0^{(\mathbb{Z}_2;-)(\mathbb{Z}_2)(++)}, & \mathcal{O}_0^{(\mathbb{Z}_2;-)(\mathbb{Z}_2)(--)}, \\
\mathcal{O}_0^{(\mathbb{Z}_2;-)(\mathrm{id})(-)}, & \mathcal{O}_0^{(\mathbb{Z}_2;-)(\mathbb{Z}_2)(+-)}, & \mathcal{O}_0^{(\mathbb{Z}_2;-)(\mathbb{Z}_2)(-+)},
\end{array}
\tag{488}
$$

where $\mathcal{O}_0^{(\mathbb{Z}_2;-)(\mathrm{id})(\pm)}$ live between two genuine line operators $\mathcal{O}_1^{(\mathbb{Z}_2)(\mathrm{id})(1)}$ and $\mathcal{O}_1^{(\mathbb{Z}_2)(\mathrm{id})(2)}$ living in multiplets

$$
\begin{aligned}
\mathcal{M}_1^{(\mathbb{Z}_2)(1)} &= \left\{ \mathcal{O}_1^{(\mathbb{Z}_2)(\mathrm{id})(1)}, \mathcal{O}_1^{(\mathbb{Z}_2)(\mathbb{Z}_2)(+)(1)}, \mathcal{O}_1^{(\mathbb{Z}_2)(\mathbb{Z}_2)(-)(1)} \right\}, \\
\mathcal{M}_1^{(\mathbb{Z}_2)(2)} &= \left\{ \mathcal{O}_1^{(\mathbb{Z}_2)(\mathrm{id})(2)}, \mathcal{O}_1^{(\mathbb{Z}_2)(\mathbb{Z}_2)(+)(2)}, \mathcal{O}_1^{(\mathbb{Z}_2)(\mathbb{Z}_2)(-)(2)} \right\},
\end{aligned}
\tag{489}
$$

both transforming in the same 1-charge $Q_2^{(\mathbb{Z}_2)}$, and $\mathcal{O}_0^{(\mathbb{Z}_2;-)(\mathbb{Z}_2)(ss')}$ transition between the $S_2^{(\mathbb{Z}_2)}$-twisted sector lines operators $\mathcal{O}_1^{(\mathbb{Z}_2)(\mathbb{Z}_2)(s)(1)}$ and $\mathcal{O}_1^{(\mathbb{Z}_2)(\mathbb{Z}_2)(s')(2)}$ in the above described multiplets. The operators $\mathcal{O}_0^{(\mathbb{Z}_2;-)(\mathrm{id})(+)}$, $\mathcal{O}_0^{(\mathbb{Z}_2;-)(\mathbb{Z}_2)(+-)}$ and $\mathcal{O}_0^{(\mathbb{Z}_2;-)(\mathbb{Z}_2)(-+)}$ are not attached to any non-identity line operators. The operator $\mathcal{O}_0^{(\mathbb{Z}_2;-)(\mathrm{id})(-)}$ is attached to $D_1^{(-)}$, and the operators $\mathcal{O}_0^{(\mathbb{Z}_2;-)(\mathbb{Z}_2)(++)}$ and $\mathcal{O}_0^{(\mathbb{Z}_2;-)(\mathbb{Z}_2)(--)}$ are attached to $D_1^{(\mathbb{Z}_2;-)}$.

The local operators $\mathcal{O}_0^{(\mathbb{Z}_2;-)(\mathbb{Z}_2)(ss')}$ can be converted into the operator $\mathcal{O}_0^{(\mathbb{Z}_2;-)(\mathrm{id})(+)}$ by sandwich construction involving condensation surface $D_2^{(\mathbb{Z}_2)}$ inside the 3d theory $\mathfrak{T}$.

A key property of the local operator $\mathcal{O}_0^{(\mathbb{Z}_2;-)(\mathrm{id})(+)}$ is that it anti-commutes with the action of $\mathbb{Z}_2$ 0-form symmetries induced on lines $\mathcal{O}_1^{(\mathbb{Z}_2)(\mathrm{id})(1)}$ and $\mathcal{O}_1^{(\mathbb{Z}_2)(\mathrm{id})(2)}$ from the $\mathbb{Z}_2$ 1-form symmetry of $\mathfrak{T}_\sigma$, as shown here

$$
\left.\begin{array}{c}
\mathcal{O}_1^{(\mathbb{Z}_2)(\mathrm{id})(2)} \\
\mathcal{O}_0^{(\mathbb{Z}_2;-)(\mathrm{id})(+)} \\
\mathcal{O}_1^{(\mathbb{Z}_2)(\mathrm{id})(1)}
\end{array}\right|_{D_1^{(-)}} \quad = \quad (-1) \times \quad \left.\begin{array}{c}
\mathcal{O}_1^{(\mathbb{Z}_2)(\mathrm{id})(2)} \\
\overline{D_1^{(-)}} \\
\mathcal{O}_0^{(\mathbb{Z}_2;-)(\mathrm{id})(+)} \\
\mathcal{O}_1^{(\mathbb{Z}_2)(\mathrm{id})(1)}
\end{array}\right| .
\tag{490}
$$

**Other 0-charges.** These are analogous to the above-discussed 0-charges, and we leave the analysis of the detailed structure of these 0-charges to interested readers.

### 5.3.3 Realization in 3d pure $\widetilde{SU}(3)$ gauge theory

From now on, we restrict ourselves to the analysis of 1-charges and their realizations. Also we only discuss realizations of these charges involving non-identity operators.

Consider 3d pure $\widetilde{SU}(3)$ gauge theory obtained by gauging $\mathbb{Z}_2$ outer-automorphism symmetry of the 3d pure $SU(3)$ gauge theory discussed in section 5.2.4. This theory has a dual $\mathbb{Z}_2$ 1-form symmetry arising from the gauging. The line operators realizing the 1-charges in the $SU(3)$ gauge theory also lie in the multiplets associated to the same respective 1-charges also in the $\widetilde{SU}(3)$ gauge theory. Let us observe this in more detail.

**1-charge $Q_2^{(\mathrm{id})}$.** A representation $R_{(n,n)}$ of $SU(3)$ descends to two representations

$$
R_{(n,n)}^{(+)}, \qquad R_{(n,n)}^{(-)},
\tag{491}
$$

of $\widetilde{SU}(3)$. These two representations are related by tensoring with the one-dimensional representation $R_{(0,0)}^{(-)}$ of $\widetilde{SU}(3)$ in which the component connected to identity in $\widetilde{SU}(3)$ acts trivially,

but the component disconnected from identity in $\widetilde{SU}(3)$ acts by a sign $-1$. That is, we have

$$R_{(n,n)}^{(+)} \otimes R_{(0,0)}^{(-)} \cong R_{(n,n)}^{(-)} . \tag{492}$$

The corresponding Wilson lines $W_1^{R_{(n,n)}^{(\pm)}}$ provide respectively the lines $\mathcal{O}_1^{(\text{id})(\pm)}$ in a multiplet $\mathcal{M}_1^{(\text{id})}$ transforming in 1-charge $Q_2^{(\text{id})}$.

In fact, the Wilson line $W_1^{R_{(0,0)}^{(-)}}$ is the generator $D_1^{(-)}$ of the $\mathbb{Z}_2$ 1-form symmetry.

**1-charge $Q_2^{(\mathbb{Z}_2)}$.**   As $\mathbb{Z}_2$ outer-automorphism relates the irreducible representations $R_{(m,n)}$ and $R_{(n,m)}$ of $SU(3)$, the representation

$$R_{(m,n)}^{(\mathbb{Z}_2)} := R_{(m,n)} \oplus R_{(n,m)} , \tag{493}$$

of $SU(3)$ descends to an irreducible representation of $\widetilde{SU}(3)$.

The corresponding Wilson line $W_1^{R_{(m,n)}^{(\mathbb{Z}_2)}}$ provides the lines $\mathcal{O}_1^{(\mathbb{Z}_2)(\text{id})}$ in a multiplet $\mathcal{M}_1^{(\mathbb{Z}_2)}$ transforming in 1-charge $Q_2^{(\mathbb{Z}_2)}$.

**1-charge $Q_2^{(V)}$.**   The vortex line $V_1$ for $\widetilde{SU}(3)$ is now a genuine line operator because the $\widetilde{SU}(3)$ is the gauge group. Moreover, since it was in twisted sector of the $\mathbb{Z}_2$ 0-form symmetry in the $SU(3)$ gauge theory, it is now charged non-trivially under the dual $\mathbb{Z}_2$ 1-form symmetry of the $\widetilde{SU}(3)$ gauge theory.

The lines

$$\mathcal{O}_1^{(V)(+)} \equiv V_1 , \qquad \mathcal{O}_1^{(V)(-)} \equiv V_1 \otimes W_1^{R_{(0,0)}^{(-)}} , \tag{494}$$

participate in a multiplet $\mathcal{M}_1^{(V)}$ transforming in 1-charge $Q_2^{(V)}$.

**1-charge $Q_2^{(V\mathbb{Z}_2)}$.**   In the $SU(3)$ gauge theory, the vortex lines $V_1^{(\omega)}$ and $V_1^{(\omega^2)}$ for $\widetilde{PSU}(3)$ are exchanged by the $\mathbb{Z}_2$ 0-form action, and moreover are in the twisted sector. Consequently, after gauging, we obtain a genuine line

$$V_1^{(\omega\omega^2)} , \tag{495}$$

in the $\widetilde{SU}(3)$ gauge theory whose pre-image in the $SU(3)$ gauge theory is the non-simple vortex line

$$V_1^{(\omega)} \oplus V_1^{(\omega^2)} . \tag{496}$$

Moreover, $V_1^{(\omega\omega^2)}$ is charged non-trivially under the $\mathbb{Z}_2$ 1-form symmetry of the $\widetilde{SU}(3)$ gauge theory.

The line $V_1^{(\omega\omega^2)}$ provides the line $\mathcal{O}_1^{(V\mathbb{Z}_2)(\text{id})}$ participating in a multiplet $\mathcal{M}_1^{(V\mathbb{Z}_2)}$ transforming in 1-charge $Q_2^{(V\mathbb{Z}_2)}$.

# 6 Conclusions and outlook

The main statement of this paper, that higher-charges of a symmetry $\mathcal{S}$ are topological defects of the SymTFT, or equivalently, elements of the Drinfeld center of $\mathcal{S}$ is applicable in any dimension. Section 3 does not make any assumptions about the dimension of the theory $\mathfrak{T}$. We

then explored the 2d and 3d theories and their symmetries, providing a construction of the SymTFT and Drinfeld center, i.e. higher charges, and gauging of symmetries from the SymTFT perspective, in sections 4 and 5 respectively. The SymTFT approach also provides us with a characterization of $\mathcal{S}$-symmetric TQFTs, as shown in section 2.8.

We have seen that the SymTFT sandwich construction is very powerful and captures the salient features of the symmetry structure of a theory, with the symmetries and the physical theory implemented in terms of boundary conditions. However, as with any sandwich, the "flavor" is in the middle, i.e. the charges come from the bulk SymTFT and its topological defects.

With symmetries $\mathcal{S}$ characterized in terms of fusion higher-categories, and their charges in terms of the Drinfeld center $\mathcal{Z}(\mathcal{S})$ of the symmetries, this prepares the floor for studying interesting applications in the IR of these symmetries and charges. In 2d many interesting results are known that utilize the power of non-invertible symmetries, see e.g. [41, 43] in 2d. We will apply the results of the current paper to such IR-constraints in [17, 18].

In $d \geq 3$ by now many examples of constructions of non-invertible symmetries exist, see for instance [4–10, 21, 22, 24, 25, 46, 47, 60–63, 74–111]. It would be exciting to study the higher-charges of this wealth of symmetries and their IR implications. It would be particularly interesting to develop this in conjunction with the classification results on higher-fusion categories, thus resulting in a comprehensive take on all symmetries and their charges.

## Acknowledgments

We especially thank David Jordan for detailed discussions on the mathematical aspects underpinning this work. We are also thankful to Fabio Apruzzi, Federico Bonetti, Lea Bottini, Mathew Bullimore, Thibault Décoppet, Andrea Ferrari, Dan Freed, Theo Johnson-Freyd, Dewi Gould, Nitu Kitchloo, Greg Moore, Daniel Pajer, Ingo Runkel, Constantin Teleman, Apoorv Tiwari, Jingxiang Wu and Matthew Yu for discussions. We are grateful to the Aspen Center for Physics, the Simons Center for Geometry and Physics, the KITP Santa Barbara, and the International Centre for Theoretical Sciences for hospitality while this work was in preparation.

**Funding information** This work is supported in part by the European Union's Horizon 2020 Framework through the ERC grants 682608 (LB, SSN) and 787185 (LB). SSN acknowledges support through the Simons Foundation Collaboration on "Special Holonomy in Geometry, Analysis, and Physics", Award ID: 724073, Schäfer-Nameki, and the EPSRC Open Fellowship EP/X01276X/1.

## A  Notation and terminology

Let us collect some key notations and terminologies used in this paper – in addition to the ones used in appendix A of [1]. Throughout $\mathfrak{T}$ is a theory in $d$ spacetime dimensions.

- $\mathcal{S}$: The Symmetry, usually a fusion higher-category.

- $D_p$: Topological defect of dimension $p$ in $\mathfrak{T}$.

- $\mathfrak{Z}(\mathcal{S})$: The Symmetry TFT for the symmetry $\mathcal{S}$.

- $\mathcal{Z}(\mathcal{S})$: The topological defects of the SymTFT $\mathfrak{Z}(\mathcal{S})$, i.e. the Drinfeld center of $\mathcal{S}$.

- $\mathfrak{B}_{\mathcal{S}}^{\mathrm{sym}}$: Topological or symmetry boundary condition for the SymTFT.

- $\mathfrak{B}_{\mathfrak{T}}^{\mathrm{phys}}$: Physical boundary condition for the SymTFT.

- $S_p$: Topological defects on the symmetry boundary $\mathfrak{B}_{\mathcal{S}}^{\mathrm{sym}}$

- $Q_{q+1}$: $(q+1)$-dimensional topological operator of the SymTFT, i.e. an element of the Drinfeld center $\mathcal{Z}(\mathcal{S})$.

- $\mathcal{E}_q$: $q$-dimensional operator on $\mathfrak{B}_{\mathcal{S}}^{\mathrm{sym}}$ at the end of $Q_{q+1}$.

- $\mathcal{M}_q$: $q$-dimensional operator (multiplet) on $\mathfrak{B}_{\mathfrak{T}}^{\mathrm{phys}}$ at the end of $Q_{q+1}$.

- $\sigma$ : Tensor functor from $\mathcal{S}$ to the symmetry category $\mathcal{S}(\mathfrak{T})$. This functor describes how the (abstract) symmetry $\mathcal{S}$ is realized as a symmetry of the theory $\mathfrak{T}$.

- $\mathfrak{T}_\sigma$ : $\mathcal{S}$-symmetric theory, i.e. a tuple $(\mathfrak{T}, \sigma)$.

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
