# Peer review of "Generalized Charges, Part II: Non-Invertible Symmetries and the Symmetry TFT"

_SciPost Physics, doi:SciPost Phys. 19, 098 (2025)_

## Round 2 · Referee Report · Anonymous (Referee 1) · 2024-3-26

Report

The paper discusses finite generalized global symmetries and their representations from the perspective of symmetry topological field theory (SymTFT). The main proposal is that there is a correspondence between the set of representations for $q$-dimensional defects, which are referred to as "$q$-charges" in the manuscript, and the set of $(q+1)$-dimensional topological defects in the corresponding SymTFT.

The authors convincingly argue why such a correspondence is physically natural, especially through the discussions in Section 3.1. The proposal is made for arbitrary spacetime dimensions, and explicit examples are given in 1+1 and 2+1 dimensions.

Perhaps it needs to be emphasized, however, that such an observation and the main physical insight in the context of fusion category symmetries of 1+1d quantum field theories had already appeared in the literature previously, in particular in https://arxiv.org/pdf/2208.05495.pdf (please see more on this below).

The main technical advance made in the current manuscript in that regard is that the similar structure is also demonstrated in the 2+1d examples with fusion 2-category symmetries discussed in Section 5.

The referee believes that the main proposal made by the authors is convincing and expected to be true in general spacetime dimensions (with the caveat that some of the necessary mathematical concepts may need to be defined properly in higher dimensions, please see below). Moreover, the paper sets a natural stage for further developments in the study of infrared phases of condensed matter systems having generalized global symmetries, some of which have already appeard in the literature. The paper therefore deserves to be published in SciPost, once the comments below are addressed.

Requested changes

(1) Throughout the paper, fusion $n$-categories and their Drinfeld centers for arbitrary values of $n$ are mentioned frequently. However, it is not very clear if such mathematical concepts have actually been properly defined for $n>2$ in the mathematics community, although the physics picture provided by the authors is clear.

If they have been defined mathematically even for $n>2$ (relevant for 3+1d and higher), it would be helpful for the interested readers if the corresponding math papers are cited. If they are not yet defined mathematically, then explicitly acknowledging this somewhere in the manuscript would help providing the readers with a better idea on the current state of the arts on the math side.

(2) In page 61, there is a following sentence: "However the perspective of 0-charges of the symmetry S as topological line defects in the SymTFT is new."

However, to the best of referee's knowledge, such an obeservation was first made in the physics literature in an earlier paper https://arxiv.org/pdf/2208.05495.pdf, based on results from the math literature in the 90's and early 2000. Maybe the above sentence needs to be modified accordingly.

In referee's opinion, the above paper by Lin et al deserves to be mentioned even around the Main Statement (i.e. around Eq. (1.1)) of the current manuscript in the Introduction, as the first paper where the same statement has been explained in physics terms for the special case of $d=2$.

(3) The normalization for the condensation defects in (1.17) and (1.18) are incorrect, which is important for the locality of the defect (e.g. to have a well-defined defect Hilbert space). The denominator must be either $|H^0 (M_2,\mathbb{Z}_N)|$ or $\sqrt{ |H_1 (M_2, \mathbb{Z}_N)| }$. The two choices are related by an Euler counterterm on the defect.

Below are minor typos:
- In the second line on page 14, the 2-morphisms must be "Topological defects of codimension-3" rather than codimension-2.
- In Eq. (1.4), maybe an $i$ is missing compared to (1.3)?

---

## Round 2 · Referee Report · Anonymous (Referee 2) · 2024-9-22

Report

This paper discusses the generalized notion of charges of non-invertible symmetries in general dimensions. The generalized charges of 2d theories has been discussed in Lin-Tachikawa-Seifnashri (LTS), while the generalization to higher dimensions is novel and interesting. The presentation is pedagogical and self-contained, and contains sufficient examples for the readers to follow. Hence I would like to recommend it for publication.

However, the discussion about generalized charges is not quite complete, at least when the QFT is 2d. In this case the SymTFT is 3d, and has both topological lines and surfaces (condensation defects) as discussed in sec 2. The authors identified the topological lines in the SymTFT as the 0-charges in sec 4. But the role of topological surfaces as 1-charges is not fully discussed (as far as the referee can tell), despite it was discussed in 3d QFTs with higher form symmetries. Since the 1-charge is already present in 2d QFTs with 0-form symmetries only, it is useful to discuss it in the minimal setup (i.e. in sec 4).

Recommendation

Ask for minor revision

---

## Round 3 · Referee Report · Anonymous (Referee 2) · 2025-9-7

Report

The authors addressed my comments, hence I would like to recommend the article for publication.

Recommendation

Publish (surpasses expectations and criteria for this Journal; among top 10%)

---

## Round 3 · Author Response

We thank the referees for reviewing this manuscript and providing their expert insights.
We have made the modifications based on their suggestions and hope that the paper would now be fit for publication. Please let us know if further modifications are needed.

For referee 2:

We have clarified why we don't study (d-1) charges in footnote in Statement 3.2.

For referee 1:

(1) We have added some citations at the beginning of section 3.2.

(2) We have modified the sentence with appropriate citation.

(3) Thank you, we have modified the equations.

Typos have also been corrected. Thank you.

---

## Editorial Decision

published